palaeontology

cephalopods, Lilliput effect, Pliensbachian–Toarcian boundary event, Toarcian oceanic anoxic event, climate warming, computed tomography

**Author for correspondence:**
Patrícia Rita
e-mail: patricia.rita@fau.de

# Mechanisms and drivers of belemnite body-size dynamics across the Pliensbachian–Toarcian crisis

Patrícia Rita[1,2], Paulina Nätscher[1], Luís V. Duarte[2,3], Robert Weis[4] and Kenneth De Baets[1]

[1]Geozentrum Nordbayern, Friedrich-Alexander-Universität Erlangen-Nürnberg, Erlangen 91054, Germany
[2]MARE (Marine and Environmental Sciences Centre), 3004-517 Coimbra, Portugal
[3]Department of Earth Sciences, University of Coimbra, 3070-790 Coimbra, Portugal
[4]National Museum of Natural History Luxembourg, Department of Palaeontology, 2160 Luxembourg, Luxembourg

PR, 0000-0002-7839-2433; KDB, 0000-0002-1651-321X

Body-size reduction is considered an important response to current climate warming and has been observed during past biotic crises, including the Pliensbachian–Toarcian crisis, a second-order mass extinction. However, in fossil cephalopod studies, the mechanisms and their potential link with climate are rarely investigated and palaeobiological scales of organization are not usually differentiated. Here, we hypothesize that belemnites reduce their adult size across the Pliensbachian–Toarcian boundary warming event. Belemnite body-size dynamics across the Pliensbachian–Toarcian boundary in the Peniche section (Lusitanian Basin, Portugal) were analysed based on the newly collected field data. We disentangle the mechanisms and the environmental drivers of the size fluctuations observed from the individual to the assemblage scale. Despite the lack of a major taxonomic turnover, a 40% decrease in rostrum volume is observed across the Pliensbachian–Toarcian boundary, before the Toarcian Oceanic Anoxic Event where belemnites go locally extinct. The pattern is mainly driven by a reduction in adult size of the two dominant species, *Pseudohastites longiformis* and *Passaloteuthis bisulcata*. Belemnite-size distribution is best correlated with fluctuations in a palaeotemperature proxy (stable oxygen isotopes); however, potential indirect effects of volcanism and carbon cycle perturbations may also play a role. This highlights the complex interplay between environmental stressors (warming, deoxygenation, nutrient input) and biotic variables (productivity, competition, migration) associated with these hyperthermal events in driving belemnite body-size.

# 1. Introduction

Body-size is a key feature of any organism, reflecting its physiology, ecology and evolutionary history, across multiple scales of biological organization [1]. Body-size reduction has been considered the third universal response to global warming, after changes in phenology and species distribution [2–4]. However, disentangling body-size responses to warming might not be straightforward due to the interactions between biotic and abiotic factors involved in climate warming episodes [5,6], even in the Jurassic, when additional factors, like human activities, were absent [6,7]. Many factors during climate warming (increased temperature, decreased oxygenation and ocean acidification) are likely to work in synergy in leading to reductions in size [6]. However, increasing nutrient supply, for example, might have the opposite effect, causing a body-size increase in some taxa [4,5].

Moreover, individual responses to warming might be even population-specific due to individual environmental requirements (availability of nutrients and oxygen) and metabolic specifications [8]. Individual living cephalopod taxa, such as squid (analogous to the fossil belemnites), as stenothermal organisms, respond rapidly to warming events, hatching at smaller size, undergoing fast growth rates over shorter lifespans and maturing younger at smaller size [9–11].

Reductions in body-size—coined the Lilliput effect [12]—have been widely reported from mass extinctions and evolutionary crises associated with environmental perturbations [13,14] including the Early Toarcian crisis [15]. However, their mechanisms and environmental drivers are still poorly understood [14], especially because the effects of environmental perturbations on an organism's body-size might depend on the biological scale of organization considered [2]. For instance, over longer evolutionary time scales, i.e. considering fossil assemblages, several mechanisms might contribute to the Lilliput effect, including increased mortality of juveniles, extinction or temporary disappearance of large taxa, preferential survival or origination of small-sized taxa or the temporary reduction in adult body-size of the surviving taxa [13,14]. All of these mechanisms could explain the Lilliput effect *sensu lato*. However, only the latter, i.e. a within-lineage body-size decrease, is considered to be the mechanism behind the Lilliput effect *sensu stricto*, which depicts a reduction in body-sizes within individual species through time [12]. Interestingly, extinction risk in marine molluscs during multiple mass extinctions has been associated with preferential extinction of smaller rather than larger species including the end-Permian and end-Cretaceous mass extinctions [7].

The Early Toarcian crisis is a multi-phased event characterized by environmental and biodiversity perturbations. One of these pulses corresponds to the Pliensbachian–Toarcian (Pli–Toa) boundary event and the second pulse corresponds to the Toarcian Oceanic Anoxic Event (T-OAE) [16,17]. The Pli–Toa boundary event (approx. 183.7 Ma, [18]) follows the cooling and regressive event of the Late Pliensbachian (Emaciatum = Spinatum Zone) in the northern Tethys and Iberia [19]. It corresponds to an increase in seawater temperature (up to 6°C, figure 1), concomitant with the beginning of a marine transgression [21,24–27] as well as a small negative carbon isotopic excursion [22,28]. The Pli–Toa boundary event records a crisis among planktonic [29,30], benthic [31] and nektonic organisms [32], expressed by extinction and changes in abundance and diversity. Despite marked extinction in ammonites [32], the few data existing on Pliensbachian European belemnite fauna do not allow recognition of the Pli–Toa boundary event as an extinction horizon, at least in northwest European basins [33–36].

After the Pli–Toa boundary event, the Polymorphum (=Tenuicostatum) Zone corresponds to a cooling phase [19,21,37] in the northern Tethys and Iberia, although it is comparatively warmer than the Late Pliensbachian. This cooling phase is followed by the T-OAE, starting at the base of the Levisoni Zone [18] and characterized by a marked increase in the seawater temperature (up to 7.5°C, figure 1), widespread anoxia and deposition of organic-rich sediments [17,21,22,38–40]. These important changes in the ocean–atmosphere system had a major impact on marine biota, causing body-size changes and extinction of particularly pelagic predators, such as ammonite species [15,32,41] and benthic suspension-feeding fauna [16,42–46]. The T-OAE also marks a major bottleneck in belemnite diversity and abundance [47] and might have triggered belemnite provincialism among Boreal–Arctic and northwest European faunas [48–50]. The increased anoxia during the T-OAE has been interpreted to contribute to the demise of many belemnite taxa, while *Acrocoelites* might have survived and radiated in its immediate aftermath [51]. The T-OAE coincides with an abrupt negative carbon isotopic excursion (CIE) which disrupts an overarching positive excursion [38,52–54], interpreted to reflect the enhanced burial of organic carbon and its preservation at the sea bottom. Although deoxygenation has been interpreted to increase since the Pli–Toa boundary, the negative CIE and widespread black shale deposition are still interpreted to reflect the peak of anoxia during the T-OAE [55].

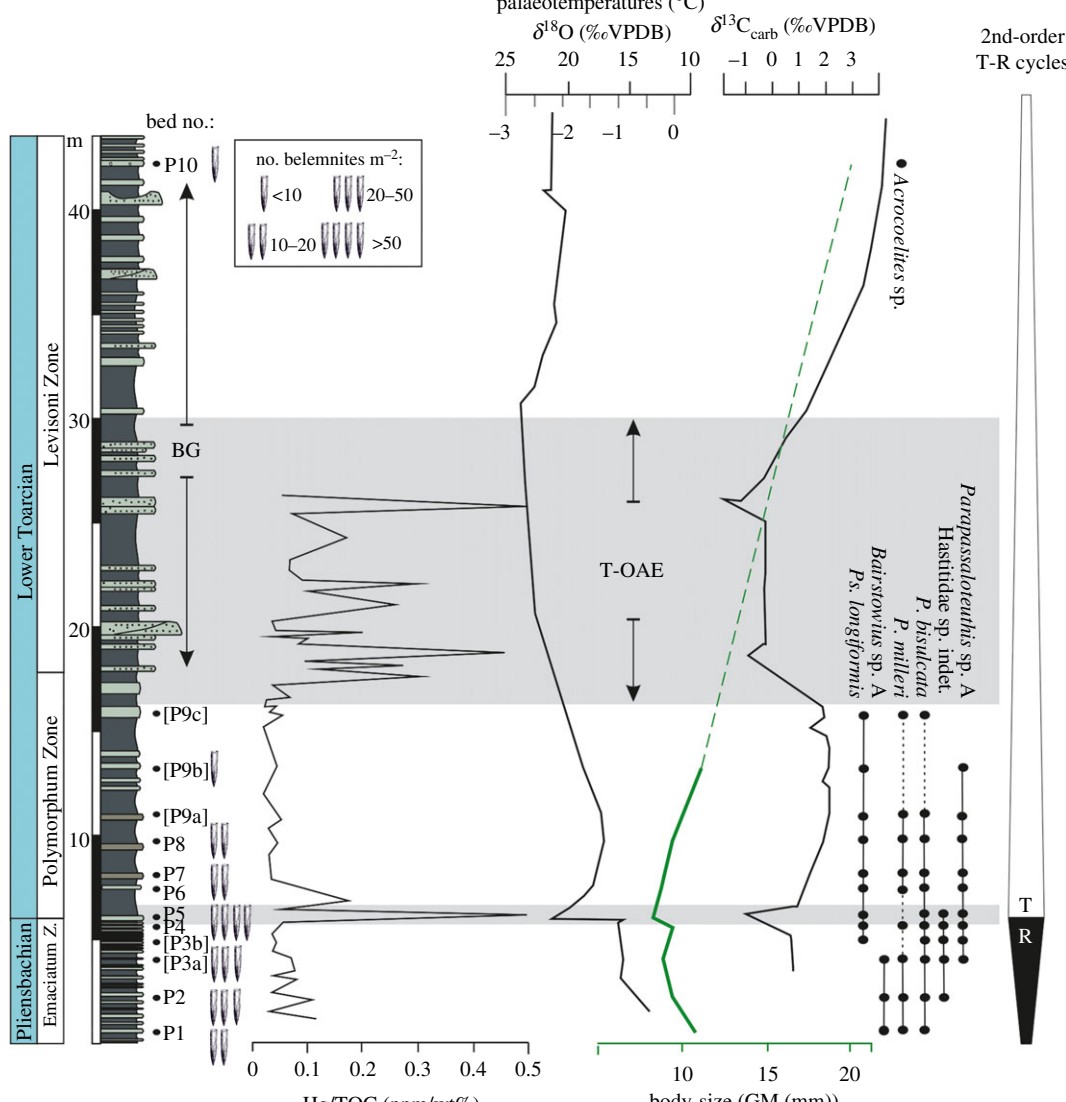

**Figure 1.** Variation of belemnite body-size (GM, assemblage scale), absolute abundance (no. belemnites m$^{-2}$) and stratigraphic ranges of species, compared with the variation of the analysed geologic proxies (mercury concentration, Hg/TOC [20], carbon and oxygen isotopes, $\delta^{13}C_{carb}$ and $\delta^{18}O$ [21,22]) and sequence stratigraphy data [23] from the Upper Pliensbachian–Lower Toarcian of Peniche. The shaded area highlights the Pli–Toa boundary event. The bed numbers in square brackets correspond to the beds that were merged due to sample size constraints, regarding the belemnite body-size analysis. See electronic supplementary material (figure S2) for details on belemnite abundance. See figure 5 for more information on lithology. Ammonite Zones: Emaciatum Zone, *Emaciaticeras emaciatum*; Polymorphum Zone, *Dactylioceras polymorphum*; Levisoni Zone, *Hildaites levisoni*. BG, belemnite gap; GM, geometric mean; T, transgressive; R, regressive.

Both warming events and the associated carbon cycle perturbations are considered a consequence of volcanic outgassing, either directly, by increasing $pCO_2$ in the seawater [56], or indirectly (i.e. by triggering other sources of isotopically light carbon [19]). In distal sections, enhanced mercury deposition is interpreted to be a proxy for constraining Karoo–Ferrar volcanism [20]. Irrespective of the controversy over the use of mercury as a proxy for volcanic outgassing [55], it is hitherto the best available proxy to temporally constrain the direct consequences of the rapid rise of $pCO_2$ in the seawater, including warming, deoxygenation and acidification.

In Peniche, mercury anomalies coincide with extreme climatic changes [20], expected to result in a body-size decrease [6,8]. By contrast, extreme climatic changes can increase weathering and precipitation, causing body-size increase instead, due to the input of nutrients [4,6]. However, despite the evidence for the increased weathering and influx of nutrients in Peniche [27], no indication of marked freshwater input or productivity increase [29] has been observed and, therefore, a reduction in body-size is expected. It is, however, worth noting that modern systems have shown us that the interaction between climate change,

nutrient input and productivity is complex and might vary regionally (e.g. [57]) and according to the scale considered [58]. Additionally, such variations might be difficult to constrain in the geological record without modelling—so far only available for a limited number of time-slices, such as at the Palaeocene–Eocene Thermal Maximum [59,60].

Belemnites are coleoids closely related to extant teuthids (such as squid). They are very abundant and diverse in the fossil record and played an important role in Jurassic ecosystems (e.g. [61]). Most of the research on Lower Jurassic belemnites was focused on the geochemistry of northwest European basins [51,62–64], temperate zones where the regional extent of anoxia was great [48,61]. The only study on belemnite body-size has been focused solely on two genera and found an ambiguous response to crisis events [15]. Therefore, a high-resolution analysis of belemnite body-size during the past episodes of environmental crisis will provide valuable insights on the response of cephalopods to the global change observed in modern marine ecosystems.

The Peniche reference section is well studied in terms of geochemistry and stratigraphy [20,22,23,27,65] and yields a highly abundant belemnite fauna, allowing for the first time the analysis of (i) the mechanisms behind belemnite body-size fluctuations, at different scales of organization (individual ontogenetic, species and assemblage), during the Pliensbachian–Toarcian interval, in subtropical latitudes and (ii) their potential environmental drivers—mainly temperature, carbon cycle and changes in $p\text{CO}_2$ associated with volcanism. We hypothesize, on the one hand, a within-lineage reduction (Lilliput effect *sensu stricto*) in belemnite body-size across the Pli–Toa boundary event and, on the other hand, a body-size increase across the T-OAE, driven by species turnover in the Peniche section.

# 2. Material and methods

## 2.1. Taxonomy and ontogeny

Belemnite species identification was based on the analysis of traditional features, such as shape (outline and profile) and the presence of grooves in the apical region  (e.g. [35]). The transverse section, the depth of penetration of the alveolus, and the apical line were all observed using CT scanning. These features were afterwards compared with the published descriptions and figures. This method also allowed us to recognize the features of each ontogenetic stage with the acquired longitudinal sections, making it possible to distinguish between adult (Neanic–Ephebic–Gerontic *sensu* [61]) and juvenile (Nepionic *sensu* [61]) specimens (electronic supplementary material, figures S1 and S2), especially by having the possibility of observing the growth increments of the rostra. The taxonomic composition of the belemnite assemblages across the studied interval is in conformity with the data from contemporaneous Tethyan sections [35,36,47,50,61,66,67]. Seven taxa compose the belemnite assemblages in the Peniche section (figure 1; electronic supplementary material, figures S1 and S2).

## 2.2. Sampling methods and belemnite body-size proxies

The specimens were collected from 13 beds (stratigraphic horizons) sampled from 45 m of the Upper Pliensbachian–Lower Toarcian sediments, corresponding to the Upper Emaciatum–Upper Levisoni ammonite zones (figure 1, approx. 2.7 Ma, [18]). As belemnites become rarer up section, the data derived from beds P9a, P9b and P9c were pooled to obtain a reasonable sample size for a quantitative analysis (figure 1). The same conservative approach was adopted for beds P3a and P3b.

We focused on quantitatively collecting from well-exposed bedding planes of limestones, marly limestones and marls. A total of 930 belemnite rostra were collected by (i) sampling all specimens within 14 one-square-meter areas and (ii) by collecting at least 30 complete specimens in the remaining bed area, i.e. outside of the quadrats. This first sampling method allowed the calculation of absolute abundance, taking into account both fragmented and complete specimens. The body-size analysis was based only on the complete specimens, regardless of the sampling method. The more complete belemnites are suspected to be a less time-averaged sample [68], because a better preservation is usually an indicator of quick burial and little transport of the fossil. Conveniently, complete specimens (with at least part of the alveolar region preserved) are easier to assign taxonomically and ontogenetically [69]. The fragmented specimens were only used to calculate the belemnite abundance per bed.

The rostrum represents a considerable part of the belemnite animal, and it is hypothesized that it acts as a counterbalance for the soft tissue and phragmocone [70], despite the increasing number of studies suggesting an original partially porous rostrum structure [71,72]. However, in all known belemnites

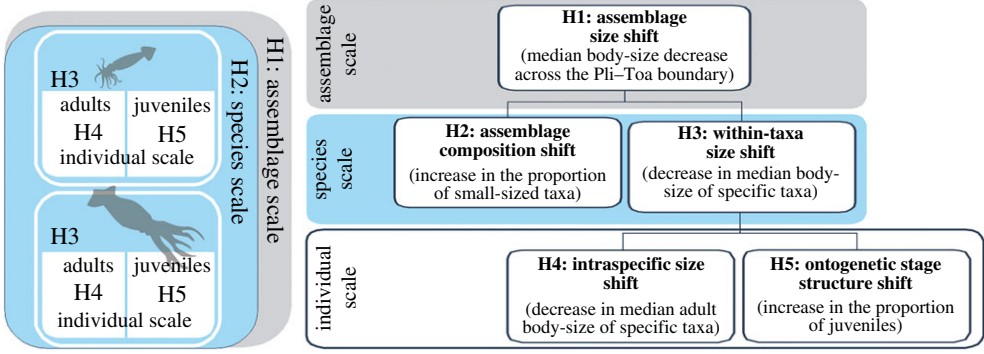

**Figure 2.** Conceptual scheme depicting the tested hypotheses regarding the mechanisms behind belemnite body-size reduction, at different palaeobiological scales of organization.

with soft tissue preservation, the fins attach to the rostrum and the soft parts closely track the outline of their internal skeleton, as in their extant relatives [73,74]. For these reasons, the rostrum can be considered a reasonable proxy for body-size in the absence of the preserved soft tissue [69]. This is also supported by a comparison between hard parts and soft parts in well-known taxa showing that larger rostra correspond to a larger mantles [75]. It is noteworthy that most of the studied specimens did not bear epirostra and, therefore, only the orthorostra were considered for the morphometric analysis. As for the epirostrum-bearing specimens, only the orthorostrum was measured.

Previous research has demonstrated that the geometric mean (GM), a unidimensional metric that combines the apical height, width and length $\left(\mathrm{GM} = \sqrt[3]{\mathrm{Dv} \times \mathrm{Dl} \times l}\right)$ is the most robust proxy to compare the body-size between morphologically different forms belonging to different species or ontogenetic stages within a species [69]. In addition, the geometric mean is more comparable to the proxies used in extant coleoids [76]. In order to calculate the GM, 277 of the complete specimens were CT scanned to allow the calculation of this metric in a non-destructive way, since the alveolus was filled with sediment, precluding direct measurement (see electronic supplementary material, table S7 for details on the settings). Additionally, 109 specimens were measured with a calliper, since the alveolus was empty. The relative change in volume better reflects changes in absolute body-size proxies used in extant taxa (e.g. weight). We, therefore, a proxy for the rostrum volume ($GM^3$) to express relative volumetric changes.

## 2.3. Mechanisms of body-size changes

The mechanisms behind the belemnite body-size fluctuations through time, from the assemblage to the individual scale of organization, were assessed by testing a set of hierarchical hypotheses (H), modified from [2] (figure 2). The first hypothesis predicts a decrease in the mean body-size at the assemblage scale, regardless of the underlying mechanisms (assemblage size shift hypothesis, H1, figure 2). If H1 is validated, there are four subsequent hypotheses that could explain this decrease [2,8]. First, an increase in the proportion of small-sized species (assemblage composition shift hypothesis, H2). Second, a decrease in the median body-size within specific taxa (within-taxa size shift hypothesis, H3), which could, in turn, be related to a decrease in adult body-size (intraspecific size shift hypothesis, H4). The last hypothesis associates the decrease in the mean body-size with an increase in the proportion of juveniles (ontogenetic stage structure shift hypothesis, H5).

The whole set of hypotheses could explain the Lilliput effect *sensu lato*, although only H4 relates to the Lilliput effect *sensu stricto*, as predicted by the temperature-size rule in extant coleoids [77]. Note that, in comparison with recent data, the interpretation of body-size changes based on fossil data is more complex because fossil assemblages reflect not only a past biotic community, but also the mortality and preservation of the organisms that were part of it.

At the assemblage scale of organization, H1 was tested by assessing the size distribution of the whole assemblage through time. The proportional change in body-size was calculated between stratigraphically consecutive beds (example: $bed_t$ and $bed_{t+1}$) in percentage of the log ratio (example: $\ln(\text{median } bed_{t+1}/\text{median } bed_t) \times 100\%$). The median belemnite body-size (geometric mean) was calculated for each bed. The whole distribution and the median were compared using the non-parametric Mann–Whitney $U$ and Kolmogorov–Smirnov tests to assess the significance

of the belemnite body-size differences between consecutive beds (electronic supplementary material, table S2). This was done at the assemblage and species scales (electronic supplementary material, table S2).

Because changes in taxonomic composition can influence the body-size patterns by the appearance and/or disappearance of taxa, we modified the within- and among-taxa approach by Rego *et al.* [78]. This method corresponds to a pair-to-pair analysis (comparison of stratigraphically successive beds, $t$ and $t+1$), focused on the taxa identified in stratigraphically successive beds (boundary crossers). This method allows division of the assemblage body-size shift (equation (2.1)) into three components: a disappearance of taxa effect (equation (2.2)), a within-lineage effect (equation (2.3)) and an appearance of new taxa effect (equation (2.4)).

$$\text{assemblage size shift} = \text{assemblage median body-size}_{\text{bed }t+1} - \text{assemblage median body-size}_{\text{bed }t}, \quad (2.1)$$

$$\text{disappearance of taxa effect} = \text{boundary crossers}_{\text{bed }t} - \text{all taxa}_{\text{bed }t}, \quad (2.2)$$

$$\text{within-lineage effect} = \text{boundary crossers}_{\text{bed }t+1} - \text{boundary crossers}_{\text{bed }t} \quad \text{and} \quad (2.3)$$

$$\text{appearance of new taxa effect} = \text{all taxa}_{\text{bed }t+1} - \text{boundary crossers}_{\text{bed }t+1}. \quad (2.4)$$

Finally, at the individual ontogenetic scale of organization, the hypotheses were tested by assessing the body-size variation within different ontogenetic stages (adults and juveniles) for individual taxa. Variation partitioning allowed us to assess the proportion of body-size (geometric mean) variation explained by ontogeny, taxonomic assignment and separation by beds and by their joint effects. If the fraction of variation corresponding to beds separation is low, this suggests that mechanisms driving differences between samples are similar and not related to particularities of the sample (e.g. lithology and age). Variation in the body-size data was partitioned using partial redundancy analysis (RDA), as implemented in the *vegan* package in R [79,80]. The significance of each fraction was assessed through the analysis of variance (ANOVA). The whole time-series was compared with the Pli–Toa boundary event.

## 2.4. Environmental drivers of body-size changes

In order to test our hypothesis of a relationship between belemnite body-size and environmental perturbations, the distribution of the GM was compared with three main geologic proxies (table 1). These are brachiopod stable oxygen isotopes [21], bulk rock carbon isotopes [22] and bulk rock mercury concentration normalized by total organic carbon [20], used as proxies for seawater palaeotemperature, negative excursions in the carbon cycle and volcanogenic outgassing, respectively (table 1). To account for the impact of sedimentary properties on body-size patterns, we corrected for the effect of lithology and belemnite absolute abundance (mean abundance of belemnites m$^{-2}$, figure 1; electronic supplementary material, figure S3) by residualizing [84]. A simple linear regression analysis is performed between the body-size and the combined effect of abundance and lithology. The residuals from this analysis are then used as outcome in the ultimate regression analysis between body-size and abiotic variables. All variables of the model were continuous with the exception of lithology. We assigned our samples a categorical variable for lithology (marl to limestone) based on carbonate/clay content. Collinearity between explanatory variables was tested by using *cor* function in R (electronic supplementary material, figure S5). Sedimentary properties and various other parameters linked with climate warming (such as strontium isotopes, total organic carbon and abundance of primary producers) could not be directly included in the models due to their high collinearity with $\delta^{13}C_{\text{carb}}$ or the other two parameters used (electronic supplementary material, figure S4); hence, the choice of the abiotic parameters is indicated in table 1.

A multiple linear regression using generalized least squares (GLS) was performed and the models were fitted by maximizing the log-likelihood (*gls* function in R). The first-order autoregressive (AR(1)) model was used, which has the property of seeking autocorrelation and of minimizing the error term in a time-series [85]. Seven models (see electronic supplementary material, R script) were compared using Akaike's second-order corrected information criterion (AICc scores), which corrects for small sample size. The power of the best model was assessed by means of an ANOVA test. The statistical significance of each coefficient of the best model was assessed by calculating the $p$-value under a $t$ approximation. The significance level was $p < 0.05$ for all the analyses, unless stated otherwise. The analyses were performed in R [80], using the packages *nlme* [86] and *qpcR* [87].

The regression analysis only included data from Emaciatum and Polymorphum zones (beds P1–P9), due to the lack of a representative sample size in the Upper Levisoni Zone (bed P10, figure 1).

**Table 1.** Environmental constraints (abiotic parameters) based on their geological proxies and their theoretical role in mediating body-size changes and their interpretation in the context of the Lusitanian Basin.

| geologic proxy [source] | environmental variable | basis | interpretation in the context of the LB | theoretical controls on body-size |
|---|---|---|---|---|
| brachiopod stable oxygen isotope ratio ($\delta^{18}O$) [21] | dominantly bottom seawater temperature | Temperature-dependent isotopic fractionation between carbonate minerals and seawater [81] | High temperatures during negative excursions | Increasing temperature through various mechanisms including the increase of both growth and development rates are expected to lead to a smaller adult body-size [8]. |
| $\delta^{13}C_{carb}$ [22] | carbon cycle perturbations related to anoxia | Isotopic fractionation between photosynthesizers and seawater. Photosynthesizers remove light $C^{12}$ from the seawater and this is incorporated into the sea bottom sediment after burial as organic carbon [82]. | Negative excursions during the Early Toarcian reflect enhanced burial of organic carbon during the zenith of anoxia [83]. They could also be influenced by increased primary production but are interpreted to reflect rather widespread oxygen depletion as primary productivity is interpreted to have dropped during these intervals [29,30]. | At the edge of an organism's temperature range, growth is usually impaired by insufficient energy or oxygen supply, decreasing both growth rate and body-size at any developmental stage [8]. |
| Hg/TOC ratio [20] | volcanogenic outgassing | Volcanism represents a source of mercury to the atmosphere. Due to rain or run-off, mercury moves from the terrestrial realm/atmosphere to the marine realm in mineral form or adsorbed in detrital organics. Hg burial is limited by low abundance and/or burial of organic matter or sulfides that would scavenge aqueous Hg [20,55]. | Hg anomalies (high Hg/TOC ratio) in the sediment are interpreted to represented markers of volcanism in distal sections. They have been interpreted to reflect volcanic outgassing, but their interpretation in some distal sections might be more complex [55]. | Increasing $pCO_2$, combined with increasing temperature, causes a decrease in both dissolved oxygen and pH of the seawater. In association with other factors related to rapid warming, such as weathering and eutrophication, these factors are expected to lead to a body-size decrease due to their direct effects on the availability of food resources [6]. |

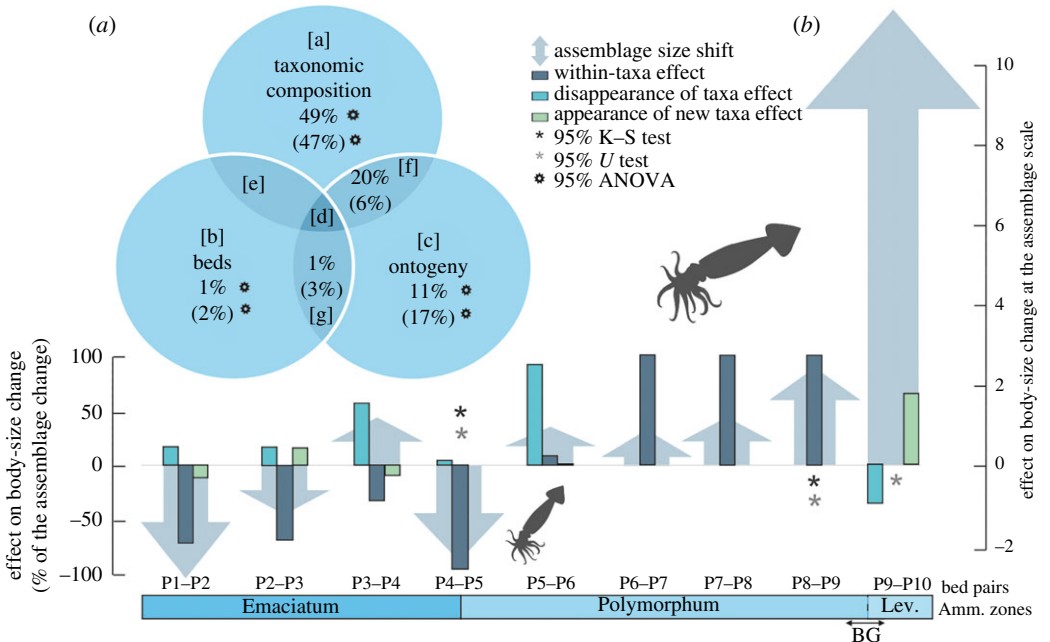

**Figure 3.** (*a*) Venn diagram depicting the partition of belemnites body-size variation between taxonomic composition, bed separation and ontogeny. The values between parentheses correspond to the whole time-series and the values without parentheses correspond to the Pli–Toa boundary. See the 'Material and methods' section for more details on the variation partitioning methodology. (*b*) Effect on body-size change at the assemblage and species scale of organization. The bed pairs in the horizontal axis represent pairwise comparisons. See electronic supplementary material (table S6) for details. For a correct interpretation of the right-side scale, the bottom of the arrowhead should be considered, rather than the tip of the arrow. Note that the *x*-axis depicts comparisons of consecutive pairs of beds (not to scale). BG, belemnite gap; *U*-test, Mann–Whitney *U*-test; K–S test, Kolmogorov–Smirnov test.

## 3. Results

### 3.1. Belemnite body-size fluctuations

#### 3.1.1. Assemblage scale

A total of seven belemnite species were identified in Peniche (figure 1). Most of the taxa range from the uppermost Pliensbachian to the Lower Toarcian (uppermost Polymorphum Zone). *Bairstowius* sp. A is exclusively represented in the uppermost Pliensbachian, being stratigraphically replaced by *Pseudohastites longiformis* in the assemblage. In the Toarcian, the Levisoni Zone is devoid of belemnites, with the exception of bed P10 (Upper Levisoni Zone) which includes two specimens assigned to *Acrocoelites* sp. (figure 1).

Three episodes of the median rostrum size decrease were recognized at the assemblage scale across the studied interval: between beds P1 and P2, between bed P2 and P3 and at the Pli–Toa boundary event (beds P4 and P5). Only the latter corresponds to a statistically significant decrease (*p*-value Kolmogorov–Smirnov test = 0.01; *p*-value Mann–Whitney *U*-test = 0.04; figure 3*b*) and corresponds to a 13% body-size decrease (figure 4). This decrease in the belemnite body-size proxy (GM) corresponds to a 40% decrease in rostrum volume (electronic supplementary material, table S2).

The assemblage body-size shift at the Pli–Toa boundary event is almost exclusively caused by the boundary crossers *Ps. longiformis*, *P. bisulcata* and Hastitidae sp. indet. (within-taxa effect; figure 3*b*). The taxonomic composition across the boundary does not change markedly, with *Ps. Longiformis*, the most abundant taxon, comprising 61.9% (bed P4) and 56.2% (bed P5) of the assemblages (figures 1 and 4). The disappearance of *Passaloteuthis milleri* and *Parapassaloteuthis* sp. at the Pli–Toa boundary event causes 4% of the decrease in the assemblage median body-size (disappearance of taxa effect; figure 3*b*).

By contrast, the other two episodes of body-size reduction (between beds P1 and P2 and between beds P2 and P3) are mainly related to the appearance (15.9 and 16.2%, respectively) and disappearance of taxa (12.1 and 15.2%, respectively), but also to the body-size decrease in specific taxa (within-taxa effect; (figure 3*b*). The within-taxa effect on the assemblage body-size shift between beds P1 and P2 and between beds P2 and P3 (72%; 69%) is, however, lower than at the Pli–Toa boundary (96%, figure 3).

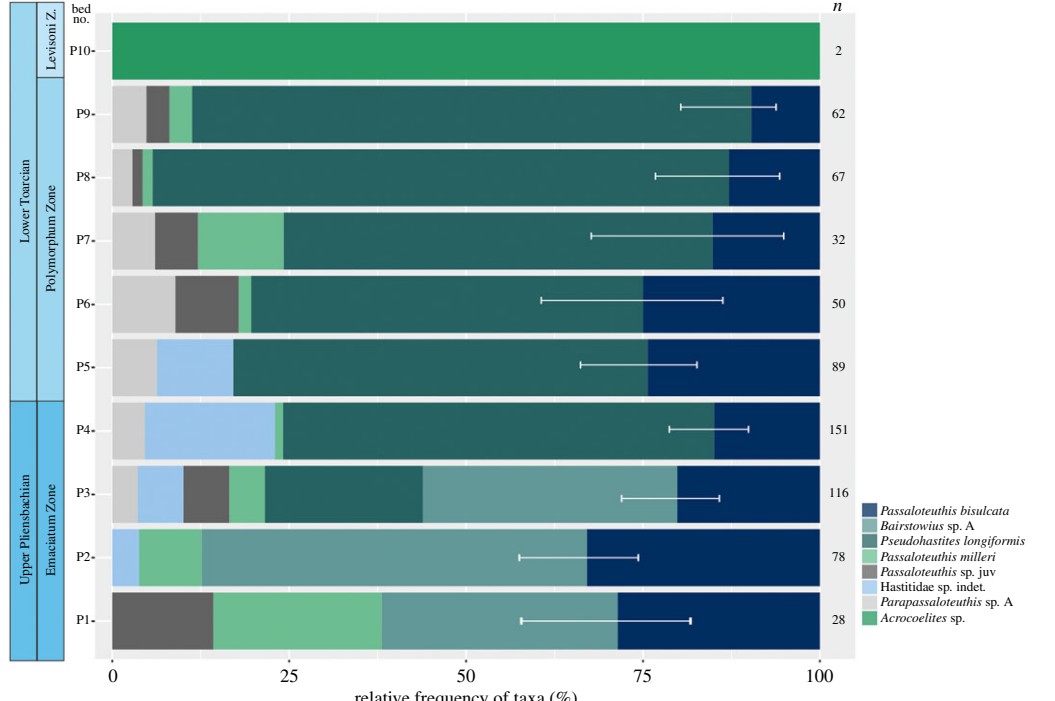

**Figure 4.** Relative frequency of the different taxa comprising the Upper Pliensbachian–Lower Toarcian Peniche belemnite assemblage. Note that determinable incomplete specimens were also taken into account and so sample size (*n*) differs between figures 4 and 5. The error bars correspond to the 95% confidence interval of the relative frequency of the species *Ps. longiformis/Bairstowius* sp. A and *P. bisulcata*, the most abundant taxa.

At the assemblage scale, an increase in the median body-size is observed between beds P3 and P4, P5 and P6, P7 and P8, P8 and P9 and between P9 and P10. However, only the bed pairs P8–P9 and beds P9–P10 correspond to significantly different body-sizes (figure 3*b*).

In sum, across the studied interval at the assemblage scale, a significant decrease in belemnite body-size is observed across the Pli–Toa boundary (beds P4 and P5). After that, from bed P5 to bed P9, a general increase in belemnite body-size is observed, followed by a belemnite gap, from bed P9 to bed P10 (figures 3*b* and 5), during which no belemnites occur.

### 3.1.2. Species scale

Three taxa (*Ps. longiformis*, *P. bisulcata* and Hastitidae sp. indet.) are recorded immediately before and immediately after the Pli–Toa boundary event (boundary crossers; figure 1). The most significant and largest reduction in size is observed in *Ps. longiformis*, the most abundant taxon (17% decrease in GM and 51% in rostrum volume; figure 6).

Despite the lack of statistical significance, *P. bisulcata*, the second most abundant taxon, also decreases in size (7% in GM and 21% in volume, figure 6) across the Pli–Toa boundary event, while Hastitidae sp. indet., slightly increases in size (8%, electronic supplementary material, figure S7). It is noteworthy that *P. bisulcata* body-size markedly decreases (109%, figure 6) immediately after the Pli–Toa boundary event, from bed P5 to bed P6 (lowermost Polymorphum Zone), which is coincident with the extirpation of Hastitidae sp. indet. (figure 1).

### 3.1.3. Individual ontogenetic scale

Considering the whole time-series, the variation partitioning analysis results revealed that taxonomic composition (47%) and ontogeny (17%) are responsible for most of the variation in belemnite body-size. The separation by bed explains only 2% of the body-size variation (figure 3*a*, values between parentheses).

At the Pli–Toa boundary event, the body-size variation partitioning into taxonomy and ontogeny is similar to what is observed for the whole time-series (49 and 11%, respectively). However, the joint effect of taxonomy and ontogeny explains 20% of the variation in body-size, in contrast with the 6% observed for the whole time-series (figure 3*a*, values without parentheses). This probably suggests that across the Pli–Toa boundary event, ontogenetic stages behave more similarly among taxa in explaining body-size variation.

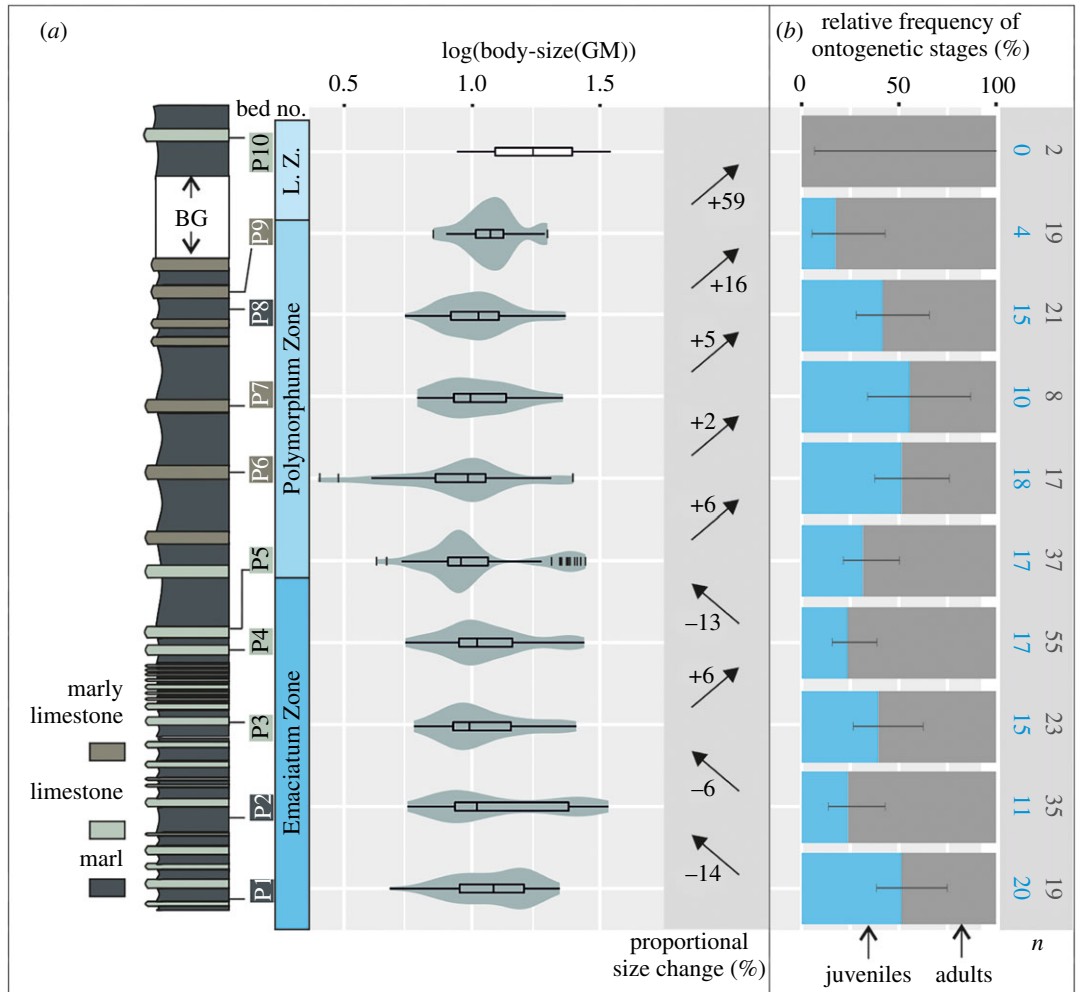

**Figure 5.** (*a*) Lithology, belemnite body-size variation (violin plots) and proportional body-size change across the studied interval in Peniche. (*b*) Relative frequency of ontogenetic stages at the assemblage scale across the studied interval in Peniche. Note that the stratigraphic log is not drawn to scale, for real thickness of beds, see figure 1. The error bars correspond to the 95% confidence interval of the juveniles and adults ratio. BG, belemnite gap; GM, geometric mean; *n*, sample size.

At the individual ontogenetic scale of organization, only the adult specimens of *Ps. longiformis* recorded a significant (Kolmogorov–Smirnov test *p*-value $= 2.61 \times 10^{-6}$; Mann–Whitney *U*-test *p*-value $= 3.98 \times 10^{-7}$) body-size decrease across the Pli–Toa boundary event (21%, electronic supplementary material, table S2). The ratio adult versus juveniles of *Ps. longiformis* does not change significantly across the Pli–Toa boundary event (31.25%, figure 6).

A reduction of 7% (although not significant) in adult body-size is observed in *P. bisulcata* specimens, across the Pli–Toa boundary event, together with a slightly increased percentage of juveniles of the same species (figure 6). Immediately after the boundary, from bed P5 to bed P6, a marked reduction (109%) in *P. bisulcata* body-size is recognized mainly among the juvenile specimens, in tandem with a decrease in their proportion (from 75 to 68.75%, figure 6).

## 3.2. Relationship with environmental parameters

After correcting for the effects of sedimentary properties (lithology and fossil abundance), the results of the regression analysis revealed that the variation in belemnite body-size is best correlated with the variation in $\delta^{18}O$ (table 2, model no. 5), at the assemblage scale. The overall model is significant (electronic supplementary material, table S3), suggesting a body-size reduction with increasing $\delta^{18}O$ values/decreasing seawater temperature values (electronic supplementary material, figure S2). However, the models combining $\delta^{18}O$ with $\delta^{13}C_{carb}$ (model no. 2, table 2) and combining Hg/TOC with $\delta^{13}C_{carb}$ (model no. 3, table 2) could not be rejected, according to the AICc scores ($\Delta < 2$). If the

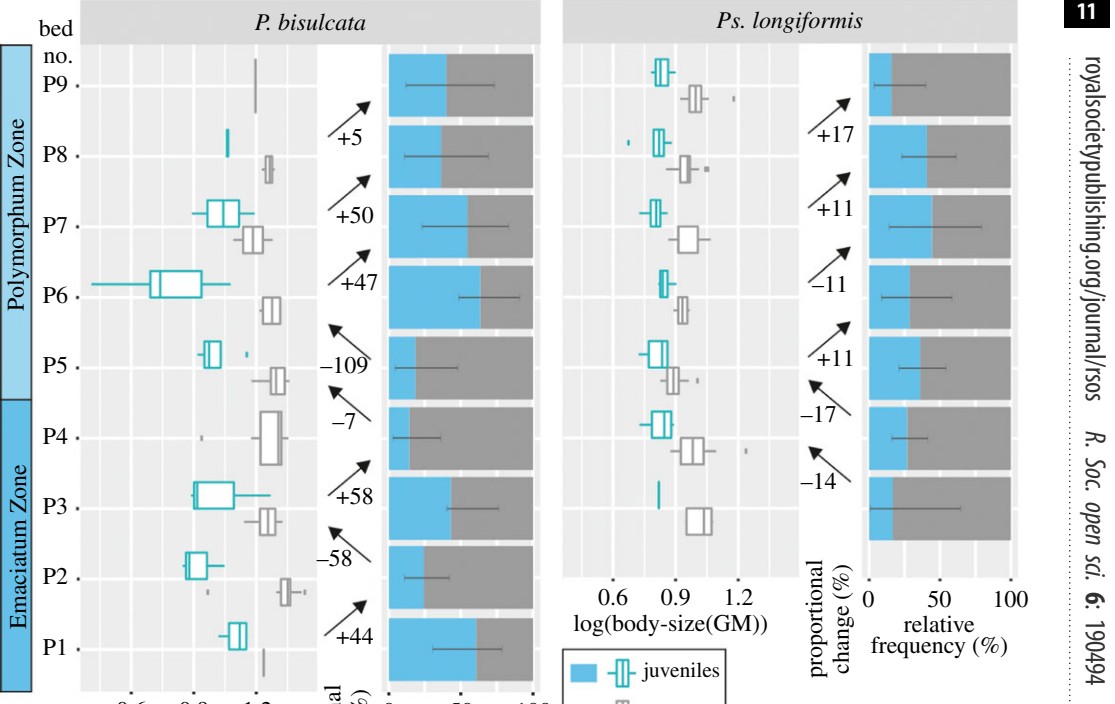

**Figure 6.** Belemnite body-size variation (GM), proportional body-size change and relative frequency of ontogenetic stages across the Upper Pliensbachian–Lower Toarcian of Peniche at species scale (*P. bisulcata* and *Ps. longiformis*). *Passaloteuthis* genus was used to calculate the relative frequency of ontogenetic stages of *P. bisulcata* due to the difficulty of a species-level classification of the juvenile specimens of *Passaloteuthis* genus. The error bars correspond to the 95% confidence interval of the juveniles and adults ratio. For sample size, see electronic supplementary material (figure S6). GM, geometric mean.

data are not corrected for the effects of the sedimentary properties, the same results are obtained, despite minor differences in significance (electronic supplementary material, table S5).

After correcting for the effects of sedimentary properties, when considering the most abundant taxon, *Ps. longiformis*, the best model explaining the body-size variation includes solely Hg/TOC (model no. 6, table 2). However, the effects of $\delta^{18}O$, $\delta^{13}C_{carb}$ and Hg/TOC cannot be discarded, since, statistically, the model nos. 5, 2 and 4 cannot be distinguished from model no. 6, according to the AICc scores. If the data are not corrected for the effects of lithology and abundance beforehand, model no. 4 (Hg/TOC + $\delta^{18}O$) best explains *Ps. longiformis* body-size variation, with highly significant results (electronic supplementary material, table S5). The role of Hg/TOC in the model is more significant than the role of $\delta^{18}O$ (electronic supplementary material, table S4b).

After correcting for the effects of lithology and fossil abundance, *P. bisulcata* body-size is best correlated with Hg/TOC (model no. 6, table 2), as well, despite the lack of significance (table 3). However, the effects of $\delta^{18}O$, $\delta^{13}C_{carb}$ and Hg/TOC cannot be disregarded, since statistically model nos. 5, 2 and 4 cannot be distinguished from model no. 6. If no correction is applied beforehand, the model no. 4 (Hg/TOC + $\delta^{18}O$) best explains the *P. bisulcata* body-size variation, with highly significant results (electronic supplementary material, table S5).

# 4. Discussion

## 4.1. Scales and mechanisms of body-size variation

### 4.1.1. Pli–Toa boundary event

Body-size reductions have been identified in several episodes of biotic/abiotic crisis, many times related to warming. Body-size fluctuations in fossil assemblages are common and can be caused by a variety of

**Table 2.** AICc ranking of models describing the effect of palaeotemperature ($\delta^{18}O$), carbon cycle perturbations ($\delta^{13}C_{carb}$) and volcanism (Hg/TOC) on belemnites body-size (GM) for the assemblage and species scale (*Ps. longiformis* and *P. bisulcata*), corrected for the effects of sedimentary properties (lithology and belemnite abundance). Only the models with $\Delta < 2$ are listed. See electronic supplementary material (table S4) for the full list of GLS models.

| | model no. | $\Delta$ | AICc scores | ANOVA *p*-value |
|---|---|---|---|---|
| assemblage | | | | |
| GM $\sim$ 1 | null | 5.09 | 1921.87 | — |
| GM $\sim \delta^{18}O$ | **5** | 0 | 1920.92 | 0.0849 |
| GM $\sim \delta^{18}O + \delta^{13}C_{carb}$ | 2 | 1.99 | 1922.91 | |
| GM $\sim$ Hg/TOC $+ \delta^{13}C_{carb}$ | 3 | 1.91 | 1922.84 | |
| *Ps. longiformis* | | | | |
| GM $\sim$ 1 | null | 2.51 | 671.37 | — |
| GM $\sim$ Hg/TOC | 6 | 0 | 668.87 | 0.0327 |
| GM $\sim \delta^{18}O + \delta^{13}C_{carb}$ | 2 | 0.87 | 669.74 | |
| GM $\sim$ Hg/TOC $+ \delta^{18}O$ | 4 | 1.31 | 670.17 | |
| GM $\sim \delta^{18}O$ | 5 | 0.50 | 669.37 | |
| *P. bisulcata* | | | | |
| GM $\sim$ 1 | null | −1.46 | 607.97 | — |
| GM $\sim$ Hg/TOC | 6 | 0 | 609.43 | 0.4306 |
| GM $\sim \delta^{13}C_{carb} +$ Hg/TOC | 3 | 1.87 | 611.30 | |
| GM $\sim \delta^{18}O$ | 5 | 0.35 | 609.78 | |
| GM $\sim \delta^{13}C_{carb}$ | 7 | 0.58 | 610.01 | |

**Table 3.** Details of the selected GLS models comparing belemnite body-size (GM) with palaeotemperature ($\delta^{18}O$), carbon cycle perturbations ($\delta^{13}C_{carb}$) and volcanism (Hg/TOC), corrected for the effects of sedimentary properties (lithology and belemnite abundance). GM, geometric mean; s.e., standard error; d.f., degrees of freedom.

| scale | model no. | coefficients | value | s.e. | *t*-value | *p*-value | d.f. | residual |
|---|---|---|---|---|---|---|---|---|
| assemblage | 5 | intercept | 2.44 | 1.48 | 1.64 | 0.1019 | 340 | 338 |
| | | $\delta^{18}O$ | 2.04 | 1.19 | 1.72 | 0.061 | — | |
| *Ps. longiformis* | 15 | intercept | 0.31 | 0.23 | 1.32 | 0.1904 | 160 | 158 |
| | | Hg/TOC | −2 | 0.93 | −2.14 | 0.0335 | — | |
| *P. bisulcata* | 5 | intercept | 0.43 | 0.94 | 0.46 | 0.6458 | 101 | 99 |
| | | Hg/TOC | −3.15 | 4.02 | −0.78 | 0.4358 | — | |

mechanisms. Our goal is to test the effect of warming and related stressors in belemnite body-size dynamics across the Pliensbachian–Toarcian crisis recoded in the Peniche section.

Statistical evaluation of our data has shown that size changes of belemnite rostra in the studied section at Peniche are significant for the Pliensbachian–Toarcian boundary event, i.e. the assemblage size shift hypothesis (H1) was validated. This size change is almost exclusively driven by a body-size reduction of single species—*Ps. longiformis*, allowing us to validate the within-taxa size shift hypothesis (H3). No marked changes in the proportion of ontogenetic stages of *Ps. longiformis* are observed at the Pli–Toa boundary event, meaning that the largest within-lineage body-size change is not due to changes in the ontogenetic structure of the population, but rather to an adult body-size reduction. In conclusion, at the individual ontogenetic scale, we were able to validate the intraspecific size-shift hypothesis (H5), across the Pli–Toa boundary event, finding a 51% decrease in rostrum volume (21% in median GM) of adult specimens of *Ps. longiformis*. By contrast, *P. bisulcata* body-size reduction is a combination of increased percentage of juveniles (increased mortality of juveniles) and reduction in adult body-size.

The Lilliput effect *sensu lato* was already identified within particular cephalopod species in the Cleveland Basin, interpreted as a response to the deteriorating environmental conditions associated with the Early Toarcian crisis [15]. However, this work does not explicitly take into account ontogeny (e.g. impact of the changes in the proportion of ontogenetic stages on body-size). Similarly, among extant squid, maturation at small size and young age is interpreted as a life-history strategy to cope with warming events [9].

The differential taxa response found in our study might indicate different environmental tolerances, related to individual physiology or life-history strategies [8]. The only important morphological difference detectable from the fossilized remains between *Ps. longiformis* and the other taxa is the presence of an epirostrum, a calcified structure which develops in late ontogenetic stages [88] as an extension of the orthorostrum. It has been interpreted that the development of an epirostrum facilitates the animals' movement in the water by gliding as counterbalance to the development of specialized reproductive organs [88,89]. The increased $pCO_2$ that characterizes the environmental crisis of the Early Toarcian in Peniche [21] could directly (through calcification) or indirectly (through nutrient availability) have affected the calcification potential of the epirostrum. However, further data need to be assembled in order to assess the physiological and environmental constraints of the development of such a structure.

According to our results, a belemnite body-size reduction is observed at the Pli–Toa boundary event in Peniche. However, no major changes in the belemnite taxonomic composition are observed (figure 4), despite being considered one of the early pulses of the Early Toarcian crisis—particularly in ammonoid cephalopods [32]. This is consistent with the data available from other regions at higher latitudes [36,37,61,67], which demonstrate that the T-OAE, by contrast, corresponds to an important turnover in belemnite species [47]. In fact, the T-OAE had greater impact on marine biota than the Pli–Toa boundary event (e.g. [16,32,41,42], namely in the Lusitanian Basin, where planktonic and benthic organisms were largely affected [29,30,90–92]. This highlights the deterioration of the environmental conditions during the Early Toarcian, starting at the Pli–Toa boundary event and culminating in the T-OAE, an interval barren of belemnites in Peniche, as indicated by our results and previous studies [65].

### 4.1.2. The aftermath of Pli–Toa boundary event

Notwithstanding the decrease in seawater temperature during the early Polymorphum Zone, this interval is characterized by palaeoenvironmental perturbations, as evidenced by the carbon isotopic record [21,22]. Moreover, seawater temperature is still higher than during the Late Pliensbachian [21]. The conditions are, however, not as severe as during the Pli–Toa boundary event, or during the onset of the T-OAE, which is also supported by the response of other groups of marine biota in the Peniche section [29,30,90–92].

*Passaloteuthis bisulcata* body-size decreases during the aftermath of the Pli–Toa boundary event (109% from bed P5 to bed P6, early Polymorphum Zone) due to an increase in the relative proportion of juveniles of that species (figure 6). This might indicate the temporary emigration of large adults [9] and/or higher juvenile mortality due to the unsuitability of the habitat, as interpreted for fish and cephalopod species [93,94]. However, we also cannot rule out the existence of smaller (stunted) adults [9] which might not develop typical adult features [95], but this could only be disentangled with further studies, namely with a sclerochronological analysis.

The decrease in *P. bisulcata* body-size in the aftermath of the Pli–Toa boundary event coincides with an increase in adult body-size of *Ps. longiformis,* and with the disappearance of Hastitidae sp. indet. It is tempting to assume that the rapid recovery of *Ps. longiformis* potentially triggered a change in the belemnite population dynamics, potentially due to the competition for food resources, causing the emigration of some species (Hastitidae sp. indet.) and the increased mortality and stress in the early growth of others (*P. bisulcata*). A shift in distribution to cooler latitudes with warming is considered one of the most important responses in modern marine ecosystems [96,97].

Furthermore, the rest of the Polymorphum Zone (i.e. from bed P6 to bed P9) is characterized by an increase in *Ps. longiformis* body-size and abundance (and less relative mortality of juveniles), relatively to *P. bisulcata.* The latter becomes less abundant and maintains its adult body-size, emphasizing a potential competition between the *Ps. longiformis* and *P. bisulcata*. However, the fact that there is no increase in the proportion of *Ps. longiformis* comparatively to the proportion of *P. bisulcata*, from bed P5 to bed P6 (figure 6), is not entirely compatible with this interpretation and highlights the constraints of testing the effects of competition and other biotic parameters on body-size in fossil assemblages.

### 4.1.3. T-OAE

The interval corresponding to the onset of the T-OAE in Peniche is barren of belemnites, starting around the Polymorphum–Levisoni zones boundary [98]. This is coincident with the beginning of the second mercury anomaly [20] and with warming [21], similar to the lowermost Toarcian conditions, although associated with stronger carbon cycle perturbations (figure 1). The belemnite gap could reflect inhospitable conditions in the Lusitanian Basin during the onset of the T-OAE. This might suggest that coleoids—even those with flexible life-history strategies, pre-adapted to such conditions, such as *Ps. longiformis*—could not cope with deteriorating conditions in subtropical epicontinental European basins, resulting in northward migration and/or local extinction during the T-OAE. This would also be consistent with the absence of belemnites in the Riff Mountains during the Levisoni Zone (compare [50,99]).

In fact, the deterioration of water-column conditions must have been worse during the T-OAE in comparison with the Pli–Toa boundary event, namely in terms of oxygenation of the water column, as evidenced by geochemical data [22], and the negative response of marine organisms in the Lusitanian Basin [29]. Moreover, a similar belemnite response was identified in coeval northwest European basins, as the belemnite gap observed in Peniche overlaps with intervals of belemnite gap, or abundant decrease, in the Cleveland, Cardigan Bay and Swabo-Franconian temperate basins. This has been previously interpreted as a response to unfavourable anoxic–euxinic water-column conditions [47,51,100].

Interestingly, the decrease in belemnite abundance, or disappearance, happens before the most severe conditions of the T-OAE are met in subtropical Peniche section (i.e. when the temperature was not the highest, figure 1), and it is not preceded by a body-size decrease. On the contrary, in the Middle and Upper Polymorphum Zone (i.e. from bed P5 to bed P9), belemnite body-size increases at the assemblage scale. However, this might also be partially related to the fact that juveniles become rare due to the unsuitability of habitat. This pattern might indicate that the combination of warming and deoxygenation affected belemnite physiology, reproduction, as well as competition for resources, even before the environmental nadir of the T-OAE. This highlights the importance of climate warming, in addition to regional anoxia, which has been argued to be a major constraint on belemnite abundance and distribution, in the northwest basins of the Tethys Ocean [47,51,100]. However, the widespread anoxia in other parts of the western Tethys (e.g. NW Europe [101]) could have indirectly affected belemnites in the non-anoxic Mediterranean domain (Lusitanian Basin and Morocco) by restricting migratory routes and/or food supply or even by inhibiting their ability to move to cooler refuges, when temperatures began to rise.

Belemnites temporarily re-appear in bed P10 (Upper Levisoni Zone), after the onset of the T-OAE, coinciding, approximately, with the end of the T-OAE-negative CIE (figure 1), with a poor record of *Acrocoelites*. This genus is known to appear and radiate in the northeast basins of the NW Tethys [36,47,48,61,66] after the T-OAE, replacing the genus *Passaloteuthis*, which has been interpreted as an evolutionary adaptation in terms of ecological preferences related to the stressful conditions of the T-OAE [51]. Moreover, the change from the Late Pliensbachian/Early Toarcian Belemnitinae-dominated assemblage to the Middle Toarcian Megateuthidinae-dominated assemblage (together with the disappearance of smaller sized species such as Hastitidae sp. indet., *Parapassaloteuthis* sp. A, *P. milleri*, *Ps. longiformis*), might be related not only to ecological preferences, but also to selective extinction, as pointed out by Payne *et al.* [7], who state that past extinction events were either non-selective, or preferentially removed smaller-bodied taxa. However, the lack of a continuous belemnite record during the Levisoni Zone in Peniche hampers a more detailed assessment. Further research in other basins is necessary to disentangle the extent of this phenomenon.

## 4.2. The relationship of sedimentary facies to belemnite body-size

Despite the fact that the magnitude of palaeoenvironmental changes relative to the rapid nature of these crisis events [102–104] is quite high on geological timescales [105], their representation in the sedimentary record is influenced by facies evolution. Sedimentary facies depend on several factors such as regional tectonics, climate and sea-level changes, which control the sedimentation rate, which is not constant across the studied interval. We use lithology to account for changes in the facies (related with relative sea-level changes) and belemnite abundance to account for changes in the sedimentation rate, since both are thought to have an effect on morphological patterns [106].

Accumulations with abundant belemnites, particularly observed in beds P4 and P5, could be related to changes in the sedimentation rate relative to the accumulation of belemnites [107]. The lowermost

Polymorphum Zone in Peniche is thought to correspond to a condensed interval [65,91,108]. Our data reveal that belemnite body-size is larger in beds with higher belemnite abundance (electronic supplementary material, figure S10). Abundance, however, does not markedly change across the boundary, where the most significant size change is observed. Regarding lithology, belemnite body-size is larger in marls in comparison with limestones, but these effects are not strong (electronic supplementary material, figure S9).

Our results indicate that when studying the effect of environmental perturbations on belemnite body-size at the scale of the assemblage, lithology and abundance do not affect the pattern observed. Before and after correction, temperature is always the best parameter in explaining the body-size variation. However, the significance is higher without correction, which is not surprising considering that residualizing might also filter out potential ecological signals contained within the sedimentary properties, in addition to the effect of preservation and collection biases. Even with the correction for the effects of sedimentary properties, and despite the existence of a significant model, the analysis failed in revealing a single model to explain belemnite body-size variation. Instead, several models are very similar to the model with the best AICc score. This probably relates to the strong interaction between abiotic parameters, for which we have reasonable proxies when driving body-size patterns during the episodes of climate warming.

However, when analysing the data at species level (*P. bisulcata* and *Ps. longiformis*), the best model for both species in explaining the body-size variation, after correction, is Hg/TOC, as opposed to temperature, without correction. This might be related to the smaller datasets, especially in the case of *P. bisulcata*, which does not allow us to distinguish between individual models. Either way, the increased warming, at the assemblage scale, and Hg/TOC anomalies, as drivers of individual species body-size variation, are not incompatible. They reveal that across the Pli-Toa boundary warming generally correlates with decreasing body-size at the assemblage scale and that the impact of 'catastrophic' environmental perturbations, triggered by volcanism, are the most severe, and probably work in concert, in driving individual species body-size decrease.

The lowermost Toarcian (base of Polymorphum Zone) in Peniche also coincides with a marine transgression [23,65] which might have impacted the size distribution, but differences in lithology and sedimentation rates could not clearly explain size differences in our results. Furthermore, deepening would rather result in finding larger species [76], which would be consistent with our observations of finding larger specimens in marls. However, a body-size decrease is seen across the boundary. Further analyses across different parts of the Lusitanian Basin would be necessary to disentangle the potentially less subtle effects of lithology and sedimentation rates in body-size patterns in shallower sections. As we have no good and/or uncorrelated proxies for other factors such as productivity or ocean acidification, we cannot assess their role in explaining belemnite body-size. However, warming is expected to have impact on all of these environmental parameters.

## 4.3. Environmental drivers of body-size fluctuations

The Pli–Toa boundary corresponds to the first episode of stress in the Early Toarcian palaeoenvironmental crisis [21], and has been associated with the first pulse of Karoo–Ferrar volcanism [20]. This had a major effect on climate, disturbing the entire ocean–atmosphere system, by causing carbon cycle perturbations, fluctuations in the seawater temperature, widespread deoxygenation and organic matter burial [83]. Our results are consistent with the hypothesis that climate is an important driver of the belemnite body-size variation at the assemblage scale, with a decreasing body-size associated with increasing seawater temperature (electronic supplementary material, figure S8). Salinity could, however, partially affect the interpretation of the oxygen isotopic data, particularly in northwest European basins, where the input of brackish and nutrient-rich Arctic surface waters seems to affect the signal [101,109]. However, decreased salinity has been deemed less significant in Peniche than warming and carbon cycle perturbations, based on phytoplankton communities analyses [30]. Furthermore, modelling suggests that, at these palaeocoordinates, salinity changes are close to zero [101,109], during the studied interval.

The reduction in adult size of *Ps. longiformis* and *P. bisulcata*—the Lilliput effect *sensu stricto*—can potentially be compared with the decrease in adult mantle length of extant squid during rapid warming events [9,11] interpreted to be related to accelerated life histories of squid, with increasing growth rates and shortening lifespans. Individual squid might also require more food per unit body-size, require more oxygen for faster metabolisms and have a reduced capacity to cope without food [77]. To what degree the decrease in adult size relates with direct effects of warming on physiology

(development rate, metabolism and respiration) or instead indirect effects on resource acquisitions, affecting growth and early development, is still under debate [8,110].

Apart from increasing seawater temperature, the mercury anomalies and carbon cycle perturbations at the Pli–Toa boundary event in Peniche, also coincide with increased weathering [22], local decreased productivity [29,30], increasing $pCO_2$ (and potentially ocean acidification) and decreasing dissolved oxygen [21]. All of these factors are expected to work in concert in driving marine organisms body-size [6], and the results of our analysis of belemnite body-size are consistent with that, especially by the strong collinearity detected between biotic and abiotic parameters (electronic supplementary material, Fig. S4).

Strontium isotopic data from Peniche support the idea of increased weathering during the Pli–Toa boundary warming event, and potentially nutrient influx [22], which is also consistent with the interpretation of enhanced mercury concentration in the sediments. However, counterintuitively, this warming event is interpreted to lead to a decrease in primary productivity, with a decrease in phytoplankton abundance [29,30,45]. This emphasizes how warming and related stressors might affect the size of primary consumers and potentially other levels of the food chain, such as predators like cephalopods. This is also consistent with the increased extinction risk for pelagic predators modelled for various hyperthermal events, including end-Permian mass extinction and the Early Toarcian crisis [7,46]. Additionally, the fact that dwarfing has been observed as a response to starving in laboratorial experiments supports this interpretation [111]. Nonetheless, macroecological studies show that spatial body-size distribution in extant cephalopods is better explained by seawater temperature than productivity [76,112].

The negative CIE observed during the Early Toarcian is interpreted to be related to volcanic outgassing, either directly, by the rapid input of isotopically light carbon into the atmosphere–hydrosphere system [56], or indirectly, by triggering various sources of isotopically light carbon [19]. This would have not only caused rapid climate warming, but also increased $pCO_2$ and decreased pH of the seawater. Cephalopods are usually interpreted to be quite resistant to ocean acidification in comparison with other marine organisms—cuttlefish even increase calcification. This is interpreted to be related to their strong acid–base regulatory abilities, and to the fact that the cuttlebone is a fully internal structure [113,114]. However, the limited data available hitherto seem to indicate that the effects of acidification might be more severe in early ontogeny, resulting in pathologies [114,115]. Nonetheless, no clear signs of aberrant early development were observed in the studied rostra. More importantly, there are no direct proxies for marked ocean acidification during the Early Toarcian in the Lusitanian Basin and only indirect evidence, such as calcareous nannofossils (Schizosphaerella) size reduction. However, this has been interpreted as an indirect consequence of $pCO_2$ increase, due to the changes in the climate and sea level [29], rather than a direct cause of high $pCO_2$.

Together with the deposition of organic-rich black shales, the Early Toarcian negative CIE has been traditionally interpreted to reflect increased anoxia in the ocean [82,116]. Irrespective of the potential local or regional overprint of increasing stratification due to basin restriction, or arctic water input, the perturbations at the Pli–Toa boundary event in Peniche are thought to reflect the start of decreasing oxygenation of seawater, while the T-OAE perturbations represent the global peak of anoxia [83]. Nonetheless, there is no evidence for bottom-water anoxia in the Lusitanian Basin [22,117], in contrast with the northwest European basins [118]. Despite that fact, less severe deoxygenation might have played a role in belemnite body-size and distribution, since increased seawater temperature results in reduced oxygen availability relative to demand [119].

The strong collinearity between biotic and abiotic factors available in the literature for the Peniche section (electronic supplementary material, figure S4) hampers an analysis of their individual effect on belemnite body-size. Nonetheless, this study allows investigation of the relationship of belemnite body-size reduction with direct (temperature) and indirect environmental stressors (e.g. deoxygenation, ocean acidification, input of nutrients) associated with climate warming, despite the complex interaction between biotic and abiotic factors, highlighting the importance of taking into account different scales of organization.

## 5. Conclusion

We document a median body-size decrease of belemnites across the Pli–Toa boundary event, at different scales of palaeobiological organization in the Peniche reference section. We find no evidence for a major taxonomic turnover and the pattern is mainly driven by a decrease in adult size of the dominant taxon

*Ps. longiformis*—Lilliput effect *sensu stricto*. This phenomenon coincides with the first pulse of the Upper Pliensbachian–Lower Toarcian palaeoenvironmental crisis, probably triggered by volcanism of the Karoo–Ferrar large igneous province. Our results indicate that climate warming best explains the body-size fluctuations observed, although the interplay with perturbations of the carbon cycle, and other environmental factors, possibly triggered by increased volcanogenic outgassing, is evident. Our results suggest that morphological responses precede extinction pulses in belemnites (i.e. during the T-OAE) in the Lusitanian Basin. They also highlight that decreasing adult body-size might rather be a life-history strategy to deal with temporarily deteriorating conditions related to warming, than being a result of taxonomic turnover. During the T-OAE, belemnites disappear from the Lusitanian Basin suggesting their local extinction and/or range shift. Changes within the belemnite assemblage, such as competition and emigration dynamics, seem to play a role in explaining belemnite body-size variation, albeit minor.

Data accessibility. The studied specimens are stored in the Science Museum (Museu da Ciência, University of Coimbra, Portugal) and the three-dimensional data are available through Zenodo (https://doi.org/10.5281/zenodo.3459233). Figures and tables and the R script used for the statistical analyses have been uploaded as part of the electronic supplementary material.

Authors' contributions. P.R. participated in the design of the study, carried out the data collection and analysis, interpreted the results and drafted the manuscript. K.D.B. designed and coordinated the study, participated in the data analysis and helped interpreting the results and drafting the manuscript. The fieldwork was carried out by P.R., K.D.B. and L.V.D. P.N. helped with data collection and preliminary data analysis. L.V.D. helped interpreting the results. R.W. helped with the taxonomic work. All authors contributed to the writing process and gave final approval for publication.

Competing interests. We declare we have no competing interests.

Funding. This is a contribution to the DFG Research Unit FOR 2332 (grant no. Ba 5148/1-1 to K.D.B.) TERSANE and to the IGCP 655 (IUGS–UNESCO).

Acknowledgements. We thank Birgit Leipner–Mata and Manuel Blank for helping in belemnite preparation, and Benjamin Gügel, Christian Schulbert and Martina Schlott for helping with scanning the specimens. We thank Manuel Steinbauer, Carl Reddin, Wolfgang Kiessling and Vanessa Roden, as well as the referees, for their valuable comments on the manuscript.

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
