## [Reviewer comments · Royal Society Open Science]

Review History

RSOS-190494.R0 (Original submission)

Review form: Reviewer 1

Is the manuscript scientifically sound in its present form?

Yes

Are the interpretations and conclusions justified by the results?

Yes

Is the language acceptable?

Yes

Is it clear how to access all supporting data?

Yes

Do you have any ethical concerns with this paper?

No

Have you any concerns about statistical analyses in this paper?

No

Recommendation?

Accept with minor revision (please list in comments)

Comments to the Author(s)

For their revised version of the article “Mechanisms and drivers of belemnite body-size dynamics across the Pliensbachian-Toarcian crisis” Rita and co-authors have addressed the comments of three reviewers. The authors put considerable effort into their analysis and employed a wide range of statistical tests to their dataset, enabling to interrogate their results in some detail. I think the authors are suitably cautious about the implications of their findings and have assembled a novel dataset which is robust enough to warrant publication in Royal Society Open Science.

I feel however, that the resulting text is still not very accessible to readers because it contains a lot of statistical nomenclature, terms, and abbreviations which are not straight forward to relate to the actual fossils and make it hard to follow the narrative. Often times I feel that by re-formulating the text using simpler language the readability would improve substantially.

E.g, I would find the first discussion paragraph (P7) much easier to read is if was written more like “Statistical evaluation of our data has shown that size changes of belemnite rostra in the studied section at Peniche are significant for the Pliensbachian-Toarcian boundary event, i.e., H1 was validated. This size change is almost exclusively driven by a body size reduction of single species...” By spelling out more directly what the different effects are and the abbreviated hypotheses mean, the reader will find it easier to follow the red thread of the text.

No clear, singular driver of the observed size change in belemnites could be identified from modelling of the morphological and geochemical data. The authors interpret this to signify that the interplay between environmental stressors associated with hyperthermals and body size of animals is complex. One could, however, also pose that factors other than those tested by the authors may have caused these changes in size or that all of the tested parameters were somehow involved. This point is amplified by the lack of quantification. How well do the produced models actually fit the observed data? Even though some of the model options are statistically significant, they might not explain much of the observed data variability.

The T-OAE and to some degree also the Pli-Toa boundary are intervals in time when numerous environmental parameters changed more or less in unison, which will make finding causalities very tricky. E.g., even though temperature might be a relatively good predictor of some of the size change data, it does not mean that temperature actually caused the size reduction but might have caused another effect which in turn led to the diminished size of the belemnites. This is to some degree acknowledged by the authors and reference is also made to modern squid ecology, but nevertheless the strong reliance on statistical evaluation in the study makes the text more prone to attack by arguments of coincidence vs causality in my opinion.

These points are not made here to criticize the approach taken by the authors but to spell out that putting slightly more emphasis on geological context in the discussion and acknowledging limitations of the modelling approach more openly in the text would in my opinion make this study even more valuable for the geoscience community.

Minor remarks (the line numbers in the manuscript seem to be shifted with respect to the text, so I tried to approximate as close as possible):

~P1L32: “fossil cephalopod studies”

~P1L40: “two dominant species”

P1 bottom: No present address is given here.

~P2L18: I think this sentence would require to be elaborated on a little further. For example, the extinction event at the Cretaceous-Paleogene boundary seems to have hit particularly the large mammals.

~P2L29: “north-west Tethyan basins”?

~P3L38: “approach was adopted”?

~P3L42: “belemnite rostra”

~P3L52: I do not think any study has shown convincingly that the belemnite rostrum was originally highly porous. Many studies have provided evidence that some of the rostrum at any given time in the belemnite’s life might have been somewhat porous, but all data I am aware of are consistent with this porosity being of subordinate importance for the bulk rostrum at best.

~P3L52: “soft tissue preservation”?

~P3L55: I agree that the rostrum is probably the best proxy one can use for belemnite size, but I think one should add a caveat here that it may not be ideal. Not all belemnites would have had the same body proportions, and some species with rather short rostra might nevertheless have been relatively big in size. E.g., in the family of the Giraffidae, body proportions differ greatly between the giraffe and the okapi. Other examples can be found across the animal kingdom.

~P4L15: Since the authors are here testing a hierarchical set of hypotheses as also shown by the tree diagram in figure 2 it might be helpful to the reader to label the hypotheses accordingly. H2-H5 are all sub-hypothesis of H1 and H4 and H5 appear to be refined versions of H3. One may thus label H2 as H1,1 and H3 as H1,2, whereas H4 would be H1,2,1 and H5 would be H1,2,2.

~P4L21: “4” should be in subscript

~P4L37: I think for the equations the prefixes “pre-” and “post-” should be deleted. It is already clear from the subscripts “bed1” and “bed2” that reference is made to different horizons and I think “pre-boundary crossers” and “post-boundary crossers” are confusing terms.

~P5L31: (see also P6L8,11) I am not sure how the authors compute rostrum volume from their morphological measurements. If the body size shrinks by 13 %, also the rostrum volume should shrink by 13 %, unless body size is meant to read as body length or other one dimensional parameter. If the latter is the case and the rostrum shape does not change, I would compute the volume as 0.87^3 , i.e., 0.66, equivalent to a volume reduction of 34 %, not 40.

~P6L13: How can something reduce its body size by 109%? It would then have a negative size.

~P6L33: A change from 75 % to 69 % to me is a reduction, not an increase.

~P6L36: Could some information be added here, how much of the observed variance in data can actually be modelled using the significant models? If only a small percentage of the size variance can be explained by the models, they might not be very useful despite being statistically significant.

~P7L37: “constraints”

~P7L43: delete “has had”

~P7L46: “an interval barren in belemnites in the Lusitanian Basin and many other sections in Iberia”.

~P8L35: “Cardigan Bay Basin sediments”

~P8L38: How was it constrained when conditions became most severe?

~P8L41: “habitats were no longer suitable”

~P9L8: I find this argument of “rapidity” debatable. As one is dealing with Mesozoic sediments temporal resolution is necessarily limited, but it is clear that the durations of both the T-OAE and the Pli-Toa boundary are not exactly short. E.g., the T-OAE likely lasts about as long as half of the Pleistocene.

~P9L57: “modelling suggests”

Review form: Reviewer 2

Is the manuscript scientifically sound in its present form?

Yes

Are the interpretations and conclusions justified by the results?

Yes

Is the language acceptable?

Yes

Is it clear how to access all supporting data?

Yes

Do you have any ethical concerns with this paper?

No

Have you any concerns about statistical analyses in this paper?

Yes

Recommendation?

Accept with minor revision (please list in comments)

Comments to the Author(s)

This is a valuable and interesting contribution to our knowledge of the Toarcian OAE and changes at the Pliensbachian-Toarcian boundary. We know that in NW Europe belemnites underwent some major changes in biogeographic distribution and this work verifies these patterns and shows that biological changes were occurring within the belemnites generally and within particular species. I like the approach (particularly considering change at different levels of organisation) and I believe it to be very thorough. Although some of the methods need to be explained more clearly – see comments below.

I also find myself asking a few questions surrounding the broader context of the work – how do you think these changes might have affected the wider food web? Would the wider anoxia would have affected belemnites in the non-anoxic Lusitanian basin e.g. by restricting migratory routes, food supply etc? Presumably it could have inhibited their ability to move to cooler refuges when temperatures began to rise. Maybe the authors have some ideas that might help us to view the changes more broadly?

I think the writing needs some polishing, it doesn't always flow and is repetitive in places. There are also some vague descriptions in the results. I think by reviewing and revising the text to tighten things up and integrate information better it could be improved.

Methods and Results:

Results are concise which is good, but they don't flow. There are many short paragraphs and obvious links aren't always made between related points. Some information is missing or ambiguous and needs to be clearer/stated more precisely. I encourage the authors to think about structure and providing more and clearer descriptions (plus see below).

I found the statistical reporting to be ambiguous. The format for writing statistics is quite standardised because we need to be very precise about what exactly a statistical test does and does not show. In this ms there is regular mention of 'significant results' and looking for a 'significant difference' without specifying what is being compared, what differed and in what direction – this applies in both the results and methods. I would like the authors to be more explicit so that the results cannot be misinterpreted. In the results you should report what test you used, what groups were compared (often the case in this ms as they are hypothesis tests) and the direction or scale of difference. The test statistics and p values must also be reported somewhere whether its in parentheses in the text or in a table.

Abbreviations (e.g. GM, AIC etc) are used regularly but seem only to be defined once – you should at least explain them once in each section of the paper to aid your reader and they should be explained in figure captions.

Figures and tables are good – but I found them a little hard to follow/not always intuitive. Can you improve the labelling and captions to make it easier for the reader to follow. I have made comments on the figures and captions in the attached pdf – explaining what needs clarification/labelling.

Discussion:

Overall, could be stronger, more integrated and with less repetition.

In my opinion the first paragraph doesn't set the context sufficiently: Why does the work matter, what's the big picture? and what did you attempt to do? Remind the reader.

I think the discussion needs to be more integrated – results are presented in one para and then context from the literature in another. Some paragraphs are very short (2-3 lines only). It would be best to combine these and remove some of the repetition.

I found reference to the ammonite zones a bit confusing (not being very familiar with the biostrat in this region) – it would be helpful to occasionally provide context relative to the event.

Detailed comments (plus see typos, issues of clarity etc on the annotated pdf (Appendix A)):

Line 31 p1: add some comment to be clear that includes this event

Line 32 p1: you could broaden your context a little to explain something about the Toarcian event - maybe use the term extinction - deoxygenation or climate change - this would make it accessible to a broader audience (including non-geologists)

Line 42 p8: see Caswell and Coe 2014 re egg laying

Line 51 p9: and widespread deoxygenation and organic matter burial

Line 18 p10: Caswell and Coe 2013 also conclude this but from secondary producers

Line 22 p10: yes but tricky because and event of this magnitude/severity has never been tested/observed for modern squid.....

Line 59 p9: interesting - what is thought to be the reason for this?

Line 28 p11: but it could effect their prey

Line 45 p11: not sure what you're considering direct vs indirect?

Line 41 p1: do you think its a direct link to SST?

Line 43 p1: I'm sure space is limited - but this is a little vague. I'm sure its complex but can you outline how you think these changes might relate to each other a little more clearly

Line 50 p1: also biotic interactions such as competition cannot be excluded- one taxon may be more successful under climate change - and species may be less competitive in the face of e.g. novel invaders

Line 51 p1: in the modern world this is likely to be even more complex - human activities such as pollution, habitat destruction and fishing will provide complex synergistic interactions as well there's refs to work on this in one of these Poloczanska refs, plus the 2016 one is a good review of what we do and don't know about the biotic impacts of climate change:

Poloczanska, E.S., Brown, C.J., Sydeman, W.J., Kiessling, W., Schoeman, D.S., Moore, P.J., Brander, K., Bruno, J.F., Buckley, L.B., Burrows, M.T., 2013. Global imprint of climate change on marine life. *Nature Climate Change*: 3, 919–925.

Poloczanska, E.S., Burrows, M.T., Brown, C.J., García Molinos, J., Halpern, B.S., Hoegh-Guldberg, O., Kappel, C.V., Moore, P.J., Richardson, A.J., Schoeman, D.S., 2016. Responses of marine organisms to climate change across oceans. *Frontiers in Marine Science*: 3, 1–21.

Line 58, p1: interesting - maybe worth using the term 'stenothermal' here which is the proper term for such sensitivity.

Line 18 p2: an example would be nice or if you prefer to do it later - maybe a broader range of refs here if possible?

Line 30 p2: are ammonite species names given in full somewhere?

Line 33 p2: I know there's a range of estimates but some quantification for SST would help the reader

Line 34 p2: and body-size changes in benthic fauna - Caswell and Coe 2013 measured thousands of bivalves in Yorkshire and found body-size shifts of comparable scale to those you report (~50%)(Caswell and Coe 2013. *Geology* 41, 1163–1166). And I think there might be some work on other taxa groups by Correia et al. 2017. *Micropaleontology* 137, 46–63

Line 52 p2: yes, but if you consider the IPCC predictions for instance - its regionally very variable -it strengthens in places and weakens in others. Also the effects vary at smaller scales - we know this from modern systems (e.g. see Caswell et al. 2018 and refs therein - this is often referred to as 'enrichment'). Also time is important because many eutrophic systems experience a boom initially followed by subsequent deoxygenation beyond the systems capacity to process all the extra organic matter (algal blooms). Geologically these are probably fairly short but across a regional scale might take some time to respond.

I suppose my point is its complex and I don't think you've quite captured the complexity here - the IPCC models/predictions for changing hydro cycle and nutrient inputs could be used to illustrate some of this complexity.

Here's the refs I mention:

Caswell, B.A., Paine, M., Frid, C.L.J., 2018. Seafloor ecological functioning over two decades of organic enrichment. *Marine Pollution Bulletin*: 136, 212–229.

IPCC, 2014. *Climate Change 2014: Synthesis Report. Contribution of Working Groups I, II and III to the Fifth Assessment Report of the Intergovernmental Panel on Climate Change*, in: Pachauri, R.K., Meyer, L.A. (Eds.). IPCC, New York.

Line 41 p3: did you look at whether preservation varied?

Line 46 p3: ok, but what do you mean by 'better' preservation? complete vs incomplete? and what did you consider complete? with phragmocone or just complete rostra?

Line 17 p4: 'community composition shift' would be better - after all a population refers to just one species whereas a community refers to all species present in the assemblage

Line 19 p4: i am curious how you can tell when maturity occurs -is it explained somewhere? in the supplement perhaps?

Line 29 p4: This is vague. The emphasis must be on comparisons - they compare groups - yes it gives you a significance value but for 'comparisons'. 'So tests were used to compare x, y parameters between z groups' is more accurate

Line 37 p4: what are 'pre-boundary crossers'?

Line 45 p4: because bed separation = temporal separation? explain why necessary

Line 56 p4: why should differences in sediment properties effect pelagic belemnites? via thier food supply? or do you mean its geochemical properties that are proxies for env change? not clear why this matters

Line 9 p 5: what metric did you use for primary producers?

Line 33 p5: ok, so what is happening at P8-P10 is this some form of recovery?

Line 37-41 p5: i would restructure this subsection to begin with this overview para

Line 8 p6: is it geometric mean length? i cant remember - remind us occasionally

Line 11 p6: if its not sig different don't try to infer meaning

Line 14 p6: family names should not be in italics - only genera and species

Line 17 p7: which metric? and what was the scale of reduction?

Line 32 p7: 'detectable from the fossilised remains' - there may have been biological differences in morphology, life history or behaviour that are not preserved on hard parts

Line 53 p7: any evidence of this from the modern literature? if so refer to it here

Lines 55 p7 to line 10 p8; this is quite repetitive and hard to follow - rework the para to be more concise and direct

Section 5.2 p10: this subsection could be much shorter

Decision letter (RSOS-190494.R0)

21-Jun-2019

Dear Professor Rita

On behalf of the Editors, I am pleased to inform you that your Manuscript RSOS-190494 entitled "Mechanisms and drivers of belemnite body-size dynamics across the Pliensbachian-Toarcian crisis" has been accepted for publication in Royal Society Open Science subject to minor revision in accordance with the referee suggestions. Please find the referees' comments at the end of this email.

The reviewers and handling editors have recommended publication, but also suggest some minor revisions to your manuscript. Therefore, I invite you to respond to the comments and revise your manuscript. Please pay special attention to the issue of 'readability' that is raised by both the reviewers and the associate editor. If a paper is not easily read, it will not be cited and the potential value of the contribution will not be realised. I would urge you to make extra effort to ensure that your work is as clearly presented as possible. Note that this will be a final condition of acceptance by the journal.

- Ethics statement

- Data accessibility

If you wish to submit your supporting data or code to Dryad (<http://datadryad.org/>), or modify your current submission to dryad, please use the following link:
<http://datadryad.org/submit?journalID=RSOS&manu=RSOS-190494>

- Competing interests

- Authors' contributions

- Acknowledgements

- Funding statement

Because the schedule for publication is very tight, it is a condition of publication that you submit the revised version of your manuscript before 30-Jun-2019. Please note that the revision deadline will expire at 00.00am on this date. If you do not think you will be able to meet this date please let me know immediately.

- 1) A text file of the manuscript (tex, txt, rtf, docx or doc), references, tables (including captions) and figure captions. Do not upload a PDF as your "Main Document";
- 2) A separate electronic file of each figure (EPS or print-quality PDF preferred (either format should be produced directly from original creation package), or original software format);

- 3) Included a 100 word media summary of your paper when requested at submission. Please ensure you have entered correct contact details (email, institution and telephone) in your user account;
- 4) Included the raw data to support the claims made in your paper. You can either include your data as electronic supplementary material or upload to a repository and include the relevant doi within your manuscript. Make sure it is clear in your data accessibility statement how the data can be accessed;
- 5) All supplementary materials accompanying an accepted article will be treated as in their final form. Note that the Royal Society will neither edit nor typeset supplementary material and it will be hosted as provided. Please ensure that the supplementary material includes the paper details where possible (authors, article title, journal name).

on behalf of Professor Stephen Hesselbo (Associate Editor) and Jon Blundy (Subject Editor)
openscience@royalsociety.org

Associate Editor Comments to Author (Professor Stephen Hesselbo):

The manuscript has been thoroughly revised from the original version, and now thoroughly re-reviewed. The reviewers have made a number of very helpful comments which, if properly implemented, will improve the manuscript greatly. In particular both reviewers suggest numerous ways in which the paper may be made more readable - in particular there should be a major effort to remove/reduce acronyms, initialisms, and short-hand references to hypotheses (H1, H2, etc.) that all require the reader to repeatedly refer back to other parts of the manuscript

or (more likely) give up. As always the responses to the comments of the reviewers should be clearly documented and if not acted upon specific justification is required.

Reviewer comments to Author:

Reviewer: 1

Comments to the Author(s)

For their revised version of the article “Mechanisms and drivers of belemnite body-size dynamics across the Pliensbachian-Toarcian crisis” Rita and co-authors have addressed the comments of three reviewers. The authors put considerable effort into their analysis and employed a wide range of statistical tests to their dataset, enabling to interrogate their results in some detail. I think the authors are suitably cautious about the implications of their findings and have assembled a novel dataset which is robust enough to warrant publication in Royal Society Open Science.

I feel however, that the resulting text is still not very accessible to readers because it contains a lot of statistical nomenclature, terms, and abbreviations which are not straight forward to relate to the actual fossils and make it hard to follow the narrative. Often times I feel that by re-formulating the text using simpler language the readability would improve substantially.

E.g, I would find the first discussion paragraph (P7) much easier to read is if was written more like “Statistical evaluation of our data has shown that size changes of belemnite rostra in the studied section at Peniche are significant for the Pliensbachian-Toarcian boundary event, i.e., H1 was validated. This size change is almost exclusively driven by a body size reduction of single species...” By spelling out more directly what the different effects are and the abbreviated hypotheses mean, the reader will find it easier to follow the red thread of the text.

No clear, singular driver of the observed size change in belemnites could be identified from modelling of the morphological and geochemical data. The authors interpret this to signify that the interplay between environmental stressors associated with hyperthermals and body size of animals is complex. One could, however, also pose that factors other than those tested by the authors may have caused these changes in size or that all of the tested parameters were somehow involved. This point is amplified by the lack of quantification. How well do the produced models actually fit the observed data? Even though some of the model options are statistically significant, they might not explain much of the observed data variability.

The T-OAE and to some degree also the Pli-Toa boundary are intervals in time when numerous environmental parameters changed more or less in unison, which will make finding causalities very tricky. E.g., even though temperature might be a relatively good predictor of some of the size change data, it does not mean that temperature actually caused the size reduction but might have caused another effect which in turn led to the diminished size of the belemnites. This is to some degree acknowledged by the authors and reference is also made to modern squid ecology, but nevertheless the strong reliance on statistical evaluation in the study makes the text more prone to attack by arguments of coincidence vs causality in my opinion.

These points are not made here to criticize the approach taken by the authors but to spell out that putting slightly more emphasis on geological context in the discussion and acknowledging limitations of the modelling approach more openly in the text would in my opinion make this study even more valuable for the geoscience community.

Minor remarks (the line numbers in the manuscript seem to be shifted with respect to the text, so I tried to approximate as close as possible):

~P1L32: “fossil cephalopod studies”

~P1L40: “two dominant species”

P1 bottom: No present address is given here.

~P2L18: I think this sentence would require to be elaborated on a little further. For example, the extinction event at the Cretaceous-Paleogene boundary seems to have hit particularly the large mammals.

~P2L29: “north-west Tethyan basins”?

~P3L38: “approach was adopted”?

~P3L42: “belemnite rostra”

~P3L52: I do not think any study has shown convincingly that the belemnite rostrum was originally highly porous. Many studies have provided evidence that some of the rostrum at any given time in the belemnite’s life might have been somewhat porous, but all data I am aware of are consistent with this porosity being of subordinate importance for the bulk rostrum at best.

~P3L52: “soft tissue preservation”?

~P3L55: I agree that the rostrum is probably the best proxy one can use for belemnite size, but I think one should add a caveat here that it may not be ideal. Not all belemnites would have had the same body proportions, and some species with rather short rostra might nevertheless have been relatively big in size. E.g., in the family of the Giraffidae, body proportions differ greatly between the giraffe and the okapi. Other examples can be found across the animal kingdom.

~P4L15: Since the authors are here testing a hierarchical set of hypotheses as also shown by the tree diagram in figure 2 it might be helpful to the reader to label the hypotheses accordingly. H2-H5 are all sub-hypothesis of H1 and H4 and H5 appear to be refined versions of H3. One may thus label H2 as H1,1 and H3 as H1,2, whereas H4 would be H1,2,1 and H5 would be H1,2,2.

~P4L21: “4” should be in subscript

~P4L37: I think for the equations the prefixes “pre-” and “post-” should be deleted. It is already clear from the subscripts “bed1” and “bed2” that reference is made to different horizons and I think “pre-boundary crossers” and “post-boundary crossers” are confusing terms.

~P5L31: (see also P6L8,11) I am not sure how the authors compute rostrum volume from their morphological measurements. If the body size shrinks by 13 %, also the rostrum volume should shrink by 13 %, unless body size is meant to read as body length or other one dimensional parameter. If the latter is the case and the rostrum shape does not change, I would compute the volume as 0.87^3 , i.e., 0.66, equivalent to a volume reduction of 34 %, not 40.

~P6L13: How can something reduce its body size by 109%? It would then have a negative size.

~P6L33: A change from 75 % to 69 % to me is a reduction, not an increase.

~P6L36: Could some information be added here, how much of the observed variance in data can actually be modelled using the significant models? If only a small percentage of the size variance can be explained by the models, they might not be very useful despite being statistically significant.

~P7L37: “constraints”

~P7L43: delete “has had”

~P7L46: “an interval barren in belemnites in the Lusitanian Basin and many other sections in Iberia”.

~P8L35: “Cardigan Bay Basin sediments”

~P8L38: How was it constrained when conditions became most severe?

~P8L41: “habitats were no longer suitable”

~P9L8: I find this argument of “rapidity” debatable. As one is dealing with Mesozoic sediments temporal resolution is necessarily limited, but it is clear that the durations of both the T-OAE and the Pli-Toa boundary are not exactly short. E.g., the T-OAE likely lasts about as long as half of the Pleistocene.

~P9L57: “modelling suggests”

Reviewer: 2

Comments to the Author(s)

This is a valuable and interesting contribution to our knowledge of the Toarcian OAE and changes at the Pliensbachian-Toarcian boundary. We know that in NW Europe belemnites underwent some major changes in biogeographic distribution and this work verifies these patterns and shows that biological changes were occurring within the belemnites generally and

within particular species. I like the approach (particularly considering change at different levels of organisation) and I believe it to be very thorough. Although some of the methods need to be explained more clearly – see comments below.

I also find myself asking a few questions surrounding the broader context of the work – how do you think these changes might have affected the wider food web? Would the wider anoxia would have affected belemnites in the non-anoxic Lusitanian basin e.g. by restricting migratory routes, food supply etc? Presumably it could have inhibited their ability to move to cooler refuges when temperatures began to rise. Maybe the authors have some ideas that might help us to view the changes more broadly?

I think the writing needs some polishing, it doesn't always flow and is repetitive in places. There are also some vague descriptions in the results. I think by reviewing and revising the text to tighten things up and integrate information better it could be improved.

Methods and Results:

Results are concise which is good, but they don't flow. There are many short paragraphs and obvious links aren't always made between related points. Some information is missing or ambiguous and needs to be clearer/stated more precisely. I encourage the authors to think about structure and providing more and clearer descriptions (plus see below).

I found the statistical reporting to be ambiguous. The format for writing statistics is quite standardised because we need to be very precise about what exactly a statistical test does and does not show. In this ms there is regular mention of 'significant results' and looking for a 'significant difference' without specifying what is being compared, what differed and in what direction – this applies in both the results and methods. I would like the authors to be more explicit so that the results cannot be misinterpreted. In the results you should report what test you used, what groups were compared (often the case in this ms as they are hypothesis tests) and the direction or scale of difference. The test statistics and p values must also be reported somewhere whether its in parentheses in the text or in a table.

Abbreviations (e.g. GM, AIC etc) are used regularly but seem only to be defined once – you should at least explain them once in each section of the paper to aid your reader and they should be explained in figure captions.

Figures and tables are good – but I found them a little hard to follow/not always intuitive. Can you improve the labelling and captions to make it easier for the reader to follow. I have made comments on the figures and captions in the attached pdf – explaining what needs clarification/labelling.

Discussion:

Overall, could be stronger, more integrated and with less repetition.

In my opinion the first paragraph doesn't set the context sufficiently: Why does the work matter, what's the big picture? and what did you attempt to do? Remind the reader.

I think the discussion needs to be more integrated – results are presented in one para and then context from the literature in another. Some paragraphs are very short (2-3 lines only). It would be best to combine these and remove some of the repetition.

I found reference to the ammonite zones a bit confusing (not being very familiar with the biostrat in this region) – it would be helpful to occasionally provide context relative to the event.

Detailed comments (plus see typos, issues of clarity etc on the annotated pdf):

Line 31 p1: add some comment to be clear that includes this event

Line 32 p1: you could broaden your context a little to explain something about the Toarcian event - maybe use the term extinction - deoxygenation or climate change - this would make it accessible to a broader audience (including non-geologists)

Line 42 p8: see Caswell and Coe 2014 re egg laying

Line 51 p9: and widespread deoxygenation and organic matter burial

Line 18 p10: Caswell and Coe 2013 also conclude this but from secondary producers

Line 22 p10: yes but tricky because an event of this magnitude/severity has never been tested/observed for modern squid.....

Line 59 p9: interesting - what is thought to be the reason for this?

Line 28 p11: but it could effect their prey

Line 45 p11: not sure what you're considering direct vs indirect?

Line 41 p1: do you think its a direct link to SST?

Line 43 p1: I'm sure space is limited - but this is a little vague. I'm sure its complex but can you outline how you think these changes might relate to each other a little more clearly

Line 50 p1: also biotic interactions such as competition cannot be excluded- one taxon may be more successful under climate change - and species may be less competitive in the face of e.g. novel invaders

Line 51 p1: in the modern world this is likely to be even more complex - human activities such as pollution, habitat destruction and fishing will provide complex synergistic interactions as well there's refs to work on this in one of these Poloczanska refs, plus the 2016 one is a good review of what we do and dont know about the biotic impacts of climate change:

Poloczanska, E.S., Brown, C.J., Sydeman, W.J., Kiessling, W., Schoeman, D.S., Moore, P.J., Brander, K., Bruno, J.F., Buckley, L.B., Burrows, M.T., 2013. Global imprint of climate change on marine life. *Nature Climate Change*: 3, 919–925.

Poloczanska, E.S., Burrows, M.T., Brown, C.J., García Molinos, J., Halpern, B.S., Hoegh-Guldberg, O., Kappel, C.V., Moore, P.J., Richardson, A.J., Schoeman, D.S., 2016. Responses of marine organisms to climate change across oceans. *Frontiers in Marine Science*: 3, 1–21.

Line 58, p1: interesting - maybe worth using the term 'stenothermal' here which is the proper term for such sensitivity.

Line 18 p2: an example would be nice or if you prefer to do it later - maybe a broader range of refs here if possible?

Line 30 p2: are ammonite species names given in full somewhere?

Line 33 p2: I know there's a range of estimates but some quantification for SST would help the reader

Line 34 p2: and body-size changes in benthic fauna - Caswell and Coe 2013 measured thousands of bivalves in Yorkshire and found body-size shifts of comparable scale to those you report (~50%)(Caswell and Coe 2013. *Geology* 41, 1163–1166). And I think there might be some work on other taxa groups by Correia et al. 2017. *Micropaleontology* 137, 46-63

Line 52 p2: yes, but if you consider the IPCC predictions for instance - it's regionally very variable - it strengthens in places and weakens in others. Also the effects vary at smaller scales - we know this from modern systems (e.g. see Caswell et al. 2018 and refs therein - this is often referred to as 'enrichment'). Also time is important because many eutrophic systems experience a boom initially followed by subsequent deoxygenation beyond the systems capacity to process all the extra organic matter (algal blooms). Geologically these are probably fairly short but across a regional scale might take some time to respond.

I suppose my point is it's complex and I don't think you've quite captured the complexity here - the IPCC models/predictions for changing hydro cycle and nutrient inputs could be used to illustrate some of this complexity.

Here are the refs I mention:

Caswell, B.A., Paine, M., Frid, C.L.J., 2018. Seafloor ecological functioning over two decades of organic enrichment. *Marine Pollution Bulletin*: 136, 212–229.

IPCC, 2014. *Climate Change 2014: Synthesis Report. Contribution of Working Groups I, II and III to the Fifth Assessment Report of the Intergovernmental Panel on Climate Change*, in: Pachauri, R.K., Meyer, L.A. (Eds.). IPCC, New York.

Line 41 p3: did you look at whether preservation varied?

Line 46 p3: ok, but what do you mean by 'better' preservation? complete vs incomplete? and what did you consider complete? with phragmocone or just complete rostra?

Line 17 p4: 'community composition shift' would be better - after all a population refers to just one species whereas a community refers to all species present in the assemblage

Line 19 p4: I am curious how you can tell when maturity occurs - is it explained somewhere? in the supplement perhaps?

Line 29 p4: This is vague. The emphasis must be on comparisons - they compare groups - yes it gives you a significance value but for 'comparisons'. 'So tests were used to compare x, y parameters between z groups' is more accurate

Line 37 p4: what are 'pre-boundary crossers'?

Line 45 p4: because bed separation = temporal separation? explain why necessary

Line 56 p4: why should differences in sediment properties affect pelagic belemnites? via their food supply? or do you mean its geochemical properties that are proxies for env change? not clear why this matters

Line 9 p 5: what metric did you use for primary producers?

Line 33 p5: ok, so what is happening at P8-P10 is this some form of recovery?

Line 37-41 p5: i would restructure this subsection to begin with this overview para

Line 8 p6: is it geometric mean length? i cant remember - remind us occasionally

Line 11 p6: if its not sig different don't try to infer meaning

Line 14 p6: family names should not be in italics - only genera and species

Line 17 p7: which metric? and what was the scale of reduction?

Line 32 p7: 'detectable from the fossilised remains' - there may have been biological differences in morphology, life history or behaviour that are not preserved on hard parts

Line 53 p7: any evidence of this from the modern literature? if so refer to it here

Lines 55 p7 to line 10 p8; this is quite repetitive and hard to follow - rework the para to be more concise and direct

Section 5.2 p10: this subsection could be much shorter

Author's Response to Decision Letter for (RSOS-190494.R0)

See Appendix B.

Decision letter (RSOS-190494.R1)

07-Oct-2019

Dear Professor Rita:

On behalf of the Editors, I am pleased to inform you that your Manuscript RSOS-190494.R1 entitled "Mechanisms and drivers of belemnite body-size dynamics across the Pliensbachian-Toarcian crisis" has been accepted for publication in Royal Society Open Science subject to minor revision in accordance with the Associate Editor's suggestions. Please find his comments at the end of this email.

The reviewers and Subject Editor have recommended publication once you have made these minor revisions to your manuscript. Therefore, I invite you to respond to the comments and revise your manuscript.

- Ethics statement

If your study uses humans or animals please include details of the ethical approval received, including the name of the committee that granted approval. For human studies please also detail

whether informed consent was obtained. For field studies on animals please include details of all permissions, licences and/or approvals granted to carry out the fieldwork.

- Data accessibility

If you wish to submit your supporting data or code to Dryad (<http://datadryad.org/>), or modify your current submission to dryad, please use the following link:
<http://datadryad.org/submit?journalID=RSOS&manu=RSOS-190494.R1>

- Competing interests

- Authors' contributions

- Acknowledgements

- Funding statement

Because the schedule for publication is very tight, it is a condition of publication that you submit the revised version of your manuscript before 16-Oct-2019. Please note that the revision deadline

will expire at 00.00am on this date. If you do not think you will be able to meet this date please let me know immediately.

on behalf of Professor Stephen Hesselbo (Associate Editor) and Jon Blundy (Subject Editor)
openscience@royalsociety.org

Associate Editor Comments to Author (Professor Stephen Hesselbo):

Associate Editor

Comments to the Author:

The authors have carried out a thorough revision of the manuscript and addressed all the issues raised by the reviewers. There were still some linguistic errors in the manuscript and a confusion between use of a hyphen versus an en-dash - I have uploaded a corrected version of the manuscript. The only revisions required are those in my corrected ms file.

Author's Response to Decision Letter for (RSOS-190494.R1)

See Appendix C.

Decision letter (RSOS-190494.R2)

06-Nov-2019

Dear Professor Rita,

I am pleased to inform you that your manuscript entitled "Mechanisms and drivers of belemnite body-size dynamics across the Pliensbachian–Toarcian crisis" is now accepted for publication in Royal Society Open Science.

To ensure your paper is prepared as rapidly as possible for publication, please can you send original files for each figure and table included in your manuscript to the editorial office as soon as possible (email below). Failure to do so may delay the preparation of the proof.

on behalf of Professor Stephen Hesselbo (Associate Editor) and Jon Blundy (Subject Editor)
openscience@royalsociety.org

Follow Royal Society Publishing on Twitter: [@RSocPublishing](https://twitter.com/RSocPublishing)
Follow Royal Society Publishing on Facebook:
<https://www.facebook.com/RoyalSocietyPublishing.FanPage/>
Read Royal Society Publishing's blog: <https://blogs.royalsociety.org/publishing/>

Appendix A**ROYAL SOCIETY
OPEN SCIENCE****Mechanisms and drivers of belemnite body-size dynamics
across the Pliensbachian-Toarcian crisis**

Journal:	Royal Society Open Science
Manuscript ID	RSOS-190494
Article Type:	Research
Date Submitted by the Author:	15-Mar-2019
Complete List of Authors:	Rita, Patrícia; GeoZentrum Nordbayern, Paleontology; Marine and Environmental Sciences Centre, 3004-517 Näschter, Paulina; GeoZentrum Nordbayern, Paleontology Duarte, Luís ; Universidade de Coimbra, Departamento de Ciências da Terra; Marine and Environmental Sciences Centre Weis, Robert; National Museum of Natural History Luxembourg de Baets, Kenneth; GeoZentrum Nordbayern, Paleontology
Subject:	Palaeontology < EARTH SCIENCES
Keywords:	cephalopods, Lilliput effect, Pliensbachian–Toarcian boundary event, Toarcian Oceanic Anoxic Event, climate warming, computed tomography
Subject Category:	Earth science

Author-supplied statements

Relevant information will appear here if provided.

Ethics

Does your article include research that required ethical approval or permits?:

This article does not present research with ethical considerations

Statement (if applicable):

CUST_IF_YES_ETHICS :No data available.

Data

It is a condition of publication that data, code and materials supporting your paper are made publicly available. Does your paper present new data?:

Yes

Statement (if applicable):

The datasets supporting this article have been uploaded as part of the Supplementary Material, together with supplementary figures and tables.

The specimens are stored in the Science Museum (Museu da Ci ncia, University of Coimbra, Portugal) and the 3D data will be stored on Zenodo.

Conflict of interest

I/We declare we have no competing interests

Statement (if applicable):

CUST_STATE_CONFLICT :No data available.

Authors' contributions

This paper has multiple authors and our individual contributions were as below

Statement (if applicable):

PR participated in the design of the study, carried out the data collection and analysis, interpreted the results and drafted the manuscript. KDB designed and coordinated the study, participated in the data analysis and helped interpreting the results and drafting the manuscript. The fieldwork was carried out by PR, KDB and LVD. PN helped with data collection and preliminary data analysis. LVD helped interpreting the results. RW helped with the taxonomic work. All authors contributed to the writing process and gave final approval for publication.

Mechanisms and drivers of belemnite body-size dynamics across the Pliensbachian–Toarcian crisis

Patrícia Rita^{1,2*}, Paulina Nätscher¹, Luís V. Duarte^{2,3}, Robert Weis⁴, Kenneth De Baets¹

¹Geozentrum Nordbayern, Friedrich–Alexander Universität Erlangen–Nürnberg, Erlangen 91054, Germany

²MARE (Marine and Environmental Sciences Centre), 3004–517 Coimbra, Portugal

³Department of Earth Sciences University of Coimbra, 3070–790 Coimbra, Portugal

⁴National Museum of Natural History Luxembourg, Department of Palaeontology, L–2160 Luxembourg

Keywords: cephalopods, Lilliput effect, Pliensbachian–Toarcian boundary event, Toarcian Oceanic Anoxic Event, climate warming, computed tomography

1. Summary

Body-size reduction is considered an important response to current climate warming and has also been observed during past biotic crises. However, in fossil cephalopods studies, the mechanisms and their potential link with climate are rarely investigated and they usually do not differentiate palaeobiological scales of organisation. We hypothesise that belemnites reduce their adult size across the Pliensbachian–Toarcian boundary warming event.

Belemnite body-size dynamics across the Pliensbachian–Toarcian in the Peniche section (Lusitanian Basin, Portugal) were analysed for the first time based on field data. We disentangle the mechanisms and the environmental drivers of the size fluctuations observed from the individual to the assemblage scale. Despite the lack of a major taxonomic turnover, a 40 % decrease in rostrum volume is observed across the Pliensbachian–Toarcian, before the Toarcian Oceanic Anoxic Event. The pattern is mainly driven by a reduction in adult size of the two-dominant species, *Pseudohastites longiformis* and *Issaloteuthis bisulcata*. Belemnite size distribution is best correlated with fluctuations in a palaeotemperature proxy, however, potential effects of volcanism and carbon cycle perturbations may also play a role. This highlights the complex interplay between environmental stressors associated with these hyperthermal events in driving belemnite body-size.

2. Introduction

Body-size is a key feature of any organism, reflecting its physiology, ecology and evolutionary history, across multiple scales of biological organisation (1). Body-size reduction has been considered the third universal response to global warming after changes in morphology and species distribution (2-4). However, disentangling body-size response to warming might not be straightforward due to interactions of the different abiotic factors during climate warming episodes (5, 6). Many factors during climate warming (increased temperature, decreased oxygenation, ocean acidification) are likely to work in synergy in leading to reductions in size (6). However, increasing nutrient supply, for example, might have the opposite effect, causing a body-size increase in some taxa (4, 5).

Moreover, individual responses to warming might be even population specific due to individual environmental requirements (availability of nutrients and oxygen) and metabolic specifications (7). Individual living cephalopod taxa, such as squid, respond rapidly to warming events, hatching at smaller size, undergoing fast growth rates over shorter life-spans and maturing younger at smaller size (8-10).

*Author for correspondence (patricia.rita@fae.de).

†Present address: Department, Institution, Address, City, Code, Country

R. Soc. open sci. article template

[revised manuscript text omitted]
 (K-S) tests were used to assess the significance of the results (Table S2). The same was done at the species scale, for each taxon. Furthermore, because changes in taxonomic composition can influence the body-size patterns by the appearance and/or disappearance of taxa, we modified the within- and among-taxa approach by Rego et al. (71). This method corresponds to a pair-to-pair analysis (comparison of successive beds) which decomposes the assemblage size shift (Eq. 1) into three components: a disappearance of taxa effect (Eq. 2), a within-lineage effect (Eq. 3) and an appearance of new taxa effect (Eq. 4).

(Eq. 1) Assemblage size shift = assemblage median body-size_{bed2} – assemblage median body-size_{bed1}

(Eq. 2) Disappearance of taxa effect = pre-boundary crossers_{bed1} – all taxa_{bed1}

(Eq. 3) Within-lineage effect = post-boundary crossers_{bed2} – pre-boundary crossers_{bed1}

(Eq. 4) Appearance of new taxa effect = all taxa_{bed2} – post-boundary crossers_{bed2}

Finally, at the individual ontogenetic scale of organisation, the hypotheses were tested by assessing body-size variation within different ontogenetic stages (adults and juveniles) for particular taxa. Variation partitioning allowed us to assess the proportion of body-size variation explained by ontogeny, taxonomic assignment and separation by beds, and by their joint effects. If the fraction of variation corresponding to beds separation is low, this suggests that mechanisms driving differences between samples are similar and not related to particularities of the sample (e.g., lithology, age). Variation in the body-size data was partitioned using partial redundancy analysis as implemented in the *vegan* package (72). The significance of each fraction was assessed through ANOVA. The whole time-series was compared with the Pli-Toa boundary event.

3.4. Environmental drivers of body-size changes

In order to test our hypothesis of a relationship between belemnite body-size and environmental perturbations, the distribution of the GM was compared with three main geologic proxies (Table 1). These are: brachiopod stable oxygen isotopes (23), bulk rock carbon isotopes (25) and mercury concentration normalised by total organic carbon (52), used as proxies for seawater palaeotemperature, negative excursions in the carbon cycle and volcanogenic outgassing, respectively (Table 1). To account for the impact of sedimentary properties on body-size patterns, we corrected for the effect of lithology and belemnite absolute abundance (mean abundance of belemnites per m², Figs. 1 and S3) by residualising (73). A simple linear regression analysis is performed between the body-size and the combined effect of abundance and lithology. The residuals from this analysis are then used as outcome in the ultimate regression analysis between body-size and abiotic variables. All variables of the model were continuous with the exception of lithology. We assigned our samples a categorical variable for lithology (marl to limestone)

R. Soc. open sci. article template

based on carbonate/clay content. Collinearity between explanatory variables was tested by using *cor* function in R (Fig. S5). Sedimentary properties as well as various other parameters linked with climate warming (such as strontium isotopes, total organic carbon and abundance of primary producers) could not be directly included in the models due to their high collinearity with $\delta^{13}\text{C}_{\text{carb}}$ or the other two used parameters (Fig. S4), hence the choice of the abiotic parameters indicated in Table 1.

A multiple linear regression using generalised least squares (GLS) was performed and the models were fitted by maximising the log-likelihood (*gls* function in R). The first-order autoregressive (AR(1)) model was used, which has the property of seeking autocorrelation and of minimising the error term in a time-series (74). Seven models (see R script, supplementary material) were compared using Akaike's second order corrected information criterion (AICc scores), which corrects for small sample size. The power of the best model was assessed by means of an ANOVA test. The statistical significance of each coefficient of the best model was assessed by calculating the *p*-value under a *t* approximation. The significance level was $p < 0.05$ for all the analyses, unless stated otherwise. The analyses were performed in R (75), using the packages *nlme* (76) and *qpcR* (77).

The regression analysis only included data from Emaciatum and Polymorphum zones (beds P1 to P9), due to the lack of a representative sample size in upper Levisoni Zone (bed P10, Fig. 1).

4. Results

4.1 Belemnite body-size fluctuations

Assemblage scale

Three episodes of median body-size decrease were recognised at the assemblage scale across the studied interval: P1–P2, P2–P3 and the Pli–Toa boundary event (P4–P5), with the latter being the only decrease with statistical significance (Fig. 3B) and corresponding to a 13 % body-size decrease (Fig. 4). This decrease in the GM corresponds to a 40 % decrease of rostrum volume (Table 2). The remaining pairs of beds correspond to increasing median body-size at the assemblage scale, with P6–P9 and P9–P10 being statistically significant (Fig. 3B).

A total of seven belemnite species were identified in Peniche (Fig. 1). Most of the taxa range from the uppermost Pliensbachian to the Lower Toarcian (uppermost Polymorphum Zone). *Bairstowius* sp. A is exclusively represented in the uppermost Pliensbachian, being stratigraphically replaced by *Pseudohastites longiformis* in the assemblage. During the Toarcian, the Levisoni Zone is devoid of belemnites, with the exception of bed P10 (upper Levisoni Zone) which includes two *Acrocoelites* sp. specimens (Fig. 1).

Assemblage body-size shift at the Pli–Toa boundary event is almost exclusively caused by a within-taxa effect (Fig. 3B) and therefore mainly driven by the boundary crossers (*Ps. longiformis*, *P. bisulcata* and *Hastitidae* sp. indet.). The taxonomic composition across the boundary does not change markedly, with *Ps. longiformis* the most abundant taxon, comprising 61.9 % (P4) and 56.2 % (P5) of the assemblages (Figs. 1 and 4). The disappearance of taxa (*Passaloteuthis milleri* and *Parapassaloteuthis* sp.) across the Pli–Toa boundary event, causes 4 % of the decrease in the assemblage median body-size (Fig. 3B).

In contrast, the other two episodes of body-size reduction (P1–P2 and P2–P3) are characterised by higher effects of appearance (15.9 % and 16.2 %, respectively) and disappearance of taxa (12.1 % and 15.2 %, respectively), with the within-taxa effect being comparatively lower at the boundary (Fig. 3B).

From P5 to P9, an increase in belemnite body-size at the assemblage scale is observed, followed by a belemnite gap, from P9 to P10 (Figs. 3B and 5).

Species scale

R. Soc. open sci. article template

Three taxa (*Ps. longiformis*, *P. bisulcata* and *Hastitidae* sp. indet.) are recorded immediately before and immediately after the Pli–Toa boundary event (Fig. 1). The most significant and largest median size reduction is observed in the most abundant taxon, *Ps. longiformis* (17 % decrease in GM and 51 % in rostrum volume, Fig. 6).

Despite the lack of statistical significance, *P. bisulcata*, the second most abundant taxon, also decreases in size (7 % in GM and 21 % in volume, Fig. 6) across the Pli–Toa boundary event, while *Hastitidae* sp. indet., on the contrary, slightly increases in size (8 %, Fig. S7). It is noteworthy that *P. bisulcata* markedly decreases in body-size (109 %, Fig. 6) immediately after the Pli–Toa boundary event, from P5 to P6 (lowermost Polymorphum Zone), which is coincident with the extirpation of *Hastitidae* sp. indet. (Fig. 1).

Individual ontogenetic scale

Considering the whole time-series, the RDA analysis results revealed that taxonomic composition (47 %) and ontogeny (17 %) are responsible for most of the variation in belemnite body-size. The separation by bed explains only 2 % of the body-size variation (Fig. 3A).

At the Pli–Toa boundary event, the body-size variation partitioning into taxonomy and ontogeny is similar to what is observed for the whole time-series. However, their joint effect explains 20 % of the variation, in contrast with the 6 % observed for the whole time-series (Fig. 3A). This suggests that across the Pli–Toa boundary event, ontogenetic stages behave more similarly among taxa in explaining body-size variation.

At the individual ontogenetic scale of organisation, focusing on *Ps. longiformis*, only adult specimens show a significant body-size decrease across the Pli–Toa boundary event (21 %, Fig. 6). The ratio adult vs juveniles of *Ps. longiformis* does not change significantly across the Pli–Toa boundary event (31.25 %, Fig. 6).

In *P. bisulcata*, a reduction of 7 % (although not significant) in adult body-size is observed across the Pli–Toa boundary event, together with a slight increased percentage of juveniles of the same species (Fig. 6). Immediately after the boundary, from P5 to P6, a marked reduction (109 %) in *P. bisulcata* body-size is recognised mainly among juveniles together with an increase in their proportion (from 75 % to 68.75 %, Fig. 6).

4.2 Relationship with environmental parameters

After correcting for the effects of sedimentary properties (lithology and fossil abundance), the results of the GLS regression analysis revealed that the variation in belemnite body-size is best correlated with the variation in $\delta^{18}\text{O}$ (Table 2, model no. 5), at the assemblage scale. The overall model is significant (Table S3), suggesting a body-size reduction with increasing $\delta^{18}\text{O}$ values/decreasing seawater temperature values (Fig. S2). However, the models combining $\delta^{18}\text{O}$ with $\delta^{13}\text{C}_{\text{carb}}$ (model no. 2, Table 2) and combining Hg/TOC with $\delta^{13}\text{C}_{\text{carb}}$ (model no. 3, Table 2) could not be rejected, according to the AICc scores ($\Delta < 2$). If the data is not corrected for the effects of the sedimentary properties the same results are obtained, despite minor differences in significance (Table S5).

After correcting for the effects of sedimentary properties, when considering the most abundant taxon, *Ps. longiformis*, the best model explaining the body-size variation includes solely Hg/TOC (model no. 6, Table 2). However, the effects of $\delta^{18}\text{O}$, $\delta^{13}\text{C}_{\text{carb}}$ and Hg/TOC cannot be discarded, since statistically the models n. 5, 2 and 4 cannot be distinguished from model no. 6, according to the AICc scores. If the data is not corrected for the effects of lithology and abundance beforehand, model no.4 (Hg/TOC + $\delta^{18}\text{O}$) best explains *Ps. longiformis* body-size variation, with highly significant results (Table S5). The role of Hg/TOC in the model is more significant than the role of $\delta^{18}\text{O}$ (Table S4b).

After correcting for the effects of lithology and fossil abundance, *P. bisulcata* body-size is best correlated with Hg/TOC (model no. 6, Table 2), as well, despite the lack of significance (Table 3). However, the effects of $\delta^{18}\text{O}$, $\delta^{13}\text{C}_{\text{carb}}$ and Hg/TOC cannot be disregarded, since statistically the models no. 5, 2 and 4 cannot be distinguished from model no. 6. If no correction is applied beforehand, the model no.4 (Hg/TOC + $\delta^{18}\text{O}$) best explains the *P. bisulcata* body-size variation, with highly significant results (Table S5).

5. Discussion

R. Soc. open sci. article template

5. 1. Scales and mechanisms of body-size variation

5. 1.1. Pli-Toa boundary event

Body-size fluctuations in fossil assemblages are common and can relate to a variety of mechanisms. The assemblage size shift hypothesis (H_1) was validated and despite the fluctuations observed, only the reduction observed at the Pli-Toa boundary event is significant. By decomposing the assemblage size shift we were able to disentangle the mechanisms of belemnite body-size change, which was almost exclusively driven by a within-lineage effect.

Out of the boundary crossers, the most abundant taxon, *Ps. longiformis* is mainly driving the observed body-size reduction, allowing us to validate the within-taxa size shift hypothesis (H_3). No changes in the proportion of ontogenetic stages of *Ps. longiformis* are observed at the Pli-Toa boundary event, meaning that the largest within-lineage body-size change is not due to changes in the ontogenetic structure of the population but rather to an adult body-size reduction. In addition, at the individual ontogenetic scale, we were able to validate the intraspecific size shift hypothesis (H_3), across the Pli-Toa boundary event, finding a 51 % decrease in rostrum volume (21 % in median GM) of adult specimens of *Ps. longiformis*. In contrast, *P. bisulcata* body-size reduction is a combination of increased percentage of juveniles (increased mortality of juveniles) and reduction of adult body-size.

Despite the fact that ontogeny was not taken into account, the Lilliput effect was already identified within particular cephalopod species in the Cleveland Basin, interpreted as a response to the deteriorating environmental conditions associated with the Early Toarcian crisis (14). Similarly, among extant squid, maturation at small size and young age is interpreted as a life-history strategy to cope with warming events (8).

The differential taxa response might indicate different environmental tolerances, related to individual physiology or life history strategies (7). The only important morphological difference between *Ps. longiformis* and the other taxa is the presence of an epirostrum, a calcified structure which develops in late ontogenetic stages (78). The increased $p\text{CO}_2$ that characterises the environmental crisis of the Early Toarcian in Peniche (23) could directly (through calcification) or indirectly (through nutrient availability) have affected the calcification potential of the epirostrum. However, further data needs to be assembled in order to assess the physiological and environmental constraints of the development of such a structure.

Despite being one of the early pulses of the Early Toarcian crisis (44) and the observed body-size reduction, the Pli-Toa boundary event in Peniche does not record major changes in the belemnite taxonomic composition (Fig. 4). This is consistent with data available from other regions at higher latitudes (34, 43, 56, 60), which demonstrate that the T-OAE, in contrast, corresponds to an important turnover in belemnite species (43). In fact, the T-OAE has had caused much more impact on marine biota than the Pli-Toa boundary event (e.g. (30, 39, 40, 79), namely in the Lusitanian Basin where planktonic and benthic organisms were largely affected (27, 28, 80-82). This highlights the deterioration of the environmental conditions during the Early Toarcian, starting at the Pli-Toa boundary event and culminating in the T-OAE, a barren interval of belemnites.

5. 1.2. The aftermath of Pli-Toa boundary event

The marked body-size decrease of *P. bisulcata* (109 %) during the aftermath of the Pli-Toa boundary event (from bed P5 to bed P6, early Polymorphum Zone) occurred in tandem with a marked increase of the relative proportion of juveniles of that species (Fig. 6). This might indicate the temporary emigration of adults and/or higher juvenile mortality, due to unsuitability of the habitat. However, we also cannot rule out the existence of stunted adults which do not develop typical adult features but this could only be disentangled with further studies, namely a sclerochronological analysis.

The decrease in *P. bisulcata* body-size in the aftermath of the Pli-Toa boundary event coincides with an increase in adult body-size of *Ps. longiformis*, and with the disappearance of *Hastitidae* sp. indet. It is tempting to assume that the rapid recovery of *Ps. longiformis* potentially triggered a change in the belemnite population dynamics, potentially due to competition for food resources, causing the emigration of some species (*Hastitidae* sp. indet.) and the increased mortality and stress in early growth of others (*P. bisulcata*). Furthermore, the rest of the Polymorphum Zone (i.e. from P6 to P9) is characterized by an increase in *Ps. longiformis* body-size (and also less relative mortality of juveniles) and abundance, relatively to *P. bisulcata*. The latter becomes less abundant and

R. Soc. open sci.

R. Soc. open sci. article template

maintains its adult body-size, emphasizing a potential competition between the *Ps. longiformis* and *P. bisulcata*. However, the fact that there is no increase in the proportion of *Ps. longiformis* comparatively to the proportion of *P. bisulcata* from P5 to P6 (Fig. 6) is not entirely compatible with this interpretation and highlights the constraints of testing the effects of biotic parameters on body-size in fossil assemblages.

Notwithstanding the decrease of seawater temperature during the early Polymorphum Zone, this interval is characterised by palaeoenvironmental perturbations as evidenced by the carbon isotopic record (23, 25). Moreover, seawater temperature is still higher than during the Late Pliensbachian. The conditions are, however, not as severe as during the Pli-Toa boundary event or during the onset of the T-OAE, which is also supported by the response of other groups of marine biota in the Peniche section (27, 28, 80-82).

5.1.3. T-OAE

The onset of the T-OAE in Peniche corresponds to a belemnite gap, starting around the Polymorphum-Levisoni zones boundary (83). This is coincident with the beginning of the second mercury anomaly (52) and with warming (23), similarly to the lowermost Toarcian conditions, although associated with stronger carbon cycle perturbations (Fig. 1).

The belemnite gap could reflect inhospitable conditions in the Lusitanian Basin during the onset of the T-OAE. This is not the case during the Pli-Toa boundary event, which only caused changes in the dynamics of the population, namely in the ontogenetic structure due to temporary migration, and body size changes. This might suggest that coleoids – even those with flexible life history strategies, pre-adapted to such conditions, such as *Ps. longiformis* – could not cope with deteriorating conditions in subtropical epicontinental European basins during the T-OAE.

In fact, the deterioration of water-column conditions must have been worse during the T-OAE in comparison with the Pli-Toa boundary event, namely in terms of oxygenation of the water column, as evidenced by geochemical data (25), and the negative response of marine organisms in the Lusitanian Basin (27). Moreover, a similar belemnite response was identified in coeval north western European basins, as the belemnite gap observed in Peniche overlaps with intervals of belemnite gap or belemnite abundance decrease in the Cleveland and Idigan Bay basins sediments (43, 46, 84), interpreted as response to unfavourable anoxic-euxinic water-column conditions.

Interestingly, the decrease in belemnite abundance, or disappearance, happens before the most severe conditions of the TOAE are met in subtropical Peniche and it is not preceded by a body-size decrease. On the contrary, during the middle and upper Polymorphum Zone (i.e., from P5 to P9) belemnite body-size increases at the assemblage scale. However, this is probably related to the fact that habitats are no longer suitable for juvenile specimens and might indicate that combination of warming and deoxygenation affected belemnite physiology, reproduction, as well as competition for resources, even before the nadir of the T-OAE. This highlights the importance of climate warming in addition to regional anoxia, which have been argued to be a major constraint on belemnites abundance and distribution in the north western basins of the Tethys ocean (43, 46, 84).

Belemnites temporarily re-appear in bed P10 (upper Levisoni Zone), after the onset of the T-OAE, coinciding approximately with the end of the T-OAE negative CIE (Fig. 1), with a poor record of *Acrocoelites*. This genus is known to appear and radiate in the north-eastern basins of the NW Tethys (34, 43, 44, 56, 59) after the T-OAE, replacing the genus *Passaloteuthis*, which has been interpreted as an evolutionary adaptation in terms of ecological preferences related to the stressful conditions of the T-OAE (46). Moreover, the change from the Late Pliensbachian/Early Toarcian Belemnitinae-dominated assemblage to the Middle Toarcian Megateuthidinae-dominated assemblage (together with the disappearance of smaller sized species such as *Hastitidae* sp. indet., *Parapassaloteuthis* sp. A, *P. milleri*, *Ps. longiformis*), might not only be related to ecological preferences but also to selective extinction, as pointed out by (15), who states that past extinction events were either nonselective or preferentially removed smaller-bodied taxa.

However, the lack of a continuous belemnite record during the Levisoni Zone in Peniche hampers a more detailed assessment. Further research in other basins is necessary to disentangle the extent of this phenomenon.

5.2. The effect of sedimentary facies on belemnite body-size

R. Soc. open sci.

R. Soc. open sci. article template

Despite the fact that both of the studied events are considered to be geologically rapid (85-87) their representation in the sedimentary record is dependent on facies evolution dependent of several factors such as regional tectonics, climate and sea level changes the role of these parameters control the sedimentation rate, which is not constant across the studied interval. We use lithology to account for changes in the facies (related with relative sea-level changes) and belemnite abundance to account for changes in the sedimentation rate, since both are thought to have an effect on morphological patterns (88).

Accumulations with abundant belemnites, particularly observed in beds P4 and P5, could be related to changes in sedimentation rate relative to the accumulation of belemnites (89). The lowermost Polymorphum Zone in Peniche is thought to correspond to a condensed interval (58, 81, 90). Our data reveals that belemnite body-size is larger in beds with higher belemnite abundance (Fig. S10). Abundance, however, does not markedly change across the boundary, where the most significant size change is observed. Regarding lithology, belemnite body-size is larger in marls in comparison with limestones but these effects are not strong (Fig. S9).

When studying the effect of environmental perturbations on belemnite body-size, at the scale of the assemblage, lithology and abundance do not affect the pattern observed. Before and after correction, temperature is always the best parameter in explaining the body-size variation. However, the significance is higher without correction which is not surprising considering that residualising might also filter out potential ecological signals contained within the sedimentary properties in addition to the effect of preservation and collection biases. Even with the correction for the effects of sedimentary properties, and despite the existence of a significant model, the analysis failed in revealing a single model to explain belemnite body-size variation. Instead, several models are very similar to the model with the best AICc score. This highlights the strong interaction between abiotic parameters in driving body-size patterns during episodes of climate warming.

However, when analysing the data at species level (*P. bisulcata* and *Ps. longiformis*), the best model explaining the body-size variation after correction is Hg/TOC as opposed to temperature without correction in both species. This might be related to the smaller datasets, especially in the case of *P. bisulcata*, which does not allow to distinguish between individual models. Either way, the increased warming, at the assemblage scale, and Hg/TOC anomalies, as drivers of individual species body-size variation, are not incompatible – it just shows that changes in warming generally correlate with decreasing sizes of belemnite assemblages across the Pli-Toa boundary and that the impact of “catastrophic” environmental perturbations triggered by volcanism are the most severe and probably work in concert in their impact on body-size decrease in individual species.

The lowermost Toarcian (base of Polymorphum Zone) in Peniche also coincides with a marine transgression (57, 58) which might have impacted on the size distribution, but differences in lithology and sedimentation rates could not clearly explain differences in our dataset. Furthermore, deepening would rather result in finding larger species (69), which would be consistent with our observations of finding larger specimens in marls. But a body size decrease is seen across the boundary. Further analyses across different parts of the basin would be necessary to disentangle its potentially less subtle effects in shallower sections.

5.3. Environmental drivers of body-size fluctuations

The Pli-Toa boundary corresponds to the first episode of stress in the Early Toarcian palaeoenvironmental crisis (23) and has been associated with the first pulse of Karroo-Ferrar volcanism (52). This had a major effect on climate, disturbing the entire ocean-atmosphere system, by causing carbon cycle perturbations and fluctuations in the seawater temperature. Our results are consistent with the hypothesis that climate is an important driver of the belemnite body-size variation at the assemblage scale, with a decreasing body-size associated with increasing seawater temperature (Fig. S8). Salinity could however partially affect the interpretation of the oxygen isotopic data, particularly in north-western European basins, where the input of brackish and nutrient-rich Arctic surface waters seems to affect the signal (91, 92). However, decreased salinity has been deemed less significant in Peniche than warming and carbon cycle perturbations based on phytoplankton communities (28). Furthermore, modelling suggest that, at this palaeocoordinates, salinity changes are close to zero (91, 92).

This reduction in adult size of *Ps. longiformis* and *P. bisulcata* – the Lilliput effect – can potentially be compared with the decrease in adult mantle length of extant squid during rapid warming events (8, 10). To what degree the decrease in adult size relates with direct effects of warming on physiology (development rate, metabolism,

R. Soc. open sci. article template

respiration) or rather indirect effects of environmental changes on resource acquisitions affecting growth and early development is still under debate (7, 93).

Apart from increasing seawater temperature, the mercury anomalies and carbon cycle perturbations at the Pli–Toa boundary event in Peniche, also coincide with increase weathering (25), local decreased productivity (27, 28), increasing $p\text{CO}_2$ (and potentially ocean acidification) and decreasing dissolved oxygen (23). All of these factors are expected to work in concert in driving marine organisms' body size (6) and our analysis of belemnite body size is consistent with that. This is also reflected in the strong collinearity of environmental and biotic proxies related to these effects.

Strontium isotopes from Peniche support the idea of increased weathering during the Pli–Toa boundary warming event and potentially nutrient influx (25), which is also consistent with the interpretation of enhanced mercury concentration in the sediments. However, contra–intuitively this warming event is interpreted to lead to a decrease in primary productivity, with a decrease in both phytoplankton abundance (27, 28). This emphasizes how warming and related stressors might in turn affect the size of primary consumers and potentially other levels of the food chain, such as predators like cephalopods. Dwarfing has been observed in response to lack of food in the laboratory, but so far not in the wild cephalopods (94). In addition, macroecological studies show that body–size distribution in extant cephalopods is better explained by seawater temperature than productivity (69).

The negative CIE observed during the Early Toarcian is interpreted to be related to volcanic outgassing either directly by the rapid input of isotopically light carbon into the atmosphere–hydrosphere system (51), or indirectly by triggering various sources of isotopically light carbon (19). This would not only have caused rapid climate warming, but also increased $p\text{CO}_2$ and decreased pH of the seawater. Cephalopods are usually interpreted to be quite resistant to ocean acidification in comparison with other marine organisms – cuttlefish even increase calcification (95, 96). However, the limited data available hitherto seem to indicate that the effects of acidification might be more severe in early ontogeny, resulting in pathologies (96, 97). Nonetheless, no clear signs of aberrant development were observed in the studied rostra. More importantly, there are no direct proxies for marked ocean acidification during the Early Toarcian in the Lusitanian Basin and only indirect evidences, such as calcareous nannofossils (*Schizosphaerella*) size reduction. However, this has been interpreted as an indirect consequence of $p\text{CO}_2$ increase, due to changes in the climate and sea–level (27), rather than a direct cause of high $p\text{CO}_2$.

Together with the deposition of organic black shales, the Early Toarcian negative CIE has been traditionally interpreted to reflect increased anoxia in the ocean (98, 99). Irrespective of the potential local or regional overprint of increasing stratification due to basin restriction or **artic** water input, the perturbations at the Pli–Toa boundary event in Peniche are thought to reflect the start of decreasing oxygenation of seawater while the T–OAE perturbations represent the global nadir of anoxia (100). Nonetheless, there is no evidence for bottom–water anoxia in the Lusitanian Basin (25, 101), in contrast with the north–western European basins (102). Despite that fact, less severe deoxygenation might have played a role in belemnite body–size and distribution, since increased seawater temperature results in reduced oxygen availability relative to demand (103).

The strong collinearity between biotic and abiotic factors available in the literature for the Peniche section (Fig. S4) hampers an analysis of their individual effect on belemnite body–size. Nonetheless, this study allows to distinguish direct and indirect effects of climate warming on belemnite body–size reduction, despite the complex interaction between biotic and abiotic factors, highlighting the importance of taking into account different scales of organization.

6. Conclusion

We document for the first time a median body–size decrease of belemnites across the Pli–Toa boundary event at different scales of palaeobiological organisation in the Peniche reference section. We find no evidence for a major taxonomic turnover (similar to results in more high–latitude sections) and the decrease is mainly driven by a decrease in adult size of the dominant taxon *Ps. longiformis* – Lilliput effect. This coincides with the onset of the first pulse of the Upper Pliensbachian – Lower Toarcian palaeoenvironmental crisis, probably triggered by

R. Soc. open sci. article template

volcanism of the Karoo Ferrar igneous province. Our results indicate that climate warming best explains the body-size fluctuations observed, although the interplay with perturbations of the carbon cycle and other factors triggered by increased volcanogenic outgassing are evident. Our results suggest that morphological responses precede extinction pulses in belemnites (i.e. during the T–OAE) in the Lusitanian Basin and highlight that decreasing adult body-size might rather be a life-history strategy to deal with temporarily deteriorating conditions related to warming than being a result of taxonomic turnover. Furthermore, changes within the belemnite assemblage, such as competition and emigration dynamics, seem to play a role in explaining belemnite body size variation, albeit minor.

Acknowledgments

We thank Birgit Leipner–Mata and Manuel Blank for helping in belemnite preparation, and Benjamin Gügel, Christian Schulbert and Martina Schlott for helping with scanning the specimens. We thank Manuel Steinbauer, Carl Reddin, Wolfgang Kiessling and Vanessa Roden for the valuable comments on the manuscript.

Funding Statement

This is a contribution to the DFG Research Unit FOR 2332 (grant number Ba 5148/1–1 to KDB) TERSANE and to the IGCP 655 (IUGS–UNESCO).

Data Accessibility

The datasets supporting this article have been uploaded as part of the Supplementary Material, together with supplementary figures and tables.

The specimens are stored in the Science Museum (Museu da Ciência, University of Coimbra, Portugal) and the 3D data will be stored on Zenodo.

Competing Interests

We declare we have no competing interests.

Authors' Contributions

PR participated in the design of the study, carried out the data collection and analysis, interpreted the results and drafted the manuscript. KDB designed and coordinated the study, participated in the data analysis and helped interpreting the results and drafting the manuscript. The fieldwork was carried out by PR, KDB and LVD. PN helped with data collection and preliminary data analysis. LVD helped interpreting the results. RW helped with the taxonomic work. All authors contributed to the writing process and gave final approval for publication.

References

1. Schmidt DN, Lazarus D, Young JR, Kucera M. Biogeography and evolution of body size in marine plankton. *Earth-Science Reviews*. 2006;78(3):239–66.

R. Soc. open sci. article template

2. Daufresne M, Lengfellner K, Sommer U. Global warming benefits the small in aquatic ecosystems. *Proceedings of the National Academy of Sciences*. 2009;106(31):12788-93.
3. Gardner JL, Peters A, Kearney MR, Joseph L, Heinsohn R. Declining body size: a third universal response to warming? *Trends in ecology & evolution*. 2011;26(6):285-91.
4. Sheridan JA, Bickford D. Shrinking body size as an ecological response to climate change. *Nature Climate Change*. 2011;1(8):401-6.

[revised manuscript text omitted]

58. Rocha RBd, Mattioli E, Duarte LV, Pittet B, Elmi S, Mouterde R, et al. Base of the Toarcian Stage of the Lower Jurassic defined by the Global Boundary Stratotype Section and Point (GSSP) at the Peniche section (Portugal). *Episodes*. 2016;39(3).
59. Doyle P. The British Toarcian (lower Jurassic) Belemnites: Part 2: Palaeontographical Society; 1992.
60. Pinard J-D, Weis R, Neige P, Mariotti N, Di Cencio A. Belemnites from the Upper Pliensbachian and the Toarcian (Lower Jurassic) of Tournadous (Causses, France). *Neues Jahrbuch für Geologie und Paläontologie-Abhandlungen*. 2014;273(2):155-77.
61. Sanders MT, Bardin J, Benzaggagh M, Cecca F. Early Toarcian (Jurassic) belemnites from northeastern Gondwana (South Riffian ridges, Morocco). *Paläontologische Zeitschrift*. 2015;89(1):51-62.

R. Soc. open sci. article template

-
62. Tomašových A, Schlögl J, Biroň A, Hudáčková N, Mikuš T. Taphonomic Clock and Bathymetric Dependence of Cephalopod Preservation in bathyal, sediment-starved environments. *PALAIOS*. 2017;32(3):135-52.
63. Rita P, De Baets K, Schlott M. Rostrum size differences between Toarcian belemnite battlefields. *Fossil Record*. 2018;21(1):171-82.
64. Monks N, Hardwick JD, Gale AS. The Function of the Belemnite Guard. *Paläontologische Zeitschrift*. 1996;70(3):425.
65. Benito MI, Reolid M, Viedma C. On the microstructure, growth pattern and original porosity of belemnite rostra: insights from calcitic Jurassic belemnites. *Journal of Iberian Geology*. 2016;42(2):201.
66. Hoffmann R, Richter DK, Neuser RD, Jöns N, Linzmeier BJ, Lemanis RE, et al. Evidence for a composite organic–inorganic fabric of belemnite rostra: Implications for palaeoceanography and palaeoecology. *Sedimentary Geology*. 2016;341:203-15.
67. Reitner J, Urlichs M. Echte Weichteilbelemniten aus dem Untertoarcium (Posidonienschiefer) Südwestdeutschlands. *Neues Jahrbuch für Geologie und Paläontologie*. 1983;165(3):450-65.
68. Klug C, Schweigert G, Fuchs D, Kruta I, Tischlinger H. Adaptations to squid-style high-speed swimming in Jurassic belemnitids. *Biology Letters*. 2016;12(1):20150877.
69. Rosa R, Gonzalez L, Dierssen HM, Seibel BA. Environmental determinants of latitudinal size-trends in cephalopods. *Marine Ecology Progress Series*. 2012;464:153-65.
70. Pecl GT, Jackson GD. The potential impacts of climate change on inshore squid: biology, ecology and fisheries. *Reviews in Fish Biology and Fisheries*. 2008;18(4):373-85.
71. Rego BL, Wang SC, Altiner D, Payne JL. Within- and among-genus components of size evolution during mass extinction, recovery, and background intervals: a case study of Late Permian through Late Triassic foraminifera. *Paleobiology*. 2012;38(4):627-43.
72. Oksanen J, Blanchet FG, Kindt R, Legendre P, Minchin P, O’Hara RB, et al. Vegan: community ecology package. R package version 2.5-2 (<https://CRAN.R-project.org/package=vegan>). 2018.
73. Jakob EM, Marshall SD, Uetz GW. Estimating fitness: a comparison of body condition indices. *Oikos*. 1996;77(1):61-7.
74. Box GEP, Jenkins GM. *Time Series Analysis: Forecasting and Control*: Holden-Day; 1994.
75. R Development Core Team. *R: a language and environment for statistical computing*. Vienna, Austria: R Foundation for Statistical Computing. Retrieved from <http://www.R-project.org>. 2018.
76. Pinheiro J BD, DebRoy S, Sarkar D, R Core Team. nlme: Linear and Nonlinear Mixed Effects Models. R package version 3.1-137 (<https://CRAN.R-project.org/package=nlme>). 2018.
77. Spiess A-N. qpcR: Modelling and Analysis of Real-Time PCR Data. R package version 1.4-1 (<https://cran.r-project.org/web/packages/qpcR/>). 2018.
78. Arkhipkin A, Weis R, Mariotti N, Shcherbich Z. ‘Tailed’ cephalopods. *Journal of Molluscan Studies*. 2015;81(3):345-55.
79. Danise S, Twitchett RJ, Little CT, Clemence ME. The impact of global warming and anoxia on marine benthic community dynamics: an example from the Toarcian (Early Jurassic). *PLoS One*. 2013;8(2):e56255.
80. Comas-Rengifo MJ, Duarte LV, Felix FF, Joral FG, Goy A, Rocha RB. Latest Pliensbachian–Early Toarcian brachiopod assemblages from the Peniche section (Portugal) and their correlation. *Episodes*. 2015;38(1):2-8.
81. Rita P, Reolid M, Duarte LV. Benthic foraminiferal assemblages record major environmental perturbations during the Late Pliensbachian–Early Toarcian interval in the Peniche GSSP, Portugal. *Palaeogeogr, Palaeoclimatol, Palaeoecol*. 2016;454:267-81.
82. Reolid M, Duarte L, Rita P. Changes in foraminiferal assemblages and environmental conditions during the T-OAE (Early Jurassic) in the northern Lusitanian Basin, Portugal. *Palaeogeography, Palaeoclimatology, Palaeoecology*. 2019.
83. Duarte LV, Comas-Rengifo MJ, Hesselbo S, Mattioli E, G. Suan, Baker S, et al. The Toarcian Oceanic Anoxic Event at Peniche. An exercise in integrated

R. Soc. open sci. article template

- stratigraphy - Stop 1.3. Field trip Guidebook: The Toarcian Oceanic Anoxic Event in the Western Iberian Margin and its context within the Lower Jurassic evolution of the Lusitanian Basin 2018. p. 54.
84. Xu W, Ruhl M, Jenkyns HC, Leng MJ, Huggett JM, Minisini D, et al. Evolution of the Toarcian (Early Jurassic) carbon-cycle and global climatic controls on local sedimentary processes (Cardigan Bay Basin, UK). *Earth and Planetary Science Letters*. 2018;484:396-411.
85. Boulila S, Galbrun B, Sadki D, Gardin S, Bartolini A. Constraints on the duration of the early Toarcian T-OAE and evidence for carbon-reservoir change from the High Atlas (Morocco). *Global and Planetary Change*. 2019;175:113-28.
86. Huang C, Hesselbo SP. Pacing of the Toarcian Oceanic Anoxic Event (Early Jurassic) from astronomical correlation of marine sections. *Gondwana Research*. 2014;25(4):1348-56.
87. Suan G, Pittet B, Bour I, Mattioli E, Duarte LV, Mailliot S. Duration of the Early Toarcian carbon isotope excursion deduced from spectral analysis: Consequence for its possible causes. *Earth and Planetary Science Letters*. 2008;267(3):666-79.
88. Holland SM. The quality of the fossil record: a sequence stratigraphic perspective. *Paleobiology*. 2000;148-68.
89. Doyle P, Macdonald DI. Belemnite battlefields. *Lethaia*. 1993;26(1):65-80.
90. Pittet B, Suan G, Lenoir F, Duarte LV, Mattioli E. Carbon isotope evidence for sedimentary discontinuities in the lower Toarcian of the Lusitanian Basin (Portugal): Sea level change at the onset of the Oceanic Anoxic Event. *Sedimentary geology*. 2014;303:1-14.
91. Ruvalcaba Baroni I, Pohl A, van Helmond NA, Papadomanolaki NM, Coe AL, Cohen AS, et al. Ocean circulation in the Toarcian (Early Jurassic): A key control on deoxygenation and carbon burial on the European Shelf. *Paleoceanography and Paleoclimatology*. 2018;33(9):994-1012.
92. Dera G, Donnadieu Y. Modeling evidences for global warming, Arctic seawater freshening, and sluggish oceanic circulation during the Early Toarcian anoxic event. *Paleoceanography and Paleoclimatology*. 2012;27(2).
93. Audzijonyte A, Barneche DR, Baudron AR, Belmaker J, Clark TD, Marshall CT, et al. Is oxygen limitation in warming waters a valid mechanism to explain decreased body sizes in aquatic ectotherms? *Global Ecology and Biogeography*. 2019;28(2):64-77.
94. Boletzky Sv. Effets de la sous-nutrition prolongée sur le développement de la coquille de *Sepia officinalis* L. (Mollusca, Cephalopoda). *Bulletin de la Société zoologique de France*. 1974;99(4):667-73.
95. Gutowska MA, Melzner F, Pörtner HO, Meier S. Cuttlebone calcification increases during exposure to elevated seawater pCO₂ in the cephalopod *Sepia officinalis*. *Marine Biology*. 2010;157(7):1653-63.
96. Dorey N, Melzner F, Martin S, Oberhänsli F, Teysse J-L, Bustamante P, et al. Ocean acidification and temperature rise: effects on calcification during early development of the cuttlefish *Sepia officinalis*. *Marine Biology*. 2013;160(8):2007-22.
97. Lacoue-Labarthe T, Reveillac E, Oberhänsli F, Teysse J-L, Jeffree R, Gattuso J. Effects of ocean acidification on trace element accumulation in the early-life stages of squid *Loligo vulgaris*. *Aquatic Toxicology*. 2011;105(1-2):166-76.
98. Ikeda M, Hori RS, Ikehara M, Miyashita R, Chino M, Yamada K. Carbon cycle dynamics linked with Karoo-Ferrar volcanism and astronomical cycles during Pliensbachian-Toarcian (Early Jurassic). *Global and Planetary Change*. 2018;170:163-71.
99. Jenkyns HC. Geochemistry of oceanic anoxic events. *Geochemistry, Geophysics, Geosystems*. 2010;11(3).
100. Them TR, Gill BC, Caruthers AH, Gerhardt AM, Gröcke DR, Lyons TW, et al. Thallium isotopes reveal protracted anoxia during the Toarcian (Early Jurassic) associated with volcanism, carbon burial, and mass extinction. *Proceedings of the National Academy of Sciences*. 2018;115(26):6596-601.
101. Rodríguez-Tovar FJ, Miguez-Salas O, Duarte LV. Toarcian Oceanic Anoxic Event induced unusual behaviour and palaeobiological changes in *Thalassinoides* tracemakers. *Palaeogeography, Palaeoclimatology, Palaeoecology*. 2017;485:46-56.

R. Soc. open sci. article template

102. McArthur JM, Algeo TJ, van de Schootbrugge B, Li Q, Howarth RJ. Basinal restriction, black shales, Re-Os dating, and the Early Toarcian (Jurassic) oceanic anoxic event. *Paleoceanography*. 2008;23(4).

103. Verberk WCEP, Atkinson D. Why polar gigantism and Palaeozoic gigantism are not equivalent: effects of oxygen and temperature on the body size of ectotherms. *Functional Ecology*. 2013;27(6):1275-85.

104. Cárdenas AL, Harries PJ. Effect of nutrient availability on marine origination rates throughout the Phanerozoic eon. *Nature Geoscience*. 2010;3(6):430.

R. Soc. open sci. article template

Tables

Table 1

Geologic proxy [source]	Environmental variable	Basis	Interpretation in the context of the LB	Theoretical controls on body-size
Brachiopod stable oxygen isotope ratio ($\delta^{18}\text{O}$) [(23)]	Dominantly temperature	Temperature-dependent isotopic fractionation between carbonate minerals and seawater (104)	High temperatures during negative excursions	Increasing temperature through various mechanisms including increase of both growth and development rates are expected to lead to a smaller adult body-size (7)
$\delta^{13}\text{C}_{\text{carb}}$ [(25)]	Carbon cycle perturbations related with anoxia	Isotopic fractionation between photosynthesisers and seawater. Photosynthesisers remove light C^{12} from the seawater and this is incorporated into the sea bottom sediment after burial as organic carbon (99).	Negative excursions during the Early Toarcian reflect enhanced burial of organic carbon during the zenith of anoxia (100). They could also be influenced by increased primary production but are interpreted to reflect rather widespread oxygen depletion as primary productivity is interpreted to have dropped during these intervals (27, 28)	At the edge of an organism's temperature range, growth is usually impaired by insufficient energy or oxygen supply, decreasing both growth rate and body size at any developmental stage (7).
Hg/TOC ratio [(52)]	Volcanogenic outgassing	Volcanism represents a source of mercury to the atmosphere. Due to rain or runoff, mercury moves from the terrestrial realm/ atmosphere to the marine realm in mineral form or adsorbed in detrital organics. Hg burial is limited by low abundance and/or burial of organic matter or sulphids that would scavenge aqueous Hg (50, 52).	Hg anomalies (high Hg/TOC ratio) in the sediment are interpreted to represent markers of volcanism in distal sections. They have been interpreted to reflect volcanic outgassing, but their interpretation in some distal sections might be more complex (50).	Increasing $p\text{CO}_2$, combined with increasing temperature, causes a decrease in both dissolved oxygen and pH of the seawater. In association with other factors related to rapid warming, such as weathering and eutrophication, these factors are expected to lead to a body-size decrease due to their direct effects on the availability of food resources (6).

Table 2

	Model no.	Δ	AIC c scores	ANOVA p-value
Assemblage				

R. Soc. open sci. article template

GM ~ 1	null	5.09	1921.87	–
GM ~ $\delta^{18}\text{O}$	5	0	1920.92	0.0849
GM ~ $\delta^{18}\text{O} + \delta^{13}\text{C}_{\text{carb}}$	2	1.99	1922.91	
GM ~ Hg/TOC + $\delta^{13}\text{C}_{\text{carb}}$	3	1.91	1922.84	
Ps. longiformis				
GM ~ 1	null	2.51	671.37	–
GM ~ Hg/TOC	6	0	668.87	0.0327
GM ~ $\delta^{18}\text{O} + \delta^{13}\text{C}_{\text{carb}}$	2	0.87	669.74	
GM ~ Hg/TOC + $\delta^{18}\text{O}$	4	1.31	670.17	
GM ~ $\delta^{18}\text{O}$	5	0.50	669.37	
P. bisulcata				
GM ~ 1	null	–1.46	607.97	–
GM ~ Hg/TOC	6	0	609.43	0.4306
GM ~ $\delta^{13}\text{C}_{\text{carb}} + \text{Hg/TOC}$	3	1.87	611.30	
GM ~ $\delta^{18}\text{O}$	5	0.35	609.78	
GM ~ $\delta^{13}\text{C}_{\text{carb}}$	7	0.58	610.01	

Table 3

scale	model no.	coefficients	value	std.error	t-value	p-value	degrees of freedom	residual
Assemblage	5	Intercept	2.44	1.48	1.64	0.1019	340	338
		$\delta^{18}\text{O}$	2.04	1.19	1.72	0.061	–	
Ps. longiformis	15	Intercept	0.31	0.23	1.32	0.1904	160	158
		Hg/TOC	–2	0.93	–2.14	0.0335	–	
P. bisulcata	5	Intercept	0.43	0.94	0.46	0.6458	101	99
		Hg/TOC	–3.15	4.02	–0.78	0.4358	–	

Figures

Figure 1

R. Soc. open sci. article template

Figure 2

Figure 3

R. Soc. open sci. article template

Figure 4

R. Soc. open sci. article template

Figure 5

Figure 6

captions

Figure and table

R. Soc. open sci. article template

Figure 1 –Variation of belemnite body-size (GM, assemblage scale), absolute abundance (no. of belemnites/m²) and stratigraphic ranges of species, compared with the variation of the analysed geologic proxies (mercury concentration, Hg/TOC (24), carbon and oxygen isotopes, $\delta^{13}\text{C}_{\text{carb}}$ and $\delta^{18}\text{O}$ (22, 23)) and sequence stratigraphy data (50) from the Upper Pliensbachian-Lower Toarcian of Peniche. The shaded area highlights the Pli-Toa boundary event. The bed numbers in square brackets correspond to the beds that were merged due to sample size constraints, regarding the belemnite body-size analysis. See Fig. S2 for details on belemnite abundance. BG=belemnite gap; T=Transgressive; R=Regressive.

Figure 2– Conceptual scheme depicting the tested hypotheses regarding the mechanisms behind belemnites body-size reduction, at different palaeobiological scales of organisation.

Figure 3 – A: Venn diagram depicting the partition of belemnites body-size variation between taxonomic composition, bed separation and ontogeny. The values between parentheses correspond to the whole time-series and the values without parentheses correspond to the Pli-Toa boundary. B: Effect on size change at the assemblage and species scale of organisation. See Table S6 for details. For a correct interpretation of the right-side scale, the bottom of the arrowhead should be considered, rather than the tip of the arrow. Note that the x-axis depicts comparisons of consecutive pairs of beds (not to scale). BG = belemnite gap.

Figure 4 – Relative frequency of the different taxa comprising the Upper Pliensbachian-Lower Toarcian Peniche belemnite assemblage. Note that determinative complete specimens were also taken into account. The error bars correspond to the 95 % confidence interval of the relative frequency of the species *Ps. longiformis* and *P. bisulcata*.

Figure 5 –Lithology, belemnite body-size variation (GM), proportional body-size change and relative frequency of ontogenetic stages at the assemblage scale across the studied interval in Peniche. Note that the stratigraphic log is not drawn to scale, for real thickness of beds, see Fig. 1. The error bars correspond to the 95 % confidence interval. BG = belemnite gap.

Figure 6 – Belemnite body-size variation (GM), proportional body-size change, relative frequency of ontogenetic stages and sample size across the Upper Pliensbachian-Lower Toarcian of Peniche at species scale (*P. bisulcata* and *Ps. longiformis*). *Passaloteuthis* genus was used to calculate the relative frequency of ontogenetic stages of *P. bisulcata* due to the difficulty of a species level classification of the juvenile specimens of *Passaloteuthis* genus. The error bars correspond to the 95 % confidence interval. The error bars correspond to the 95 % confidence interval. For sample size, see Fig. S6.

R. Soc. open sci. article template

Table 1 – Environmental constraints (abiotic parameters) based on their geological proxies and their theoretical role in mediating body–size changes and their interpretation in the context of the Lusitanian Basin (LB).

Table 2 – AICc ranking of models describing the effect of palaeotemperature ($\delta\delta^{18}\text{O}$), carbon cycle perturbations ($\delta\delta^{13}\text{C}_{\text{carb}}$) and volcanism (Hg/TOC) on belemnites body–size (GM) for the assemblage and species scale (*Ps. longiformis* and *P. bisulcata*), corrected for the effects of sedimentary properties (lithology and belemnite abundance). Only the models with $\Delta < 2$ are listed. See TS4 for the full list of GLS models.

Table 3 – Details of the selected GLS models comparing belemnite body–size (GM) with palaeotemperature ($\delta\delta^{18}\text{O}$), carbon cycle perturbations ($\delta\delta^{13}\text{C}_{\text{carb}}$) and volcanism (Hg/TOC), corrected for the effects of sedimentary properties (lithology and belemnite abundance).

R. Soc. open sci. article template

Supplementary figure and table captions

Table S1 – Measured body-size parameters on belemnite specimens from the Upper Pliensbachian-Toarcian interval of the Peniche section and selected environmental parameters from the literature (ESM file 1).

Table S2 – Log ratio, sample size and results of the statistical tests on the differences in the median of pairs of beds (ESM file 2).

Table S3 – Partitioning analysis results highlighting the relative variation fractions (proportions) of belemnites body size explained by taxonomic composition, beds and ontogeny for the whole time-series and for the pair-to-pair analysis. Bold indicates a minimum of 90 % statistical significance level (ESM file 3).

Table S4 – Details of GLS regression analysis comparing belemnite body-size and the environmental parameters: a – details of the selected GLS models, after correcting for lithology and abundance; b – Details of the selected GLS models, without correcting for lithology and abundance (raw data); c– Full list of the analysed models for assemblage and species scales, with AICc scores and delta values indicated, together with the p-values for the models with the smaller AICc score (ESM file 4).

Table S5 – AICc ranking of models describing the effect of palaeotemperature ($\delta^{18}\text{O}$), carbon cycle perturbations ($\delta^{13}\text{C}_{\text{carb}}$) and volcanism (Hg/TOC) on belemnites body-size for the assemblage and species scale (*Ps. longiformis* and *P. bisulcata*), not corrected for the effects of sedimentary properties (lithology and belemnite abundance). Only the models with $\Delta < 2$ are listed. See TS4 for the full list of GLS model (ESM file 5).

Table S6 – Detailed values of the components of the assemblage size shift calculated with Rego et al. method (disappearance of taxa effect, appearance of new taxa effect and within-lineage effect) (see ESM file 6). Table S7 – Micro-CT phoenix v|tome|x s 240 (Research Edition) scanner settings for the scanned belemnite specimens. The reconstruction was made with the GEDatos|x 2.4 software. Subsequent image stack processing (e.g. subsampling), as well as the measurements and volume acquisition, was derived using Studio Volume Graphics Max™ v 3.0 software (Heidelberg) (ESM file 6).

Table S7 – Micro-CT phoenix v|tome|x s 240 (Research Edition) scanner settings for the scanned belemnite specimens. The reconstruction was made with the GEDatos|x 2.4 software. Subsequent image stack processing (e.g. subsampling), as well as the measurements and volume acquisition, was derived using Studio Volume Graphics Max™ v 3.0 software (Heidelberg) (ESM file 7).

R. Soc. open sci. article template

Figure S1 – Selected belemnites from the Upper Pliensbachian-Lower Toarcian of Peniche: A1 - *Bairstowius* sp. A (adult from bed P1); A2- *Bairstowius* sp. A (adult from bed P2); A3- *Bairstowius* sp. A (juvenile from bed P1); B1- *Pseudohastites longiformis* (adult from bed P4); B2 and B3 - *Pseudohastites longiformis* (juveniles from bed P4); C1 and C2- *Hastitidae* sp. indet. (from bed P5); D1 and D2 - *Passaloteuthis milleri* (adults from bed P2). The left side corresponds to lateral view and right side corresponds to dorsal/ventral view (ESM file 8).

Figure S2 – Selected belemnites from the Upper Pliensbachian-Lower Toarcian of Peniche: E1- *Parapassaloteuthis* sp. A (adult from bed P3); E2 - *Parapassaloteuthis* sp. A (juvenile from bed P3); F - *Passaloteuthis* sp. juv. (from bed P1); G1-G2 - *Passaloteuthis bisulcata* (adults from bed P5); G3 - *Passaloteuthis bisulcata* (adult from bed P2). The left side corresponds to lateral view and right side corresponds to dorsal/ventral view (ESM file 8).

Figure S3 – Belemnite absolute abundance (no. of belemnites collected per m²). The interrupted black line connects the mean (black dots) for the beds where more than one quadrat were analysed. Each red dot represents a quadrat of 1 m². The error bars represent the standard deviation of the mean (ESM file 8).

Figure S4 – Correlation matrix depicting the relation between the different environmental variables available in the literature for the Peniche section. Note that only $\delta^{18}\text{O}$, $\delta^{13}\text{C}_{\text{carb}}$, Hg/TOC, lithology and abundance were included as explanatory variables in the regression analysis due to the high collinearity among the remaining ones. See TS1 for details (ESM file 8).

Figure S5– Correlation matrix depicting the relation between the different explanatory variables used in the performed regression analysis between belemnite body-size and abiotic parameters (ESM file 8).

Figure S6 – Belemnite body-size variation (GM), proportional body-size change, relative frequency of ontogenetic stages and sample size (n) across the Upper Pliensbachian-Lower Toarcian of Peniche at species scale (*P. bisulcata* and *Ps. longiformis*). *Passaloteuthis* genus was used to calculate the relative frequency of ontogenetic stages of *P. bisulcata* due to the difficulty of a species level classification of the juvenile specimens of *Passaloteuthis* genus. The error bars correspond to the 95 % confidence interval. Note that the sample size (n) corresponds to species level (ESM file 8).

Figure S7 – Body-size variation (GM) and sample size (n) of Pliensbachian-Toarcian boundary-crossers (*P. bisulcata*, *Ps. longiformis* and *Hastitidae* sp. indet.) of the Peniche section. Although *Parapassaloteuthis* sp. A also crosses the Pli-Toa boundary, we have no complete specimens across this interval and therefore, no accurate estimate of body-size (ESM file 8).

Figure S8 – Relationship between the seawater palaeotemperature proxy ($\delta^{18}\text{O}$) and belemnite body-size (GM) during the Upper Pliensbachian-Lower Toarcian of Peniche. The grey area corresponds to the 95 % confidence interval (ESM file 8).

Figure S9 – Relationship between belemnite body-size (GM) and lithology for the studied interval in Peniche. Adjusted R²=0.0009292; p-value=0.4083. The grey area corresponds to the 95 % confidence interval (ESM file 8).

Figure S10 – Relationship between belemnite body-size (GM) and belemnite absolute abundance (no. of belemnites/ m²) for the studied interval in Peniche. The grey area corresponds to the 95 % confidence interval. Adjusted R²=9.356e-05; p-value=0.3105 (ESM file 8).

Mechanisms and drivers of belemnite body-size dynamics across the
Pliensbachian-Toarcian crisis

Patrícia Rita^{1,2}, Paulina Nätscher¹, Luís V. Duarte^{2,3}, Robert Weis⁴, Kenneth
De Baets¹

¹Geozentrum Nordbayern, Friedrich-Alexander Universität Erlangen-Nürnberg, Erlangen 91054,
Germany

²MARE (Marine and Environmental Sciences Centre), 3004-517 Coimbra, Portugal

³Department of Earth Sciences University of Coimbra, 3070-790 Coimbra, Portugal

⁴National Museum of Natural History Luxembourg, Department of Palaeontology, L-2160 Luxembourg

Correspondence to: Patrícia Rita (patricia.rita@fau.de)

~~Body-size reduction is considered an important response to current climate warming~~
~~and has also been observed during past mass-extinction episodes of biotic crisis biotic crises.~~
~~Body-size reduction has been attributed to climate warming under scenarios of mass~~
~~extinction episodes. However, in fossil cephalopods studies concerning belemnites, the~~
~~mechanisms and environmental drivers of those changes their potential link with climate are~~
~~rarely investigated and they usually do not differentiate biological-palaeobiological scales of~~
~~organisation (individual, population and assemblage). We hypothesize that belemnites reduce~~
~~their adult body-size across the Pliensbachian-Toarcian (Pli-Toa) boundary warming event~~
~~(Pli-Toa).~~

Belemnite body-size dynamics across the Pliensbachian-Toarcian Pli-Toa in the
Peniche section (Lusitanian Basin, Portugal, Portugal) were analysed for the first time based on
field data. We disentangle the mechanisms and the environmental drivers of the size
fluctuations observed from the individual to the assemblage scale of organisation. Despite the
lack of a major taxonomic turnover, a 40 % there is a significant decrease in rostrum volume is
observed of 13% in median rostrum size (40 % in volume) across the Pliensbachian-
Toarcian Pli-Toa, before the Toarcian Oceanic Anoxic Event, and local belemnite extirpation.
The decreasing body-size pattern is mainly driven by a reduction in adult size of the two-
dominant species in the assemblages, *Pseudohastites longiformis* and *Passaloteuthis bisulcata*.
Belemnite size distribution is best correlated with fluctuations in a palaeotemperature proxy,
although however, potential effects of volcanism and carbon cycle perturbations and other
consequences of volcanic outgassing, may also play a role as well as sedimentary
architecture (fossil abundance and lithology), cannot be ignored. This highlights the complex
interplay between environmental stressors associated with these is hyperthermal events the

Commented [PR1]: Vanessa was suggesting to replace it by "may also play a role" because it is a more positive way but it changes the meaning right?

consequences of volcanism-related major environmental perturbations within the ocean-atmosphere system in driving belemnites body-size.

Keywords: belemnites, body-size, Pliensbachian–Toarcian boundary event, Peniche reference section, Lusitanian Basin

1 Introduction

Body-size is a key feature of any organism, reflecting its physiology, ecology and evolutionary history, across multiple scales of biological organisation (1). Body-size reduction has been considered the third universal response to global warming after changes in phenology and species distribution (2-4). ~~However, However, disentangling body-size response to warming might not be so straightforward due to individual responses might be taxon-specific (Ohlberger 2013) and synergies/interactions of the different abiotic factors during climate warming episodes influencing it (Calosi (Calosi, Putnam et al. 2018) et al., 2018) have to be considered (5, 6). Many factors during climate warming (increased temperature, decreased oxygenation, ocean acidification) are expected/likely to work in synergy in leading to reductions in size (6). However, increasing nutrient supply, for example, might have the opposite effect, rather than causing a body-size reduction, an increase might be observed in some regions or taxa (O’Gorman (4, 5) et al., 2017 ; Gardner and Bickford 2011).~~

Moreover, individual responses to warming might be ~~taxon-specific or even population specific . However, due to individual nutritional/environmental requirements (availability of nutrients and oxygen/reduced capacity to deal taxa might also require without food) and metabolic specifications , require more oxygen for faster metabolisms and have a reduced capacity to deal without food. (7). –~~ Furthermore, individual responses to warming might therefore be taxon-specific or even population-specific (Ohlberger 2013). Individual living cephalopod taxa, such as ~~Many squids, respond rapidly to warming events, and are expected to hatching out at smaller size, undergoing fast growth rates over shorter life-spans and and maturing younger and at smaller size during such events (8-10).~~ Furthermore, individual responses to warming might be taxon-specific (Ohlberger 2013). ~~can show rapid life history responses to warming events and associated environmental changes (Moreno, Azevedo et al. 2007, Hoving, Gilly et al. 2013), while other organisms are less affected or respond differently. Cephalopods are ectotherms because ectotherms constitute the vast majority of organism biomass and about 99% of all species worldwide is of particular importance to understand how they respond to warming.~~

~~More importantly, the effects of warming-related stresses visible in assemblages might depend~~
~~on the biological scale of organisation considered~~ (Daufresne, Lengfellner et al. 2009).
~~Individual living cephalopod taxa, for example, can show rapid life history responses to~~
~~warming events and associated environmental changes~~ (Moreno, Azevedo et al. 2007, Hoving,
Gilly et al. 2013).
Reductions in body ~~size~~ – ~~often dubbed~~ coined the Lilliput effect (11) – have been widely
reported from mass extinctions and evolutionary crises associated with environmental
perturbations (12, 13) ~~including the Early Toarcian crisis~~ (14). ~~However, their mechanisms~~
~~ecological and evolutionary significance and environmental drivers are still poorly~~
~~understood~~ (13), especially because ~~the effects of environmental perturbations on an~~
~~organism's body size might depend on the biological scale of organisation considered~~ (2). Over
longer evolutionary time ~~scales, i.e. considering fossil assemblages, several additional~~
~~mechanisms might contribute to the Lilliput effect~~ patterns observed between fossil
assemblages, including increased mortality of juveniles. Among fossil organisms, the lack of
knowledge is even larger. Different mechanisms have been proposed to explain the Lilliput
effect, such as extinction or temporary disappearance of large taxa, preferential survival or
origination of small ~~sized taxa~~ or the temporary reduction in adult body ~~size~~ of surviving
taxa (Twitchett (12, 13), 2007; Harries and Knorr, 2009). All of these mechanisms could explain
the Lilliput effect *sensu lato*. However, only the latter, i.e., a within ~~lineage body size~~
decrease, is considered to be the mechanism behind the Lilliput effect *sensu stricto*, which
depicts a reduction in body ~~sizes~~ within individual species through time and it corresponds to
the original term Urbanek (11) (1993). Interestingly, extinction during past events might also be
associated with preferential extinction of smaller rather than larger species (Payne (15) ~~et al.~~
2016).
~~including the Pli-Toa crisis~~ (Mortén and Twitchett 2009). ~~However, their ecological and~~
~~evolutionary significance is still poorly understood~~ (Harries and Knorr 2009).
The Early Toarcian Pli-Toa ~~crisis~~ is a multi ~~phased~~ event characterised by
palaeo ~~environmental and biodiversity~~ perturbations (16, 17). One of these pulses corresponds
to the Pli ~~Toa~~ (Pliensbachian ~~Toarcian~~) boundary event and the second pulse corresponds to
the Toarcian Oceanic Anoxic Event (T ~~OAE~~).
The Pli ~~Toa~~ boundary event (XXX1823.7 Ma, Ogg (18), 2004) follows the cooling and
regressive event of the Late Pliensbachian (Emaciatum = ~~Spinatum~~ Zone) in the northern
Tethys and Iberia (e.g. (19) Ruebsam et al., 2019). It corresponds to an ~~abrupt~~ increase of
seawater temperature, concomitant with the beginning of a marine transgression ((20-
24) Korte & Hesselbo, 2016, Rosales et al. 2004, 2018, Suan et al. 2008, 2010) as well as a small

negative carbon isotopic excursion (25, 26) Hesselbo et al., 2007; (Littler, Hesselbo et al.
 2010) Littler et al., 2010). The Pli–Toa boundary event records a crisis among planktonic ((27,
 28) Mattioli et al., 2008; Correia et al., 2017), benthic ((29) Aberhan & Fürsich, 1997) and
 nektonic organisms ((30) Dera et al., 2010) organisms, expressed by extinction and changes in
 abundance and diversity. Despite marked extinction in ammonites ((30) Dera et al., 2010 ~~XXX~~),
 the few data existing on Pliensbachian belemnite fauna do not allow to distinguish the Pli–Toa
 boundary event as an extinction horizon, at least in ~~northwestern~~north–western basins ((31–
 34) Little 1995, Caswell and Coe, 2009; Schlegelmilch, 1998; Weis et al., 2018). ~~belemnites only~~
 seem to show a minor taxonomic turnover (Caswell and Coe XXX) – although comparable little
 data is known from South European sections. **Namely in belemnites.**

 After the Pli–Toa boundary event, the Polymorphum Zone (= Tenuicostatum) – corresponds to
 a cooling phase (e.g. (19, 23, 35) Suan et al., 2008; Ruebsan et al., 2019), ~~although~~although
 comparatively warmer than the Late Pliensbachian. This cooling phase is followed by the T–
 OAE, starting at the base of the Levisoni Zone (= Serpentinum = Falciferum, 1822.78 Ma,
 (18) Dgg, 2004, GTS 2018) and characterized by a gradual marked gradual increase of the
 seawater temperature, widespread anoxia and as well as widespread deposition of organic–
 rich sediments – reflecting in a negative (e.g. Suan et al., 2008; , anoxia and the consequent
 deposition of organic-rich sediments (17, 23, 25, 36–38). These important changes in the
 ocean–atmosphere system had a major impact on marine biota, causing body–size changes
 and extinction of ammonite species ((14, 30, 39) Morten & Twitchett, 2009; Cecca and
 Macchioni, 2004, Dera et al., 2011) and extinction of benthic fauna (e.g. (16, 40–42) Little and
 Benton, 1995, Harries and Little, 1999, García Joral and Goy, 2000, Danise et al., 2013). The T–
 OAE also marks a major ~~another~~ shift/bottleneck in belemnite diversity and abundance (43)
 ((Caswell, Coe, and Cohen 2009) Caswell and Coe, 2009; Ullmann et al., 2014; Dzyuba et al.,
 2015), and triggered belemnite provincialism amongst Boreal–Arctic and Northwest European
 faunas ((44, 45) Dera et al., 2016, Doyle, 1987, 1994; Doyle et al., 1997; Dzyuba et al., 2015).
 The increased anoxia during the T–OAE has been ~~estimated~~interpreted to contribute to the
 demise of many belemnite taxa, while *Acrocoelites* might have survived and radiated in its
 immediate aftermath ((46) Ullmann et al., 2014). This event – e T–OAE coincides ~~During this~~
 ~~event~~with an abrupt negative carbon isotopic excursion (CIE) which disrupts an overarching
 positive excursion ((36, 47–49) Jenkyns and Clayton, 1986, 1997; Hesselbo et al., 2000, Jenkyns,
 2003, Hermoso et al., 2009a), reflecting enhanced burial of organic carbon and its preservation
 at the sea bottom. Although deoxygenation ~~has~~ve been interpreted to increase since the
 Pliensbachian–Toarcian boundary, the negative CIE and widespread black shale deposition are

~~still interpreted to reflect the nadir of anoxia during the T-OAE ((50)Them (Them, 2019 #101)~~
~~(Them, 2019 #101)et al., 2018).~~

~~Belemnites are interpreted to have had occupied different ecological niches in order to cope~~
~~with the environmental stress conditions caused by the T-OAE (diversification) and to have~~
~~suffered important morphological dynamics on rostra.~~

~~TheseBoth two warming events are thought to be a consequence of volcanic outgassing~~
~~(massive input of CO₂ in the atmosphere) related to the Karoo-Ferrar igneous Province (József~~
~~Pálffy, 2000), which is supported by the enhanced mercury composition of the sediments in~~
~~distal sections (Percival et al., 2015). Despite some controversy of the usage of mercury as a~~
~~proxy for volcanic outgassing (Them et al., 2019)(Them II et al., 2019), in Peniche the spikes on~~
~~mercury concentration are coincident with the abovementioned perturbations, both at the Pli-~~
~~Toa boundary event and during the onset of the T-OAE ((Percival et al., 2015)Percival et al.~~

~~2015). Both warming events and the associated carbon cycle perturbations are considered a~~
~~direct consequence of volcanic outgassing, either directly, by increase of pCO₂ in the seawater~~
~~(51) or indirectly (i.e. by triggering other sources of light carbon, (19)Ruebsam et al., 2019)~~
~~(Pálffy and Smith 2000). In distal sections, enhanced mercury deposition is interpreted to be a~~
~~Our best proxy for constraining the Karoo-Ferrar volcanism is enhanced mercury deposition~~
~~in distal sections (52). Irrespective of the controversy of the usage of mercury as a proxy for~~
~~volcanic outgassing (50), it is hitherto remains our the best available proxy to temporally~~
~~constrain the direct consequences of rapid rise of pCO₂ in the seawater, including warming,~~
~~deoxygenation and increasing seawater temperature, decrease dissolved oxygen and decrease~~
~~of pH (ocean acidification).~~

~~(Suan et al. 2010)~~

~~In Peniche, mercury anomalies coincide with extreme climatic changes (52)Percival et al.~~
~~2015(Suan et al. 2010), both expected to result in a body-size decrease (6, 7) (Calosi et al.~~
~~2019, Ohlberger et al.). On the contrary, extreme climatic changes can increase weathering and~~
~~precipitation (Calosi et al. 2019)et al. 2019), causing body-size increase instead, due to the~~
~~input of nutrients (4, 6). However, despite the evidence for increased weathering and influx of~~
~~nutrients in Peniche (Suan (24) et al. 2010), no indication of marked freshwater input or~~
~~productivity increase (27) have been observed and, therefore, a reduction in body-size is~~
~~expected. (Mattioli et al., 2009).~~

176
177
178
~~Belemnites are coleoids closely related to extant teuthids. They are very abundant in the fossil~~
~~record and played an important role are abundant fossils, important component ofin Jurassic~~
~~ecosystems. and closely relate with extant teuthids~~Most of the research on Lower Jurassic
~~belemnites was focused on the geochemistry of north-western European basins (46, 53-55),~~
~~temperate zones where the regional extent of anoxia wasis widespread (44, 56). The only~~
~~study on belemnite body-size was focused solely on two genera and found an ambiguous~~
~~response to crisis events (14). and focuses on areas where regional extent of anoxia is~~
~~widespread~~Therefore, a high-resolution analysis of belemnite body-size during past
~~episodes of environmental crisis will provide valuable insights on the cephalopods-response of~~
~~cephalopods to the global change observed in modern marine ecosystems.~~
189 ~~making them ideally suited to study their body size dynamics across these event on longer~~
~~time scales.~~ associated with rapid warming, anoxia and the consequent deposition of
~~organic-rich sediments (Jenkyns 1988, Hesselbo, Grocke et al. 2000, Wignall, Newton et al.~~
~~2005, Hesselbo, Jenkyns et al. 2007, Caruthers, Smith et al. 2013). Multiple extinction pulses~~
~~among nektonic and benthic organisms characterize this period (!!! INVALID CITATION !!! [18-~~
~~21]). This crisis is assumed to be a consequence of the massive input of CO2 in the atmosphere~~
~~by volcanic activity in the Karoo-Ferrar Igneous Province (József Pálfy 2000), which is~~
~~supported by the enhanced mercury composition of distal sections sediments (Percival, Witt et~~
~~al. 2015). The effects of volcanism have been observed across the Pli-Toa boundary and during~~
~~the onset of the Toarcian-Oceanic Anoxic Event (T-OAE, base of the Levisoni ammonite Zone),~~
~~the two main pulses of this crisis, associated with increasing seawater palaeotemperature~~
~~(Suan, Mattioli et al. 2008).~~
201 ~~Most of the research on Pli-Toa belemnites has focused on north-western European basins in~~
~~temperate latitudes (Doyle 1990, Dera, Toumoulin et al. 2016), often focusing on geochemistry~~
~~(van de Schootbrugge, McArthur et al. 2005, McArthur, Doyle et al. 2007, Harazim, Van De~~
~~Schootbrugge et al. 2013, Ullmann, Thibault et al. 2014). The only body-size study including~~
~~belemnite data focused on two genera and did not find an unambiguous response to crisis~~
~~events (Morten and Twitchett 2009), which might be potentially related with the selected~~
~~body-size proxy (Rita, De Baets et al. 2018).The Peniche reference section is well studied in~~
~~terms of geochemistry and stratigraphy (24, 25, 52, 57, 58) and yields a highly abundant~~

~~belemnite fauna, allowing for the first time the analysis of (i) the mechanisms behind~~
~~belemnite body-size fluctuations at different scales of organisation (individual ontogenetic,~~
~~species and assemblage) during the Pliensbachian–Toarcian interval in subtropical latitudes~~
~~and (ii) their potential environmental drivers.~~

~~We hypothesise, on one hand, a within-lineage reduction (Lilliput effect) in belemnite body-~~
~~size across the Pli–Toa boundary warming event and, on the other hand, a body-size increase~~
~~across the T–OAE driven by species turnover, disappearance both *Passaloteuthis* and~~
~~*Pseudohastites* (subfamily Belemnitinae) *elites* (subfamily *Megateuthididae*) in the Peniche~~
~~section while we predict rather an increase across TOAE driven by the extinction of~~
~~*Passaloteuthidae* and replacement with *Acrocoelites*. The highly abundant belemnite fauna of~~
~~the Peniche reference section is well studied in terms of geochemistry and stratigraphy~~
~~(Percival et al. 2015; Hesselbo et al. 2007; Suan et al. 2010; Duarte 2007; Rocha et al. 2016),~~
~~allowing for the first time the analysis of (i) the mechanisms behind belemnite size fluctuations~~
~~at different scales of organisation (individual, population and assemblage) during the~~
~~Pliensbachian–Toarcian interval at the Pli–Toa transition in subtropical latitudes and (ii) their~~
~~potential environmental drivers.~~

2 Material and methods

2.1. Taxonomy and ontogeny

~~Belemnite species identification was based on the analysis of traditional features, such as~~
~~shape (outline and profile) and the presence of grooves in the apical region (e.g. Schlegelmilch~~
~~(33), 1998). The transverse section, the depth of penetration of the alveolus and the apical line~~
~~were observed using the CT-scanning. All of these features were afterwards compared with~~
~~published descriptions and figures. This method also allowed us to recognise the features of~~
~~each ontogenetic stage with the acquired longitudinal sections, beingmaking it possible to~~
~~distinguish between adult (*Neanic–Ephebic–Gerontic sensu* (56) Doyle, 1990) and juvenile~~
~~(*Nepionic–Neanic sensu* (56) Doyle, 1990) specimens (Figs. S1 and S2). Across the studied~~
~~interval, belemnite assemblage was composed of seven taxa (Fig. 1) The taxonomic~~
~~composition of the belemnite assemblages across the studied interval is, being in conformity~~
~~with the taxa found in data from contemporaneous Tethyan sections ((33, 34, 43, 56, 59–~~
~~61) Caswell and Coe, 2009; Doyle, 1990, 1992; Schlegelmilch 1998; Pinard et al., 2014; Sanders~~
~~et al., 2015; Weis et al., 2018). Seven taxa compose the belemnite assemblages in the Peniche~~
~~section (Figs. 1, S1 and S2).~~

2.2 Sampling methods and belemnite body-size proxies

2.1 Mechanisms of body-size changes

The specimens ~~come~~ were collected from 13 beds sampled ~~from~~ of 45 m of Upper Pliensbachian – Lower Toarcian sediments, corresponding to ~~the Upper~~ Emaciatum – Upper Levisoni ammonite zones (Fig. 1, approximately ~~xx~~ 2.7 Ma, Ogg (18) ~~et al., 2016~~ Fig. 1). ~~As belemnites become rarer up section, the data derived from beds P9a, P9b and P9c was pooled to obtain a reasonable sample size for a quantitative analysis (Fig. 1). The same conservative approach was done for beds P3a and P3b.~~

We focused on quantitatively collecting from well-exposed bedding planes of limestones, ~~marly limestones~~ and marls. A total of 930 belemnites rostra were collected by (1) sampling ~~all~~ ~~all specimens within~~ ~~in~~ 1414 one-square-meter ~~1m²-~~ areas the specimens found within known areas (1 m²) and (2) by collecting at least 30 complete specimens in the remaining bed area, i.e., outside of the quadrants. This first sampling method ~~was~~ allowed the calculation of ~~belemnite~~ absolute abundance, taking into account both fragmented and complete specimens. The ~~body-size~~ analysis was based only on the complete specimens, regardless of the sampling method. ~~The more complete belemnites are suspected to be a less time-averaged sample (62) because a better preservation is usually an indicator~~ ~~synonym~~ of quick burial and little transport of the fossil. Conveniently, complete specimens are easier to assign taxonomically and ontogenetically (63). The fragmented specimens were only used to calculate the belemnite abundance per bed.

~~from both~~ Only the complete specimens were used for the body-size analysis. ~~As belemnites become rarer up section, the data derived from beds P9a, P9b and P9c was pooled to obtain a reasonable sample size for a quantitative analysis (Fig. 1). The same was done for beds P3a and P3b.~~ The rostrum represents a considerable part of the belemnite animal, and it is hypothesised that it acts as a counterbalance for the soft-parts and phragmocone (Monks (64) ~~et al., 1996~~), ~~despite~~ despite the increasing amount of studies suggesting an original porous rostrum structure (or at least less calcified) (Spaeth, 1975; Benito (65, 66) ~~et al., 2016~~; Hoffmann ~~et al., 2016~~). However, in all known belemnites with soft-parts preservation where it known, in most belemnites, the fins attach to the rostrum and the soft-parts ~~rear~~ soft-parts closely track the rostrum outline of their internal skeleton, as it is the case in their extant relatives (Reitner (67, 68) ~~and~~ ~~Ulrichs, 1983~~; Klug ~~et al., 2016~~). Plus, the rostrum is

276 ~~thought to be the attachment of the fins in belemnoids in general (Reitner and Urlichs, 1983;~~
277 ~~Klug et al., 2016).~~ For these reasons, the rostrum can be considered a reasonable proxy for
~~body size in the absence continuously coverage of preserved soft—parts (Rita (63) et al., 2018).~~
Previous research has demonstrated that the geometric mean (GM) of the apical height, width
and length is the most robust proxy ~~to compare body size between morphologically different~~
~~forms belonging to different species or ontogenetic stages within a species for belemnite~~
~~body-size (63). In addition, it is and~~ more comparable to proxies used in extant coleoids (69). ~~In~~
~~order to calculate the GM, For this purpose,~~ 277 of the ~~most~~ complete specimens were CT—
scanned to allow the calculation of this metric in a non—destructive way, ~~since the alveolus~~
~~was filed with sediment, precluding the measuring process.~~ Additionally, ~~in addition, in~~ 109
specimens ~~were measured with a calliper, since—since these metric could be measured—the~~
~~apical height, width and length was measured with a calliper the alveolus was empty.~~
~~The more complete belemnites are suspected to be a less time-averaged sample~~
(Tomašových, Schlögl et al. 2017) ~~—and little transport Conveniently and easier to assign~~
~~taxonomically and ontogenetically (Rita, De Baets et al. 2018).~~ The analyses were performed
~~at the bed scale.~~

Figure 1 – Variation of belemnite body-size (GM Geometric mean, assemblage scale), absolute abundance (no. of belemnites/m²) and species stratigraphic ranges of species through time, compared with the variation of the analysed geologic proxies of (mercury concentration, Hg/TOC (52), carbon and oxygen isotopes, $\delta^{13}\text{C}_{\text{org}}$ and $\delta^{18}\text{O}$ (23, 25)) and sequence stratigraphy data (2nd-order T-R cycles, (Rocha, 2016 #10) (Rocha, 2016 #10) (Rocha, 2016 #10)) – from the Upper Pliensbachian-Lower Toarcian of Peniche. The shaded area highlights the first pulse of the upper Pliensbachian-lower Toarcian boundary event-eris. The bed numbers in square brackets correspond to the beds that were merged due to sample size constraints, regarding the belemnite body-size analysis. See Fig. S2 for details on belemnite abundance. BG=belemnite gap; T=Transgressive; R=Regressive. Bed numbers, from P1 to P10, indicate the levels sampled with a black dot. The bed numbers in square brackets correspond to the beds that were merged due to sample size constraints. The interrupted lines connect beds where the species were missing in between

2.4.3 Mechanisms of body-size changes

~~In order to assess the~~ The mechanisms behind the belemnite body-size fluctuations ~~through~~
 ~~time,~~ from the assemblage to the individual scale of organisation ~~through time,~~ ~~were assessed~~
 ~~by testing~~ a set of hierarchical hypotheses (H), modified from (2) ~~,-was tested-~~(Fig. 2).

*Figure 2— Conceptual scheme depicting the tested hypotheses regarding the mechanisms behind belemnites body-*
 *size reduction, at different palaeobiological scales of organisation.*

The first hypothesis predicts a decrease in mean body-size at the assemblage scale, ~~whatever~~
 ~~regardless of~~ the underlying mechanisms (assemblage size shift hypothesis, H_1 , Fig. 2). If H_1 is
 validated, there are four subsequent hypotheses that could explain this decrease (2, 7). First,
 an increase in the proportion of small-sized species (population composition shift hypothesis,
 H_2). Second, a decrease in median body-size within specific taxa (within-taxa size shift
 hypothesis, H_3), which could, in turn, be related to a decrease in adult body-size (intraspecific
 size shift hypothesis, H_{4a}). The last hypothesis associates the decrease in mean body-size with
 an increase in the proportion of juveniles (ontogenetic stage structure shift hypothesis, H_{4b}).
 The whole set of hypotheses could explain the Lilliput effect *sensu lato*, although only H_{4b}
 relates to the Lilliput effect *sensu stricto*, as predicted by the temperature size rule in extant
 coleoids (70). ~~Note that, in comparison with recent data, The interpretation of body-size~~
 ~~changes in~~ based on fossil assemblages data is slightly more complex ~~as~~ ~~because fossil~~
 ~~assemblages reflect not only a past biotic community but also the mortality and preservation~~
 ~~of the organisms that were part of it. in living assemblages~~

~~as the composition of samples does not only depend on their relative abundance during life~~
 ~~but also relative mortality and preservation.~~

At the assemblage scale of organisation, H_1 was tested by assessing the size distribution of the
 whole assemblage through time. The proportional change in body-size ~~was calculated~~
 between beds in percentage of the log ~~ratio~~return (example: $\ln \frac{\text{median bed2}}{\text{median bed1}} * 100\%$). The non-
 parametric Mann-Whitney U and Kolmogorov-Smirnov (K-S) tests were used to assess the
 significance of the results (Table S2). The same was done at the ~~population-species~~ scale, for
 each taxon. Furthermore, because changes in taxonomic composition can influence the body-
 size patterns by the appearance and/or disappearance of taxa, we modified the within- and
 among-taxa approach by Rego et al. (71). This method corresponds to a pair-to-pair
 analysis (comparison of successive beds) which decomposes the assemblage size shift (Eq. 1)
 into three components: a disappearance of taxa effect (Eq. 2), a within-lineage effect (Eq. 3)
 and an appearance of new taxa effect (Eq. 4).

**(Eq. 1) Assemblage size shift** = assemblage median body-size_{bed2} - assemblage median body-
 size_{bed1}

**(Eq. 2) Disappearance of taxa effect** = pre-boundary crossers_{bed1} - all taxa_{bed1}

**(Eq. 3) Within-lineage effect** = post-boundary crossers_{bed2} - pre-boundary crossers_{bed1}

**(Eq. 4) Appearance of new taxa effect** = all taxa_{bed2} - post-boundary crossers_{bed2}

Finally, at the individual ontogenetic scale of organisation, the hypotheses were tested by
 ~~assessing body-size variation within comparing~~ different ontogenetic stages (adults and
 juveniles) ~~forwithin the particular taxa population by their variation in body-size~~. Variation
 partitioning allowed us to assess the proportion of body-size variation explained by ontogeny,
 taxonomic assignment and separation by beds, and by their joint effects. If the fraction of
 variation corresponding to beds separation is low, this suggests that mechanisms driving
 differences between samples are similar and not related to particularities of the sample (e.g.,
 lithology, age). Variation in the body-size data was partitioned using partial redundancy
 analysis as implemented in the *vegan* package (72). The significance of each fraction was
 assessed through ANOVA. The whole time-series was compared with the Pli-Toa boundary
 event.

2.42 Environmental drivers of body-size changes

In order to test our hypothesis of a relationship between belemnites body-size and
 environmental perturbations ~~and sedimentary properties~~, the distribution of the GM was

compared with several three main geologic proxies (Table 1). These are: Bbrachiopod stable
oxygen isotopes ($\delta^{18}\text{O}$) ($\delta^{18}\text{O}$: 23), bulk rock carbon isotopes ($\delta^{13}\text{C}_{\text{carb}}$) ($\delta^{13}\text{C}_{\text{carb}}$: 25) and mercury
concentration normalised by total organic carbon (Hg/TOC) (Hg/TOC: 52), ~~were~~ used as
proxies for seawater palaeotemperature, ~~perturbations negative excursions~~ in the carbon cycle
~~(often interpreted to reflect widespread anoxia or oxygen depletion??)~~ and volcanogenic
~~outgassing coinciding with effects of rapid changes in pCO₂ activity~~, respectively (Table 1). To
account for the impact of sedimentary
properties on body-size patterns, we correct~~ed~~g for the effect of lithology and belemnite
absolute abundance (mean abundance of belemnites per m², (Figs. 1 and S3) by residualizing
(e.g. Jakob (73) et al., 1996). A simple linear regression analysis is performed between the
body-size and the combined effect of abundance and lithology. The residuals from this
analysis are then used as outcome ~~in the and afterwards used in the~~ ultimate regression
analysis between body-size and abiotic variables. All variables of the model were continuous
with the exception of lithology. We assigned our samples a categorical variable for lithology
(marl to limestone) based on carbonate/clay content~~their grain size and calcium carbonate~~
~~content~~. Collinearity between explanatory variables was tested by using *cor* function in R (Fig.
S5). Sedimentary properties as well as various other parameters linked with climate warming
(such as strontium isotopes, total organic carbon and abundance of primary producers) could
not be directly included in the models due to their high collinearity with $\delta^{13}\text{C}_{\text{carb}}$ or the other
two used parameters (Table Figs. S3-S4S1), hence the choice of the abiotic parameters
indicated in Table 1. ~~The impact of C changes in sedimentary properties on body size patterns~~
~~were~~ assessed using lithology and mean abundance of belemnites per m² (Fig. 1). ~~We used~~
~~to different lithologies that we evaluated based on grain size and calcium carbonate content~~
(marl to limestone). ~~Due to high collinearity between the sedimentary properties and $\delta^{13}\text{C}_{\text{carb}}$~~
~~carbon cycles perturbations (Table S51), these two variables could not be included in the same~~
~~model. Therefore, body size was corrected beforehand for the effects of sedimentary~~
~~properties by residualizing (e.g. Jakob et al., 1996). A simple linear regression analysis is~~
~~performed between the body size and the combined effect of abundance and lithology. The~~
~~residuals from this analysis are then used as outcome and afterwards used in the ultimate~~
~~regression analysis between body size and abiotic variables. sedimentary properties. Despite~~
~~lithology, used as a categorical variable, the remaining variables of the model were continuous.~~
~~ed ultimate~~

~~We used a~~ multiple linear regression using generalised least squares (GLS) was performed
~~and the. The~~ models were fitted by maximising the log-likelihood (gls function in R). The
first-order autoregressive (AR(1)) model was used, which has the property of seeking
autocorrelation and of minimising the error term in a time-series (74). ~~Twenty-seven~~
models (see R script, supplementary material) were compared using Akaike's second order
corrected information criterion (AICc scores), which corrects for small sample size. The power
of the best model was assessed by means of an ANOVA test. The statistical significance of each
coefficient of the best model was assessed by calculating the p-value under a t
approximation. The significance level was $p < 0.05$ for all the analyses, unless if stated
otherwise. The analyses were performed in R (75), using the packages "nlme" (76) and "qpcR"
(77).

The regression analysis only included data from Emaciatum and Polymorphum zones (beds P1
to P9, Fig. 1), due to the lack of a representative sample size in Upper Levisoni Zone (bed P10,
Fig. 1). For this analysis, each belemnite was assigned the value of its respective beds
(abiotic factors, environmental proxies and sedimentary properties).

Table 1 – Environmental constraints (abiotic factors, parameters) based on their geological proxies and their
theoretical role in mediating body-size changes and their interpretation in the context of the ~~LB~~ Lusitanian Basin
(LB).

Geologic proxy (source) Environmental variable Basis Interpretation references Effect on body-size Theoretical control(s)

Geologic proxy [source]	Environmental variable	Basis	Interpretation in the context of the LB	Theoretical control(s) on body—size
$\delta^{18}\text{O}$ (Suan et al., 2008) Brachiopod stable oxygen isotope ratio ($\delta^{18}\text{O}$) [Suan (23) et al., 2008]	Dominantly temperature ((Cárdenas, 2010 #105)ref) Carbon cycle perturbations related with anoxia (Jenkyns, 2010)	Temperature—dependent isotopic fractionation between carbonate minerals and seawater (78)	High/low temperatures during negative/positive excursions	negative When the temperature are not extreme there is an increase in Increasing temperature through various mechanisms including, but not extreme temperatures, cause an increase of both growth and development rates are expected to lead to a smaller adult body—size (e.g. Ohlberger (7) et al., 2011)
$\delta^{13}\text{C}_{\text{carb}}$ (Hesselbo (25) et al., 2007) Hg/TOC ratio (Percival (52) et al., 2015)	Carbon cycle perturbations related with anoxia (Jenkyns, 2010) Volcanogenic outgassing (Them II, 2019)	Isotopic fractionation between organic matter and seawater (80) et al. remove light C^{12} from the seawater and there are influence on but are into the sea bottom cycle in the sediment after burial as organic carbon in the atmosphere (79). Changes in sea water temperature or oxygen availability or pCO_2 affect the isotopic signal. Volcanism represents a source of mercury to the atmosphere. Due to rain or runoff, mercury moves from the terrestrial realm/ atmosphere to the marine realm in mineral form or adsorbed in detrital organics. Hg burial is limited by low abundance and/or burial of organic matter or sulphids that would scavenge aqueous Hg (50, 52)	Negative excursions during the Pliensbachian-Torjón reflect enhanced burial of organic carbon during the Pliensbachian-Torjón. This is interpreted to have dropped water temperature during these intervals (Correa et al., 2008) wide spread oxygen depletion as primary productivity is (79). Changes in sea water temperature or oxygen availability or pCO_2 affect the isotopic signal. Hg anomalies (high Hg/TOC ratio) in the sediment are interpreted to represented markers of volcanism in distal sections. They have been interpreted to reflect volcanic outgassing, correspond to phases of volcanic outgassing, but enhanced input of terrestrially sourced materials and local redox variability also play a role (which can geologically be more or less instantaneous) their interpretation in some somedistal sections might be more complex Them I (50); 2019.	At the edge of an organisms' temperature range, growth is usually impaired by insufficient energy or oxygen supply, decreasing both growth rate and body size at any developmental stage, leading to a reverse temperature—size relationship (Ohlberger (7), 2011). Oxygen depletion of the sea bottom reflect major changes in the seawater column chemistry that may lead to unsuitable conditions for marine organisms. Furthermore, a drop in primary productivity reduces belemnite food resources availability and, therefore, their body size. Increasing pCO_2, combined with increasing temperature, causes in seawater a decrease in both dissolved oxygen and pH of the seawater. In association with other factors related to rapid warming, such as weathering and eutrophication, these factors are expected to lead to a body—size decrease due to their direct effects on the availability of food resources (Calosi (6) et al. 2018). Increased terrestrial input and nutrient influx could also lead to enhanced productivity and body size increase in some taxa (not yet near their tolerance limits, Sheridan & Bickford 2011). However, increased productivity is not observed in Peniche and effects of eutrophication and associated deoxygenation would rather lead to a body size decrease in belemnite cephalopods as they are adapted to normal marine conditions. Even if burial is further enhanced by decreasing oxygen as well increasing terrestrial input, these are also expected to decrease body size in most cephalopod taxa.
lithology abundance			1999 detrital organics. Hg burial is limited by low abundance and/or burial of OM or sulphids that would scavenge aqueous Hg.	

3 Results

3.1 Belemnite body-size fluctuations

Assemblage scale

Three episodes of median body-size decrease were recognised at the assemblage scale across the studied interval (Figs. 23B-4): P1-P2, P2-P3 and at the Pli-Toa boundary event (P4-P5), with the latter being the only decrease with statistical significance (Fig. 3B) and corresponding to a 13% body-size decrease (Fig. 34). This decrease in the GM corresponds to a 40% decrease of rostrum volume decrease (Table S2). The remaining pairs of beds correspond to an increasing median body-size at the assemblage scale of belemnite assemblages, with P8-P9 and P9-P10 being statistically significant (Fig. 3B).

Figure 3 – A: Venn diagram depicting the partition of belemnites body-size variation between taxonomic composition, bed separation and ontogeny. The values *between* in *brackets/parentheses* correspond to the whole time-series and the values *without parentheses/brackets* correspond to the Pli–Toa boundary. *The shaded area highlights the first pulse of the upper-Pliensbachian–lower-Toarcian crisis*. B: Effect on size change at the assemblage and *species/population* scales of organisation. See *Table S2 and T556* for details. For a correct interpretation of the right-side scale, the bottom of the arrowhead should be considered, rather than the tip of the arrow. Note that the x-axis depicts comparisons of consecutive pairs of beds (not to scale). BG = belemnite gap.

Population scale

A total of seven belemnite species were identified in Peniche (Fig. 1). Most of the taxa range
 from the uppermost Pliensbachian ~~until to~~ the Lower Toarcian (uppermost Polymorphum
 Zone). *Bairtowius* sp. A is exclusively represented in the uppermost Pliensbachian, being
 ~~stratigraphically~~ replaced by *Pseudohastites longiformis* in the assemblage. During the
 Toarcian, the ~~uppermost Polymorphum Zone – Levisoni Zone~~ interval is ~~barren devoid from of~~
 belemnites, with the exception ~~of of the the four Acrocoelites sp. specimens found in bed P10~~
 ~~(Upper Levisoni Zone), Fig. 1) which includes two Acrocoelites sp. specimens (Fig. 1) P133.~~

 *Figure 4 – Relative frequency of the different taxa comprising the Upper Pliensbachian-Lower Toarcian Peniche*
 *belemnite assemblage. Note that determinable incomplete specimens were also taken into account. The error bars*
 *correspond to the 95 % confidence interval of the relative frequency of the species *Pseudohastites longiformis* and*
 **Passaloteuthis bisulcata*.*

The assemblage body-size shift at the Pli-Toa boundary event is almost exclusively caused by
 a within-taxa effect (Fig. 3B) and therefore mainly driven by the boundary crossers (*Ps.*
 *longiformis*, *P. bisulcata* and *Hastitidae sp. indet.*). The taxonomic composition immediately
 before and after across the boundary does not change markedly (Fig. 1), with *Ps. longiformis*
 the most abundant taxon, comprising 61.9 % (P4) and 56.2 % (P5) of the assemblages (Figs. 1
 and S14). The disappearance of taxa effect (*Passaloteuthis milleri* and *Parapassaloteuthis sp.*)

~~from P4 to P5~~across the Pli–Toa boundary event, causes 4% of the decrease in the
assemblage median body-size (Fig. 3B2).

In contrast, the other two episodes of body-size reduction (P1–P2 and P2–P3) are
characterized by higher effects of ~~taxon~~-appearance (15.9% and 16.2%, respectively) and
disappearance of taxa (12.1% and 15.2%, respectively), with the within-taxa effect being
comparatively lower than at the boundary (Fig. 3B).

From P5 to P9, an increase in belemnite body-size at the assemblage scale is observed,
followed by a belemnite gap, from P9 to P10 (Figs. 3B4 and 45).

Figure 54 –Lithology, B belemnite body–size variation (GM), lithology and proportional body–size change and
 relative frequency of ontogenetic stages at the assemblage scale across the studied interval in Peniche at the
 assemblage and population/species scales (P. bisulcata and Ps. longiformis). Note that the stratigraphic log is not
 drawn to scale, for real thickness of beds, see Fig. 1. The error bars correspond to the 95 % confidence interval. For
 sample size, see Fig. S65 and Table S1. BG = belemnite gap.

Individual scale Species scale

Three taxa (*Ps. longiformis*, *P. bisulcata* and *Hastitidae* sp. indet.) are recorded immediately before and immediately after the Pli-Toa boundary event across the Pli-Toa boundary event (Fig. 1), in the sense that they are present right before and right after the Pli-Toa boundary event. The most significant and largest median size reduction (17% of decrease, Table S1) is observed in the most abundant taxon, *Ps. longiformis* (17% of decrease in GM and 51% in rostrum volume, Fig. 465). Despite the lack of statistical significance, *P. bisulcata*, the second most abundant taxon, *P. bisulcata*, also show a slight but insignificant decreases in size (7% in GM and 21% in volume, Fig. 465), across the Pli-Toa boundary event, while *Hastitidae* sp. indet., on the contrary, slightly increases in median-size (8%, Fig. S7S5). It is noteworthy that *P. bisulcata* body-size shows a marked median size markedly decreases in body-size (109%, Fig. 65) immediately after the Pli-Toa boundary event, from P5 to P6 (lowermost Polymorphum Zone, Fig. 3), which is coincident with the disappearance-extirpation of *Hastitidae* sp. indet. (Fig. 12), enhanced Hg/TOC and low $\delta^{18}\text{O}$ (Fig. 1).

*Figure 5 – Body-size variation of cross-borders (*P. bisulcata*, *Ps. longiformis* and *Hastitidae* sp. indet.). Although*
*Parapassaloteuthis* sp. A also crosses the boundary, we have no complete specimens across this interval and
*therefore, no accurate estimate of body size*

Individual ontogenetic scale

Considering the whole time-series, the RDA analysis results revealed that taxonomic
composition (47%) and ontogeny (17%) are responsible for most of the variation in
belemnites body-size variation (Fig. 3A). The separation by bed (fraction b) only explains only
2% of the body-size variation. (Fig. 3A).

At the Pli-Toa boundary event, the body-size variation partitioning into for separate
taxonomy and ontogeny is similar to what is observed for the whole time-series. However,
their joint effect of ontogeny (fraction c) and taxonomy (fraction f) explains 20% of the
variation, in contrast with the 6% observed for the whole time-series (Fig. 3A). This suggests
that across the Pli-Toa boundary event, ontogenetic stages behave more similarly among taxa
in explaining body-size variation.

Figure 6 – Belemnite body-size variation (GM), proportional body-size change, relative frequency of ontogenetic
 stages and sample size (n) across the Upper Pliensbachian-Lower Toarcian of Peniche at species scale (*P. bisulcata*
 and *Ps. longiformis*). *Passaloteuthis* genus was used to calculate the relative frequency of ontogenetic stages of *P.*
 *bisulcata* due to the difficulty of a species level classification of the juvenile specimens of *Passaloteuthis* genus. The
 error bars correspond to the 95 % confidence interval. Body-size variation of *P. bisulcata* and *Ps. longiformis* through
 the studied interval with emphasis on the ontogeny. The error bars correspond to the 95 % confidence interval. For
 sample size, see Fig. S6.

At the individual ontogenetic scale of organisation, focusing on *Ps. longiformis*, only adult
 specimens show a significant body-size decrease across the Pli-Toa boundary event
 (decrease of 21 %, Table S1 Fig. 6). The same is observed for *P. bisulcata*, which also decreases
 7% in size. In addition, immediately after the boundary, from P5 to P6, *P. bisulcata* markedly
 decreases in size (109%, Figs. 4 and S4). The ratio adult vs juveniles of *Ps. longiformis* is
 constant (does not change significantly) across the Pli-Toa boundary event (31.25 %, Fig. 6).

In what concerns *P. bisulcata*, a reduction of 7 % (although not significant) in adult body-size
 is observed across the Pli-Toa boundary event, together with a slight increased percentage of
 juveniles of the same species (Fig. 6). Immediately after the boundary, from P5 to P6, a marked
 reduction (109 %) in *P. bisulcata* body-size is recognized mainly among juveniles together
 with an increase in their proportion (from 75 % to 68.75 %, Fig. 6).

3.2 Relationship with environmental parameters

After correcting for the effects of lithological sedimentary properties (lithology and fossil
 abundance), the results of the GLS regression analysis revealed that the variation in
 belemnites body-size is best correlated with the variation in \$\delta^{18}\text{O}\$ (Table 12, model no. 235),
 at the assemblage scale. The overall model is significant (Table S3), suggesting a body-size
 reduction with decreasing \$\delta^{18}\text{O}\$ values/decreasing seawater temperature values (Fig. S2).
 However, the models combining \$\delta^{18}\text{O}\$ with \$\delta^{13}\text{C}_{\text{carb}}\$ (model no. 2, Table 2) and combining
 Hg/TOC with \$\delta^{13}\text{C}\$ \$\delta^{13}\text{C}_{\text{carb}}\$ (model no. 3, Table 2) could not not be ruled out be rejected (models
 no. 2, 3 and 5, Table 1), according to the AICc scores (\$\Delta < 2\$ ). If the data is not corrected for the
 effects of the sedimentary properties the same results are obtained, despite minor differences
 in the significance (Table S5).

After correcting for the effects of sedimentary lithology, when
 considering the most abundant taxon, *Ps. longiformis*, the best model (model no. 15) combines
 $\delta^{13}\text{C}$, $\delta^{18}\text{O}$, and lithology to explaining the body-size variation includes solely Hg/TOC (model
 no. 6, Table 2). However, the effects of $\delta^{18}\text{O}$, $\delta^{13}\text{C}$ and Hg/TOC cannot be discarded,
 since statistically the models n. 5, 2 and 4 cannot only be poorly be distinguished from model
 no. 6, according to the AICc scores. If the data is not corrected for the effects of lithology and
 abundance beforehand, the model no.4 (Hg/TOC + $\delta^{18}\text{O}$) best explains the best *Ps. longiformis*
 body-size variation, with highly significant being the results highly significant (Table S5). The
 role of Hg/TOC in the model is more significant than the role of $\delta^{18}\text{O}$ (Table S4b). The overall
 model is significant (Table 1). However, when considering the different coefficients, only the
 effect of the lithology and palaeotemperature are statistically significant (Table S3). The
 lithology of the studied samples does not change across the boundary (Fig. 3). The models
 combining $\delta^{13}\text{C}$, $\delta^{18}\text{O}$, lithology, Hg/TOC and abundance of specimens (models no. 4, 18 and
 20) cannot be ruled out, according to the AICc scores (Table 1). Nevertheless, the abundance
 of specimens does not change markedly across the boundary (Fig. 1).

 After correcting for the effects of lithology and fossil abundance, ical properties for *P.*
 *bisulcata*, body-size is best correlated with Hg/TOC as well $\delta^{18}\text{O}$ and abundance (model no.
 56, Table 2), as well, despite the lack of however it is not significance (Table 3), after
 correcting for the effects of lithology and fossil abundance. The overall model is significant
 (Table 1) but only the effect of abundance is significant (Table S3), being proportional to the
 body-size variation. However, the effects of $\delta^{18}\text{O}$, $\delta^{13}\text{C}$ and Hg/TOC cannot be
 disregarded, since statistically the models no. 5, 2 and 4 cannot be distinguished from
 model no. 6. If no correction is applied beforehand, the model no.4 (Hg/TOC + $\delta^{18}\text{O}$) best
 explains the best *P. bisulcata* body-size variation, with highly significant results being the
 results highly significant (Table S5).

 The models that combines Hg/TOC and abundance (models no. 8 and 27) could not be ruled
 out, according with the AICc scores ($\Delta < 2$).

 Table 2 – AICc ranking of models describing the effect of palaeotemperature ($\delta^{18}\text{O}$), carbon cycle perturbations
 ($\delta^{13}\text{C}_{\text{carb}}$) and volcanism (Hg/TOC) on belemnites body-size (GM) for the assemblage and species scale (*Ps.*
 *longiformis* and *P. bisulcata*), corrected for the effects of sedimentary properties (lithology and belemnite
 abundance). Only the models with $\Delta < 2$ are listed. See TS4 for the full list of GLS models.

Table 221 – AICc-ranking of models describing the effect of $\delta^{18}\text{O}$, $\delta^{13}\text{C}$ and Hg/TOC on belemnites body-size (GM) for the assemblage and population-species scale (*Ps. longiformis* and *P. bisulcata*), with the effect of lithology and abundance corrected. Only the models with $\Delta < 2$ are listed. See TS4 for the full list of GLS models. The models that best explain the belemnite body-size variation are in bold.

	Model no.	Δ	AIC-c scores	ANOVA p-value
Assemblage				
GM ~ 1	null	5.1	1925.5	-
GM ~ $\delta^{18}\text{O}$	23	0	1920.5	0.008
GM ~ $\delta^{18}\text{O}$ + Hg/TOC	2	1.1	1921.6	0.0172
GM ~ $\delta^{18}\text{O}$ + abundance	5	1.8	1922.3	0.0243
GM ~ $\delta^{18}\text{O}$ + $\delta^{13}\text{C}$	3	2	1922.5	0.0277
Ps. longiformis				
GM ~ 1	null	13	681.4	-
GM ~ $\delta^{13}\text{C}$ + $\delta^{18}\text{O}$ + lithology	15	0	684.9	0.0002
GM ~ $\delta^{18}\text{O}$ + lithology	4	0.9	685.9	0.0003
GM ~ Hg/TOC + $\delta^{13}\text{C}$ + lithology + abundance	20	1.5	686.5	0.0005
GM ~ $\delta^{13}\text{C}$ + $\delta^{18}\text{O}$ + lithology + abundance	18	1.6	686.6	0.0005
P. bisulcata				
GM ~ 1	null	4.0912	459.9	-
GM ~ $\delta^{18}\text{O}$ + abundance	5	0	455.8	0.0152
GM ~ Hg/TOC + abundance	8	1.1	456.9	0.0266
GM ~ abundance	27	1	456.9	0.0232

	Model no.	Δ	AIC c scores	ANOVA p-value
Assemblage				
GM ~ 1	null	5.090944	1921.871865	=
GM ~ $\delta^{18}\text{O}$	5	0	1920.9212	0.00849849
GM ~ $\delta^{18}\text{O}$ + $\delta^{13}\text{C}_{\text{carb}}$ GM ~ $\delta^{13}\text{C}$ + Hg/TOC	32	1.991915	1922.83691	-
GM ~ Hg/TOC + $\delta^{13}\text{C}_{\text{carb}}$ GM ~ $\delta^{18}\text{O}$ + $\delta^{13}\text{C}$	32	1.9891	1922.9184	-
Ps. longiformis				
GM ~ 1	null	2.5081	671.3736	=
GM ~ Hg//TOC	6	0	668.86567	0.0327
GM ~ $\delta^{18}\text{O}$ + $\delta^{13}\text{C}_{\text{carb}}$ GM ~ $\delta^{18}\text{O}$	25	0.8705047	669.746693703	-
GM ~ Hg/TOC + $\delta^{18}\text{O}$ GM ~ Hg/TOC + $\delta^{13}\text{C}$	24	1.3108699	670.176697355	-
GM ~ $\delta^{18}\text{O}$ GM ~ Hg/TOC + $\delta^{18}\text{O}$	45	0.5013052	669.376701708	-
P. bisulcata				
GM ~ 1	null	-1.4609	607.97	=

GM ~ Hg/TOC	6	0	609.4309	0.4306
GM ~ $\delta^{13}C_{carb} + Hg/TOC$	3	1.87	611.30	-
GM ~ $\delta^{18}O$	5	0.3537	609.7846	-
GM ~ $\delta^{13}C\delta^{13}C_{carb}$	7	0.58758	610.01967	-
GM ~ $\delta^{13}C + Hg/TOC$	3	1.8668	611.2977	-

Table 3 – Details of the selected GLS models comparing belemnite body–size (GM) with palaeotemperature ($\delta^{18}O$), carbon cycle perturbations ($\delta^{13}C_{carb}$) and volcanism (Hg/TOC), and the environmental parameters corrected for the effects of sedimentary properties (lithology and belemnite abundance).

Model	Coefficients	-	-	-	-	-	-	-
assemblage	model no.	coefficients	value	std. error	t value	p value	degrees of freedom	residual
Assemblage #5	5	Intercept	2.44	1.48	1.64	0.1019	340	338
		$\delta^{18}O$	2.04	1.19	1.72	0.061	=	-
Ps. longiformis #15	15	Intercept	0.31	0.23	1.32	0.1904	160	158
		Hg/TOC	-2	0.93	-2.14	0.0335	=	-
P. bisulcata #5	5	Intercept	0.43	0.94	0.46	0.6458	101	99
		Hg/TOC	-3.15	4.02	-0.78	0.4358	=	-

945GM ~ $\delta^{18}O + \delta^{13}C$	2	1.12	672.67
2972.93GM ~ $\delta^{18}O + Hg/TOC + 4031.98GM ~ \delta^{18}O + \delta^{13}C$	3	1.09	672.64
GM ~ Hg/TOC + $\delta^{13}C$			

GM ~ \$\delta^{18}O\$ 51.322.8760.161.7174 4 Discussion

**1. Scales and mechanisms of body–size variation**

**1.1. Pli–Toa boundary event**

Body–size fluctuations in fossil assemblages are common and can relate to a variety of
 mechanisms. The assemblage size shift hypothesis (H_1) was validated and despite the
 fluctuations observed, only the reduction observed at the Pli–Toa boundary event is
 significant. By decomposing the assemblage size shift we were able to disentangle the
 mechanisms of belemnite body–size change, which ~~revealed to be~~ was almost exclusively
 driven by a within–lineage effect.

Out of the boundary crossers, the most abundant taxon, *Ps. longiformis* is mainly driving the
observed body-size reduction, allowing us to validate the within-taxa size shift hypothesis
(H_3). No changes in the proportion of ontogenetic stages of *Ps. longiformis* are observed at the
Pli-Toa boundary event, meaning that the largest within-lineage body-size change is not
due to changes in the ontogenetic structure of the population but rather to an adult body-
size reduction (size-at-age reduction). In conclusion, at the individual ontogenetic scale, we
were able to validate the intraspecific size shift hypothesis (H_2), across the Pli-Toa boundary
event, by recognising finding a 51 % decrease in rostrum volume (21 % in median GM) of adult
specimens of *Ps. longiformis*. (Morten and Twitchett 2009) (Hoving et al. 2013) In contrast, *P.*
*bisulcata* body-size reduction is a combination of increased percentage of juveniles (increased
mortality of juveniles) and reduction of adult body-size.

Despite the fact that ontogeny was not taken into account, the Lilliput effect was already
identified within particular cephalopod species in the Cleveland Basin, interpreted as a
response to the deteriorating environmental conditions associated with the Early Toarcian
crisis (14). Similarly, among extant squid, maturation at small size and young age has been
interpreted as a life-history strategy have been recognized to cope with warming events (8).

The differential taxa response might indicate different environmental tolerances, related to
individual physiology or life history strategies (7). The only important morphological difference
between *Ps. longiformis* and the other taxa is the presence of an epirostrum, a calcified
structure which develops in late ontogenetic stages (81). The increased pCO_2 that
characterizes the environmental crisis of the Early Toarcian in Peniche (23) [24] could directly
(through calcification) or indirectly (through nutrient availability) have affected the
calcification potential of the epirostrum. However, further data needs to be assembled in
order assess the physiological and environmental constrains of the development of such a
structure.

Despite being one of the early pulses of the Early Toarcian crisis (44) and the observed body-
size reduction-observed, the Pli-Toa boundary event in Peniche does not record major
changes in the belemnite taxonomic composition (Fig. 4). This is consistent with data available
from other regions at higher latitudes (34, 43, 56, 60), which demonstrate that the T-OAE, in
contrast, corresponds to an important turnover in belemnite species (43). In fact, the T-OAE
has been shown to have had caused much more impact on marine biota than the Pli-Toa
boundary event (e.g. (30, 39, 40, 82) Cecca and Macchioni, 2004, Dera et al., 2011, Little and
Benton, 1995, Danise et al., 2013), namely in the Lusitanian Basin where planktonic and

benthic organisms were largely affected (e.g. Mattioli (27, 28, 83-85) ~~et al., 2008,~~(Correia,
Riding, Duarte, et al. 2017) Correia et al., 2017, Comas (Comas-Rengifo et al. 2015) and Rengifo
et al., 2015, Pinto et al., Rita et al., 2016). This highlights the deterioration of the
environmental conditions during the Early Toarcian, starting at the Pli-Toa boundary event
and culminating in the T-OAE, a barren interval of belemnites.

1.2. The aftermath of Pli-Toa boundary event

The marked body-size decrease of *P. bisulcata* (109 %) during the aftermath of the Pli-Toa
boundary event (from bed P5 to bed P6, Early Polymorphum Zone) ~~happens~~occurred in
tandem with a marked increase of the relative proportion of juveniles of that species (Fig. 6).
This might indicate the temporary emigration of adults and/or higher juvenile mortality, due to
unsuitability of the habitat. However, we also cannot rule out the existence of stunted adults
which do not develop typical adult features ~~(Hoving, 2013 #28) But this could can only be~~
disentangled with further studies, namely a sclerochronological analysis, ~~which would allow~~
the recognition of stunted adults.

The decrease in *P. bisulcata* body-size in the aftermath of the Pli-Toa boundary event
coincides with an increase in adult body-size of *Ps. longiformis*, and with the disappearance of
*Hastitidae* sp. indet. It is tempting to assume that the rapid recovery of *Ps. longiformis*
potentially triggered a change in the belemnite population dynamics, potentially due to
competition for food resources, causing the emigration of some species (*Hastitidae* sp. indet.)
and the increased mortality and stress in early growth of others (*P. bisulcata*). Furthermore,
the rest of the Polymorphum Zone (i.e. from P6 to P9) is characterized by an increase in *Ps.*
*longiformis* body-size (and also less relative mortality of juveniles), and also in its abundance,
relatively to *P. bisulcata*. The latter, which becomes less abundant and maintains its adult
body-size, emphasizing a potential competition between the *Ps. longiformis* and *P.*
*bisulcata* two species. However, the fact that ~~from P5 to P6~~ there is no increase in the
proportion of *Ps. longiformis* comparatively to the proportion of *P. bisulcata* ~~from P5 to P6~~
(Fig. 6) is not entirely compatible with this interpretation and highlights the constraints of
testing the effects of biotic parameters on ~~coinciding~~ body-size in fossil assemblages.

Notwithstanding the decrease of seawater temperature during the Early Polymorphum Zone,
this interval is characterized by palaeoenvironmental perturbations as evidenced by the
carbon isotopic record ((23, 25) Fig. 1). Moreover, seawater temperature is still higher than
during the Late Pliensbachian. The lack of mercury anomalies might indicate however, that

~~†~~The conditions are, however, not so severe as during the Pli–Toa boundary event or during
 the onset of the T–OAE, which is also supported by the response of other groups of marine
 biota in the Peniche section (27, 28, 83–85) (Mattioli, Correia, Rita).

*P. bisulcata* size dynamics in the aftermath of the Pli–Toa boundary event coincides with the
 recovery phase of *Ps. longiformis*, which records an increase in adult body size, and with the
 disappearance of *Hastitidae sp. indet.* It is tempting to ... The rapid recovery of *Ps. longiformis*
 potentially triggered a change in the belemnite population dynamics, potentially due to
 competition for food resources, causing the emigration of some species (*Hastitidae sp. indet.*)
 and the increased mortality and stress in early growth of others (*P. bisulcata*). The event (P5–
 P6) probably triggered the larger scale recovery of *P. longiformis*, which becomes larger and
 more abundant (and also less mortality of juveniles) during the rest of Polymorphum Zone, in
 contrast with *P. bisulcata* (adults remain the same size through time), which reduces its
 relative abundance, emphasizing a potential competition between the two species.

This is however hard to test and unlikely because *Ps. Longi* does not increase relative to
 *Passaloteutis bisulcata*.

The event (P5–P6) probably triggered the larger scale recovery of *P. longiformis*, which
 becomes larger and more abundant (and also less mortality of juveniles) during the rest of
 Polymorphum Zone, in contrast with *P. bisulcata* (adults remain the same size through time),
 which reduces its relative abundance, emphasizing a potential competition between the two
 species.

1.3. T–OAE

The onset of the T–OAE in Peniche corresponds to a belemnite gap, starting at ca. 30 cm
 below the Polymorphum–Levisoni zones boundary. This is coincident with the beginning of
 the second mercury anomaly and with warming, similarly to the lowermost Toarcian
 conditions, although associated with stronger carbon cycle perturbations (Fig. 1).

The belemnite gap could reflect inhospitable conditions in the Lusitanian Basin during the
 onset of the T–OAE. This is not the case, which did not happen during the Pli–Toa boundary
 event, which only caused changes in the dynamics of the population, namely in the
 ontogenetic structure due to temporary migration, and body size changes. This might suggest
 that coleoids – even those with flexible life history strategies, pre-adapted to such conditions,

Commented [PR2]: Check throughout

such as *Ps. longiformis* — could not cope with deteriorating conditions in subtropical
epicontinental European basins during the T—OAE.

In fact, the deterioration of water—column conditions must have been worse during the T—
OAE in comparison with the Pli—Toa boundary event, namely in terms of oxygenation of the
water column, as evidenced by geochemical data (25) (Hesselbo et al., 2007), and the
negative response of marine organisms in the Lusitanian Basin (e.g. Mattioli (27) et al., 2008,
Correia et al., 2017, Comas and Rengifo et al., 2015, Pinto et al., Rita et al., 2016). Moreover, a
similar belemnite response was identified in coeval north western European basins, as the
belemnite gap observed in Peniche overlaps with intervals of belemnite gap or belemnite
abundance decrease in the Cleveland and Cardigan Bay basins sediments ((43, 46, 86) Xu et al.,
2018; Caswell and Coe, Ullmann et al., 2014), interpreted as response to unfavourable anoxic—
euxinic water—column conditions.

Interestingly, the decrease in belemnite abundance, or disappearance, happens before the
most direct severe conditions of the TOAE are met in subtropical Peniche section and it is not
preceded by a body—size decrease. On the contrary, during the middle and upper
Polymorphum Zone (i.e., from P5 to P9) belemnite the body—size of both juvenile and adult
specimens increases during the middle and upper Polymorphum Zone (i.e., from P5 to P9), at
the assemblage scale. However, this can be related is probably related
to the fact that habitats are no longer suitable for juvenile specimens, and might indicate that
combination of warming and deoxygenation deteriorating environmental conditions even
before the onset of the T—OAE affected belemnite physiology, reproduction, as well as
competition for resources, even before the onset nadir of the T—OAE—. This highlights the
importance of climate warming in addition to regional anoxia—, which have been argued to be
a major constraint on belemnites abundance and distribution in the north western basins of
the Tethys ocean (43, 46, 86) which might have impacted their physiology, reproduction, as
well as competition for resources. Ref. (xu et al, Ullmann)

tracefossil Luis rabacal, they disappear before the onset of the toae and before the highest
values of TOC

Belemnites temporarily re—appear in bed P10 (Upper Levisoni Zone), after the onset of the T—
—OAE, coinciding approximately with the end of the T—OAE negative CIE (Fig. 1), with a poor
record of four specimens of *Acrocoelites* genus. This genus is known to appear and radiate in
the north—eastern basins of the NW Tethys ((34, 43, 44, 56, 59) Doyle, 1990, 1992, Caswell and
Coe, 2014, Dera et al., 2016, Weis et al., 2018) after the T—OAE, replacing the genus

Commented [PR3]: I think it is more true for *Ps. Longiformis* actually

Commented [KD4R4]: It seem to be the case for both in the graphs.

Commented [PR5R4]:

Commented [PR6R4]:

*Passaloteuthis*, which has been interpreted as an evolutionary adaptation in terms of
ecological preferences related to the stressful conditions of the T-OAE (46). Moreover, the
change from the Late Pliensbachian/Early Toarcian Belemnitinae-dominated assemblage to
the Middle Toarcian Megateuthidinae-dominated assemblage (together with the
disappearance of smaller sized species such as *Hastitidae* sp. indet., *Parapassaloteuthis* sp. A,
*P. milleri*, *Ps. longiformis*), might not only be related to with ecological preferences but also
with selective extinction, as pointed out by (15) Payne et al., 2016, who states that past
extinction events were either nonselective or preferentially removed smaller-bodied taxa.

However, the lack of a continuous belemnite record during the Levisoni Zone in Peniche
hampers a more detailed n-appropriate assessment. Further research in other basins is
necessary to disentangle the extent of this phenomenon.

4.2. The effect of sedimentary facies on belemnite body-size

Despite the fact that both of the studied events are considered to be geologically rapid (120–
620 kyr, (87{Boullila, 2019 #133, 88})), their representation in the sedimentary record is
dependent on facies evolution dependent of several factors such as regional tectonics, climate
and sea level changes the role of these parameters control the sedimentation rate, which is
not constant across the studied interval. We use lithology to account for changes in the facies
(related with relative sea-level changes) and belemnite abundance to account for changes in
the sedimentation rate, since both are thought to have an effect on morphological patterns
(89).

Accumulations with abundant belemnites, particularly observed in beds P4 and P5, are
known could be to be related to changes in sedimentation rate relative to the accumulation of
belemnites (90). The lowermost Polymorphum Zone in Peniche is thought to correspond to a
condensed interval (58, 84, 91), but is better represented as many other sections (Morard et
al., 2003). Our data reveals that belemnite body-size is larger in beds with higher belemnite
abundance (Fig. S10). Abundance, however, does not markedly change across the boundary,
where the most significant size change is observed. Regarding lithology, belemnite body-size is
larger in marls in comparison with limestones but this is not significant but these effects are not
strong (Fig. S9).

When studying the effect of environmental perturbations on belemnite body-size, at the scale
 of the assemblage, lithology and abundance do not affect the pattern observed. Before and
 after correction, temperature is always the best parameter in explaining the body-size
 variation. However, the significance is higher without correction which is not surprising
 considering that residualising might also filter out potential ecological signals contained within
 the sedimentary properties in addition to the effect of preservation and collection biases. Even
 with the correction for the effects of sedimentary properties, and despite the existence of a
 significant model, the analysis failed in revealing a single model to explain belemnite body-size
 variation. Instead, several models are very similar to the model with the best AICc score. This
 highlights the strong interaction between abiotic parameters in driving body-size patterns
 during episodes of climate warming.

However, when analysing the data at species level (*P. bisulcata* and *Ps. longiformis*), the best
 model explaining the body-size variation after correction is Hg/TOC as opposed to
 temperature without correction in both species. This might be related to the smaller -datasets,
 especially in the case of *P. bisulcata*, which does not allow to distinguish between individual
 models. Either way, the increased warming, at the assemblage scale, and Hg/TOC anomalies,
 as drivers of individual species body-size variation, are not incompatible – it just shows that
 changes in warming generally correlate with decreasing sizes of belemnite assemblages across
 the Pli-Toa boundary and that the impact of “catastrophic” environmental perturbations
 triggered by volcanism are the most severe and probably work in concert in their impact on
 body-size decrease in individual species.

The Pli-Toa boundary in Peniche also coincides with a marine transgression (58) which might
 have impacted on the size distribution-data, but differences in lithology and sedimentation
 rates could not clearly explain differences in our dataset. Furthermore, deepening
 shouldwould rather results in finding larger specimenspecies (69), which would be consistent
 with our observations of finding larger specimens in marls. But{Shi, 2016-#132}a body size
 decrease is seen ,but the opposite is seen across the boundary. Further analyses across
 different parts of the basin would be necessary to disentangle its potentially moreless subtle
 effects in shallower sections.

 Increased warming as well as sea-level changes during the upper Pliensbachian-Lower Toarcian
 are also related with marked changes in facies (e.g., lithology) and sedimentation rates.
 Increased warming as well as sea-level changes during the upper Pliensbachian-Lower Toarcian
 are also related with marked changes in facies (e.g., lithology) and sedimentation rates. This
 can affect fossil abundance as well as size and other morphological patterns (Holland 2000).

Commented [KD7]: Suan et al. 2010 and/or mattioli

Luis?

Commented [PR8R8]: I think pittet et al., 2014 is probably better

The lowermost Polymorphum Zone is thought to correspond to a condensed interval based on
 the accumulation of micro and macrofossils (Rita, 2016 #8218; Rocha, 2016 #10; Pittet, 2014
 #122). Pittet et al., 2014. Accumulations with abundant belemnites, particularly beds P4 and
 P5, are known to be related to changes in sedimentation rate relative to the accumulation of
 fossils (Doyle and Macdonald 1993).
 In our study, however, at the scale of the assemblage, lithology and abundance do not affect
 the variation of body size explained by the environmental variables. Before and after
 correction, temperature is always the best parameter in explaining the body size variation.
 However, the significance is higher when we don't correct which is not surprising considering
 that residualizing might also filter out potential ecological signals contained within the
 sedimentary properties in addition to the effect of preservation and collection biases. Even
 with the correction for the effects of sedimentary properties, and despite the existence of a
 significant model, the analysis failed in revealing a single model to explain belemnite body size
 variation. Instead, several models are equivalent and couldn't be distinguished from the model
 with the best AICc score. This highlights the strong interaction between abiotic parameters in
 driving body size patterns during episodes of climate warming.
 However, when analysing the data at species level (*P. bisulcata* and *Ps. longiformis*), the best
 model explaining the body size variation after correction is Hg/TOC as opposed to temperature
 without correction in both species. This might be related to the small dataset, especially in the
 case of *Passaloteuthis bisulcata*, that makes it difficult to distinguish between individual
 models. Either way, the increased warming at the assemblage and Hg/TOC as drivers of
 individual species body size variation are not incompatible – it just shows that changes in
 warming generally correlate with decreasing sizes of belemnite assemblages across the
 Pliensbachian-Toarcian boundary and that the impact of “catastrophic” environmental
 perturbations triggered by volcanism are the most severe and probably work in synergy in
 their impact on body size decrease in individual species.

Commented [PR9]: And also the presence of unconformities Pittet et al 2014

Commented [KD10]: Yes, but I would mention that changes in lithology and abundance poorly explain body size metrics.

**4.3.**

**Environmental drivers of body size fluctuations**

The Pli–Toa boundary event corresponds to the first episode of stress in the Early Toarcian
 palaeoenvironmental crisis (23) and it has been associated with the a first pulse of Karroo–
 Ferrar volcanism (52). This had a major effect on climate, disturbing the whole entire ocean–
 atmosphere system, by causing carbon cycle perturbations and fluctuations in the seawater
 temperature. Karroo–Ferrar volcanism through direct outgassing and potential triggering other
 sources of greenhouse gases This resulted in global climate warming and apart from warmer

seawater temperature coincided with decreasing dissolved oxygen, increase weathering and
increasing $p\text{CO}_2$ (potentially ocean acidification) and locally decreased productivity in Peniche
(Suan, 2008 #11) Suan et al. 2008) which all might have affected body size in synergy (Calosi et
al., 2019). Our results are consistent with the hypothesis that climate seawater
palaeotemperature ($\delta^{18}\text{O}$) is an important driver of the belemnite body size variation at the
assemblage level scale, with a decreasing body size associated with increasing seawater
temperature at the assemblage level (Fig. S8). Salinity could however partially effect affect the
interpretation of the oxygen isotopic data, particularly in north-western European basins,
where particularly higher amplitude of changes in northwestern north-western basins have
been attributed to freshwater the input of brackish and nutrient-rich Arctic surface waters
seems to affect the signal (92, 93). However, decreased salinity has been deemed less
significant in Peniche than warming and carbon cycle perturbations based Salinity could also
affect the interpretation of the oxygen isotope data, although decreased salinity has been
argued to be less significant in Peniche than warming and carbon cycle perturbations, based on
phytoplankton communities (94). Furthermore, modelling suggest that, at this
palaeocoordinates, salinity and modelling changes are close to zero (92, 93) and moreover the
effects on salinity. Other effects might have influenced the signal, but brachiopods analysed
by data from Suan et al. 2008xx have been screened for diagenetic overprint and climate
warming is consistent with independent observations such as northward migration of
ammonoids.

such as seasonality and diagenesis can affect the signal but....

This reduction in adult size of *Ps. longiformis* and *P. bisulcata* — the Lilliput effect — can
potentially be compared with the decrease in adult mantle length of extant of extant squid
during episodes of warming rapid warming events (8, 10). The large decrease observed
during the Pliensbachian-Toarcian warming event as well correlation with of with events
interpreted to reflect rapid warming.

To what degree the decrease in adult size relates with direct effects of warming on physiology
(development rate, metabolism, respiration) or rather indirect effects of environmental
changes Warming causes body size reduction because directly affects biochemical reactions
important for metabolism and on resource acquisitions affecting therefore the increase growth
and early development and development rates is still under debated, leading to a smaller adult
size (e.g. Ohlberger (Ohlberger, 2013 #29; Audzijonyte, 2019 #135) et al., 2011).

Apart from increasing seawater temperature, the mercury anomalies and carbon cycle
 perturbations at the Pli–Toa boundary event in Peniche, also reflect coincide with increase
 weathering ((25)Hesselbo et al., 2007), and local decreased productivity (27, 28)(Mattioli et al.
 2009; Correia et al., 2017), and increasing $p\text{CO}_2$ (and potentially ocean acidification) and
 decreasing dissolved oxygen (23)(Suan, 2008 #11). They are all reflected in the. All of these
 factors are expected to work in concert in driving marine organisms body size (6) and our
 analysis of belemnite body size is consistent with that. This is also reflected in the strong
 collinearity of environmental and biotic proxies related to these effects they are all reflected in
 the.

 The correlation of (Correia, 2017 #93) with Sr/Strontium isotopes from Peniche support the idea
 of increased weathering during the Pli–Toa boundary warming event and potentially nutrient
 influx (25), which is also consistent with the interpretation of enhanced mercury concentration
 in the sediments. support the idea of increased weathering during these warming events and
 potentially nutrient influx. However, contra-intuitively these warming events are is
 interpreted to lead to a decrease in primary productivity, with a decrease in both
 phytoplankton abundance (Correia, 2017 #93; Mattioli, 2009 #92). which might also have been
 related with direct effects of $p\text{CO}_2$ increase. This is also reflecting by the good negative
 correlation of temperature with phytoplankton abundance (Correia, 2017 #93) and, diversity
 and size (Mattioli, 2009 #92). This emphasizes how warming and related stressors as well
 reduced after might in turn also effect the size of primary consumers and potentially
 other levels of the food chain, such as predators like as well their predators — cephalopods.
 Dwarfing has been observed in response to lack of food in the laboratory, but so far not in the
 wild cephalopods (Boletzky 1974 (95)). In addition, macroecological studies show that body—
 size distribution in extant cephalopods is better explained by seawater temperature than
 productivity ((69)Rosa et al. 2012).

 The negative carbon cycle isotopic excursions CIE observed during the Early Toarcian are
 interpreted to be related to volcanic outgassing either directly by now thought the to relate to
 rapid input of isotopically light carbon into the atmosphere–hydrosphere system (51), or
 indirectly related to direct volcanic outgassing or indirect by triggering of various sources of
 isotopically light carbon ((19)ref). This would not only have caused rapid climate warming, but
 also increased $p\text{CO}_2$ and decreased pH of the seawater. In the Lusitanian Basin a size reduction

~~has been observed among phytoplankton in relation with the direct effects of $p\text{CO}_2$ increase~~
~~(Mattioli, 2009 #92). However, cephalopods are usually interpreted to be quite~~
~~robust/resistant to ocean acidification than other marine/in comparison with other marine~~
~~organisms – cuttlefish even increase calcification (Dorey, 2013 #68)(96, 97)(Gutowska, 2010~~
~~#125; Dorey, 2013 #68) (Dorey et al., 2013). However, so far only the limited data is available~~
~~hitherto and do seem to indicate that the effects of acidification might be indirect (Guerra et~~
~~al., 2011) or direct but more severe in early ontogeny, resulting in pathologies (97, 98)(Dorey~~
~~et al., 2013; Lacoue-Labarthe et al., 2011). Nonetheless, no clear signs of aberrant/berrant~~
~~development were observed in the studied specimens/rostra. More importantly, there are no~~
~~direct proxies for marked ocean acidification during the Pliensbachian-Toarcian boundary~~
~~event-Early Toarcian in the Lusitanian Basin and only indirect evidences, for it during the TOAE~~
~~such as calcareous nannofossils (*Schizosphaerella*) –size reduction. However, this has been~~
~~interpreted as an indirect consequence of $p\text{CO}_2$ increase, due to changes in the climate and~~
~~sea-level (27), rather than a direct cause of high $p\text{CO}_2$. (Mattioli, 2009 #92;ref).~~

~~Together with the deposition of organic black shales, However, our results indicate that other~~
~~factors implicated in rapid warming such volcanic outgassing or release from other such~~
~~sources of light carbon reflected in negative carbon cycle perturbations and decreased~~
~~oxygenation exert an effect on belemnites body size, emphasizing the potential synergy of the~~
~~different factors affecting the ocean-atmosphere system.~~

~~† the Early Toarcian negative carbon cycle perturbations/isotopic excursions/CIE and associated~~
~~black shales have been traditionally interpreted to reflect widespread anoxia and roughly~~
~~correlated with phases of increased anoxia in the ocean (e.g. (79, 99) Ikeda et al., 2018).~~
~~Irrespective of the potential local or regional overprint of increasing stratification due to basin~~
~~restriction or freshwater/artic water input, the first perturbations at the Pli-Toa boundary~~
~~event in Peniche iares thought to reflect the start of decreasing oxygenation of seawater while~~
~~the second – the T-OAE perturbations during the T-OAE areis thought to represent the global~~
~~nadir of anoxia ((80) Them et al., 2017). Nonetheless, in Peniche, there is however no evidence~~
~~for bottom-water anoxia in the Lusitanian Basin (25, 100), in contrast as opposed to basins in~~
~~with the northwestern/north-western European basins (101). Despite that fact, less severe~~
~~de/oxygenation might have played a role in belemnite body-size and distribution, since~~
~~increased seawater temperature results in reduced oxygen availability relative to demand~~
~~(102). either way as finer proxies reflecting oxygen decrease globally which effect might be~~
~~more exegeratted as oxygen demand is considered higher than supply in warmer waters.~~

~~The negative carbon cycle perturbations are now thought to relate to rapid input of light~~
 ~~carbon related to direct volcanic outgassing or indirect triggering of various sources of light~~
 ~~carbon. This would not only have caused rapid climate warming, but also increased $p\text{CO}_2$ and~~
 ~~decreased pH of the seawater. However, cephalopods are usually interpreted to be quite~~
 ~~robust to ocean acidification than other mollusksmolluscs – some even increase calcification.~~
 ~~However, so far only limited data is available and do indicate effects to be more severe in early~~
 ~~ontogeny resulting in pathologies – no clear signs could be seen in our specimens. More~~
 ~~importantly, there is so far no clear evidence for marked ocean acidificationacidification during~~
 ~~the Pliensbachian Toarcian event and only indirect evidence for it during the TOAE.~~

~~Sea-level change associated with warming events might impacted on the weathering and~~
 ~~habitat distribution and more indirectly resulting in less limestone as well as lower~~
 ~~sedimentation rate which impact the accumulation and sampling opportunity of belemnites.~~
 ~~The Pliensbachian-Toarcian also reflects marked changes from more limestones to black shales~~
 ~~which are closely linked with changes in CaCO_3 related to $p\text{CO}_2$ (Mattioli et al., Suan et al., 2008).~~
 ~~However, when correcting for the effects of sedimentary properties changes, as mentioned~~
 ~~above, it, the implication of climate warming on driving belemnite body size changes remained~~
 ~~obvious. (see paragraph) The Pliensbachian-Toarcian also reflects marked changes from more~~
 ~~limestones to black shales which are closely linked with changes in CaCO_3 related to $p\text{CO}_2$.~~

~~There is a clear~~The strong collinearity between biotic and abiotic factors available in the
 literature for the Peniche section various factors implicated in size for which we have proxies
 in Peniche (see supplementary materialTableFig. S4x) hampers an analysis of their individual
 effect on belemnite body size), which makes it hard to separate relative effect of factors
 which more or less changed in concert. Nonetheless, this study allows to distinguish direct and
 indirect effects of climate warming on belemnite body size reduction, despite theThis reveals
 complex interaction between biotic and abiotic factorsthe complexity of body size patterns,
 highlighting the importance of taking into account different scales of organization. and their
 biotic and abiotic drivers and how do they work in concert/complex interaction by affecting
 not only directly the water column conditions but also the biotic communities that live there
 and also by indirectly affecting them due to the effects of volcanism and ultimately climate
 change in the atmosphere. With our analyses, we can however implicate direct or indirect
 effects of climate warming in the reduction of size within belemnite and we highlight the
 importance of analysis at different scales of organization. Furthermore,

Commented [PR11]: I don't think this paragraph is that important... I have the feeling it doesn't add much...we already talk about lithology in the above section, where we mention the effect of sedimentary properties on body size

Commented [PR12R12]: Put in the facies part

Commented [KD13]: Ok, I mentioned the collinearity before here.

~~The disruption of the ocean-atmosphere system which characterizes the Early Toarcian is not~~
~~only known by increasing seawater temperature but also by perturbations of the carbon cycle,~~
~~potentially related with volcanic outgassing (the input of light isotopically carbon in the~~
~~seawater). These perturbations are intrinsically connected with availability of nutrients~~
~~(primary productivity) and oxygen. The Early Toarcian in Peniche corresponds to a decrease in~~
~~the primary production and an increasing $p\text{CO}_2$. A drop in primary productivity reduces food~~
~~resources availability and, therefore, is interpreted to reduce organisms body size. A decrease~~
~~in oxygen availability is also thought to have a negative on body size by increasing organisms~~
~~metabolic rate (Sheridan & Bickford 2011, Baudron et al, 2014). All these factors as well as~~
~~other factors associated with warming like increased weathering which in turn can lead to~~
~~increased eutrophication are expected to work in synergy to lead to a decrease in body size~~
~~(Calosi et al. 2018). At the edge of organisms' temperature range, growth is usually impaired~~
~~by insufficient energy or oxygen supply, decreasing both growth rate and body size at any~~
~~developmental stage, leading to a reverse temperature-size relationship (Ohlberger, 2011).~~

~~The complex interconnection between all these factors during the interval studied cause a~~
~~collinearity to happen between them in terms of statistics (see supplementary material).~~
~~Therefore, it is not possible with the current approach assess the effects of those variables on~~
~~belemnite body size. Other factors such pH could have an influence but no good proxy is~~
~~available at the moment. the rapid warming event related with volcanic release or triggering of~~
~~light carbon had clear effects on adult size in species. The size of nannofossils could be used~~
~~but it doesn't cover the whole time series we analysed, specially the Pliensbachian. Moreover,~~
~~the lack of data for the some of the upper Pliensbachian beds would also hamper a proper~~
~~analysis.~~

~~Sr isotopes: nutrient input from weathering of continental rocks. High trophic resources would~~
~~enhance the success of belemnite population. Furthermore, they may increase the abundance~~
~~of primary producers, and subsequently the food web complexity~~

~~Sea level: increasing in marine niche differentiation by highly developed stratification in the~~
~~water column and/or latitudinal marine isolating barriers. Furthermore, lowstands increase in~~
~~the area available for erosion and nutrient input.~~

Ph ($p\text{CO}_2$) belemnite are suppose to be able to increase the ability to calcify their rostra by
comparison with living analogous such as Sepia.

primary productivity : dinoflageltes are photosyntetic marine plankton. as primary producers
(plankton, photosintetic) organisms, a higher % of dinoflagelates in the water column indicate
that more nutrients are available for belemnites prey, such as fish, crustaceous and other
organisms feeding on the primary producers. If belemnite food resources availability increase,
belemnite body size at maturity increase because their growth rate will also be higher.

Oxygenation of the water column

**Scales and mechanisms of body-size variation**

Body-size fluctuations in assemblages are not uncommon and can relate with a variety of
mechanisms. The assemblage size shift hypothesis (H_4) was validated and despite the
fluctuations observed, only the reduction observed at the Pli-Toa boundary is significant. By
decomposing the assemblage size shift we were able to disentangle the mechanisms of body-
size change, which revealed to be almost exclusively driven by a within-lineage effect.

Out of the boundary crossers, the most abundant taxon, *Ps. longiformis* is mainly driving the
observed body-size reduction, allowing us to validate the within-taxa size shift hypothesis.

Finally, at scale of individuals, we were able to validate the intraspecific size shift hypothesis
(H_5), by recognising a size decrease of 21% among adult specimens of *Ps. longiformis* across
the Pli-Toa boundary. This Lilliput effect is similar to significant changes in adult mantle length
observed in extant jumbo squid during episodes of warming (Hoving, Gilly et al. 2013). and/
which is also supported by data from other groups rapid recovery and stress in early growth of
others. Interestingly, the decrease in belemnite abundance is not preceded by a body size
decrease. On the contrary, the body size of juvenile and adult specimens increases during the
final part of which could relate to the fact that habitats are no longer suitable for juvenile
specimens.

Belemnites do however dissappear and decrease in diversity and ontogenetic stages
represented before the most direst conditions of the TOAE are met in subtropical Peniche the
highlighting also the importance of climate warming in opposition to regional anoxia — which
might have impacted their physiology, reproduction, as well as competition for resources.

~~Irrespective, if this effect is exacerbated by the fact we only found larger (?adult) specimens, similar turnover patterns are expected in other European basins., Weis et al., 2018 Belemnitinae Megateuthidinae et Hastitidae sp. indet., Parapassaloteuthis sp. A, P. milleri, Ps. longiformis selective extinction~~

~~The effect of sedimentary facies on belemnites body size~~

~~Changes in facies (e.g. lithology) and sedimentation rate can affect fossil abundance and morphological patterns (Holland 2000). The lowermost Polymorphum Zone is thought to correspond to a condensed interval based on the accumulation of micro and macrofossils (Rita, Reolid et al. 2016, Rocha, Mattioli et al. 2016). Accumulations with abundant belemnites (Doyle and Macdonald 1993), particularly beds P4 and P5, are known to be related to changes in sedimentation rate relative to the production of fossils.~~

~~The model combining palaeotemperature (Suan, Mattioli et al. 2008) with belemnite abundance is worse than the model which only considers temperature when explaining belemnite size.~~

~~However, no significant changes in abundance are recorded across the Pli-Toa boundary, making it possible to conclude that, at least the most significant change on body size is not caused by sedimentary condensation. On the other hand, the effects of the lithology are might, for example, be more evident from beds P1 to P2 (uppermost Pliensbachian), for the whole assemblage. This non-significant size reduction might be related to the fact that these two beds correspond to more marly intervals (transgressive cycles of 4th order) and they possibly allow a better sampling process (Duarte 2007).~~

~~In what concerns *Ps. longiformis*, the best model explaining body size variation includes lithology and the effect of belemnites abundance could not be entirely ruled out. However, at the Pli-Toa boundary, these factors cannot explain the size reduction observed, since no major changes in lithology nor belemnites abundance are observed. It is, therefore, safe to say that the effect of lithology and abundance is less significant than the effect of temperature on *Ps. longiformis*.~~

~~However, when considering *P. bisulcata*, the best model includes abundance and seawater temperature. Since belemnites abundance significantly changes from P5 to P6 (lower Polymorphum Zone), it is not possible to rule out the effect of this factor in the body size pattern.~~

~~Environmental drivers of body size fluctuations~~

1103 The Pli-Toa boundary corresponds to the first episode of stress in the Early Toarcian
palaeoenvironmental crisis (Suan, Mattioli et al. 2008) and it was associated with a pulse of
volcanic activity. Our results are consistent with the hypothesis that seawater
palaeotemperature is an important driver of the belemnite body-size variation, with a
decreasing body-size associated with increasing seawater temperature (Fig. S2). Salinity could
also affect the interpretation of the oxygen isotope data, although decreased salinity has been
argued to be less significant in Peniche than warming and carbon cycle perturbations, based on
phytoplankton communities (Correia, Riding et al. 2017) and modelling (Dera and Donnadieu
2012). The body-size dynamics observed is partially driven by carbon cycle perturbations
(Suan, Mattioli et al. 2008) or other factors linked with increased volcanic activity, suggested
by enhanced mercury concentration (Percival, Witt et al. 2015), emphasizing the potential
synergy of the different factors affecting the ocean-atmosphere system.

Despite being one of the early pulses of the Early Toarcian crisis (Dera, Toumoulin et al. 2016),
the Pli-Toa boundary in Peniche does not record major changes in the taxonomic composition
(Fig. 4), which is consistent with the results of the studies from other regions at higher
latitudes (Doyle 1990, Caswell and Coe 2014, Pinard, Weis et al. 2014, Weis, Neige et al. 2018).
It has been shown that the major turnover is observed during the T-OAE instead (Caswell
and Coe 2014).

Besides the reduction in adult size of *Ps. longiformis* across the Pli-Toa boundary, immediately
after it, from bed P5 to bed P6, the size of *P. bisulcata* specimens markedly decreases (109%).
This coincides with the disappearance of *Hastitidae* and with palaeoenvironmental
perturbations (Fig. 1), such as palaeotemperature, carbon cycle perturbations and volcanism
(Suan, Mattioli et al. 2008, Percival, Witt et al. 2015). Therefore, we consider the main crisis to
span bed P5 to P6, which coincide with the range of the first pulse of the Upper
Pliensbachian-Lower Toarcian crisis (Figs. 1 and 3, shaded area). *P. longiformis* seems to be
the most affected taxon, despite its quick recovery in terms of size, apparently becoming more
robust over time. This response could potentially be attributed to a life history strategy to cope
with warming events, equivalent to the one known from extant squids [6, 35]. *P. bisulcata*
seems to respond differently—mainly by the increased abundance and reduced size of
juveniles in fossil samples (Fig. S4)—potentially indicating the temporary migration of larger
juveniles/adults out of the basin, simultaneously with the recovery of adults of *Ps. longiformis*.
It has been argued that *P. bisulcata* was mainly affected by widespread anoxia during the T-
OAE (Ullmann, Thibault et al. 2014).

The differential response of the taxa might indicate different environmental tolerances,
related to individual physiology or life history strategies (Ohlberger 2013). The only important
morphological difference between *Ps. longiformis* and the other taxa is the presence of an
epirostrum, a calcified structure which develops late in ontogeny (Arkhipkin, Weis et al. 2015).
The increased $p\text{CO}_2$ that characterizes the environmental crisis of the Early Toarcian [24] could
have affected its calcification potential **or its food availability or habitat** in Peniche. However,
further data needs to be assembled to assess the relationship between the development of an
epirostrum and environmental conditions.

Belemnites disappear from the Peniche section during the onset of the T-OAE, ca. 30 cm below
the *Polymorphum Levisoni* zones boundary. This is coincident with the beginning of the
second mercury anomaly and with warming, similarly to the lowermost Toarcian conditions
(Fig. 4). However, the decrease in belemnites abundance or disappearance is not preceded by
a body size decrease. This could reflect inhospitable conditions in the Lusitanian Basin during
the onset of the T-OAE, what did not happen during the lowermost Toarcian. This might
suggest that coleoids – even those with flexible life history strategies, pre-adapted to such
conditions – could not cope with deteriorating conditions in subtropical epicontinental
European basin during the T-OAE. Further research in other basins is necessary to disentangle
the extent of this phenomenon. Belemnites temporarily re-appear in bed P10 (upper Levisoni
Zone), after the onset of the T-OAE, but with a poor record of four large specimens of
*Acrocoelites* genus, driving the increase in median size of belemnite assemblages during the
uppermost Lower Toarcian. This genus is known to appear and radiate in various environments
after the T-OAE and might have benefitted from its consequences (Ullmann, Thibault et al.
2014). Increased warming as well sea level changes during the Pliensbachian-Toarcian and
TOAE are also relate with marked changes in facies (e.g., decrease in carbonate lithologies,
increase of organic shales) and sedimentation rates. as well as size and other morphological
temperature is always the best parameter in explaining the body size variation, although there
are. However, the significance is higher when we don't correct which is not surprising
considering that residualizing might also filter out potential ecological signals contained within
lithology and abundance in addition to affect of preservation and collection biases. arameters,
the best model explaining the body size variation after correction is Hg/TOC as opposed to
temperature without correction in both species. At the species level, smaller size datasets,
makes it difficult to distinguish between individual models particularly in *Passaloteuthis*
*bisulcata* for which we have less datapoints. in *Passaloteuthis* and might overinterpret the
data warming decreasing triggered by volcanism probably workspecies

5 Conclusions

We document for the first time a median body size decrease of belemnites across the Pli-
Toa boundary event at different scales of palaeobiological/ecological organisation in the

1177 Peniche reference section. We find no evidence for a major taxonomic turnover (similar to
1178 results in more high-latitude sections) and the decrease is mainly driven by a decrease in
adult size of the dominant taxon *Ps. longiformis* – Lilliput effect. This coincides with the onset
of the first pulse of the Upper Pliensbachian – Lower Toarcian palaeoenvironmental crisis,
probably ~~related with~~triggered by ~~by~~ volcanogenic outgassing ~~volcanism of activity in~~ the
Karoo Ferrar igneous province. Our results indicate that climate warming best explains the
body-size fluctuations observed, although the interplay with perturbations of the carbon
cycle and other factors ~~associated triggered by~~with increased volcanogenic outgassing volcanic
~~activity~~ are evident. Our results suggest that morphological responses precede extinction
pulses ~~in~~ belemnites (i.e. during the T-OAE in the Lusitanian Basin) and highlight that
decreasing adult ~~body-size~~ might rather be a life-history strategy to deal with temporarily
deteriorating conditions ~~related to warming~~ than being ~~a result associated with of~~ marked
taxonomic turnover. ~~Furthermore, changes within the belemnite assemblage, such as~~
competition and emigration dynamics, seem to play a role in explaining belemnite body size
variation, albeit minor. ~~or extinctions.~~

**Data accessibility:** The specimens are stored in the Science Museum (*Museu da Ciência*,
University of Coimbra, Portugal) and the 3D data will be stored on Zenodo.

Permission to carry out fieldwork: The field work was authorized by the encharged
authorities.

**Funding:** This is a contribution to the DFG Research Unit FOR 2332 (grant number Ba 5148/1-1
to KDB) TERSANE and to the IGCP 655 (IUGS-UNESCO).

Competing interests: We declare we have no competing interests.

**Author's contributions:** PR participated in the design of the study, carried out the data
collection and analysis, interpreted the results and drafted the manuscript. KDB designed and
coordinated the study, participated in the data analysis and helped interpreting the results and
drafting the manuscript. The fieldwork was carried out by PR, KDB and LVD. PN helped with
data collection and preliminary data analysis. LVD helped interpreting the results. RW helped

with the taxonomic work. All authors contributed to the writing process and gave final
approval for publication.

~~Competing interests: We declare we have no competing interests.~~

~~Funding: This is a contribution to the DFG Research Unit FOR 2332 (grant number Ba 5148/1-1~~
~~to KDB) TERSANE and to the IGCP 655 (IUGS-UNESCO).~~

**Acknowledgements:** We thank Birgit Leipner—Mata and Manuel Blank for helping in
belemnite preparation, and Benjamin Gügel, Christian Schulbert and Martina Schlott for
helping with scanning the specimens. We thank Manuel Steinbauer, Carl Reddin, ~~and~~ Wolfgang
Kiessling ~~and Vanessa Roden~~ for the valuable comments on the ~~data analysis.~~

~~manuscript.~~

References

1. Schmidt DN, Lazarus D, Young JR, Kucera M. Biogeography and evolution of body size in marine plankton. *Earth-Sci Rev.* 2006;78(3–4):239–66.
2. Daufresne M, Lengfellner K, Sommer U. Global warming benefits the small in aquatic ecosystems. *Proceedings of the National Academy of Sciences.* 2009;106(31):12788–93.
3. Gardner JL, Peters A, Kearney MR, Joseph L, Heinsohn R. Declining body size: a third universal response to warming? *Trends Ecol Evol.* 2011;26(6):285–91.
4. Sheridan JA, Bickford D. Shrinking body size as an ecological response to climate change. *Nature Climate Change.* 2011;1(8):401–6.

[revised manuscript text omitted]

1373 *Sciences Journal*. 2007;16.

58. Rocha RBd, Mattioli E, Duarte LV, Pittet B, Elmi S, Mouterde R, et al. Base of the
Toarcian Stage of the Lower Jurassic defined by the Global Boundary Stratotype Section and
Point (GSSP) at the Peniche section (Portugal). *Episodes*. 2016;39(3).
- 59. Doyle P. *The British Toarcian (Lower Jurassic) Belemnites: Part 2: Palaeontographical*
*Society*; 1992.
- 60. Pinard J-D, Weis R, Neige P, Mariotti N, Di Cencio A. Belemnites from the Upper
Pliensbachian and the Toarcian (Lower Jurassic) of Tournadous (Causse, France). *Neues*
*Jahrbuch für Geologie und Paläontologie-Abhandlungen*. 2014;273(2):155-77.
- 61. Sanders MT, Bardin J, Benzaggagh M, Cecca F. Early Toarcian (Jurassic) belemnites
from northeastern Gondwana (South Riffian ridges, Morocco). *Paläontologische Zeitschrift*.
2015;89(1):51-62.
- 62. Tomašových A, Schlögl J, Biroň A, Hudáčková N, Mikuš T. Taphonomic Clock and
Bathymetric Dependence of Cephalopod Preservation in bathyal, sediment-starved
environments. *PALAIOS*. 2017;32(3):135-52.
- 63. Rita P, De Baets K, Schlott M. Rostrum size differences between Toarcian belemnite
battlefields. *Fossil Record*. 2018;21(1):171-82.
- 64. Monks N, Hardwick JD, Gale AS. The Function of the Belemnite Guard.
*Paläontologische Zeitschrift*. 1996;70(3):425.
- 65. Benito MI, Reolid M, Viedma C. On the microstructure, growth pattern and original
porosity of belemnite rostra: insights from calcitic Jurassic belemnites 2016.
- 66. Hoffmann R, Richter DK, Neuser RD, Jöns N, Linzmeier BJ, Lemanis RE, et al. Evidence
for a composite organic-inorganic fabric of belemnite rostra: Implications for
palaeoceanography and palaeoecology. *Sedimentary Geology*. 2016;341:203-15.
- 67. Reitner J, Urlichs M. Echte Weichteilbelemniten aus dem Untertoarcium
(Posidonienschiefer) Südwestdeutschlands. *Neues Jahrbuch für Geologie und Paläontologie*.
1983;165(3):450-65.
- 68. Klug C, Schweigert G, Fuchs D, Kruta I, Tischlinger H. Adaptations to squid-style high-
speed swimming in Jurassic belemnitids. *Biology Letters*. 2016;12(1):20150877.
- 69. Rosa R, Gonzalez L, Dierssen HM, Seibel BA. Environmental determinants of latitudinal
size-trends in cephalopods. *Marine Ecology Progress Series*. 2012;464:153-65.
- 70. Pecl G, Jackson G. The potential impacts of climate change on inshore squid: biology,
ecology and fisheries. *Rev Fish Biol Fish*. 2008;18(4):373-85.
- 71. Rego BL, Wang SC, Altiner D, Payne JL. Within- and among-genus components of size
evolution during mass extinction, recovery, and background intervals: a case study of Late
Permian through Late Triassic foraminifera. *Paleobiology*. 2012;38(4):627-43.
- 72. Oksanen J, Blanchet FG, Kindt R, Legendre P, Minchin P, O'Hara RB, et al. Vegan:
community ecology package. R package version 2.5-2 ([https://CRAN.R-](https://CRAN.R-project.org/package=vegan)
[project.org/package=vegan](https://CRAN.R-project.org/package=vegan)). 2018.
- 73. Jakob EM, Marshall SD, Uetz GW. Estimating fitness: a comparison of body condition
indices. *Oikos*. 1996;77(1):61-7.
- 74. Box GEP, Jenkins GM. *Time Series Analysis: Forecasting and Control: Holden-Day*; 1994.
- 75. R Development Core Team. R: a language and environment
for statistical computing. Vienna, Austria: R Foundation for Statistical Computing. Retrieved
from <http://www.R-project.org>. 2018.
- 76. Pinheiro J BD, DebRoy S, Sarkar D, R Core Team. nlme: Linear and Nonlinear Mixed
Effects Models. R package version 3.1-137 (<https://CRAN.R-project.org/package=nlme>). 2018.
- 77. Spiess A-N. qpcR: Modelling and Analysis of Real-Time PCR Data. R package version
1.4-1 (<https://cran.r-project.org/web/packages/qpcR/>). 2018.
- 78. Cárdenas AL, Harries PJ. Effect of nutrient availability on marine origination rates
throughout the Phanerozoic eon. *Nature Geoscience*. 2010;3(6):430.

79. Jenkyns HC. Geochemistry of oceanic anoxic events. *Geochemistry, Geophysics,*
*Geosystems.* 2010;11(3).
80. Them TR, Gill BC, Caruthers AH, Gerhardt AM, Gröcke DR, Lyons TW, et al. Thallium
isotopes reveal protracted anoxia during the Toarcian (Early Jurassic) associated with
volcanism, carbon burial, and mass extinction. *Proceedings of the National Academy of*
*Sciences.* 2018;115(26):6596-601.
81. Arkhipkin A, Weis R, Mariotti N, Shcherbich Z. 'Tailed' cephalopods. *J Molluscan Stud.*
2015;81:345-55.
82. Danise S, Twitchett RJ, Little CT, Clemence ME. The impact of global warming and
anoxia on marine benthic community dynamics: an example from the Toarcian (Early Jurassic).
*PLoS One.* 2013;8(2):e56255.
83. Comas-Rengifo MJ, Duarte LV, Felix FF, Joral FG, Goy A, Rocha RB. Latest
Pliensbachian-Early Toarcian brachiopod assemblages from the Peniche section (Portugal) and
their correlation. *Episodes.* 2015;38(1):2-8.
84. Rita P, Reolid M, Duarte LV. Benthic foraminiferal assemblages record major
environmental perturbations during the Late Pliensbachian–Early Toarcian interval in the
Peniche GSSP, Portugal. *Palaeogeogr, Palaeoclimatol, Palaeoecol.* 2016;454:267-81.
85. Reolid M, Duarte L, Rita P. Changes in foraminiferal assemblages and environmental
conditions during the T-OAE (Early Jurassic) in the northern Lusitanian Basin, Portugal.
*Palaeogeography, Palaeoclimatology, Palaeoecology.* 2019.
86. Xu W, Ruhl M, Jenkyns HC, Leng MJ, Huggett JM, Minisini D, et al. Evolution of the
Toarcian (Early Jurassic) carbon-cycle and global climatic controls on local sedimentary
processes (Cardigan Bay Basin, UK). *Earth and Planetary Science Letters.* 2018;484:396-411.
87. Huang C, Hesselbo SP. Pacing of the Toarcian Oceanic Anoxic Event (Early Jurassic)
from astronomical correlation of marine sections. *Gondwana Research.* 2014;25(4):1348-56.
88. Boulila S, Galbrun B, Sadki D, Gardin S, Bartolini A. Constraints on the duration of the
early Toarcian T-OAE and evidence for carbon-reservoir change from the High Atlas (Morocco).
*Global and Planetary Change.* 2019;175:113-28.
89. Holland SM. The quality of the fossil record: a sequence stratigraphic perspective.
*Paleobiology.* 2000:148-68.
90. Doyle P, Macdonald DI. Belemnite battlefields. *Lethaia.* 1993;26(1):65-80.
91. Pittet B, Suan G, Lenoir F, Duarte LV, Mattioli E. Carbon isotope evidence for
sedimentary discontinuities in the lower Toarcian of the Lusitanian Basin (Portugal): Sea level
change at the onset of the Oceanic Anoxic Event. *Sedimentary geology.* 2014;303:1-14.
92. Ruvalcaba Baroni I, Pohl A, van Helmond NA, Papadomanolaki NM, Coe AL, Cohen AS,
et al. Ocean circulation in the Toarcian (Early Jurassic): A key control on deoxygenation and
carbon burial on the European Shelf. *Paleoceanography and Paleoclimatology.* 2018;33(9):994-
1012.
93. Dera G, Donnadieu Y. Modeling evidences for global warming, Arctic seawater
freshening, and sluggish oceanic circulation during the Early Toarcian anoxic event.
*Paleoceanography and Paleoclimatology.* 2012;27(2).
94. Correia VF, Riding JB, Fernandes P, Duarte LV, Pereira Z. The palynology of the lower
and middle Toarcian (Lower Jurassic) in the northern Lusitanian Basin, western Portugal. *Rev*
*Palaeobot Palynol.* 2017;237:75-95.
95. Boletzky Sv. Effets de la sous-nutrition prolongée sur le développement de la coquille
de *Sepia officinalis* L.(Mollusca, Cephalopoda). *Bulletin de la Société zoologique de France.*
1974;99(4):667-73.
96. Gutowska MA, Melzner F, Pörtner HO, Meier S. Cuttlebone calcification increases
during exposure to elevated seawater pCO₂ in the cephalopod *Sepia officinalis*. *Marine*
*Biology.* 2010;157(7):1653-63.

- 97. Dorey N, Melzner F, Martin S, Oberhänsli F, Teyssié J-L, Bustamante P, et al. Ocean
acidification and temperature rise: effects on calcification during early development of the
cuttlefish *Sepia officinalis*. *Marine Biology*. 2013;160(8):2007-22.
- 98. Lacoue-Labarthe T, Reveillac E, Oberhänsli F, Teyssié J-L, Jeffree R, Gattuso J. Effects of
ocean acidification on trace element accumulation in the early-life stages of squid *Loligo*
*vulgaris*. *Aquatic Toxicology*. 2011;105(1-2):166-76.
- 99. Ikeda M, Hori RS, Ikehara M, Miyashita R, Chino M, Yamada K. Carbon cycle dynamics
linked with Karoo-Ferrar volcanism and astronomical cycles during Pliensbachian-Toarcian
(Early Jurassic). *Global and Planetary Change*. 2018;170:163-71.
- 100. Rodríguez-Tovar FJ, Miguez-Salas O, Duarte LV. Toarcian Oceanic Anoxic Event induced
unusual behaviour and palaeobiological changes in *Thalassinoides* tracemakers.
*Palaeogeography, Palaeoclimatology, Palaeoecology*. 2017;485:46-56.
- 101. McArthur JM, Algeo TJ, van de Schootbrugge B, Li Q, Howarth RJ. Basinal restriction,
black shales, Re-Os dating, and the Early Toarcian (Jurassic) oceanic anoxic event.
*Paleoceanography*. 2008;23(4).
- 102. Verberk WCEP, Atkinson D. Why polar gigantism and Palaeozoic gigantism are not
equivalent: effects of oxygen and temperature on the body size of ectotherms. *Functional*
*Ecology*. 2013;27(6):1275-85.

 **Figure 1**— Variation of belemnite body size (geometric mean), abundance and species ranges through time. Variation of Hg/TOC (52), carbon and oxygen isotopes (23, 25). The lower shaded area highlights the first pulse of the upper Pliensbachian—lower Toarcian crisis. Bed numbers, from P1 to P10, indicate the levels sampled with a black dot. The bed numbers in square brackets correspond to the beds that were merged due to sample size constraints. The interrupted lines connect beds where the species were missing in between. BG=belemnite gap.

*Figure 2—Conceptual scheme depicting the tested hypotheses regarding the mechanisms behind belemnites body-*
*size reduction, at different palaeobiological scales of organisation.*

Figure 3 – A: Venn diagram depicting the partition of belemnites body-size-variation between taxonomic composition, bed separation and ontogeny. The values in brackets correspond to the whole time-series and the values without brackets correspond to the Pli-Toa boundary. The shaded area highlights the first pulse of the upper Pliensbachian – lower Toarcian crisis. B: Effect on size change at the assemblage and population-species scales of organisation. See TS2 and TS5 for details. For a correct interpretation of the right-side scale, the bottom of the arrowhead should be considered, rather than the tip of the arrow. Note that the x-axis depicts comparisons of consecutive pairs of beds (not to scale). BG = belemnite gap.

*Figure 4—Belemnite body-size variation (GM=geometric mean), lithology and proportional body-size change across*
*the studied interval at the assemblage and SPECIES population scales (P. bisulcata and Ps. longiformis). Note that*
*the stratigraphic log is not drawn to scale, for real thickness of beds, see Fig. 1. For sample size, see Fig. S6 and TS1.*
*BG = belemnite gap.*

*Figure 5—Body-size variation of cross-borders (P. bisulcata, Ps. longiformis and Hastitidae sp. indet.). Although*
*Parapassaloteuthis sp. A also crosses the boundary, we have no complete specimens across this interval and*
*therefore, no accurate estimate of body size*

Commented [PR14]: Not sure if we need this figure in the paper or in the supplement

Commented [KD15]: Could be in if we discuss it somewhere. I feel the one with data for Passaloteuthis and Acrocoelites would be more important thought.

Figure 6 – relative abundance of juvenile vs adult specimens at the belemnite assemblage scale from the upper Pliensbachian to the Middle Levisoni Zone in the Peniche section.

Supplementary material

- Table S1 – Measured body-size parameters on belemnites specimens from the Upper Pliensbachian-Toarcian interval ranging from Emaciatum to Levisoni Zone of the Peniche section and selected environmental parameters from the literature (see ESM file 1).
- Table S2 – Log ratio return, sample size and results of the statistical tests on the differences in the median of pairs of beds (ESM file 28).
- Table S3 – Partitioning analysis results highlighting the relative variation fractions (proportions) of belemnites body size explained by taxonomic composition, beds and ontogeny for the whole time-series and for the pair-to-pair analysis. Bold indicates a minimum of 90% statistical significance level (ESM file 3).
- Table S4 – Details of GLS regression analysis comparing belemnite body-size and the environmental parameters: a – details of the selected GLS models, after correcting for lithology and abundance; b – Details of the selected GLS models, without correcting for lithology and abundance (raw data); c – Full list of the analysed models for assemblage and species scales, with AICc scores and delta values indicated, together with the p-values for the models with the smaller AICc score (ESM file 4).

			From-interval	To-interval	Phase1
assemblage	median-GM	P925-(Emaciatum)	P949-(Emaciatum)	background	
assemblage	median-GM	P949-(Emaciatum)	P961-(Emaciatum)	background	
assemblage	median-GM	P961-(Emaciatum)	P982-(Emaciatum)	background	
assemblage	median-GM	P982-(Emaciatum)	P984-(Polymorphum)	background	
assemblage	median-GM	P984-(Polymorphum)	P8-(Polymorphum)	crisis	
assemblage	median-GM	P8-(Polymorphum)	P12-(Polymorphum)	survival	
assemblage	median-GM	P12-(Polymorphum)	P14-(Polymorphum)	recovery	
assemblage	median-GM	P14-(Polymorphum)	P20-(Polymorphum)	recovery	
assemblage	median-GM	P20-(Polymorphum)	P133-(Levisoni)	recovery	
Passaloteuthis	median-apical-length	P925-(Emaciatum)	P949-(Emaciatum)	background	
Passaloteuthis	median-apical-length	P949-(Emaciatum)	P961-(Emaciatum)	background	
Passaloteuthis	median-apical-length	P961-(Emaciatum)	P982-(Emaciatum)	background	
Passaloteuthis	median-apical-length	P982-(Emaciatum)	P984-(Polymorphum)	background	
Passaloteuthis	median-apical-length	P984-(Polymorphum)	P8-(Polymorphum)	crisis	
Passaloteuthis	median-apical-length	P8-(Polymorphum)	P12-(Polymorphum)	survival	
Passaloteuthis	median-apical-length	P12-(Polymorphum)	P14-(Polymorphum)	recovery	
Passaloteuthis	median-apical-length	P14-(Polymorphum)	P20-(Polymorphum)	recovery	
Pseudohastites-longiformis	median-apical-length	P961-(Emaciatum)	P982-(Emaciatum)	background	
Pseudohastites-longiformis	median-apical-length	P982-(Emaciatum)	P984-(Polymorphum)	background	
Pseudohastites-longiformis	median-apical-length	P984-(Polymorphum)	P8-(Polymorphum)	crisis	
Pseudohastites-longiformis	median-apical-length	P8-(Polymorphum)	P12-(Polymorphum)	survival	
Pseudohastites-longiformis	median-apical-length	P12-(Polymorphum)	P14-(Polymorphum)	recovery	
Pseudohastites-longiformis	median-apical-length	P14-(Polymorphum)	P20-(Polymorphum)	recovery	

Table S53 – Detailed values of the components of the assemblage size shift calculated with Rego et al. method
 (disappearance of taxa effect, appearance of new taxa effect and within-lineage effect) (see ESM file 2).

Table S34 — Partitioning analysis results highlighting the relative variation fractions (proportions) of belemnites body size explained by taxonomic composition, beds and ontogeny for the whole time series and for the pair to pair analysis. Bold indicates a minimum of 90 % statistical significance level (ESM file 6).

Beds pairs	Residuals	fraction-a Taxonomic composition	fraction-b Beds	fraction-c Ontogeny	fractions d+e+f+g Interaction
P1-P2	0.21	0.62	0.01	0.19	0.04
P2-P3	0.25	0.54	0.01	0.17	0.02
P3-P4	0.25	0.46	0	0.17	0.12
PT-bounday (P4-P5)	0.2	0.49	0.01	0.11	0.21
	-	p-value=0.001	p-value=0.023	p-value=0.001	-
P5-P6	0.29	0.4	0.01	0.2	0.12
P6-P7	0.49	0.2	0.02	0.33	0
P7-P8	0.39	0.48	0	0.21	0
P8-P9	0.35	0.45	0.02	0.2	0.04
P9-P10	0.18	0.27	0	0.09	0.45
Whole time-series	0.26	0.47	0.02	0.17	0.07
	-	p-value=0.001	p-value=0.001	p-value=0.001	-

Ta

Table S5 – AICc ranking of models describing the effect of palaeotemperature ($\delta^{18}\text{O}$), carbon cycle perturbations
 ($\delta^{13}\text{C}$ and $\delta^{13}\text{C}_{\text{carb}}$) and volcanism (Hg/TOC) on belemnites body-size for the assemblage and species scale (*Ps.*
 *longiformis* and *P. bisulcata*), not corrected for the effects of sedimentary properties (lithology and belemnite
 abundance). Only the models with $\Delta < 2$ are listed. See TS4 for the full list of GLS model (ESM file 5).

*s.* – The models that best explain the belemnite body-size variation are in bold.

	Model no.	Δ	AIC-c scores	ANOVA p-value
Assemblage				
	null	5.09	1926.04	-
GM ~ $\delta^{18}\text{O}$	5	0	1920.95	0.0077
GM ~ $\delta^{18}\text{O} + \delta^{13}\text{C}$	2	1.97	1922.93	-
GM ~ $\delta^{18}\text{O} + \text{Hg/TOC}$	4	1.03	1921.98	-
Ps. longiformis				
	null	9.83	681.38	-
GM ~ $\delta^{18}\text{O} + \text{Hg/TOC}$	4	0	671.55	0.0009
GM ~ $\delta^{18}\text{O} + \delta^{13}\text{C}$	2	1.12	672.67	-
GM ~ Hg/TOC + $\delta^{13}\text{C}$	3	1.09	672.64	-
GM ~ $\delta^{18}\text{O}$	5	1.32	672.87	-
GM ~ Hg/TOC	6	0.16	671.71	-
P. bisulcata				
	null	6.17	621.14	-
GM ~ $\delta^{18}\text{O} + \text{Hg/TOC}$	4	0	614.97	0.0056
GM ~ Hg/TOC + $\delta^{13}\text{C} + \delta^{18}\text{O}$	1	1.66	616.64	-

Table S4 – Details of the selected GLS models comparing belemnites body-size and the
 environmental parameters (ESM file 4).

Model	Coefficients						
		Value	Std. Error	t-value	p-value	degrees-of-freedom	residual
#23	Intercept	14.518	1.477	9.829	0.000	340	338
	$\delta^{18}\text{O}$	3.152	1.180	2.671	0.008	-	-
Assemblage							
#15	Intercept	12.373	1.553	7.950	0.000	161	156
Ps. longiformis	$\delta^{13}\text{C}$	1.069	0.609	1.765	0.080	-	-
	$\delta^{18}\text{O}$	3.872	0.918	4.203	0.000	-	-
	lithology_marl	-2.273	1.299	-1.995	0.048	-	-

	Lithology:marly limestone	-0.268	1.116	-0.253	0.801	-	
#5	Intercept	17.515	3.265	3.364	0.000	76	73
P.-bisulcata	$\delta^{18}O$	4.228	2.281	1.853	0.068	-	
	abundance	0.070	0.029	2.417	0.018	-	

1571 Table S6 – Detailed values of the components of the assemblage size shift calculated with Rego et al. method
 (disappearance of taxa effect, appearance of new taxa effect and within-lineage effect) (see ESM file 6).

~~Table S5 – Detailed values of the components of the assemblage size shift calculated with Rego et al. method~~
 ~~(disappearance of taxa effect, appearance of new taxa effect and within-lineage effect) (see ESM file 2).~~

Table S67 – Micro-CT phoenix v|tome|x s 240 (Research Edition) scanner settings for the scanned belemnite
 specimens. The reconstruction was made with the GEDatos|x 2.4 software. Subsequent image stack processing (e.g.
 subsampling), as well as the measurements and volume acquisition, was derived using Studio Volume Graphics
 Max™ v 3.0 software (Heidelberg) (see ESM file 37).

Note: The following figures will be displayed in "Supplementary figures" in the ESM file 87.

*Figure S1* – Selected belemnites from the Upper Pliensbachian-Lower Toarcian of Peniche: A1 - *Bairstowius* sp. A
 (adult from bed P1)Plet; A2- *Bairstowius* sp. A (adult from bed P2); A3- *Bairstowius* sp. A (juvenile from bed P1); B1-
 *Pseudohastites longiformis* (adult from bed P4); B2 and B3 - *Pseudohastites longiformis* (juveniles from bed P4); C1
 and C2- *Hastitidae*, sp. indet. (from bed P5); D1 and D2 - *Passaloteuthis milleri* (adults from bed P2).

e. The left side corresponds to lateral view and right side corresponds to dorsal/ventral view.

*Figure S32 – Selected belemnites from the Upper Pliensbachian-Lower Toarcian of Peniche: E1- Parapassaloteuthis*
 *sp. A (adult from bed P3); E2 - Parapassaloteuthis sp. A (juvenile from bed P3); F - Passaloteuthis sp. juv. (from bed*
 *P1); G1-G2 - Passaloteuthis bisulcata (adults from bed P5); G3 - Passaloteuthis bisulcata (adult from bed P2.) The*
 *left side corresponds to lateral view and right side corresponds to dorsal/ventral view. Plate left corresponds to*
 *lateral and right corresponds to dorsal/ventral side*

Figure S23 – Belemnite absolute abundance (Numberno. of belemnites collected per square meter² (known-area collection). The interrupted black line connects the mean (black dots) for the beds where more than one quadrat were analysed. Each red dot represents a quadrat of 1 m². The error bars represent the standard deviation of the mean. ~~diag.~~

Figure 1 Correlation matrix depicting the relation between the different explanatory variables used in the statistical models.

Figure S34 — Correlation matrix depicting the relation between the different environmental variables available in the literature for the Peniche section. Note that only $\delta^{18}\text{O}$, $\delta^{13}\text{C}$, Hg.TOC, lithology and abundance were included as explanatory variables in the regression analysis due to the high collinearity among the remaining ones. See TS1 for details used in the statistical models and additional ones that we didn't use.

*Figure S54— Correlation matrix depicting the relation between the different explanatory variables used in the*
*performed statistical models: regression analysis between belemnite body-size and abiotic parameters.*

Figure S56 -- Belemnite body-size variation (GM), abundance of belemnites per m² and proportional body-size change, relative frequency of ontogenetic stages and sample size (n) across the studied interval Upper Pliensbachian-Lower Toarcian of Peniche at the assemblage and species scales (*P. bisulcata* and *Ps. longiformis*). Note that the stratigraphic log is not drawn to scale, for real thickness of beds, see Fig. 1. *Passaloteuthis* genus was used to calculate the relative frequency of ontogenetic stages of *P. bisulcata* due to the difficulty of a species level classification of the juvenile specimens of *Passaloteuthis* genus. The error bars correspond to the 95 % confidence interval. Note that the sample size (n) corresponds to species level. BG = belemnite gap.

Figure S15 – Cumulative frequency of the different taxa comprising the Peniche-belemnite
 assemblage. Using all the data we have and not only GM

Commented [KD16]: Yes, could be added. However, confidence intervals would be good to have have. Furthermore – this is actually called relative abundance in another graph – please decide on name for such graph and be consistent

Commented [PR17R17]: Add total sample size
 Add in the results section

Figure S7—GM was used, neanic and adults were merged.

Commented [KD18]: Careful that always same bed numbers are used. These are still old numbers. Also again error bars as you did for the assemblage would be useful here.

Commented [KD19]: I would add them for completeness sake.

**Figure 2 – L was used adults and neanic merged**

**Figure 3 – GM was used.**

Figure S7 – Body-size variation (GM) and sample size (n) of Pliensbachian-Toarcian boundary-crossers (*P. bisulcata*,
*Ps. longiformis* and *Hastitidae* sp. indet.) of the Peniche section. Although *Parapassaloteuthis* sp. A also crosses the
Pli-Toa boundary, we have no complete specimens across this interval and therefore, no accurate estimate of body-
size.

Figure S852 – Relationship between the effect of the seawater palaeotemperature proxy ($\delta^{18}O$) and belemnite body size (GM) during the Upper Pliensbachian-Lower Toarcian of Peniche. The grey area corresponds to the 95% confidence interval.

*Figure S9 – Relationship between belemnite body-size (GM) and lithology for the studied interval in Peniche.*
 *Adjusted $R^2 = -0.0009292$; p -value = 0.4083. The grey area corresponds to the 95 % confidence interval.*

*Figure S10 – Relationship between belemnite body-size (GM) and belemnite absolute abundance (no. of belemnites/*
 *m²) for the studied interval in Peniche. The grey area corresponds to the 95 % confidence interval. Adjusted*
 *$R^2 = 9.356e-05$; p -value = 0.3105.*

Figure S3—Number of belemnites collected per square meter (known area collection). The black line connects the median.

Figure S4 — Body-size variation of *P. bisulcata* and *Ps. longiformis* through the studied interval with emphasis on the ontogeny.

*Figure S5* – Body-size variation of cross-borders (*P. bisulcata*, *Ps. longiformis* and *Hastitidae sp. indet.*). Although
*Parapassaloteuthis sp. A* also crosses the boundary, we have no complete specimens across this interval and
therefore, not accurate proper body size.

*Figure S6* – Belemnite body size variation, abundance of belemnites per m² and proportional body size change across
the studied interval at the assemblage and population scales (*P. bisulcata* and *Ps. longiformis*). Note that the
stratigraphic log is not drawn to scale, for real thickness of beds, see Fig. 1. BG = belemnite gap.

Response letter

We thank reviewers for their constructive criticism, and time spent to analyse this manuscript. The responses, and explanations related to their comments are listed below:

Referee #1

Referee's comment: *The way data are merged is not convincing. As explained better below, in 2 instances occurrences and sizes from different beds have been merged, making it unlikely that the results of the analysis would show true population and individual size changes through time.*

Author's reply: We understand the point of the reviewer but in fact, subzone or zone scale is a common scale in fossil body-size studies (e.g. Rego et al., 2012; Atkinson et al., 2019) which is even coarser than the scale that we use in our study (bed scale). Ideally, we would analyse the belemnite assemblage bed by bed without pooling any data. However, this kind approach is not possible in the Peniche outcrop (and possibly in anywhere else that we are aware of) because the number of belemnites decreases dramatically from the Pli-Toa boundary event upwards. Additionally, due to outcrop conditions (the bed surfaces are not well exposed), the Toarcian, in comparison with the Pliensbachian, is not easy to sample and therefore, it is not possible to collect a reasonable number of complete specimens per bed. Without merging we would have less than 10 specimens per bed in some cases (and not all of them were complete), which is not a significant number of specimens in a body-size analysis, especially when we want to perform a robust taxonomical and ontogenetical approach. Moreover, when the regression analysis was performed, each bed was assigned the value of the respective geological proxy for that particular bed, according to the data available in the literature, without merging any bed.

Referee's comment: *Besides that, the manuscript is extremely difficult to read. The introduction is confusing (e.g., the Pl-To event and the T-OAE are not clearly distinguished and defined; and the cooling and warming intervals through the interval not even mentioned), the methods are poorly described, as some parts of the results.*

**Author's reply:** We have improved the clarity of the introduction by clearly separating the Pli-Toa
boundary event from the T-OAE. Additional paragraphs were also included in the methods and
results in order to better explain the details of our study.

**Referee's comment:** *The discussion in many instances sounds like a repletion of the results, and fail*
*to critically discuss the data, to compare them with previous studies, and to put them in a wider*
*context.*

**Author's reply:** The discussion is now clearer; the ideas were further developed and put in a wider
context by comparing the taxonomic assemblages from Peniche with contemporaneous ones. We
also use the wider context in terms of the cephalopods context, comparing fossil with recent data.
Additionally, we clearly discuss the belemnite gap in Peniche during the T-OAE and how it compares
with the data from north-eastern Tethyan coeval basins. However, it is worth to mention that we
present for the first time a high-resolution body-size analysis on Pliensbachian-Toarcian belemnite
assemblages on different scales of organization. In general, the few existing studies have a much
larger temporal scale of analysis and are usually focused on diversity, taxonomy and abundance
patterns, except for Morten and Twitchett (2009), which found an ambiguous belemnite body-size
response). Therefore, the comparison with the existing data on Lower Jurassic belemnites is
sometimes not so straightforward.

**Referee's comment:** *The use of the chosen abiotic factors to explain body-size change is not justified,*
*the meaning of the proxies is not explained (apart from $\delta^{18}O$), and their possible relationship with*
*body-size change not clarified.*

**Author's reply:** We have included Table 1 in the methods section (2.4 Environmental drivers of body-
size changes) with the list of the different geologic proxies used, the environmental variables that
they represent and their theoretical controls on body-size. The latter is further developed in the
discussion section (4.1. Environmental drivers of body-size fluctuations). Additionally, we have
included in the introduction section a paragraph explaining how we expect belemnite body-size to be
affected by changes in abiotic factors.

**Referee's comment:** *The ecology (life style and diet) has never been mentioned in the entire*
*manuscript. Is it not an important factor potentially linked to body-size change? For instance, could*
*not body size be related to food availability? Even if abiotic factors are the main topic of the*
*manuscript, it does not mean that biotic ones should be totally neglected – considering that they are*
*the second important cause of change in body size after temperature - as pointed out by Gardner et*
*al. 2011 – cited in the introduction.*

**Author's reply:** Despite the complexity of assessing the effects of biotic factors on body-size when
studying a fossil community, we agree with the referee in the sense that they cannot be neglected.
Therefore, the influence of competition and food availability/nutrient influx within belemnite
assemblages are now discussed (4.1. Scales and mechanisms of body-size variation and 4.3.
Environmental drivers of body-size fluctuations). However, it is noteworthy that belemnite ecology is
a topic that lacks a detailed analysis. The little data available derives mainly from the interpretation
of geochemical analysis (e.g. Ullmann et al., 2014). Therefore, further interpretation would be
speculative.

**Referee's comment:** *The low significance of the results and the way the data are described and*
*discussed, does not make this manuscript of high impact or of interest for a wide public, and it is thus*
*not suitable for publication in this journal.*

**Author's reply:** No comment.

**Referee's comment** (lines 35-42): *This is a very general statement, and is fine at the beginning of the*
*manuscript. However, you should probably mention why it is important to know more about the*
*response of cephalopods to climate change, in terms of body size.*

**Author's reply:** We agree with the referee and, therefore, this idea is now further developed in the
introduction.

**Referee's comment** (lines 46-55): *The so-called Pli-Toa crisis needs to be delineated better. It is*
*important to distinguish between the Pl-To event and the Early Toarcian (eTo) Event – in term of type*
*of environmental perturbation and pattern of extinction. It is never mentioned that in the late*
*Pliensbachian (Emaciatum-Spinatum Zone) there is a cooling event; as observed in the northern*

*Tethys (e.g., Korte & Hesselbo, 2016) and in the Iberian Peninsula (e.g., Rosales et al. 2004; Rosales et*
*al., 2018 –Peniche section included (Suan et al., 2008; Suan et al., 2010). Rapid warming occurs at the*
*Pl- To boundary, and successive cooling has been hypothesized for the first ammonite zone of the*
*Toarcian (Polymorphous-Tenuicostatum: e.g. Suan et al. 2008; Ruebsan et al. 2019), as also shown in*
*Figure 1. This environmental background should be clearly described to give the reader a correct*
*understanding of the event(s). In fact, this should also be considered when referring to the extinction*
*event. For instance, Dera et al 2010 (cited by the authors), relate ammonite extinction the end of the*
*Late Pliensbachian to strong cooling of seawater.*

**Author's reply:** As mentioned above, we have improved the clarity of the introduction by clearly
separating the Pli-Toa boundary event from the T-OAE, in terms of palaeoenvironmental
perturbations but also in terms of the effects on marine biota. The references suggested were added
as well. Additionally, in both results and discussion sections, we clearly separate both events.

**Referee's comment** (lines 60-61): *This sentence needs to be reworded: is the section well studied in*
*term of stratigraphy and geochemistry or the fauna?*

**Author's reply:** We meant well studied in terms of geochemistry. The sentence was reworked.

**Referee's comment** (lines 68-69): *Can you please specify the approximate time-span covered by the*
*study?*

**Author's reply:** Approximately 2.75 Ma, according to Ogg et al. (2016). This information is now
included.

**Referee's comment** (line 74): *Similar trends of what? Body size? Is this sufficient to decide to merge*
*samples collected with different methods? This needs to be justified better.*

**Author's reply:** We collected samples using two different methods with two distinct purposes. The
data collected within quadrats is useful to calculate belemnite absolute abundance. However, since
the quadrats do not cover enough area to guarantee an appropriate number of complete specimens

1
2
3 per bed, it was necessary to collect additionally from that same bed. This is now explained more
clearly in the methods section.

**Referee's comment** (lines 76-77): *The interval included in beds P9a-b-c corresponds to about a half of*
*the Polymorphum Zone. The time-equivalent zone in the northern Tethys (Semicelatum Zone) has an*
*estimated length of 0.302 Ma (McArthur et al. 2000 - Earth and Planetary Science Letters 179, 269-*
*285). How is it possible to obtain any population-level information from merging fossils that come*
*from a period of 0.15 million of years (half of the time-interval)? Same observation for beds P3a and*
*P3b. The Emaciatum zone has a duration of 1.21 Ma (Spinatum Zone – McArthur et al. 2010).*

**Author's reply:** As mentioned above, subzone or zone scale is a common scale in fossil body-size
studies which is even coarser than the scale that we use in our study (bed scale). The need for a
significant number of complete specimens, representative of different species, made us using the
approach of pooling some of the analysed beds.

**Referee's comment** (lines 80-82): *for a total of 277+199 measures? Please make it clear.*

**Author's reply:** 199 is never mentioned. We scanned 277 of the 930 collected rostra. 109 specimens
were measured with the calliper, also from the 930 collected rostra. This paragraph is now improved.

**Referee's comment** (lines 82-83): *What does this sentence mean? Did you use the more complete*
*specimens for identification? This sentence needs to be re-worded. What monographs-museum*
*collections did you use for identification? Did you only use the more complete specimens for*
*ontogenetic analysis? Where are the methods that explain how you divided juvenile from adult*
*specimens?*

**Author's reply:** One paragraph is now included in the material and methods section (2.1. Taxonomy
and ontogeny) explaining how did we classify the specimens and how did we distinguish between
ontogenetic stages. The literature used is also mentioned.

**Referee's comment** (lines 83-84): *What analyses? So far you have never described any analysis*
*except CT-scan and calliper size measurements.*

**Author's reply:** We are referring to the body-size analysis. The sentence is now changed.

**Referee's comment** (line 127): *Again, it is not mentioned how the different ontogenetic stages were*
*identified.*

**Author's reply:** As mentioned above, this information is now included in a new paragraph in the
material and methods section.

**Referee's comment** (line 131): *Why do you refer to differences between samples now, if you were*
*speaking about populations? Did not your samples contain more than one species? Please be sharp*
*clear when describing methods.*

**Author's reply:** When we mention "sample" we mean a bed sample, that is, an assemblage of
belemnites collected from a specific stratigraphic level, identified in Fig. 1 as "bed no.", on the left
side, right next to the stratigraphic log (black dots). Fig. 1 depicts also the stratigraphic distribution of
the different species across the studied interval, emphasising the height/position of different
samples with black dots.

**Referee's comment** (line 141): *Why perturbation in the carbon cycle and volcanic activity should*
*control belemnite body size? Please explain better what the variation of each proxy indicates (its*
*increase and decrease) and motivate its use.*

**Author's reply:** As mentioned above, this information is now included in Table 1, mentioned in the
introduction sections and further discussed in the discussion section (4.3. Environmental drivers of
body-size fluctuations).

**Referee's comment** (lines 142-143): *Do you mean that lithology and belemnite abundance were*
included as extra parameters in the model? Please explain better. Was lithology included as the only

*categorical variable – while the others are continuous variables? Please specify. Did you check for*
*possible correlations between variables before running the model?*

**Author's reply:** Collinearity between variable was tested by using *cor* function in R, as mentioned in
the material and methods section (2.4 Environmental drivers of body-size changes). Due to the high
correlation between sedimentary properties (lithology and fossil and abundance) and abiotic
parameters (see correlation matrix) a new methodological approach was developed. This is now
described in detail in the in the material and methods section (2.4 Environmental drivers of body-size
changes). We also included extra information about the different variables considered, namely if
they are continuous or categorical.

**Referee's comment** (lines 171-172): *What does it mean that Bairstowius sp. has been replaced by P.*
*longiformis? Ecologically? How can you affirm it? Please explain.*

**Author's reply:** Fig. 1 depicts the stratigraphic distribution of the belemnite taxa across the studied
interval. From Fig. S1 we can see that Bairstowius sp. A is the most abundant taxa during the upper
Pliensbachian (beds P1, P2 and P3). From bed P4 upwards, *Ps. longiformis* is the most abundant taxa
and *Bairstowius sp.* disappears. In that sense, we consider that *Ps. longiformis* replaces
stratigraphically *Bairstowius sp. A*. However, the data available does not allow further conclusions,
namely if this stratigraphic replacement reflects an ecological replacement.

**Referee's comment** (line 191): *what about Parapassaloteuthis sp. A and P. milleri? Are they not also*
*crossing the boundary?*

**Author's reply:** These two taxa are present in both Toarcian and Pliensbachian assemblages, but we
don't find them in the last Pliensbachian bed and in the first Toarcian bed (beds 4 and 5) – despite
considerable sample size. In this sense, *Parapassaloteuthis sp. A* and *P. milleri* are not boundary-
crossers. This can be seen in Fig. 1, where the interrupted line represents the absence of these
species in these specific beds. This sentence was rewritten.

**Referee's comment** (lines 271-278): *This part seems a mere repetition of the results. I do not see any*
*discussion of the data here.*

**Author's reply:** This part was removed from the discussion.

**Referee's comment (lines 269-270):** *Not clear: a better sampling process (marly beds) is related to*
*the non-significant size reduction? What does it mean? If sequence stratigraphic interpretation is*
*available, it should be added to Figure 1, highlighting transgressive and regressive cycles, condensed*
*horizons and so on.*

**Author's reply:** This is no longer discussed due to the lack of high-resolution sequence stratigraphic
data.

**Referee's comment (lines 287-290):** *What is the % of explained variation in the models by 13C?*

**Author's reply:** The details of the 7 models analysed are given in the supplementary material. The %
of explained variation is given by adjusted r-squared. However, since we used GLS models, we cannot
obtain this value. The gls function in the nlme package fits using maximum likelihood (or restricted
maximum likelihood) rather than looking at sums of squares, so an adjusted r-squared is not a direct
result like in ordinary least squares. The idea of r-squared does not really translate well to models
beyond ordinary least squares, so adjusted r-squared would not either. There are other measures of
overall model fit that penalize or adjust for the number of terms in the model, e.g. AIC and BIC. The
AIC scores were provided before but the BIC are also provided now.

**Referee's comment (lines 287-290):** *How did changes in the carbon isotopes could have affected*
*changes in belemnite body size? Did you expect a decrease in body size with a carbon negative*
*excursion? Why? Again, what is the role of volcanic activity (any toxicity of the water?).*

**Author's reply:** The potential effect of volcanic activity and the changes in the carbon isotopes on
body-size are explained in Table 1. Furthermore, they are now discussed in detail the discussion
section (4.3. Environmental drivers of body-size changes).

**Referee's comment (lines 295-298):** *This part is extremely confusing. The interval between P5 and P6,*
*where *P. bisulcata* size markedly decrease, is characterized by a decrease in temperature! More*
*positive $\delta^{18}O$. Why this is not discussed? It seems pretty relevant.*

**Author's reply:** This is now discussed in the discussion section (4.1. Scales and mechanisms of body-
size variation) when we clearly separate the Pli-Toa boundary event, the aftermath of Pli-Toa
boundary event and T-OAE.

**Referee's comment (lines 306-307):** *I am surprised that the hypothesis of Ulman about the shift of*
*belemnite habitat (life in the upper part of the water column) during the OAE is not mentioned or*
*discussed.*

**Author's reply:** This author and this specific hypothesis are (and were) mentioned two times in the
discussion when we refer to the appearance of genus *Acrocoelites* after the T-OAE and when we
refer that it has been argued that *P. bisulcata* was mainly affected by widespread anoxia during the
T-OAE (lines 342 and 374 of the original manuscript).

**Referee's comment (line 318):** *The warming of the T-OAE is gradual, while the one at the PI-TO*
*seems abrupt. Is this difference true, or is a consequence of the stratigraphic condensation of the PI-*
*To interval? Please discuss it somewhere in the text. The two warming events do not look similar to a*
*reader with no stratigraphic knowledge.*

**Author's reply:** This is now discussed in greater detail in the discussion section (4.2. The effect of
sedimentary facies on belemnite body-size).

**Referee's comment (line 335):** *Above you said that the main crisis also included the beds (P5-P6)*
*where there is the decrease in size of *P. bisulcata*. Why you do not mention it in the conclusions?*

**Author's reply:** This idea is now better developed in the discussion section (4.1. Scales and
mechanisms of body-size variation) and we had named the interval ranging from bed P5 to P6 “the
aftermath of the Pli-Toa boundary event” (section 4.1.2).

**Referee's comment** (Figure 1): *The carbon isotope curve is missing. Please also add the extension of*
*the T-OAE.*

**Author's reply:** The carbon isotope curve is now in the figure.

**Referee's comment** (Figure 4): *The third plot has no legend!*

**Author's reply:** The legend of all plots is located on the left side of this figure. The label of the x axis
was missing for the 1st and 3rd plots but it is now depicted.

**Referee's comment** (line 73): *The mean of what?*

**Author's reply:** The mean of the abundance of belemnites per m². This is now clearly stated in the
text.

**Referee's comment** (line 157): *Typo in Fig. number.*

**Author's reply:** Accepted and corrected.

Referee # 2

Referee's comment: I really liked this paper. I think that you could beef up your discussion and conclusions more by discussing WHY there might be a size change across this interval, but it's really nice to see a solid empirical dataset being used to address an interesting question. Most of my edits/suggestions in the attached document are grammatical.

Author's reply: The discussion was improved by clearly stating possible causes of belemnite body-size, such as changes in the development rate and growth rate, and how they can be affected by environmental parameters.

Referee's comment (line 30): *should read 'belemnite' not 'belemnites'*

Author's reply: Accepted and changed.

Referee's comment (lines 38-39): *What do you mean by 'synergies', and why should they be considered? (I'm just looking for an extension to the sentence that explains how these abiotic factors might affect the system).*

Author's reply: A sentence was added explaining how the combination of abiotic factors can affect the body-size pattern observed.

Referee's comment (line 44): *Either 'widely' is overstating the case given you only list two references, or there are more relevant references that you've omitted here.*

Author's reply: Extra references were added.

Referee's comment (lines 51-52): *Instead of "distal sections sediments" (grammatically incorrect) say "the sediments in distal sections" or other phrase.*

**Author's reply:** Accepted and changed.

**Referee's comment (line 58):** *Instead of 'did not find an unambiguous response' say "found an*
*ambiguous response" (remove the double negative)*

**Author's reply:** Accepted and changed.

**Referee's comment (line 61):** *How do well-studied geochem and strat allow you to study mechanisms*
*of size change? I think you maybe mean drivers? Stratigraphy and geochemistry are not mechanisms*
*of phenotypic change but they can certainly be controls on its preservation in the fossil record (strat)*
*and environmental conditions that caused it (geochem).*

**Author's reply:** A well-studied section in terms of geochemistry and stratigraphy provides is useful in
the sense that it allows us the identify the abiotic drivers of belemnite body-size changes. The
sentence was rewritten to avoid confusion.

**Referee's comment (line 69):** *Insert 'the' before 'Emaciatum'.*

**Author's reply:** Accepted and changed.

**Referee's comment (lines 71-72):** *Instead of 'found within known areas' and giving the area size,*
*consider saying "(1) sampling all specimens within [n] 1m² areas."*

**Author's reply:** Accepted and changed.

**Referee's comment (line 72):** *Where did the 30 complete specimens come from? Within the same 1m*
*quadrats as the total sampled specimen set, or from the entire bed, or somewhere else?*

**Author's reply:** As explained above, this is now clarified in the text.

**Referee's comment** (line 72): *The mean of what? And were you not calculating means unless you*
*sampled more than one 1m section?*

**Author's reply:** As explained above, the mean of the abundance of belemnites per m². This is now
clear in the text.

**Referee's comment** (lines 73-74): *Which samples were merged? The total-specimen 1m samples +*
*the complete-specimen samples, per bed, or just the complete specimens from the total-specimen 1m*
*samples + the complete-specimen samples? Could you please reword this to be a little clearer?*

**Author's reply:** This paragraph is now better organized, with a clear separation between the two
sampling methods used and the number of specimens of each.

**Referee's comment** (line 74): *If you only used complete specimens, why are you reporting the bulk*
*samples?*

**Author's reply:** Because the incomplete specimens were used to calculate belemnite absolute
abundance. The complete specimens were used for the body-size analysis. This is now described in
detail in the material and methods section.

**Referee's comment** (line 82): *Why do you assume less time-averaging for complete specimens? I*
*realise you reference this but a one-sentence explanation of the background of this assumption would*
*be helpful.*

**Author's reply:** The idea is now further developed in the text.

**Referee's comment** (lines 99-106): *Your set of hypotheses is nicely explained! I appreciate the clarity*
*of this section very much. However, you list H5 before H4, which is a little confusing. It would be*
*helpful if you could rearrange this to have the hypotheses go in numeric order, to help the reader.*

**Author's reply:** Thank you. Accepted and changed in both figure and text.

**Referee's comment (lines 120-124):** *The clarity of this would be improved if you assigned symbols to*
*the terms of your equations instead of writing them out in words.*

**Author's reply:** Since we do not use the terms of the equations in the text, we do not think it is useful
to assign symbols to them. It would rather make it more confusing in our opinion.

**Referee's comment (line 126):** *What is variation partitioning?*

**Author's reply:** It was already explained in the text that variation partitioning allowed us to assess
the proportion of body-size variation explained by ontogeny, taxonomic assignment and separation
by beds, and by their joint effects. Additionally, we mentioned that the variation in the body-size
data was partitioned using partial redundancy analysis. We do not think further explanation is
necessary.

**Referee's comment (line 137):** *Should read 'belemnite' not 'belemnites'.*

**Author's reply:** Accepted and changed.

**Referee's comment (line 142-143):** *Which sedimentary properties are assessed by lithology and*
*mean abundance of belemnites? I assume you mean something like grainsize and carbonate*
*proportion? Could you explicitly state which sedimentary variables you use?*

**Author's reply:** Lithologies were distinguished based on the carbonate/clay content. This is now
referred in the text.

**Referee's comment (line 173):** *'barren from belemnites' should read 'barren of belemnites'*

**Author's reply:** Accepted and changed.

**Referee's comment** (line 190, and elsewhere): *'Individual scale' is a little misleading – this is more*
*like per-taxon per-bed scale, if I understand your analysis correctly. I would try and find a different*
*name for this scale of analysis.*

**Author's reply:** Accepted. This is now changed for "individual ontogenetic scale".

**Referee's comment** (line 199): *'belemnites' should read 'belemnite'.*

**Author's reply:** Accepted and changed.

**Referee's comment** (line 212): *'belemnites' should read 'belemnite'*

**Author's reply:** Accepted and changed.

**Referee's comment** (line 229): *'combines' should read 'combine'*

**Author's reply:** Accepted and changed.

**Referee's comment** (line 230): *'according with' should read 'according to'*

**Author's reply:** Accepted and changed.

**Referee's comment** (line 245): *'can relate with' should read 'can relate to'*

**Author's reply:** Accepted and changed.

**Referee's comment** (line 256): *'belemnites' should read 'belemnite'*

**Author's reply:** Accepted and changed.

**Referee's comment** (lines 269-270): *Do you mean that marls are physically easier to sample because*
*they're softer?*

**Author's reply:** Yes, that's what we mean. However, this part was removed due to the lack of high-
resolution sequence stratigraphic data.

**Referee's comment:** Line 271: *"In what concerns" should read "Concerning" or "Looking only at"*

**Author's reply:** Accepted and changed.

**Referee's comment** (line 272): *'belemnites' should read 'belemnite'.*

**Author's reply:** Accepted and changed.

**Referee's comment** (lines 274-275): *I follow your logic about how temperature is the dominant*
*factor, but can you add any explanation about why lithology and abundance are being picked up by*
*the model despite not varying across this specific boundary?*

**Author's reply:** This happened due to the high collinearity between sedimentary properties and
environmental parameters. As explained before, we decided to change our methodological approach
in order to correctly combine sedimentary properties and environmental parameters in the same
analysis. The new approach is explained in detail in the material and methods section.

**Referee's comment** (line 274): *'belemnites' should read 'belemnite'*

**Author's reply:** Accepted and changed.

**Referee's comment** (line 277): *'belemnites' should read 'belemnite'*

**Author's reply:** Accepted and changed.

**Referee's comment** (line 285): *This is the first time you've mentioned salinity as a potential factor:
can you explain more about how it could affect the O isotopes and how you've ruled it out? It might
be worth moving this part to the introduction, if you're just ruling it out rather than incorporating it
into discussion of your findings.*

**Author's reply:** We keep the influence of salinity in the discussion with some additional explanations.
Although we are just ruling it out, we think it makes more sense to include it in the discussion, rather
than in the introduction.

**Referee's comment** (line 288): *Either 'dynamics' should read 'dynamic' or the 'is' should be 'are'*

**Author's reply:** Accepted and changed.

**Referee's comment** (Line 319): *'belemnites' should read 'belemnite'*

**Author's reply:** Accepted and changed.

**Referee's comment** (Line 321): *'what' should read 'which'*

**Author's reply:** Accepted and changed.

**Referee's comment** (Figure 1): *You give three scales (paleotemperature/delta 18O, delta 13C, and
size, but there are only two lines on the figure – it looks like you're not reporting any delta 13C data.
Either remove the scale or put the actual data on the figure*

**Author's reply:** The curve is now added to the figure.

**Referee's comment** (Line 548): *'belemnites' should read 'belemnite'*

**Author's reply:** Accepted and changed.

**Referee's comment** (Line 553): *'belemnites' should read 'belemnite'*

**Author's reply:** Accepted and changed.

**Referee's comment** (Table S4): *you need spaces between 'lithology' and the next word on lines 9 and
10.*

**Author's reply:** This is no longer in the table.

**Referee's comment** (lines 582-583): *should read "and therefore no accurate estimate of body size"*

**Author's reply:** Accepted and changed.

Referee #3

Referee's comment: *The authors do not distinguish at all between the belemnite and its internal calcitic hard part, the rostrum. What they set out to study is the evolution of the animal's size and what they measure is the size of the calcitic remains, which do not necessarily change in size in the same proportion as the animals. I think it needs some further explanation why the authors think that the size of the rostrum is a valid proxy for the size of the belemnite. Is there a study on material where the outline of the soft tissue of the belemnite is still preserved and the ratio of rostrum size versus belemnite body size could be quantified?*

Author's reply: We did not find any remains of belemnite soft parts in Peniche. There are only a few examples in the world where soft-tissues of belemnites are preserved and therefore can not be used for quantitative studies. In the material and methods section (2.2 Sampling methods and belemnite body-size proxies), we now have a paragraph dedicated to the usage of the rostrum as a proxy for belemnite body-size.

Referee's comment: *The authors pose a series of different hypotheses for body size changes that could be tested. These include abundance of juveniles and the size of adult rostra. Could the authors elaborate on how juveniles were differentiated from adults? I assume this cannot only be done on the basis of size as this would lead to circular reasoning but there may be morphological characters of the rostra that allow for this distinction to be made.*

Author's reply: A completely new subsection - 2.1. Taxonomy and ontogeny- was included in the material and methods section explaining this point.

Referee's comment: *With respect to the size of the adults there are two ways for a body size reduction: 1) belemnites grow less rapidly 2) belemnites die younger on average. These are two quite different options that have distinct implications. Could the authors comment on the likelihood of either of the two?*

**Author's reply:** In order to address those questions in detail, one would have to perform a
sclerochronological analysis to assess changes in the growth rate during ontogeny. This is not the
goal of our work and we cannot address these questions within the scope of this study. Our goal is to
assess the mechanisms and environmental drivers of belemnite body size, not the causes of it.
However, we have included in Table 1 the theoretical controls of environmental parameters on body-
size and we further develop those ideas in the discussion section. Regarding the mortality of
juveniles, this question is now partially addressed in the discussion section by the analysing the
differences observed in the proportion of juveniles vs adults throughout the studied interval for
particular species (*Ps. longiformis* and *P. bisulcata*).

**Referee's comment:** *Could the authors place their findings on belemnite size and abundance in some*
*more context? Is there some published literature on other sites that these data can be compared to?*
*Xu et al. (2018) for example report quantitative data on belemnite abundance in the Mochras drill*
*core from the late Pliensbachian to late Toarcian and compare this to qualitative information from*
*elsewhere. Work on faunas, (sometimes including belemnite abundance) has also been done on*
*outcrops in other European basins (e.g., Caswell and Coe, 2013; Danise et al., 2015) and some*
*interesting data have been presented on benthos in Iberian sections (Gahr, 2005).*

**Author's reply:** Some remarks on how the belemnite gap from Peniche compares with data from UK
sections were added to the discussion section, citing the suggested references and some additional
ones.

**Referee's comment:** *Many interesting points are mentioned, but the authors very rarely explore them*
*in detail and in dedicated sections. Rather, one sometimes needs to collect information from*
*throughout the text to bring together evidence for some of the studied aspects. For example, "the*
*evolution of belemnite abundance through time" could be one dedicated paragraph. How does it*
*change, is this a function of sedimentation rate alone or are there other factors, are known levels of*
*condensation related to larger fractions of fractured and bioeroded material or not? Where do the*
*belemnites disappear to when they become rare? Are there known sections where numbers do not*
*change so they would just migrate, or are their numbers diminished regionally?*

**Author's reply:**

The organization of the manuscript was improved by clearly separating different aspects of the
discussion. The discussion is separated into three subsections: “Scales and mechanisms of body-size
variation”, “The effect of sedimentary facies on belemnite body-size” and “Environmental drivers of
body-size fluctuations”. In the first subsection, we discuss individually the Pli-Toa boundary event,
the aftermath of the Pli-Toa boundary event and the T-OAE and we discuss the relationship between
belemnite abundance and migration. The relationship between abundance and condensation
intervals is addressed in the second subsection (4.2. The effect of sedimentary facies on belemnite
body-size).

**Referee’s comment:** *The authors employ a wide range of statistical tests and models and report on*
*the significance of the outcomes of these. Often I feel this part of the study gets too much weight and*
*the implications of the findings are then discussed in too little detail. The list of arguments that*
*Mercury = volcanism, volcanism = temperature increase, Hg spikes = Karoo Ferrar, temperature*
*change = defining stress for belemnites are all controversial and worth some further discussion I*
*think. Is Hg indeed a robust proxy for volcanism or are there other ways of increasing Hg/TOC? Why*
*would one have to normalise against TOC? I am not aware of any study that has shown that Hg is*
*indeed associated with organic matter. It could equally be carried in sulphides.*

*Volcanism occurs not only in Karoo-Ferrar in the Early Jurassic and there may be other volcanic*
*centres nearer by but of much lesser importance for global climate parameters that could have*
*caused Hg spikes?*

**Author’s reply:** These questions are now address in detail throughout the manuscript. We discuss
other sources of mercury in the marine sediments and also the theoretical controls of them on
belemnite body-size. Karoo-Ferrar is used in the manuscript an abbreviation of Karoo-Ferrar-Chon
Aike LIP, as in Percival et al., 2015. Irrespective of their ultimate controls, they coincide with rapid
warming (triggered by volcanism) and are, therefore, a good means to test if those phases
correspond to more rapid changes in body-size.

**Referee’s comment:** *The Suan brachiopod d18O curve from the Lusitanian Basin is very valuable but*
*based on very few analyses and questions remain whether all the samples analysed only show a*
*robust averaged temperature signal or whether complicating effects such as seasonality, diagenesis*
*and palaeosalinity might have played a measurable role as well. Considering that even analytical*
*uncertainty of the d18O analyses can explain quite a large fraction of the range of 1.2 permil*

*(probably about 0.4 permil assuming typical analytical uncertainty of +/- 0.2 permil), a substantial*
*amount of noise is to be expected when trying to identify a palaeotemperature signal.*

**Author's reply:** Despite the open questions regarding Suan et al. (2007) paleotemperature data, it is
hitherto the best dataset available for the Peniche section. We acknowledge the complex effects
such as seasonality, diagenesis and palaeosalinity on oxygen isotopes, and so the authors, but our
study does not allow us to tackle with it. The way our results are discussed in the manuscript is now
more conservative, giving more value to the interplay of the different environmental parameters,
rather than temperature alone in driving the pattern of belemnite body-size.

**Referee's comment:** *The models find that temperature is an important factor defining the size*
*evolution of belemnites, but I think it may be useful to discuss why that might be. What is it that*
*makes belemnites grow smaller? Is it a change in water chemistry associated with climate change or*
*maybe changes in food source? The authors point out these options, but could this be explored in a*
*little more detail?*

**Author's reply:** As mentioned before, in order to address those questions, one would have to
perform a sclerochronological analysis to assess changes in the growth rate during ontogeny. This is
not the goal of our work and we cannot address these questions within the scope of this study. Our
goal is to assess the mechanisms and environmental drivers of belemnite body size and not the
causes of it. However, the potential causes of body size change, such as changes in the growth and
development rates, are clearly stated in Table 1 (theoretical controls on body-size) and discussed in
the discussion section.

**Referee's comment (line 30):** *Here and elsewhere I think it should read "belemnite body size"*

**Author's reply:** Accepted and corrected.

**Referee's comment (line 229):** *"models that combine"*

**Author's reply:** Accepted and corrected.

**Referee's comment (line 230):** *"according to"*

**Author's reply:** Accepted and corrected.

**Referee's comment** (line 274, 277): "*belemnite abundance*"

**Author's reply:** Accepted and corrected.

**Referee's comment** (line 288): "*are partially driven*"

**Author's reply:** This sentence is not no longer in the manuscript.

**Referee's comment** (line 299): "*first pulse*"

**Author's reply:** Accepted and corrected.

**Referee's comment** (line 302): *Can another word than "robust" be found here? "robust" is often used to describe the morphology of certain belemnite rostra and I assume this is not what the authors want to address here.*

**Author's reply:** Accepted and changed.

**Referee's comment** (line 311): "*2*" in *pCO₂* should be in subscript.

**Author's reply:** Accepted and corrected.

**Referee's comment** (line 312): *Is ocean acidification suggested to affect complex organisms forming internal hard parts as badly as single-celled phytoplankton? I would assume that animals such as squid have more efficient means to alter the chemical parameters in their calcification space and might be able to cope with pH and chemical changes in seawater better.*

**Author's reply:** Indeed the mechanisms are different. This is now discussed in detail in the discussion section (4.3. Environmental drivers of body-size fluctuations).

**Referee's comment (line 312):** *Is there any literature looking into the changes in faunal assemblages*
*or depositional environments in Peniche at the time to allow at least for some tentative answers to*
*these points?*

**Author's reply:** These points are now further discussed and new references are included.

**Referee's comment (line 323):** *"epicontinental European basins"*

**Author's reply:** Accepted and corrected.

**Referee's comment (line 342):** *"temporarily"*

**Author's reply:** Accepted and corrected.

**Referee's comment (line 426):** *Author names seem to be wrongly formatted*

**Author's reply:** Accepted and corrected.

**Referee's comment:** *References 34 and 37 are duplicates of 30 and 24.*

**Author's reply:** These references were indeed duplicates and therefore deleted.

**Referee's comment (lines 484-490):** There seems to have been some formatting issue with these
references. What are the open brackets at the end standing for?

**Author's reply:** This was a formatting issue related to the software used. Corrected.

**Referee's comment (Figure 1):** *I assume the long-dotted line represents a connection of size data*
*from the pre T-OAE interval to Acrocoelites? Maybe it would be worth adding a data point at the level*
*where Acrocoelites was studied. At present one is drawn to the biostratigraphy.*

**Author's reply:** A black point indicating the level where *Acrocoelites sp.* was collected was already
depicted in the figure (on the right side of the log, indicated as P10).

**Referee's comment** (Figure 1): *Also, there do appear to be no carbon isotope ratios in the plot despite*
*the axis next to the palaeotemperature scale and these data being mentioned in the caption. Maybe*
*it would be worth adding the Hesselbo et al. (2007, EPSL 253, 455-470) wood and bulk rock carbon*
*isotope data to contextualize the low-resolution brachiopod data from Suan et al.?*

**Author's reply:** The carbon isotope curve was missing but it is now included in the figure.

**Referee's comment:** *The supplementary figures and tables could benefit from some more work on*
*the style, e.g., italicising species names, using the same font sizes throughout on axis descriptions,*
*adding error bars to Figure S2, cutting some of the decimal places in the reported statistics in Table*
*S2.*

**Author's reply:** The species names are now italicised in figures. The fonts of the supplementary
figures were homogenized. Fig. S2 (now Fig. S8) includes confidence intervals now. The decimal
places in Table S2 were cut.

Mechanisms and drivers of belemnite body size dynamics across the Pliensbachian-Toarcian crisis

Patrícia Rita^{1,2}, Paulina Nätscher¹, Luís V. Duarte^{2,3}, Robert Weis⁴, Kenneth De Baets¹

¹Geozentrum Nordbayern, Friedrich-Alexander Universität Erlangen-Nürnberg, Erlangen 91054, Germany

²MARE (Marine and Environmental Sciences Centre), 3004-517 Coimbra, Portugal

³Department of Earth Sciences University of Coimbra, 3070-790 Coimbra, Portugal

⁴National Museum of Natural History Luxembourg, Department of Palaeontology, L-2160 Luxembourg

Correspondence to: Patrícia Rita (patricia.rita@fau.de)

Body-size reduction has been attributed to climate warming under scenarios of mass extinction episodes. However, concerning belemnites, the mechanisms and environmental drivers of those changes are rarely investigated and they usually do not differentiate biological scales of organisation (individual, population and assemblage). We hypothesize that belemnites reduce their adult body-size across the Pliensbachian-Toarcian (Pli-Toa) boundary.

Belemnite body-size dynamics across the Pli-Toa in the Peniche section (Lusitanian Basin, Portugal) were analysed for the first time based on field data. We disentangle the mechanisms and the environmental drivers of the size fluctuations observed from the individual to the assemblage scale of organisation. Despite the lack of a major taxonomic turnover, there is a significant decrease of 13% in median rostrum size across the Pli-Toa, before the Toarcian Oceanic Anoxic Event, and local belemnite extirpation. The decreasing body-size pattern is mainly driven by a reduction in adult size of the two-dominant species in the assemblages, *Pseudohastites longiformis* and *Passaloteuthis bisulcata*. Belemnite size distribution is best correlated with fluctuations in a palaeotemperature proxy, although potential effects of carbon cycle perturbations and other consequences of volcanism, as well as sedimentary architecture (fossil abundance and lithology), cannot be ignored. This highlights the interplay between the consequences of volcanism-related major environmental perturbations within the ocean-atmosphere system in belemnites body-size.

Keywords: belemnites, body-size, Pliensbachian-Toarcian, Peniche, Lusitanian Basin

1 Introduction

Body-size is a key feature of any organism, reflecting its physiology, ecology and evolutionary history, across multiple scales of biological organisation [1]. Body-size reduction has been considered the third universal response to global warming after changes in phenology and species distribution [2-4]. However, individual responses might be taxon-specific [5] and synergies of different abiotic factors have to be considered [6]. More importantly, the effects of warming-related stresses visible in assemblages might depend on the biological scale of organisation considered [2]. Individual living cephalopod taxa, for example, can show rapid life history responses to warming events and associated environmental changes [7, 8].

Reductions in body-size – often dubbed the Lilliput effect [9] - have been widely reported from mass extinctions and evolutionary crises associated with environmental perturbations [10, 11], including the Pli-Toa crisis [12]. However, their ecological and evolutionary significance is still poorly understood [11]. The Pli-Toa crisis is a multi-phased event characterized by palaeoenvironmental perturbations associated with rapid warming, anoxia and the consequent deposition of organic-rich sediments [13-17]. Multiple extinction pulses among nektonic and benthic organisms characterize this period [18-21]. This crisis is assumed to be a consequence of the massive input of CO₂ in the atmosphere by volcanic activity in the Karoo-Ferrar Igneous Province [22], which is supported by the enhanced mercury composition of distal sections sediments [23]. The effects of volcanism have been observed across the Pli-Toa boundary and during the onset of the Toarcian Oceanic Anoxic Event (T-OAE, base of the Levisoni ammonite Zone), the two main pulses of this crisis, associated with increasing seawater palaeotemperature [24].

Most of the research on Pli-Toa belemnites has focused on north-western European basins in temperate latitudes [20, 25], often focusing on geochemistry [26-29]. The only body-size study including belemnite data focused on two genera and did not find an unambiguous response to crisis events [12], which might be potentially related with the selected body-size proxy [30].

The highly abundant belemnite fauna of the Peniche reference section is well studied in terms of geochemistry and stratigraphy [16, 23, 31-33], allowing for the first time the analysis of (i) the mechanisms behind belemnite size fluctuations at different scales of organisation (individual, population and assemblage) at the Pli-Toa transition in subtropical latitudes and (ii) their potential environmental drivers.

2 Material and methods

2.1 Mechanisms of body-size changes

The specimens come from 13 beds sampled from 45 m of Upper Pliensbachian - Lower Toarcian sediments, corresponding to Emaciatum – Levisoni ammonite zones (Fig. 1). We focused on quantitatively collecting from well-exposed bedding planes of limestones and marls. A total of 930 belemnites rostra were collected by (1) sampling all the specimens found within known areas (1 m²) and (2) by collecting 30 complete specimens. When it was possible to sample more than one square meter, the mean was calculated. As the complete specimens of both sampling methods showed similar trends, these samples were merged by bed. Only the complete specimens were used for the body-size analysis. As belemnites become rarer up section, the data derived from beds P9a, P9b and P9c was pooled to obtain a reasonable sample size for a quantitative analysis (Fig. 1). The same was done for beds P3a and P3b. Previous research has demonstrated that the geometric mean (GM) of the apical height, width and length is the most robust proxy for belemnite size [34] and more comparable to proxies used in extant coleoids [35]. For this purpose, 277 of the most complete specimens were CT-scanned to allow the calculation of this metric in a non-destructive way. In addition, in 109 specimens the apical height, width and length was measured with a calliper. The more complete belemnites are suspected to be less time-averaged [36] and easier to assign taxonomically and ontogenetically [30]. The analyses were performed at the bed scale.

Figure 1 – Variation of belemnite body-size (geometric mean), abundance and species ranges through time. Variation of Hg/TOC [23], carbon and oxygen isotopes [16, 37]. The shaded area highlights the first pulse of the upper Pliensbachian - lower Toarcian crisis. The bed numbers in square brackets correspond to the beds that were merged due to sample size constraints.

In order to assess the mechanisms behind the belemnite body-size fluctuations from the assemblage to the individual scale of organisation through time, a set of hierarchical hypotheses (H), modified from [2], was tested (Fig. 2).

Figure 2- Conceptual scheme depicting the tested hypotheses regarding the mechanisms behind belemnites body-size reduction, at different palaeobiological scales of organisation.

The first hypothesis predicts a decrease in mean body-size at the assemblage scale, whatever the
 underlying mechanisms (assemblage size shift hypothesis, H_1 , Fig. 2). If H_1 is validated, there are
 four subsequent hypotheses that could explain this decrease [2, 5]. First, an increase in the
 proportion of small-sized species (population composition shift hypothesis, H_2). Second, a
 decrease in median body-size within specific taxa (within- taxa size shift hypothesis, H_3), which
 could, in turn, be related to a decrease in adult body-size (intraspecific size shift hypothesis, H_5).
 The last hypothesis associates the decrease in mean body-size with an increase in the proportion
 of juveniles (ontogenetic stage structure shift hypothesis, H_4).
 The whole set of hypotheses could explain the Lilliput effect *sensu lato*, although only H_5 relates to
 the Lilliput effect *sensu stricto*, as predicted by the temperature size rule in extant coleoids [38].
 At the assemblage scale of organisation, H_1 was tested by assessing the size distribution of the
 whole assemblage through time. The proportional change in body-size between beds in
 percentage of the log return (example: $\ln \frac{\text{median bed2}}{\text{median bed1}} * 100\%$). The non-parametric Mann-Whitney
 U and Kolmogorov–Smirnov (K-S) tests were used to assess the significance of the results (Table
 S2). The same was done at the population scale, for each taxon. Furthermore, because changes in
 taxonomic composition can influence the body-size patterns by the appearance and/or
 disappearance of taxa, we modified the within and among taxa approach by Rego et al. [39]. This
 method corresponds to a pair-to-pair analysis (comparison of successive beds) which decomposes
 the assemblage size shift (Eq. 1) into three components: a disappearance of taxa effect (Eq. 2), a
 within-lineage effect (Eq. 3) and an appearance of new taxa effect (Eq. 4).

**(Eq. 1) Assemblage size shift** = assemblage median body-size_{bed2} - assemblage median body-size

**(Eq. 2) Disappearance of taxa effect** = pre-boundary crossers_{bed1} - all taxa_{bed1}

**(Eq. 3) Within-lineage effect** = post-boundary crossers_{bed2} - pre-boundary crossers_{bed1}

**(Eq. 4) Appearance of new taxa effect** = all taxa_{bed2} - post-boundary crossers_{bed2}

Finally, at the individual scale of organisation, the hypotheses were tested by comparing different
 ontogenetic stages within the population by their variation in body-size. Variation partitioning
 allowed us to assess the proportion of body-size variation explained by ontogeny, taxonomic
 assignment and separation by beds, and by their joint effects. If the fraction of variation
 corresponding to beds separation is low, this suggests that mechanisms driving differences
 between samples are similar and not related to particularities of the sample (e.g., lithology, age).

Variation in the body-size data was partitioned using partial redundancy analysis as implemented
in the *vegan* package [40]. The significance of each fraction was assessed through ANOVA. The
whole time-series was compared with the Pli-Toa boundary.

2.2 Environmental drivers of body-size changes

In order to test our hypothesis of a relationship between belemnites body-size and environmental
perturbations and sedimentary properties, the distribution of the GM was compared with several
proxies. Brachiopod stable oxygen isotopes ($\delta^{18}\text{O}$)[37], bulk rock carbon isotopes ($\delta^{13}\text{C}$)[16, 37] and
mercury concentration normalized by total organic carbon (Hg/TOC) [23] were used as proxies for
seawater palaeotemperature, perturbations in the carbon cycle and volcanic activity, respectively.
Changes in sedimentary properties were assessed using lithology and mean abundance of
belemnites per m^2 (Fig. 1). We used a multiple linear regression using generalised least squares
(GLS). The model was fitted by maximizing the log-likelihood. The first-order autoregressive
(AR(1)) model was used, which has the property of seeking autocorrelation and of minimizing the
error term in time-series [41]. Twenty-seven models (see R script, supplementary material) were
compared using Akaike's second order corrected information criterion (AICc scores), which
corrects for small sample size. The power of the best model was assessed by means of an ANOVA
test. The statistical significance of each coefficient of the best model was assessed by calculating
the p-value under a t approximation. The significance level was $p < 0.05$ for all the analyses. The
analyses were performed in R [42], using the packages "nlme" [43] and "qpcR" [44].

3 Results

3.1 Size fluctuations

Assemblage scale

Three episodes of median body-size decrease were recognised at the assemblage scale across the
studied interval (Figs. 23-4): P1-P2, P2-P3 and at the Pli-Toa boundary (P4-P5), with the latter
being the only decrease with statistical significance and corresponding to a 13% body-size
decrease (Fig. 3). The remaining pairs of beds correspond to an increasing median size of
belemnite assemblages, with P8-P9 and P9-P10 being statistically significant (Fig. 3).

*Figure 3 – A: Venn diagram depicting the partition of belemnites body-size variation between taxonomic composition,*
*bed separation and ontogeny. The values in brackets correspond to the whole time-series and the values without*

brackets correspond to the Pli-Toa boundary. The shaded area highlights the first pulse of the upper Pliensbachian - lower
Toarcian crisis. B: Effect on size change at the assemblage and population scales of organisation. See TS2 and TS5 for
details. For a correct interpretation of the right-side scale, the bottom of the arrowhead should be considered, rather
than the tip of the arrow. Note that the x-axis depicts comparisons of consecutive pairs of beds (not to scale). BG =
belemnite gap.

Population scale

A total of seven belemnite species were identified in Peniche (Fig. 1). Most of the taxa range from
the uppermost Pliensbachian until the Lower Toarcian (uppermost Polymorphum Zone).
*Bairstowius* sp. A is exclusively represented in the uppermost Pliensbachian, being replaced by
*Pseudohastites longiformis* in the assemblage. During the Toarcian, the uppermost Polymorphum
Zone - Levisoni Zone interval is barren from belemnites, with the exception of the four
*Acrocoelites* sp. specimens found in bed P133.

The assemblage body-size shift at the Pli-Toa boundary is almost exclusively caused by a within-
taxa effect (Fig. 3) and therefore mainly driven by the boundary crossers. The taxonomic
composition immediately before and after the boundary does not change markedly (Fig. 1), with
*Ps. longiformis* the most abundant taxon, comprising 61.9% (P4) and 56.2% (P5) of the
assemblages (Fig. S1). The disappearance of taxa effect (*Passaloteuthis milleri* and
*Parapassaloteuthis* sp.) from P4 to P5, causes 4% of the decrease in the assemblage median body-
size (Fig. 2). In contrast, the other two episodes of body-size reduction (P1-P2 and P2-P3) are
characterized by higher effects of taxon appearance (15.9% and 16.2%, respectively) and
disappearance of taxa (12.1% and 15.2%, respectively), with the within-taxa effect being
comparatively lower than at the boundary (Fig. 3).

*Figure 4 – Belemnite body-size variation, lithology and proportional body-size change across the studied interval at the*
*assemblage and population scales (*P. bisulcata* and *Ps. longiformis*). Note that the stratigraphic log is not drawn to scale,*
*for real thickness of beds, see Fig. 1. For sample size, see Fig. S6 and TS1. BG = belemnite gap.*

Individual scale

Three taxa (*Ps. longiformis*, *P. bisulcata* and *Hastitidae* sp. *indet.*) cross the Pli-Toa boundary (Fig.
1). The most significant and largest median size reduction (17% of decrease, Table S1) is observed
in the most abundant taxon, *Ps. longiformis* (Fig. 4). The second most abundant taxon, *P. bisulcata*,
also show a slight but insignificant decrease (7%, Fig. 4), while *Hastitidae* sp. *indet.*, on the
contrary, slightly increases in median size (8%, Fig S5). It is noteworthy that *P. bisulcata* body-size

shows a marked median size decrease from P5 to P6 (lowermost Polymorphum Zone, Fig. 3),
coincident with the disappearance of *Hastitidae* sp. *indet.*, enhanced Hg/TOC and low $\delta^{18}\text{O}$ (Fig. 1).

Considering the whole time-series, the RDA analysis results revealed that taxonomic composition
(47%) and ontogeny (17%) are responsible for most of the variation in belemnites body-size
variation (Fig. 3A). The separation by bed (fraction b) only explains 2% of the body-size variation.
At the Pli-Toa boundary, the body-size variation partitioning for separate taxonomy and ontogeny
is similar to what is observed for the whole time-series. However, the joint effect of ontogeny
(fraction c) and taxonomy (fraction f) explains 20% of the variation, in contrast with the 6%
observed for the whole time-series (Fig. 3A). This suggests ontogenetic stages behave more
similarly among taxa in explaining body size variation.

At the individual scale of organisation, focusing on *Ps. longiformis*, only adult specimens show a
significant body-size decrease across the Pli-Toa boundary (decrease of 21%, Table S1). The same
is observed for *P. bisulcata*, which also decreases 7% in size. In addition, immediately after the
boundary, from P5 to P6, *P. bisulcata* markedly decreases in size (109%, Figs. 4 and S4).

3.2 Relationship with environmental parameters

The results of the GLS regression analysis reveal that the variation in belemnites body-size is best
correlated with the variation in $\delta^{18}\text{O}$ (Table 1, model no. 23), at the assemblage scale. The overall
model is significant (Table S3), suggesting a body-size reduction with decreasing $\delta^{18}\text{O}$ values (Fig.
S2). However, the models combining $\delta^{18}\text{O}$ with $\delta^{13}\text{C}$ and Hg/TOC could not be ruled out (models
no. 2, 3 and 5, Table 1), according to the AICc scores ($\Delta < 2$).

When considering the most abundant taxon, *Ps. longiformis*, the best model (model no. 15)
combines $\delta^{13}\text{C}$, $\delta^{18}\text{O}$, and lithology to explain the body-size variation. The overall model is
significant (Table 1). However, when considering the different coefficients, only the effect of the
lithology and palaeotemperature are statistically significant (Table S3). The lithology of the studied
samples does not change across the boundary (Fig. 3). The models combining $\delta^{13}\text{C}$, $\delta^{18}\text{O}$, lithology,
Hg/TOC and abundance of specimens (models no. 4, 18 and 20) cannot be ruled out, according to
the AICc scores (Table 1). Nevertheless, the abundance of specimens does not change markedly
across the boundary (Fig. 1).

For *P. bisulcata*, body-size is best correlated with $\delta^{18}\text{O}$ and abundance (model no. 5). The overall
 model is significant (Table 1) but only the effect of abundance is significant (Table S3), being
 proportional to the body-size variation. The models that combines Hg/TOC and abundance
 (models no. 8 and 27) could not be ruled out, according with the AICc scores ($\Delta < 2$).

*Table 1 – AICc ranking of models describing the effect of $\delta^{18}\text{O}$, $\delta^{13}\text{C}$ and Hg/TOC on belemnites body-size for the*237 *assemblage and population scale (*Ps. longiformis* and *P. bisulcata*). Only the models with $\Delta < 2$ are listed. See TS4 for the*238 *full list of GLS models. The models that best explain the belemnite body-size variation are in bold.*

	Model no.	Δ	AIC c scores	ANOVA p-value
Assemblage				
GM ~ 1	null	5.1	1925.5	-
GM ~ $\delta^{18}\text{O}$	23	0	1920.5	0.008
GM ~ $\delta^{18}\text{O}$ + Hg/TOC	2	1.1	1921.6	0.0172
GM ~ $\delta^{18}\text{O}$ + abundance	5	1.8	1922.3	0.0243
GM ~ $\delta^{18}\text{O}$ + $\delta^{13}\text{C}$	3	2	1922.5	0.0277
Ps. longiformis				
GM ~ 1	null	13	681.4	-
GM ~ $\delta^{13}\text{C}$ + $\delta^{18}\text{O}$ + lithology	15	0	684.9	0.0002
GM ~ $\delta^{18}\text{O}$ + lithology	4	0.9	685.9	0.0003
GM ~ Hg/TOC + $\delta^{13}\text{C}$ + lithology + abundance	20	1.5	686.5	0.0005
GM ~ $\delta^{13}\text{C}$ + $\delta^{18}\text{O}$ + lithology + abundance	18	1.6	686.6	0.0005
P. bisulcata				
GM ~ 1	null	4.0912	459.9	-
GM ~ $\delta^{18}\text{O}$ + abundance	5	0	455.8	0.0152
GM ~ Hg/TOC + abundance	8	1.1	456.9	0.0266
GM ~ abundance	27	1	456.9	0.0232

4 Discussion

Scales and mechanisms of body-size variation

Body-size fluctuations in assemblages are not uncommon and can relate with a variety of
 mechanisms. The assemblage size shift hypothesis (H_1) was validated and despite the fluctuations
 observed, only the reduction observed at the Pli-Toa boundary is significant. By decomposing the

248 assemblage size shift we were able to disentangle the mechanisms of body-size change, which
revealed to be almost exclusively driven by a within-lineage effect.

Out of the boundary crossers, the most abundant taxon, *Ps. longiformis* is mainly driving the
observed body-size reduction, allowing us to validate the within-taxa size shift hypothesis.

Finally, at scale of individuals, we were able to validate the intraspecific size shift hypothesis (H_5),
by recognising a size decrease of 21% among adult specimens of *Ps. longiformis* across the Pli-Toa
boundary. This Lilliput effect is similar to significant changes in adult mantle length observed in
extant jumbo squid during episodes of warming [7].

The effect of sedimentary facies on belemnites body-size

Changes in facies (e.g. lithology) and sedimentation rate can affect fossil abundance and
morphological patterns [45]. The lowermost Polymorphum Zone is thought to correspond to a
condensed interval based on the accumulation of micro and macrofossils [33, 46]. Accumulations
with abundant belemnites [47], particularly beds P4 and P5, are known to be related to changes in
sedimentation rate relative to the production of fossils.

The model combining palaeotemperature [37] with belemnite abundance is worse than the model
which only considers temperature when explaining belemnite size.

However, no significant changes in abundance are recorded across the Pli-Toa boundary, making it
possible to conclude that, at least the most significant change on body-size is not caused by
sedimentary condensation. On the other hand, the effects of the lithology are, for example,
evident from beds P1 to P2 (uppermost Pliensbachian), for the whole assemblage. This non-
significant size reduction might be related to the fact that these two beds correspond to more
marly intervals (transgressive cycles of 4th order) and they possibly allow a better sampling process
[32].

In what concerns *Ps. longiformis*, the best model explaining body-size variation includes lithology
and the effect of belemnites abundance could not be entirely ruled out. However, at the Pli-Toa
boundary, these factors cannot explain the size reduction observed, since no major changes in
lithology nor belemnites abundance are observed. It is, therefore, safe to say that the effect of
lithology and abundance is less significant than the effect of temperature on *Ps. longiformis*.

However, when considering *P. bisulcata*, the best model includes abundance and seawater
temperature. Since belemnites abundance significantly changes from P5 to P6 (lower
Polymorphum Zone), it is not possible to rule out the effect of this factor in the body-size pattern.

Environmental drivers of body-size fluctuations

The Pli-Toa boundary corresponds to the first episode of stress in the Early Toarcian
palaeoenvironmental crisis [24] and it was associated with a pulse of volcanic activity. Our results
are consistent with the hypothesis that seawater palaeotemperature is an important driver of the
belemnite body-size variation, with a decreasing body-size associated with increasing seawater
temperature (Fig. S2). Salinity could also affect the interpretation of the oxygen isotope data,
although decreased salinity has been argued to be less significant in Peniche than warming and
carbon cycle perturbations, based on phytoplankton communities[48] and modelling[49]. The
body-size dynamics observed is partially driven by carbon cycle perturbations [37] or other factors
linked with increased volcanic activity, suggested by enhanced mercury concentration [23],
emphasizing the potential synergy of the different factors affecting the ocean-atmosphere system.

Despite being one of the early pulses of the Early Toarcian crisis [20], the Pli-Toa boundary in
Peniche does not record major changes in the taxonomic composition (Fig. 4), which is consistent
with the results of the studies from other regions at higher latitudes [25, 50-52]. It is has been
shown that the major turnover is observed during the T-OAE instead [51].

Besides the reduction in adult size of *Ps. longiformis* across the Pli-Toa boundary, immediately
after it, from bed P5 to bed P6, the size of *P. bisulcata* specimens markedly decreases (109 %). This
coincides with the disappearance of *Hastitidae* and with palaeoenvironmental perturbations (Fig.
1), such as palaeotemperature, carbon cycle perturbations and volcanism [23, 37]. Therefore, we
consider the main crisis to span bed P5 to P6, which coincide with the range of the fist pulse of the
Upper Pliensbachian-Lower Toarcian crisis (Figs. 1 and 3, shaded area). *P. longiformis* seems to be
the most affected taxon, despite its quick recovery in terms of size, apparently becoming more
robust over time. This response could potentially be attributed to a life history strategy to cope
with warming events, equivalent to the one known from extant squids [6, 35]. *P. bisulcata* seems
to respond differently - mainly by the increased abundance and reduced size of juveniles in fossil
samples (Fig. S4) - potentially indicating the temporary migration of larger juveniles/adults out of

the basin, simultaneously with the recovery of adults of *Ps. longiformis*. It has been argued that *P.*
*bisulcata* was mainly affected by widespread anoxia during the T-OAE [27].

The differential response of the taxa might indicate different environmental tolerances, related to
individual physiology or life history strategies [5]. The only important morphological difference
between *Ps. longiformis* and the other taxa is the presence of an epirostrum, a calcified structure
which develops late in ontogeny [53]. The increased $p\text{CO}_2$ that characterizes the environmental
crisis of the Early Toarcian [24] could have affected its calcification potential or its food availability
or habitat in Peniche. However, further data needs to be assembled to assess the relationship
between the development of an epirostrum and environmental conditions.

Belemnites disappear from the Peniche section during the onset of the T-OAE, ca. 30 cm below the
Polymorphum-Levisoni zones boundary. This is coincident with the beginning of the second
mercury anomaly and with warming, similarly to the lowermost Toarcian conditions (Fig. 4).
However, the decrease in belemnites abundance or disappearance is not preceded by a body-size
decrease. This could reflect inhospitable conditions in the Lusitanian Basin during the onset of the
T-OAE, what did not happen during the lowermost Toarcian. This might suggest that coleoids –
even those with flexible life history strategies, pre-adapted to such conditions - could not cope
with deteriorating conditions in subtropical epicontinental European basin during the T-OAE.
Further research in other basins is necessary to disentangle the extent of this phenomenon.
Belemnites temporarily re-appear in bed P10 (upper Levisoni Zone), after the onset of the T-OAE,
but with a poor record of four large specimens of *Acrocoelites* genus, driving the increase in
median size of belemnite assemblages during the uppermost Lower Toarcian. This genus is known
to appear and radiate in various environments after the T-OAE and might have benefitted from its
consequences[27].

5 Conclusions

We document for the first time a median body-size decrease of belemnites across the Pli-Toa
boundary at different scales of palaeoecological organisation in the Peniche reference section. We
find no evidence for a major taxonomic turnover (similar to results in more high-latitude sections)
and the decrease is mainly driven by a decrease in adult size of *Ps. longiformis*. This coincides with
the onset of the first pulse of the Upper Pliensbachian – Lower Toarcian palaeoenvironmental

crisis, probably related with volcanic activity in the Karoo Ferrar igneous province. Our results
indicate that warming best explains the body-size fluctuations observed, although the interplay
with perturbations of the carbon cycle and other factors associated with increased volcanic
activity are evident. Our results suggest that morphological responses precede extinction pulses in
belemnites and highlight that decreasing adult size might rather be a life history strategy to deal
with temporary deteriorating conditions than being associated with marked taxonomic turnover
or extinctions.

**Data accessibility:** The specimens are stored in the Science Museum (*Museu da Ciência*, University
of Coimbra, Portugal) and the 3D data will be stored on Zenodo.

**Author's contributions:** PR participated in the design of the study, carried out the data collection
and analysis, interpreted the results and drafted the manuscript. KDB designed and coordinated
the study, participated in the data analysis and helped interpreting the results and drafting the
manuscript. The fieldwork was carried out by PR, KDB and LVD. PN helped with data collection and
preliminary data analysis. LVD helped interpreting the results. RW helped with the taxonomic
work. All authors contributed to the writing process and gave final approval for publication.

**Competing interests:** We declare we have no competing interests.

**Funding:** This is a contribution to the DFG Research Unit FOR 2332 (grant number Ba 5148/1-1 to
KDB) TERSANE and to the IGCP 655 (IUGS-UNESCO).

**Acknowledgements:** We thank Birgit Leipner-Mata and Manuel Blank for helping in belemnite
preparation, and Benjamin Gügel, Christian Schulbert and Martina Schlott for helping with
scanning the specimens. We thank Manuel Steinbauer, Carl Reddin and Wolfgang Kiessling for the
valuable comments on the data analysis.

References

1. Schmidt D.N., Lazarus D., Young J.R., Kucera M. 2006 Biogeography and evolution of body size in marine plankton. *Earth-Sci Rev* **78**(3–4), 239-266. (doi:<http://dx.doi.org/10.1016/j.earscirev.2006.05.004>).
2. Daufresne M., Lengfellner K., Sommer U. 2009 Global warming benefits the small in aquatic ecosystems. *Proceedings of the National Academy of Sciences* **106**(31), 12788-12793. (doi:10.1073/pnas.0902080106).
3. Gardner J.L., Peters A., Kearney M.R., Joseph L., Heinsohn R. 2011 Declining body size: a third universal response to warming? *Trends Ecol Evol* **26**(6), 285-291. (doi:<http://dx.doi.org/10.1016/j.tree.2011.03.005>).
4. Sheridan J.A., Bickford D. 2011 Shrinking body size as an ecological response to climate change. *Nature Climate Change* **1**(8), 401-406. (doi:10.1038/nclimate1259).
5. Ohlberger J. 2013 Climate warming and ectotherm body size – from individual physiology to community ecology. *Funct Ecol* **27**(4), 991-1001. (doi:10.1111/1365-2435.12098).
6. O’Gorman E.J., Zhao L., Pichler D.E., Adams G., Friberg N., Rall Björn C., Seeney A., Zhang H., Reuman D.C., Woodward G. 2017 Unexpected changes in community size structure in a natural warming experiment. *Nature Climate Change* **7**(9), 659-663. (doi:10.1038/nclimate3368).
7. Hoving H.J., Gilly W.F., Markaida U., Benoit-Bird K.J., Brown Z.W., Daniel P., Field J.C., Parassenti L., Liu B., Campos B. 2013 Extreme plasticity in life-history strategy allows a migratory predator (jumbo squid) to cope with a changing climate. *Glob Chang Biol* **19**(7), 2089-2103. (doi:10.1111/gcb.12198).
8. Moreno A., Azevedo M., Pereira J., Pierce G.J. 2007 Growth strategies in the squid *Loligo vulgaris* from Portuguese waters. *Mar Biol Res* **3**(1), 49-59. (doi:10.1080/17451000601129115).
9. Urbanek A. 1993 Biotic crises in the history of Upper Silurian graptoloids: A Palaeobiological model. *Historical Biology* **7**(1), 29-50. (doi:10.1080/10292389309380442).
10. Twitchett R.J. 2007 The Lilliput effect in the aftermath of the end-Permian extinction event. *Palaeogeography, Palaeoclimatology, Palaeoecology* **252**(1-2), 132-144. (doi:10.1016/j.palaeo.2006.11.038).
11. Harries P.J., Knorr P.O. 2009 What does the ‘Lilliput Effect’ mean? *Palaeogeogr, Palaeoclimatol, Palaeoecol* **284**(1–2), 4-10. (doi:<http://dx.doi.org/10.1016/j.palaeo.2009.08.021>).
12. Morten S.D., Twitchett R.J. 2009 Fluctuations in the body size of marine invertebrates through the Pliensbachian–Toarcian extinction event. *Palaeogeography, Palaeoclimatology, Palaeoecology* **284**(1-2), 29-38. (doi:10.1016/j.palaeo.2009.08.023).
13. Hesselbo S.P., Grocke D.R., Jenkyns H.C., Bjerrum C.J., Farrimond P., Morgans Bell H.S., Green O.R. 2000 Massive dissociation of gas hydrate during a Jurassic oceanic anoxic event. *Nature* **406**(6794), 392-395.
14. Jenkyns H.C. 1988 The early Toarcian (Jurassic) anoxic event; stratigraphic, sedimentary and geochemical evidence. *American Journal of Science* **288**(2), 101-151. (doi:10.2475/ajs.288.2.101).
15. Wignall P.B., Newton R.J., Little C.T.S. 2005 The timing of paleoenvironmental change and cause-and-effect relationships during the early Jurassic mass extinction in Europe. *Am J Sci* **305**(10), 1014-1032. (doi:10.2475/ajs.305.10.1014).
16. Hesselbo S.P., Jenkyns H.C., Duarte L.V., Oliveira L.C.V. 2007 Carbon-isotope record of the Early Jurassic (Toarcian) Oceanic Anoxic Event from fossil wood and marine carbonate (Lusitanian

Basin, Portugal). *Earth and Planetary Science Letters* **253**(3-4), 455-470.
(doi:10.1016/j.epsl.2006.11.009).
- 17. Caruthers A.H., Smith P.L., Gröcke D.R. 2013 The Pliensbachian–Toarcian (Early Jurassic)
extinction, a global multi-phased event. *Palaeogeography, Palaeoclimatology, Palaeoecology* **386**,
104-118. (doi:10.1016/j.palaeo.2013.05.010).
- 18. Danise S., Twitchett R.J., Little C.T.S., Clémence M.-E. 2013 The Impact of Global Warming
and Anoxia on Marine Benthic Community Dynamics: an Example from the Toarcian (Early
Jurassic). *PLoS ONE* **8**(2), e56255. (doi:10.1371/journal.pone.0056255).
- 19. Dera G., Neige P., Dommergues J.-L., Fara E., Laffont R., Pellenard P. 2010 High-resolution
dynamics of Early Jurassic marine extinctions: the case of Pliensbachian–Toarcian ammonites
(Cephalopoda). *Journal of the Geological Society* **167**(1), 21-33. (doi:10.1144/0016-76492009-068).
- 20. Dera G., Toumoulin A., De Baets K. 2016 Diversity and morphological evolution of Jurassic
belemnites from South Germany. *Palaeogeography, Palaeoclimatology, Palaeoecology* **457**, 80-97.
(doi:10.1016/j.palaeo.2016.05.029).
- 21. García Joral F., Gómez J.J., Goy A. 2011 Mass extinction and recovery of the Early Toarcian
(Early Jurassic) brachiopods linked to climate change in Northern and Central Spain.
*Palaeogeography, Palaeoclimatology, Palaeoecology* **302**(3), 367-380.
(doi:<https://doi.org/10.1016/j.palaeo.2011.01.023>).
- 22. József Pálffy P.L.S. 2000 Synchrony between Early Jurassic extinction, oceanic anoxic event,
and the Karoo-Ferrar flood basalt volcanism. *Geology* **28**(8), 747-750. (doi:10.1130/0091-
7613(2000)28<747:SBEJEO>2.0.CO;2).
- 23. Percival L.M.E., Witt M.L.I., Mather T.A., Hermoso M., Jenkyns H.C., Hesselbo S.P., Al-
Suwaidi A.H., Storm M.S., Xu W., Ruhl M. 2015 Globally enhanced mercury deposition during the
end-Pliensbachian extinction and Toarcian OAE: A link to the Karoo–Ferrar Large Igneous Province.
*Earth and Planetary Science Letters* **428**, 267-280. (doi:10.1016/j.epsl.2015.06.064).
- 24. Suan G., Mattioli E., Pittet B., Mailliot S., Lécuyer C. 2008 Evidence for major
environmental perturbation prior to and during the Toarcian (Early Jurassic) oceanic anoxic event
from the Lusitanian Basin, Portugal. *Paleoceanography* **23**(1), PA1202.
(doi:10.1029/2007PA001459).
- 25. Doyle P. 1990 *The British Toarcian (lower Jurassic) Belemnites: Part 1*, Palaeontographical
Society.
- 26. Harazim D., Van De Schootbrugge B.A.S., Sorichter K., Fiebig J., Weug A., Suan G.,
Oschmann W. 2013 Spatial variability of watermass conditions within the European Epicontinental
Seaway during the Early Jurassic (Pliensbachian–Toarcian). *Sedimentology* **60**(2), 359-390.
(doi:10.1111/j.1365-3091.2012.01344.x).
- 27. Ullmann C.V., Thibault N., Ruhl M., Hesselbo S.P., Korte C. 2014 Effect of a Jurassic oceanic
anoxic event on belemnite ecology and evolution. *Proceedings of the National Academy of*
*Sciences* **111**(28), 10073-10076. (doi:10.1073/pnas.1320156111).
- 28. McArthur J.M., Doyle P., Leng M.J., Reeves K., Williams C.T., Garcia-Sanchez R., Howarth
R.J. 2007 Testing palaeo-environmental proxies in Jurassic belemnites: Mg/Ca, Sr/Ca, Na/Ca, $\delta^{18}\text{O}$
and $\delta^{13}\text{C}$. *Palaeogeogr, Palaeoclimatol, Palaeoecol* **252**(3–4), 464-480.
(doi:<http://dx.doi.org/10.1016/j.palaeo.2007.05.006>).
- 29. van de Schootbrugge B., McArthur J.M., Bailey T.R., Rosenthal Y., Wright J.D., Miller K.G.
2005 Toarcian oceanic anoxic event: An assessment of global causes using belemnite C isotope
records. *Paleoceanography* **20**(3), PA3008. (doi:10.1029/2004PA001102).
- 30. Rita P., De Baets K., Schlott M. 2018 Rostrum size differences between Toarcian belemnite
battlefields. *Foss Rec* **21**(1), 171-182. (doi:10.5194/fr-21-171-2018).

31. Suan G., Mattioli E., Pittet B., Lécuyer C., Suchéras-Marx B., Duarte L.V., Philippe M.,
Reggiani L., Martineau F. 2010 Secular environmental precursors to Early Toarcian (Jurassic)
extreme climate changes. *Earth Planet Sci Lett* **290**(3–4), 448-458.
(doi:<http://dx.doi.org/10.1016/j.epsl.2009.12.047>).
32. Duarte L. 2007 Lithostratigraphy, sequence stratigraphy and depositional setting of the
Pliensbachian and Toarcian series in the Lusitanian Basin, Portugal. *Ciências da Terra/Earth
Sciences Journal* **16**.
- 33. Rocha R.B.d., Mattioli E., Duarte L.V., Pittet B., Elmi S., Mouterde R., Cabral M.C., Comas-
Rengifo M.J., Gomez J.J., Goy A., et al. 2016 Base of the Toarcian Stage of the Lower Jurassic
defined by the Global Boundary Stratotype Section and Point (GSSP) at the Peniche section
(Portugal). *Episodes* **39**(3). (doi:10.18814/epiugs/2016/v39i3/99741).
- 34. Rita P., De Baets K., Schlott M. 2018 Rostrum size differences between Toarcian belemnite
battlefields. *Fossil Record* **21**(1), 171-182. (doi:10.5194/fr-21-171-2018).
- 35. Rosa R., Gonzalez L., Dierssen H.M., Seibel B.A. 2012 Environmental determinants of
latitudinal size-trends in cephalopods. *Mar Ecol Prog Ser* **464**, 153-165. (doi:10.3354/meps09822).
- 36. Tomašových A., Schlögl J., Biroň A., Hudáčková N., Mikuš T. 2017 Taphonomic Clock and
Bathymetric Dependence of Cephalopod Preservation in bathyal, sediment-starved environments.
*Palaeos* **32**(3), 135-152. (doi:10.2110/palo.2016.039).
- 37. Suan G., Mattioli E., Pittet B., Mailliot S., Lécuyer C. 2008 Evidence for major
environmental perturbation prior to and during the Toarcian (Early Jurassic) oceanic anoxic event
from the Lusitanian Basin, Portugal. *Paleoceanography* **23**(1), n/a-n/a.
(doi:10.1029/2007pa001459).
- 38. Pecl G., Jackson G. 2008 The potential impacts of climate change on inshore squid: biology,
ecology and fisheries. *Rev Fish Biol Fish* **18**(4), 373-385. (doi:10.1007/s11160-007-9077-3).
- 39. Rego B.L., Wang S.C., Altiner D., Payne J.L. 2012 Within- and among-genus components of
size evolution during mass extinction, recovery, and background intervals: a case study of Late
Permian through Late Triassic foraminifera. *Paleobiology* **38**(4), 627-643. (doi:10.1666/11040.1).
- 40. Oksanen J., Blanchet F.G., Kindt R., Legendre P., Minchin P., O'Hara R.B., Simpson G.,
Solymos P., Stevens M.H.H., Wagner H. 2018 Vegan: community ecology package. R package
version 2.5-2 (<https://CRAN.R-project.org/package=vegan>). (
- 41. Box G.E.P., Jenkins G.M. 1994 *Time Series Analysis: Forecasting and Control*, Holden-Day.
42. R Development Core Team. 2018 R: a language and environment
for statistical computing. Vienna, Austria: R Foundation for Statistical Computing. Retrieved from
<http://www.R-project.org>. (
- 43. Pinheiro J B.D., DebRoy S, Sarkar D, R Core Team. 2018 nlme: Linear and Nonlinear Mixed
Effects Models. R package version 3.1-137 (<https://CRAN.R-project.org/package=nlme>). (
- 44. Spiess A.-N. 2018 qpcR: Modelling and Analysis of Real-Time PCR Data. R package version
1.4-1 (<https://cran.r-project.org/web/packages/qpcR/>).
- 45. Holland S.M. 2000 The quality of the fossil record: a sequence stratigraphic perspective.
*Paleobiology*, 148-168.
- 46. Rita P., Reolid M., Duarte L.V. 2016 Benthic foraminiferal assemblages record major
environmental perturbations during the Late Pliensbachian–Early Toarcian interval in the Peniche
GSSP, Portugal. *Palaeoecography, Palaeoclimatology, Palaeoecology* **454**, 267-281.
- 47. Doyle P., Macdonald D.I. 1993 Belemnite battlefields. *Lethaia* **26**(1), 65-80.
- 48. Correia V.F., Riding J.B., Fernandes P., Duarte L.V., Pereira Z. 2017 The palynology of the
lower and middle Toarcian (Lower Jurassic) in the northern Lusitanian Basin, western Portugal. *Rev
Palaeobot Palynol* **237**, 75-95. (doi:<https://doi.org/10.1016/j.revpalbo.2016.11.008>).

49. Dera G., Donnadiou Y. 2012 Modeling evidences for global warming, Arctic seawater
freshening, and sluggish oceanic circulation during the Early Toarcian anoxic event.
*Paleoceanography* **27**(2), PA2211. (doi:10.1029/2012PA002283).
50. Weis R., Neige P., Dugué O., Cencio A.D., Thuy B., Numberger-Thuy L., Mariotti N. 2018
Lower Jurassic (Pliensbachian-Toarcian) belemnites from Fresney-le-Puceux (Calvados, France):
taxonomy, chronostratigraphy and diversity. *Geodiversitas* **40**(4), 87-113.
51. Caswell B.A., Coe A.L. 2014 The impact of anoxia on pelagic macrofauna during the
Toarcian Oceanic Anoxic Event (Early Jurassic). *Proc Geol Assoc* **125**(4), 383-391.
(doi:<https://doi.org/10.1016/j.pgeola.2014.06.001>).
52. Pinard J.-D., Weis R., Neige P., Mariotti N., Di Cencio A. 2014 Belemnites from the Upper
Pliensbachian and the Toarcian (Lower Jurassic) of Tournadous (Causse, France). *Neues Jahrbuch*
*für Geologie und Paläontologie-Abhandlungen* **273**(2), 155-177.
53. Arkhipkin A., Weis R., Mariotti N., Shcherbich Z. 2015 'Tailed' cephalopods. *J Molluscan*
*Stud* **81**, 345-355. (doi:10.1093/mollus/eyu094).

*Figure 1 – Variation of belemnite body-size (geometric mean), abundance and species ranges through time. Variation of*
 *Hg/TOC [23], carbon and oxygen isotopes [16, 37]. The shaded area highlights the first pulse of the upper Pliensbachian -*
 *lower Toarcian crisis. The bed numbers in square brackets correspond to the beds that were merged due to sample size*
 *constraints.*

Figure 2- Conceptual scheme depicting the tested hypotheses regarding the mechanisms behind belemnites body-size reduction, at different palaeobiological scales of organisation.

Figure 3 – A: Venn diagram depicting the partition of belemnites body-size variation between taxonomic composition,
 bed separation and ontogeny. The values in brackets correspond to the whole time-series and the values without
 brackets correspond to the Pli-Toa boundary. The shaded area highlights the first pulse of the upper Pliensbachian - lower
 Toarcian crisis. B: Effect on size change at the assemblage and population scales of organisation. See TS2 and TS5 for
 details. For a correct interpretation of the right-side scale, the bottom of the arrowhead should be considered, rather
 than the tip of the arrow. Note that the x-axis depicts comparisons of consecutive pairs of beds (not to scale). BG =
 belemnite gap.

*Figure 4 – Belemnite body-size variation, lithology and proportional body-size change across the studied interval at the*
 *assemblage and population scales (P. bisulcata and Ps. longiformis). Note that the stratigraphic log is not drawn to scale,*
 *for real thickness of beds, see Fig. 1. For sample size, see Fig. S6 and TS1. BG = belemnite gap.*

Supplementary material

*Table S1 – Measured body size parameters on belemnites specimens from the interval ranging from Emaciatum to*
 *Levisoni Zone of the Peniche section (see ESM file 1).*

*Table S2 – Log return, sample size and results of Statistical tests on the differences in the median of pairs of beds (ESM*
 *file 8).*

			From interval	To interval	Phase1	Phase2	s1	s2
assemblage	median GM	P925 (Emaciatum)	P949 (Emaciatum)	background	background	10.80049	9.42723	
assemblage	median GM	P949 (Emaciatum)	P961 (Emaciatum)	background	background	9.427231	8.86356	
assemblage	median GM	P961 (Emaciatum)	P982 (Emaciatum)	background	background	8.863563	9.44640	
assemblage	median GM	P982 (Emaciatum)	P984 (Polymorphum)	background	crisis	9.446406	8.26931	
assemblage	median GM	P984 (Polymorphum)	P8 (Polymorphum)	crisis	survival	8.269314	8.75732	
assemblage	median GM	P8 (Polymorphum)	P12 (Polymorphum)	survival	recovery	8.757321	8.93768	
assemblage	median GM	P12 (Polymorphum)	P14 (Polymorphum)	recovery	recovery	8.937686	9.4400	
assemblage	median GM	P14 (Polymorphum)	P20 (Polymorphum)	recovery	recovery	9.44006	11.1200	
assemblage	median GM	P20 (Polymorphum)	P133 (Levisoni)	recovery	recovery	11.12009	20.0590	
Passaloteuthis	apical length median	P925 (Emaciatum)	P949 (Emaciatum)	background	background	29.97	51.24	
Passaloteuthis	apical length median	P949 (Emaciatum)	P961 (Emaciatum)	background	background	51.245	24.4	
Passaloteuthis	apical length median	P961 (Emaciatum)	P982 (Emaciatum)	background	background	24.42	43.6	
Passaloteuthis	apical length median	P982 (Emaciatum)	P984 (Polymorphum)	background	crisis	43.67	39.5	
Passaloteuthis	apical length median	P984 (Polymorphum)	P8 (Polymorphum)	crisis	survival	39.56	14.	
Passaloteuthis	apical length median	P8 (Polymorphum)	P12 (Polymorphum)	survival	recovery	14.5	17.8	
Passaloteuthis	apical length median	P12 (Polymorphum)	P14 (Polymorphum)	recovery	recovery	17.87	30.9	
Passaloteuthis	apical length median	P14 (Polymorphum)	P20 (Polymorphum)	recovery	recovery	30.99	31.	
Pseudohastites	apical length median	P961 (Emaciatum)	P982 (Emaciatum)	background	background	23.555	26.6	
Pseudohastites	apical length median	P982 (Emaciatum)	P984 (Polymorphum)	background	crisis	26.66	22.6	
Pseudohastites	apical length median	P984 (Polymorphum)	P8 (Polymorphum)	crisis	survival	22.68	24.2	
Pseudohastites	apical length median	P8 (Polymorphum)	P12 (Polymorphum)	survival	recovery	24.23	21.1	
Pseudohastites	apical length median	P12 (Polymorphum)	P14 (Polymorphum)	recovery	recovery	21.19	24.08	
Pseudohastites	apical length median	P14 (Polymorphum)	P20 (Polymorphum)	recovery	recovery	24.085	30.5	

*Table S3 – Partitioning analysis results highlighting the relative variation fractions (proportions) of belemnites body size*
 *explained by taxonomic composition, beds and ontogeny for the whole time-series and for the pair-to-pair analysis. Bold*
 *indicates a minimum of 90 % statistical significance level (ESM file 6).*

Beds pairs	Residuals	fraction a	fraction b	fraction c	fractions
		Taxonomic composition	Beds	Ontogeny	d+e+f+g
					Interaction
P1-P2	0.21	0.62	0.01	0.19	0.04
P2-P3	0.25	0.54	0.01	0.17	0.02
P3-P4	0.25	0.46	0	0.17	0.12
PT boundary (P4-P5)	0.2	0.49	0.01	0.11	0.21
	-	p value = 0.001	p value = 0.023	p value = 0.001	-
P5-P6	0.29	0.4	0.01	0.2	0.12
P6-P7	0.49	0.2	0.02	0.33	0
P7-P8	0.39	0.48	0	0.21	0
P8-P9	0.35	0.45	0.02	0.2	0.04
P9-P10	0.18	0.27	0	0.09	0.45
Whole time-series	0.26	0.47	0.02	0.17	0.07
	-	p value = 0.001	p value = 0.001	p value = 0.001	-

*Table S4 – Details of the selected GLS models comparing belemnites body-size and the environmental parameters (ESM*
 *file 4).*

Model	Coefficients						
		Value	Std.Error	t-value	p-value	degrees of freedom	residual
#23 Assemblage	Intercept	14.518	1.477	9.829	0.000	340	338
	$\delta^{18}\text{O}$	3.152	1.180	2.671	0.008	-	
#15 Ps. longiformis	Intercept	12.373	1.553	7.950	0.000	161	156
	$\delta^{13}\text{C}$	1.069	0.609	1.765	0.080	-	
	$\delta^{18}\text{O}$	3.872	0.918	4.203	0.000	-	
	lithologymarl	-2.273	1.299	-1.995	0.048	-	
	lithologymarly limestone	-0.268	1.116	-0.253	0.801	-	
#5 P. bisulcata	Intercept	17.515	3.265	3.364	0.000	76	73
	$\delta^{18}\text{O}$	4.228	2.281	1.853	0.068	-	
	abundance	0.070	0.029	2.417	0.018	-	

*Table S5 – Detailed values of the components of the assemblage size shift calculated with Rego et al. method*
 *(disappearance of taxa effect, appearance of new taxa effect and within-lineage effect) (see ESM file 2).*

*Table S6 – Micro-CT phoenix v|tome|x s 240 (Research Edition) scanner settings for the scanned belemnite specimens.*
 *The reconstruction was made with the GEDatos|x 2.4 software. Subsequent image stack processing (e.g. subsampling),*

as well as the measurements and volume acquisition, was derived using Studio Volume Graphics Max™ v 3.0 software
(Heidelberg) (see ESM file 3).

Note: The following figures will be displayed in “Supplementary figures” in the ESM file 7.

*Figure S1 – Cumulative frequency of the different taxa comprising the Peniche belemnite assemblage.*

Figure S2 – The effect of seawater palaeotemperature on belemnites body-size.

Figure S3 – Number of belemnites collected per square meter (known area collection). The black line connects the median.

Figure S4 – Body-size variation of *P. bisulcata* and *Ps. longiformis* through the studied interval with emphasis on the ontogeny.

*Figure S5 – Body-size variation of cross-borders (*P. bisulcata*, *Ps. longiformis* and *Hastitidae sp. indet.*). Although*
 *Parapassaloteuthis sp. A also crosses the boundary, we have no complete specimens across this interval and therefore,*
 *not a proper body size.*

*Figure S6 - Belemnite body-size variation, abundance of belemnites per m² and proportional body size change across the*
*studied interval at the assemblage and population scales (*P. bisulcata* and *Ps. longiformis*). Note that the stratigraphic*
*log is not drawn to scale, for real thickness of beds, see Fig. 1. BG = belemnite gap.*

Appendix B

Associate Editor Comments to Author (Professor Stephen Hesselbo)

The manuscript has been thoroughly revised from the original version, and now thoroughly re-reviewed. The reviewers have made a number of very helpful comments which, if properly implemented, will improve the manuscript greatly. In particular both reviewers suggest numerous ways in which the paper may be made more readable - in particular there should be a major effort to remove/reduce acronyms, initialisms, and short-hand references to hypotheses (H1, H2, etc.) that all require the reader to repeatedly refer back to other parts of the manuscript or (more likely) give up. As always, the responses to the comments of the reviewers should be clearly documented and if not acted upon specific justification is required.

Author's answer: An effort was made in order to reduce acronyms and initialisms. The ones which are necessary are always explained in the beginning of each section and also in the figure captions. Some of the acronyms are very commonly used in this kind of study like ANOVA or AICc scores and therefore they were kept.

Reviewers comments to Author

Reviewer #1

Reviewer's comment: *I feel however, that the resulting text is still not very accessible to readers because it contains a lot of statistical nomenclature, terms, and abbreviations which are not straight forward to relate to the actual fossils and make it hard to follow the narrative. Often times I feel that by re-formulating the text using simpler language the readability would improve substantially. E.g, I would find the first discussion paragraph (P7) much easier to read if it was written more like "Statistical evaluation of our data has shown that size changes of belemnite rostra in the studied section at Peniche are significant for the Pliensbachian-Toarcian boundary event, i.e., H1 was validated. This size change is almost exclusively driven by a body size reduction of single species..." By spelling out more directly what the different effects are and the abbreviated hypotheses mean, the reader will find it easier to follow the red thread of the text.*

Author's answer: An effort was made in order to make the text easier to read by removing some acronyms and abbreviations. We also changed many sentences and paragraphs especially in the discussion and results sections to make the text easier to follow. The organization of the sentences in paragraphs was carefully reviewed.

Reviewer's comment: *No clear, singular driver of the observed size change in belemnites could be identified from modelling of the morphological and geochemical data. The authors interpret this to signify that the interplay between environmental stressors associated with hyperthermals and body size of animals is complex. One could, however, also pose that factors other than those tested by the authors may have caused these changes in size or that all of the tested parameters were somehow involved. This point is amplified by the lack of quantification. How well do the produced models actually fit the observed data? Even though some of the model options are statistically significant, they might not explain much of the observed data variability.*

Author's answer: We do acknowledge that other factors might be involved in explaining the body-size variation and we now clearly say it in the discussion section. We also express ourselves in a careful way in order to avoid draw conclusions which are not possible to draw from our data. Regarding the quantification, as mentioned in the previous revision of this manuscript, R squares are not calculated for GLS regression analyses.

Reviewer's comment: *The T-OAE and to some degree also the Pli-Toa boundary are intervals in time when numerous environmental parameters changed more or less in unison, which will make finding causalities very tricky. E.g., even though temperature might be a relatively good predictor of some of the size change data, it does not mean that temperature actually caused the size reduction but might have caused another effect which in turn led to the diminished size of the belemnites. This is to some degree acknowledged by the authors and reference is also made to modern squid ecology, but nevertheless the strong reliance on statistical evaluation in the study makes the text more prone to attack by arguments of coincidence vs causality in my opinion. These points are not made here to criticize the approach taken by the authors but to spell out that putting slightly more emphasis on geological context in the discussion and acknowledging limitations of the modelling approach more openly in the text would in my opinion make this study even more valuable for the geoscience community.*

Author's answer: The discussion section was changed in order to accommodate sentences that acknowledge the limitations of our models and we also changed some other parts in order to be more conservative. We think that by having a section fully dedicated to sedimentary facies in the discussion section we are giving a good emphasis on the geological context. In that way we did not make this section shorter as suggested by the reviewer #2.

Reviewer's comment: ~P1L32: "fossil cephalopod studies"

Author's answer: Accepted and changed.

Reviewer's comment: ~P1L40: "two dominant species"

Author's answer: Accepted and changed.

Reviewer's comment: P1 bottom: No present address is given here.

Author's answer: Address added.

Reviewer's comment: ~P2L18: *I think this sentence would require to be elaborated on a little further. For example, the extinction event at the Cretaceous-Paleogene boundary seems to have hit particularly the large mammals.*

Author's answer: Accepted and changed.

Reviewer's comment: ~P2L29: "north-west Tethyan basins"?

Author's answer: Accepted and changed.

Reviewer's comment: ~P3L38: "approach was adopted"?

Author's answer: Accepted and changed.

Reviewer's comment: ~P3L42: "belemnite rostra"

Author's answer: Accepted and changed.

Reviewer's comment: ~P3L52: *I do not think any study has shown convincingly that the belemnite rostrum was originally highly porous. Many studies have provided evidence that some of the rostrum at any given time in the belemnite's life might have been somewhat porous, but all data I am aware of are consistent with this porosity being of subordinate importance for the bulk rostrum at best.*

Author's answer: Accepted. The word "partially" was added.

Reviewer's comment: ~P3L52: "soft tissue preservation"?

Author's answer: Accepted and changed.

Reviewer's comment: ~P3L55: *I agree that the rostrum is probably the best proxy one can use for belemnite size, but I think one should add a caveat here that it may not be ideal. Not all belemnites would have had the same body proportions, and some species with rather short rostra might nevertheless have been relatively big in size. E.g., in the family of the Giraffidae, body proportions differ greatly between the giraffe and the okapi. Other examples can be found across the animal kingdom.*

Author's answer: Accepted. The paragraph includes now a sentence acknowledging this and a suitable reference.

Reviewer's comment: ~P4L15: *Since the authors are here testing a hierarchical set of hypotheses as also shown by the tree diagram in figure 2 it might be helpful to the reader to label the hypotheses accordingly. H2-H5 are all sub-hypothesis of H1 and H4 and H5 appear to be refined versions of H3. One may thus label H2 as H1,1 and H3 as H1,2, whereas H4 would be H1,2,1 and H5 would be H1,2,2.*

Author's answer: We think that this change would make the manuscript harder to read. The figure 2 is very clear and I think that the set of hypotheses tested is very clearly explained in the materials and methods section. However, the figure is changed to emphasize the relation between the hypotheses.

Reviewer's comment: ~P4L21: "4" should be in subscript.

Author's answer: Accepted and corrected.

Reviewer's comment: ~P4L37: *I think for the equations the prefixes "pre-" and "post-" should be deleted. It is already clear from the subscripts "bed1" and "bed2" that reference is made to*

different horizons and I think “pre-boundary crossers” and “post-boundary crossers” are confusing terms.

Author’s answer: Accepted and changed.

Reviewer’s comment: *~P5L31: (see also P6L8,11) I am not sure how the authors compute rostrum volume from their morphological measurements. If the body size shrinks by 13 %, also the rostrum volume should shrink by 13 %, unless body size is meant to read as body length or other one dimensional parameter. If the latter is the case and the rostrum shape does not change, I would compute the volume as 0.87^3 , i.e., 0.66, equivalent to a volume reduction of 34 %, not 40.*

Author’s answer: The geometric mean corresponds to the cubic root of the product of Dv, DI and I and therefore is unidimensional parameter. The formula and the explanation of how it relates to volume is now included in the materials and methods section. The change in volume is now added to the supplementary material (Table S2).

Reviewer’s comment: *~P6L13: How can something reduce its body size by 109%? It would then have a negative size.*

Author’s answer: It is a proportional change. It only means that the size has been reduced below the initial level. Please check the table S2 where the values are calculated with the respective formulas used.

Reviewer’s comment: *~P6L33: A change from 75 % to 69 % to me is a reduction, not an increase.*

Author’s answer: Accepted and corrected.

Reviewer’s comment: *~P6L36: Could some information be added here, how much of the observed variance in data can actually be modelled using the significant models? If only a small percentage of the size variance can be explained by the models, they might not be very useful despite being statistically significant.*

Author’s answer: As mentioned before in this letter of response and also in the previous revision of the manuscript, this method does not allow the computation of the r squares and therefore no quantification is possible in that sense. We acknowledge the limitations of our models and therefore we express ourselves in a very cautious way.

Reviewer’s comment: *~P7L37: “constraints”.*

Author’s answer: Accepted and changed.

Reviewer’s comment: *~P7L43: delete “has had”.*

Author’s answer: Accepted and changed.

Reviewer's comment: ~P7L46: *"an interval barren in belemnites in the Lusitanian Basin and many other sections in Iberia"*.

Author's answer: Here we are focusing on our results. The way this gap in Peniche compares with other coeval basins is explored more in detail in the discussion section.

Reviewer's comment: ~P8L35: *"Cardigan Bay Basin sediments"*.

Author's answer: Accepted and changed.

Reviewer's comment: ~P8L38: *How was it constrained when conditions became most severe?*

Author's answer: We mean when the temperature of the water was the highest. See Fig. 1. This is now clearer in the text.

Reviewer's comment: ~P8L41: *"habitats were no longer suitable"*.

Author's answer: Accepted and corrected.

Reviewer's comment: ~P9L8: *I find this argument of "rapidity" debatable. As one is dealing with Mesozoic sediments temporal resolution is necessarily limited, but it is clear that the durations of both the T-OAE and the Pli-Toa boundary are not exactly short. E.g., the T-OAE likely lasts about as long as half of the Pleistocene.*

Author's answer: What is geologically rapid can be differently defined. We feel that we already express ourselves in a cautious way and cite appropriate refs. The sentence was changed.

Reviewer's comment: ~P9L57: *"modelling suggests"*.

Author's answer: Accepted and changed.

Reviewer #2

Reviewer's comment: *I also find myself asking a few questions surrounding the broader context of the work – how do you think these changes might have affected the wider food web? Would the wider anoxia would have affected belemnites in the non-anoxic Lusitanian basin e.g. by restricting migratory routes, food supply etc? Presumably it could have inhibited their ability to move to cooler refuges when temperatures began to rise. Maybe the authors have some ideas that might help us to view the changes more broadly?*

Author's answer: Accepted. We now include some considerations about migrations and cited appropriate references in the discussion section.

Reviewer's comment: I think the writing needs some polishing, it doesn't always flow and is repetitive in places. There are also some vague descriptions in the results. I think by reviewing and revising the text to tighten things up and integrate information better it could be improved.

Author's answer: The results are less vague now and an effort was made to make the text more fluid, by reducing abbreviations and acronyms and to use a more incisive language style. This was especially done in the results and in the discussion.

Methods and Results

Reviewer's comment: *Results are concise which is good, but they don't flow. There are many short paragraphs and obvious links aren't always made between related points. Some information is missing or ambiguous and needs to be clearer/stated more precisely. I encourage the authors to think about structure and providing more and clearer descriptions (plus see below).*

Author's answer: An effort was made to better organize the results by incorporating the reviewers suggestions and by changing some others parts of the text. The results were written in a more direct and concise way. In terms of structure, we organized the sentences in paragraphs in a different way in the results and in the discussion sections.

Reviewer's comment: *Found the statistical reporting to be ambiguous. The format for writing statistics is quite standardised because we need to be very precise about what exactly a statistical test does and does not show. In this ms there is regular mention of 'significant results' and looking for a 'significant difference' without specifying what is being compared, what differed and in what direction – this applies in both the results and methods. I would like the authors to be more explicit so that the results cannot be misinterpreted. In the results you should report what test you used, what groups were compared (often the case in this ms as they are hypothesis tests) and the direction or scale of difference. The test statistics and p values must also be reported somewhere whether its in parentheses in the text or in a table.*

Author's answer: The test used and the significance considered are described in the methods, as well as how were the comparisons made. The p values are all given in the supplementary material. A table might be added if necessary but since there are already so many tables in the main text we decided to keep in the supplementary material, for now. I now include in the text the p values for the most important results described. If all of them are included the text, the ideas will flow less. An effort was made to make everything more clear.

Reviewer's comment: *Abbreviations (e.g. GM, AIC etc) are used regularly but seem only to be defined once – you should at least explain them once in each section of the paper to aid your reader and they should be explained in figure captions.*

Author's answer: The abbreviations and acronyms are nor defined at the beginning of each chapter and also in the figure captions. However, I would like to point out that it is common practice to refer to ANOVA and AICc without explaining all the meaning, as those are often used terms.

Reviewer's comment: *Figures and tables are good – but I found them a little hard to follow/not always intuitive. Can you improve the labelling and captions to make it easier for the reader to follow. I have made comments on the figures and captions in the attached pdf – explaining what needs clarification/labelling.*

Author's answer: We changed the figures according to the comments and we also made some other adjustments in order to make the figure easier to follow, especially regarding the labels.

Discussion:

Reviewer's comment: *Overall, could be stronger, more integrated and with less repetition.*

Author's answer: The discussion includes now some new remarks and an effort was made in order to make it less repetitive. An effort was also made to make a more broader context hopefully contributing to a stronger discussion.

Reviewer's comment: *In my opinion the first paragraph doesn't set the context sufficiently: Why does the work matter, what's the big picture? and what did you attempt to do? Remind the reader.*

Author's answer: Accepted. The first paragraph included a broader context and reminds the reader of the goals of the study.

Reviewer's comment: *I think the discussion needs to be more integrated – results are presented in one para and then context from the literature in another. Some paragraphs are very short (2-3 lines only). It would be best to combine these and remove some of the repetition.*

Author's answer: Accepted and changed. The paragraphs are better organized now.

Reviewer's comment: *I found reference to the ammonite zones a bit confusing (not being very familiar with the biostrat in this region) – it would be helpful to occasionally provide context relative to the event.*

Author's answer: I think that figure 1 gives a good context to the reader in terms of ammonite zonation context. I do think that if instead of using ammonite zones, as often made in palaeontology publications, we refer to the T-OAE and the Pli-Toa boundary it will even make it hard to understand. We added the full names of the ammonites used in the chronozone in the Fig. 1 caption.

Reviewer's comment: *Line 31 p1: add some comment to be clear that includes this event.*

Author's answer: Accepted and changed.

Reviewer's comment: *Line 32 p1: you could broaden your context a little to explain something*

about the Toarcian event - maybe use the term extinction - deoxygenation or climate change - this would make it accessible to a broader audience (including non-geologists).

Author's answer: We already talk about warming but the extinction is now referred.

Reviewer's comment: *Line 42 p8: see Caswell and Coe 2014 re egg laying.*

Author's answer: Caswell and Coe 2014 is already cited.

Reviewer's comment: *Line 51 p9: and widespread deoxygenation and organic matter burial*

Author's answer: Accepted and added.

Reviewer's comment: *Line 18 p10: Caswell and Coe 2013 also conclude this but from secondary producers*

Author's answer: ref added.

Reviewer's comment: *Line 22 p10: yes but tricky because and event of this magnitude/severity has never been tested/observed for modern squid.....*

Author's answer: It is true but it does not mean that we cannot try to address the problem.

Reviewer's comment: *Line 59 p9: interesting - what is thought to be the reason for this?*

Author's answer: Sentence added with further explanation.

Reviewer's comment: *Line 28 p11: but it could effect their prey*

Author's answer: Sentence changed.

Reviewer's comment: *Line 45 p11: not sure what you're considering direct vs indirect?*

Author's answer: **accepted.** This is now explained in detail.

Reviewer's comment: *Line 41 p1: do you think its a direct link to SST?*

Author's answer: The paleotemperature proxy we use is the stable oxygen isotopes from brachiopod shells, as done by Suan et al., 2008. They interpret it as bottom water temperature, as they are benthic organisms. This is now clear in the manuscript, but we do not find it relevant for the abstract.

Reviewer's comment: *Line 43 p1: I'm sure space is limited - but this is a little vague. I'm sure its complex but can you outline how you think these changes might relate to each other a little more clearly*

Author's answer: Accepted. Sentence changed with more concrete terms.

Reviewer's comment: *Line 50 p1: also biotic interactions such as competition cannot be excluded- one taxon may be more successful under climate change - and species may be less competitive in the face of e.g. novel invaders*

Author's answer: Accepted and changed.

Reviewer's comment: *Line 51 p1: n the modern world this is likely to be even more complex - human activities such as pollution, habitat destruction and fishing will provide complex synergistic interactions as well there's refs to work on this in one of these Poloczanska refs, plus the 2016 one is a good review of what we do and dont know about the biotic impacts of climate change:*

Poloczanska, E.S., Brown, C.J., Sydeman, W.J., Kiessling, W., Schoeman, D.S., Moore, P.J., Brander, K., Bruno, J.F., Buckley, L.B., Burrows, M.T., 2013. Global imprint of climate change on marine life. Nature Climate Change: 3, 919–925.

Poloczanska, E.S., Burrows, M.T., Brown, C.J., García Molinos, J., Halpern, B.S., Hoegh-Guldberg, O., Kappel, C.V., Moore, P.J., Richardson, A.J., Schoeman, D.S., 2016. Responses of marine organisms to climate change across oceans. Frontiers in Marine Science: 3, 1–21.

Author's answer: Accepted, sentence changed and refs added.

Reviewer's comment: *Line 58, p1: interesting - maybe worth using the term 'stenothermal' here which is the proper term for such sensitivity.*

Author's answer: Added.

Reviewer's comment: *Line 18 p2: an example would be nice or if you prefer to do it later - maybe a broader range of refs here if possible?*

Author's answer: Accepted and example is now given.

Reviewer's comment: *Line 30 p2: are ammonite species names given in full somewhere?*

Author's answer: Now they are provided in the figure 1 caption.

Reviewer's comment: *Line 33 p2: I know there's a range of estimates but some quantification for SST would help the reader.*

Author's answer: The sentence is not changed. We use the estimation of Suan et al., 2008 for the bottom water temperature. The reader can also see that in Fig. 1., which is now cited.

Reviewer's comment: *Line 34 p2: and body-size changes in benthic fauna - Caswell and Coe 2013 measured thousands of bivalves in Yorkshire and found body-size shifts of comparable scale to those you report (~50%)(Caswell and Coe 2013. Geology 41, 1163–1166). And i think there might be some work on other taxa groups by Correia et al. 2017. Micropaleontology 137, 46-63*

Author's answer: Accepted. Refs added.

Reviewer's comment: *Line 52 p2: yes, but if you consider the IPCC predictions for instance - its regionally very variable -it strengthens in places and weakens in others. Also the effects vary at smaller scales - we know this from modern systems (e.g. see Caswell et al. 2018 and refs therein - this is often referred to as 'enrichment'). Also time is important because many eutrophic systems experience a boom initially followed by subsequent deoxygenation beyond the systems capacity to process all the extra organic matter (algal blooms). Geologically these are probably fairly short but across a regional scale might take some time to respond.*

I suppose my point is its complex and i don't think you've quite captured the complexity here - the IPCC models/predictions for changing hydro cycle and nutrient inputs could be used to illustrate some of this complexity.

Heres the refs i mention:

Caswell, B.A., Paine, M., Frid, C.L.J., 2018. Seafloor ecological functioning over two decades of organic enrichment. Marine Pollution Bulletin: 136, 212–229.

IPCC, 2014. Climate Change 2014: Synthesis Report. Contribution of Working Groups I, II and III to the Fifth Assessment Report of the Intergovernmental Panel on Climate Change, in: Pachauri, R.K., Meyer, L.A. (Eds.). IPCC, New York.

Author's answer: We agree with the reviewer and therefore a few sentences were added and some of the suggested references were cited.

Reviewer's comment: *Line 41 p3: did you look at whether preservation varied?*

Author's answer: The outcrop conditions were quite similar during the whole interval studied and we can also see that the preservation is very similar in most of the specimens studied.

Reviewer's comment: *Line 46 p3: ok, but what do you mean by 'better' preservation? complete vs incomplete? and what did you consider complete? with phragmocone or just complete rostra?*

Author's answer: We mean with at least part of the alveolar region preserved. The sentence is now changed.

Reviewer's comment: *Line 17 p4: 'community composition shift' would be better - after all a population refers to just one species whereas a community refers to all species present in the assemblage.*

Author's answer: "Assemblage" is a term often used to refer to a fossil community. We think that the term community in this context can be misleading so we would like to keep assemblage. But we agree that population can also be misleading so we changed it.

Reviewer's comment: *Line 19 p4: i am curious how you can tell when maturity occurs -is it explained somewhere? in the supplement perhaps?*

Author's answer: By observing the growth increments of the rostra observed through the longitudinally cut specimens and/or scans. This is now added to the methods section.

Reviewer's comment: Ok for incremental change but how do you compare bed 1 with bed 6?

Author's answer: It is always possible to compare any bed because it is a proportional change.

Reviewer's comment: *Line 29 p4: This is vague. The emphasis must be on comparisons - they compare groups - yes it gives you a significance value but for 'comparisons'. 'So tests were used to compare x, y parameters between z groups' is more accurate*

Author's answer: Accepted. A paragraph was added to explain how were the comparisons made.

Reviewer's comment: *Line 37 p4: what are 'pre-boundary crossers'?*

Author's answer: This is explained in the materials and methods sections but as both reviewers find it confusing we changed it by removing the pre and post prefixes.

Reviewer's comment: *Line 45 p4: because bed separation = temporal separation? explain why necessary.*

Author's answer: Accepted and changed.

Reviewer's comment: *Line 56 p4: why should differences in sediment properties effect*

pelagic belemnites? via their food supply? or do you mean its geochemical properties that are proxies for env change? not clear why this matters.

Author's answer: Sentence changed.

Reviewer's comment: Line 9 p 5: what metric did you use for primary producers?

Author's answer: Abundance of dinoflagellates. This is provided in the appendix.

Reviewer's comment: What was the difference and what was compared?

Author's answer: In the methods there is now an explanation for how exactly the test between beds were made. I think if we explain it here makes the results really hard to read.

Reviewer's comment: *Line 33 p5: ok, so what is happening at P8-P10 is this some form of recovery?*

Author's answer: We discuss this in the discussion. I don't think the results are the place for that.

Reviewer's comment: *Line 37-41 p5: i would restructure this subsection to begin with this overview para*

Author's answer: Accepted and changed.

Reviewer's comment: *Line 8 p6: is it geometric mean length? i cant remember - remind us occasionally*

Author's answer: The meaning of GM is now more clearly explained in the materials and methods and we explain the meaning of the acronym at the beginning of each section and also in the figure captions.

Reviewer's comment: *Line 11 p6: if its not sig different don't try to infer meaning*

Author's answer: I cannot agree with this. Although not enough to be statistically significant, de the small sample size, I still think it is relevant and therefore it is also discussed in the discussion section.

Reviewer's comment: *Line 14 p6: family names should not be in italics - only genera and species*

Author's answer: Accepted and changed.

Reviewer's comment: *Line 17 p7: which metric? and what was the scale of reduction? data added to the sentence.*

Author's answer: Accepted and changed.

Reviewer's comment: *Line 32 p7: 'detectable from the fossilised remains' - there may have been biological differences in morphology, life history or behaviour that are not preserved on hard parts.*

Author's answer: Accepted and changed.

Reviewer's comment: *Line 53 p7: any evidence of this from the modern literature? if so refer to it here*

Author's answer: Sentence changed.

Reviewer's comment: *Lines 55 p7 to line 10 p8; this is quite repetitive and hard to follow - rework the para to be more concise and direct*

Author's answer: Accepted and changed. An attempt was made to be more concise and direct by adding references and concrete examples.

Reviewer's comment: *Section 5.2 p10: this subsection could be much shorter*

Author's answer: Many of the considerations made in this section were asked in the previous revision of the manuscript. The reviewer said we need to focus on the geological context of our study and therefore I cannot delete information from this section because it is very relevant for the discussion of our results. I could make it shorter if the reviewer give me some concrete points where I am being repetitive in this section.

PDF comments

Page 2 (of 161)

Reviewer's comment: *Line 40: would be best to name the proxy here*

Author's answer: Accepted and changed.

Reviewer's comment: *Line 40: Presumably indirectly?*

Author's answer: Accepted and changed.

Reviewer's comment: *Line 49: delete 'so'*

Author's answer: Accepted and changed.

Reviewer's comment: *Line 56: even be*

Author's answer: Accepted and changed.

Reviewer's comment: *Line 58: add 'that are somewhat analogous to belemnites' or similar to accommodate your wider audience*

Author's answer: Accepted and added.

Page 3 (of 161)

Reviewer's comment: *Line 9: hyphenate as in other places - check all instances*

Author's answer: Accepted and changed throughout.

Reviewer's comment: *Line 11: "For instance, over longer..."*

Author's answer: Accepted and added.

Reviewer's comment: *Line 17: yellow highlighted text not needed here*

Author's answer: Accepted and deleted.

Reviewer's comment: *Line 21: refs for this?*

Author's answer: Accepted and added.

Reviewer's comment: *Line 28: European?*

Author's answer: Accepted and added.

Reviewer's comment: *Line 31: in which region/site ?*

Author's answer: Info added.

Reviewer's comment: *Line 38: and mention the P-T boundary shift*

Author's answer: We already say in the introduction section "Despite marked extinction in ammonites (30), the few data existing on Pliensbachian European belemnite fauna do not allow to distinguish the Pli-Toa boundary event as an extinction horizon, at least in north-west European basins (31-34)." This summarizes that data available for the Pli-Toa boundary regarding belemnites. I am not sure if I understand what des the reviewer mean with this comment. Please clarify.

Reviewer's comment: *Line 57: add 'squid'*

Author's answer: Accepted and added.

Reviewer's comment: Line 58: I think there's also been some work on their evolution around this time? Maybe in Doyle 1990 or a Hallam ref?

Author's answer: Doyle 1990 was already cited.

Page 4 (of 161)

Reviewer's comment: *Line 11: and diverse?*

Author's answer: Accepted and added.

Reviewer's comment: *Line 13: insert 'the'*

Author's answer: Not clear.

Reviewer's comment: *Line 51 singular*

Author's answer: Accepted and changed.

Reviewer's comment: *Line 52: "soft-parts" highlighted*

Author's answer: Corrected.

Reviewer's comment: *Line 60: any special settings?*

Author's answer: These are in the appendix. The table is now cited in the text.

Page 5 (of 161)

Reviewer's comment: Line 6: filled instead of filed.

Author's answer: Accepted and corrected.

Reviewer's comment: Line 20: add)

Author's answer: Accepted and added.

Reviewer's comment: Line 22: not italic

Author's answer: Accepted and changed.

Reviewer's comment: Line 32: add "stratigraphically"

Author's answer: Accepted and added.

Reviewer's comment: Line 36: ok for incremental change but how do you compare bed 1 and bed 6?

Author's answer: This is just an example to compare. This method makes them all comparable. It allows to compare 2 assemblages. We changed the sentence to make it more clear.

Reviewer's comment: Line 40: as above

Author's answer: Changed.

Reviewer's comment: Line 43: variance partitioning

Author's answer: No, the method is called variation partitioning. Add some brief explanation in the methods to make it more clear.

Reviewer's comment: Line 47: see my comment above re groups - what was compared?

Author's answer: we now explain clearly in the methods how were the pairwise comparisons made and testes.

Reviewer's comment: *Line 48: of R presumably?*

Author's answer: Accepted and the ref was included.

Reviewer's comment: *Line 49: meaning of ANOVA*

Author's answer: Accepted and changed.

Reviewer's comment: *Line 54: of bulk rock? (of mercury)*

Author's answer: Yes. This is now added to the text.

Page 6 of 161

Reviewer's comment: *Line 8: what metric for primary producers?*

Author's answer: Dinoflagelates abundance but i don't think it is worthy to mentioned it here cause this is a rather general statement.

Reviewer's comment: *Line 17: delete "if".*

Author's answer: Accepted and changed.

Reviewer's comment: *Line 29: replace body by rostrum.*

Author's answer: Accepted and changed.

Reviewer's comment: *Line 30: between beds...instead of P1-P2.*

Author's answer: Accepted and added.

Reviewer's comment: *Line 30: 'statistically significant decrease' instead.*

Author's answer: Accepted and changed.

Reviewer's comment: *Line 31: quote the statistical test.*

Author's answer: Accepted and added.

Reviewer's comment: *Line 33: it is not sufficient to say it was significant - what was the difference? what was compared?*

Author's answer: Accepted. It is now more clear.

Reviewer's comment: *Line 33: "having significantly different body-sizes" instead.*

Author's answer: Accepted and changed.

Reviewer's comment: *Line 33: add "beds"*

Author's answer: Accepted and changed.

Reviewer's comment: *Line 44: this should follow on from para 1*

Author's answer: Accepted. Par 1 and 2 swapped

Reviewer's comment: *Line 49: delete "of the"*

Author's answer: This cannot be changed because it would change the sense of the sentence.

Reviewer's comment: *Line 52: how much lower?*

Author's answer: This can be seen in the figure (Fig. 2) and the figure is cited. I added the values in brackets.

Reviewer's comment: *Line 56: during which no belemnites occur*

Author's answer: Accepted and added.

Page 7

Reviewer's comment: *Line 15: some odd para breaks*

Author's answer: In this case the space is separating different subsections of the "belemnite body-size fluctuations" section. I don't think it makes sense to remove the space.

Reviewer's comment: *Line 18: abbrev in full here*

Author's answer: This is explained in the methods now. In our opinion it adds confusion to write it in full.

Reviewer's comment: *Line 21: and the other parameters you considered?*

Author's answer: The parameters are explained in the methods.

Reviewer's comment: *In what?*

Author's answer: In body-size. Added.

Reviewer's comment: *Line 23: which was what? remind us*

Author's answer: Accepted and added.

Reviewer's comment: *Line 25: how did they behave?*

Author's answer: Unfortunately I don't understand this comment. I think the sentence is self-explanative, however I am happy to change it afterwards if I understand the question.

Reviewer's comment: *Line 27: results should all be in past tense - 'showed'. Put this text in the above paragraph as its directly relevant.*

Author's answer: Accepted and changed.

Reviewer's comment: *Line 32: of the assemblage? of the population?*

Author's answer: I am referring to the change in size occurred in *P. bisulcata*, so I don't mean the assemblage size shift, but within taxa size shift. If I don't specify the species, I say "assemblage". Please see figure 2 to clarify this, which is also clearly stated in the methods.

Reviewer's comment: *Line 39: which body-size metric?*

Author's answer: We only use GM, as stated in the methods. The formula is now included.

Reviewer's comment: *Line 43: meaning of AICc*

Author's answer: This is explained in the methods, I think it adds confusion if I replace it by "Akaike's second order corrected information criterion". This is a term very often used in this kind of study.

Page 8

Reviewer's comment: *Line 10: i would begin with broader context - why does it matter? then what did you aim to do, then present your main result.*

Author's answer: Added.

Reviewer's comment: *Line 17: which boundary? remind us what this means*

Author's answer: Pli-Toa added.

Reviewer's comment: *Line 22: remind us what it is (H5 hypothesis)*

Author's answer: Accepted and added.

Reviewer's comment: *Line 23: a median geometric mean?*

Author's answer: The geometric mean as explained in the methods section combines D_v , D_l and l . The formula is now included in the methods. In the same way you can calculate the median of the diameter or length, you can also calculate the median of the geometric mean.

Reviewer's comment: *Line 25: 'shown to be an important factor for these species'? you did account for it in your analyses - or are you referring to this other work - not clear.*

Author's answer: The ontogeny was not considered at all in the work cited. Sentence changed.

Reviewer's comment: *Line 27: need to directly link to your data - needs to be integrated within the above.*

Author's answer: Accepted and changed.

Reviewer's comment: *Line 30: add "found in the present study".*

Author's answer: Accepted and added.

Reviewer's comment: *Line 32: ok, where and what is its supposed purpose?*

Author's answer: Sentence added.

Reviewer's comment: *Line 36: "constraints" instead*

Author's answer: Accepted and changed.

Reviewer's comment: *Line 38: reorganise sentence to emphasise your results more - 'Changes in body size were observed at xx despite.....!'*

Author's answer: Accepted and changed.

Reviewer's comment: *Line 39: and not P-T*

Author's answer: Not clear what the reviewer means. Please clarify.

Reviewer's comment: *Line 42: delete yellow and add 'greater'*

Author's answer: Accepted and changed.

Reviewer's comment: *Line 45: 'an interval barren of'*

Author's answer: Accepted and changed.

Reviewer's comment: *Line 49: "in" instead of "of"*

Author's answer: Accepted and changed.

Reviewer's comment: *Line 56: "Hastitidae sp. indet." shouldn't be italic*

Author's answer: Accepted and changed.

Reviewer's comment: *Line 59: integrate with existing knowledge and refer to it*

Author's answer: Accepted and changed.

Page 9 (out of 161)

Reviewer's comment: *Line 13: refs*

Author's answer: References added.

Reviewer's comment: *Line 19: thickness or time interval?*

Author's answer: Interval. Sentence changed.

Reviewer's comment: *Line 25: delete yellow - confusing and already said above*

Author's answer: Accepted and deleted.

Reviewer's comment: *Line 32: combine with 2 above paragraphs*

Author's answer: Accepted and changed.

Reviewer's comment: *Line 35: add 'temperate' to help with distinction from your subtropical data*

Author's answer: Changed.

Reviewer's comment: *Line 40: can you remind us where this is in terms of the event - its hard to keep track*

Author's answer: Accepted and changed.

Reviewer's comment: *Line 44: Ocean instead of ocean.*

Author's answer: Accepted and changed throughout.

Reviewer's comment: *Line 45: hyphenate as below? check for consistency*

Author's answer: Accepted and changed throughout

Page 9

Reviewer's comment: *Line 23: refs? its not clae that this is your data*

Author's answer: "According to our results" was added to the sentence to emphasize that we refer to our data..

Page 10

Reviewer's comment: *Line 37: artic highlighted*

Author's answer: Replaced by "arctic".

Figure 2

Reviewer's comment: *Reviewer's comment: see my comment on the methods 'community' is preferable to 'population'*

Author's answer: We changed it for assemblage, as previously explained.

Reviewer's comment: *It would be more intuitive if you moved H4 and H5 to the right to sit under H3 - and put the individual scale label on the left with the others*

Author's answer: Accepted and changed.

Figure 3 A

Reviewer's comment: *It is unclear on what the % are, and im also not clear on the different groups 'beds', 'ontogeny' etc. Can you be clearer on the plot and in the caption.*

Author's answer:

We provide better explanation about this method (variation partitioning) in the methods section and we also changed it here.

Figure 3B

Reviewer's comment: *Is this body-size?*

Author's answer: Yes. Left Y axys label changed.

Reviewer's comment: *Is this body-size?*

Author's answer: Yes. Right Y axis label changed.

Reviewer's comment: *Are these organised based on statistical comparisons? if so 'comparisons' or 'pairwise comparisons' would be clearer (x axis).*

Author's answer:

They do correspond to pairwise comparisons. This is now explained in the figure caption to make the figure easier to read.

Figure 4

Reviewer's comment: *Font sizes are unnecessarily small, plenty of space to use larger fonts*

Author's answer: Accepted and changed.

Reviewer's comment: *What are the error bars?*

Author's answer: The figure was changed as well as the caption.

Figure 5

Reviewer's comment: *See below re log - its no longer mm once you log it - i assume you mean you log transformed the data.*

Author's answer: by this notation i just means that i log transformed the geometric mean data (in mm). Label changed to make it more clear.

Reviewer's comment: *What is the scale of the log?*

Author's answer: As mentioned in the figure caption, the log is not drawn to scale.

Reviewer's comment: *What are the error bars?*

Author's answer: The meaning is explained in the figure caption.

Reviewer's comment: *What does the grey area represent?*

Author's answer: The left graph of fig. 5 represents a violin plots. This kind a plot combines box plots with the kernel density plots. the caption now explains it and the figure was also changed to make more clear the different graphs represented.

Figure 6

Reviewer's comment: *Add label for 'ammonite zones'.*

Author's answer: Accepted and added.

Reviewer's comment: *Its not clear to me - how sample size is shown, or what the 'error bars' on the frequency data show. Can you clarify in the caption?*

Author's answer: Figure caption changed.

Reviewer's comment: *If its the log it doesnt have mm units.*

Author's answer: By this notation i just means that i log transformed the geometric mean data (in mm). Label changed to make it clearer.

Reviewer's comment: *Proportional change in juveniles vs adults? not clear to me what this is*

Author's answer: Yes. Figure changed.

Figure 3 caption

Reviewer's comment: what are the values?

Author's answer: The materials and methods section explains in detail the variation partitioning method. A reference to this section of the paper is now included in the caption.

Reviewer's comment: *is it 'effect on size'? as in body-size? or is it an 'effect size' as in the magnitude of the effect on a mixture of parameters? I find the figure hard to follow so i think the caption needs expanding.*

Author's answer: We mean effect on body size change, as it is. We follow the notation of Rego et al., 2012. Figure changed.

Reviewer's comment: *Can you explain what the different tests were and what the symbols indicate - is it significance at $p < 0.05$?? And what is the 'U test'? is it the 'Mann-Whitney U'? (the latter is the proper name).*

Author's answer: The materials and methods section already included the sentence: "The statistical significance of each coefficient of the best model was assessed by calculating the p-value under a t approximation. The significance level was $p < 0.05$ for all the analyses, unless stated otherwise". This is now included in the caption of the appendix table S2. 'U test' means Mann-Whitney U test and this is now corrected in the table S6. This is also in fig. 3 to make the legend smaller but it is explained in the caption.

Figure 4 caption

Reviewer's comment: *and so n differs from fig x and y?*

Author's reply: Yes, that information is now added to make it clearer.

Reviewer's comment: *Around which value? and what is the error?*

Author's reply: the figure is changed to make it more obvious. The error can be read in % on the x axis and it was calculated based on the % of the most abundant taxa (*P. bisulcata* vs *P. longiformis*/*Bairistowius* sp. A).

Figure 5 caption

Reviewer's comment: *Are the captions for figs 5 and 6 the wrong way around?*

Author's reply: No, they are not. Fig 5 refers to the whole assemblage and fig. 6 refers to the species *P. bisulcata* and *Ps. longiformis*.

Reviewer's comment: *On which data - this is confusing because you have bars on the frequencies as well - be specific*

Author's reply: The error bars are only represented in the relative frequencies. The left side of the figure corresponds to a box and whisker graph which does not depict any error bars. The figure and caption were changed and this is more clear now.

Reviewer's comment: *Ok but what is the scale? there must be one for the lithology*

Author's reply: The lithology legend is indicated in fig 5, this is not referred in the caption of Fig. 1.

Figure 6 caption

Reviewer's comment: *Geometric mean? explain all abbreviations*

Author's answer: "Geometric mean" added to the caption.

Reviewer's comment: *I can't see n anywhere on the figure.*

Author's answer: Sample size (n) is depicted in figure S6 (supplementary figure) and not in Fig. 6, as mentioned in Fig. 6 caption.

Reviewer's comment: *"Passaloteuthis" in italics.*

Author's answer: Accepted and changed.

Reviewer's comment: *You said this twice - typo?*

Author's answer: Yes. Deleted.

Appendix C

R. Soc. open sci. article template

ROYAL SOCIETY
OPEN SCIENCE

R. Soc. open sci.

doi:10.1098/not yet assigned

Mechanisms and drivers of belemnite ~~body-~~ ~~size~~body-size dynamics across the Pliensbachian–Toarcian crisis

Patrícia Rita^{1,2*}, Paulina Nätscher¹, Luís V. Duarte^{2,3}, Robert Weis⁴,
Kenneth De Baets¹

¹Geozentrum Nordbayern, Friedrich–Alexander Universität Erlangen–Nürnberg, Erlangen 91054, Germany

²MARE (Marine and Environmental Sciences Centre), 3004–517 Coimbra, Portugal

³Department of Earth Sciences University of Coimbra, 3070–790 Coimbra, Portugal

⁴National Museum of Natural History Luxembourg, Department of Palaeontology, L–2160 Luxembourg

Keywords: cephalopods, Lilliput effect, Pliensbachian–Toarcian boundary event, Toarcian Oceanic Anoxic Event, climate warming, computed tomography

1. Summary

~~Body size~~Body-size reduction is considered an important response to current climate warming and has also been observed during past biotic crises, including the Pliensbachian–Toarcian crisis, a second–order mass extinction. However, in fossil cephalopods studies, the mechanisms and their potential link with climate are rarely investigated and they usually do not differentiate palaeobiological scales of organisation are usually not differentiated. Here, w~~We~~ hypothesise that belemnites reduce their adult size across the Pliensbachian–Toarcian boundary warming event.

Belemnite ~~body size~~body-size dynamics across the Pliensbachian–Toarcian boundary in the Peniche section (Lusitanian Basin, Portugal) were analysed for the first time based on newly collected field data. We disentangle the mechanisms and the environmental drivers of the size fluctuations observed from the individual to the assemblage scale. Despite the lack of a major taxonomic turnover, a 40 % decrease in rostrum volume is observed across the Pliensbachian–Toarcian boundary, before the Toarcian Oceanic Anoxic Event where belemnites go locally extinct. The pattern is mainly driven by a reduction in adult size of the two–dominant species, *Pseudohastites longiformis* and *Passaloteuthis bisulcata*. Belemnite size distribution is best correlated with fluctuations in a palaeotemperature proxy (stable oxygen isotopes); however, potential indirect effects of volcanism and carbon cycle perturbations may also play a role. This highlights the complex interplay between environmental stressors ~~stressors~~–(warming, deoxygenation, nutrient input~~nutrient input~~) and biotic ~~factors~~variables (productivity, competition, migration)

~~environmental stressors~~ associated with these hyperthermal events in driving belemnite ~~body-size~~ ~~body-size~~.

2. Introduction

~~Body-size~~ ~~Body-size~~ is a key feature of any organism, reflecting its physiology, ecology and evolutionary history, across multiple scales of biological organisation (1). ~~Body-size~~ ~~Body-size~~ reduction has been considered the third universal response to global warming after changes in phenology and species distribution (2-4). However, disentangling ~~body-size~~ ~~body-size~~ responses to warming might not be so straightforward due to interactions ~~between~~ ~~of the different biotic and~~ ~~abiotic factors~~ ~~during~~ ~~involved~~ climate warming episodes ~~and also due to biotic factors~~ (5, 6), ~~even in a Jurassic, when additional factors like the human activities were absent~~ (6, 7). Many factors during climate warming (increased temperature, decreased oxygenation, ocean acidification) are likely to work in synergy in leading to reductions in size (6). However, increasing nutrient supply, for example, might have the opposite effect, causing a ~~body-size~~ ~~body-size~~ increase in some taxa (4, 5).

Moreover, individual responses to warming might ~~be~~ ~~even~~ ~~be~~ population specific – due to individual environmental requirements (availability of nutrients and oxygen) and metabolic specifications (8). Individual living cephalopod taxa, such as squid (~~–analogous to the fossil belemnites~~), ~~as stenothermal organisms~~, respond rapidly to warming events, hatching at smaller size, undergoing fast growth rates over shorter life-~~–~~spans and maturing younger at smaller size (9-11).

Reductions in ~~body-size~~ ~~body-size~~ – coined the Lilliput effect (12) – have been widely reported from mass extinctions and evolutionary crises associated with environmental perturbations (13, 14) including the Early Toarcian crisis (15). However, their mechanisms and environmental drivers are still poorly understood (14), especially because the effects of environmental perturbations on an organism's ~~body-size~~ ~~body-size~~ might depend on the biological scale of organisation considered (2). ~~For instance, o~~ ~~Over~~ longer evolutionary time-~~–~~scales, i.e. considering fossil assemblages, several mechanisms might contribute to the Lilliput effect, including increased mortality of juveniles, extinction or temporary disappearance of large taxa, preferential survival or origination of small-sized taxa or the temporary reduction in adult ~~body-size~~ ~~body-size~~ of surviving taxa (13, 14). All of these mechanisms could explain the Lilliput effect *sensu lato*. However, only the latter, i.e., a within-~~–~~lineage ~~body-size~~ ~~body-size~~ decrease, is considered to be the mechanism behind the Lilliput effect *sensu stricto*, which depicts a reduction in ~~body-size~~ ~~body-size~~ within individual species through time ~~and it corresponds to the original term~~ (12). Interestingly, extinction ~~risk in marine mollusks during multiple mass extinctions has been associated during past events might also be associated~~ with preferential extinction of smaller rather than larger species ~~including the end-Permian, end-Triassic and end-Cretaceous mass extinctions (7)–such as for example Cretaceous-Paleogene boundary seems to have hit particularly the large mammals (ref)~~.

The Early Toarcian crisis is a multi-phased event characterised by environmental and biodiversity perturbations. ~~One of these pulses corresponds to the Pliensbachian-Toarcian (Pli-~~

Formatted: Font: (Default) Palatino-Roman, Font color: Auto, Pattern: Clear

Formatted: Font: (Default) Palatino-Roman, Font color: Auto, Pattern: Clear

Formatted: Font: (Default) Palatino-Roman, Font color: Auto, Pattern: Clear

Formatted: Font: (Default) Palatino-Roman, Font color: Auto, Pattern: Clear

Formatted: Font: (Default) Palatino-Roman, Font color: Auto, Pattern: Clear

Formatted: Space Before: 12 pt, After: 8 pt

Field Code Changed

Formatted: Not Highlight

Formatted: Space Before: 12 pt

Formatted: German (Germany)

Formatted: German (Germany)

*Author for correspondence (patricia.rita@fae.u.de).

†Present address: Geozentrum Nordbayern, Loewenichstrasse 28, 91054 Erlangen, Germany Department, Institution, Address, City, Code, Country, R. Soc. open sci.

~~the Pli-Toa boundary event and the second pulse corresponds to the Toarcian Oceanic Anoxic Event (T-OAE) (16, 17). One of these pulses corresponds to the Pliensbachian-Toarcian (Pli-Toa) boundary event and the second pulse corresponds to the Toarcian Oceanic Anoxic Event (T-OAE).~~

The Pli-Toa boundary event (~183.7 Ma, (18)) follows the cooling and regressive event of the Late Pliensbachian (Emaciatum = Spinatum Zone) in the northern Tethys and Iberia (19). It corresponds to an increase of seawater temperature (up to 6°C, Fig. 1), concomitant with the beginning of a marine transgression (20-24) as well as a small negative carbon isotopic excursion (25, 26). The Pli-Toa boundary event records a crisis among planktonic (27, 28), benthic (29) and nektonic organisms (30), expressed by extinction and changes in abundance and diversity. Despite marked extinction in ammonites (30), the few data existing on Pliensbachian European belemnite fauna do not allow to distinguish recognition of the Pli-Toa boundary event as an extinction horizon, at least in north-western European basins (31-34).

Formatted: Not Highlight

Field Code Changed

Formatted: Space Before: 12 pt

After the Pli-Toa boundary event, the Polymorphum (= Tenuicostatum) Zone —corresponds to a cooling phase (19, 23, 35) in the northern Tethys and Iberia, although it is comparatively warmer than the Late Pliensbachian. This cooling phase is followed by the T-OAE, starting at the base of the Levisoni Zone (18) and characterised by a marked increase of the seawater temperature (up to 7.5°C, Fig. 1), widespread anoxia and deposition of organic-rich sediments (17, 23, 25, 36-38). These important changes in the ocean-atmosphere system had a major impact on marine biota, causing body-size changes and extinction of particularly pelagic predators such as ammonite species (15, 30, 39) and extinction of benthic suspension-feeding fauna (16, 28, 40-44). Modelling suggest that particularly The T-OAE also marks a major bottleneck in belemnite diversity and abundance (45), and might have triggered belemnite provincialism amongst Boreal-Arctic and North-west European faunas (46-48). The increased anoxia during the T-OAE has been interpreted to contribute to the demise of many belemnite taxa, while *Acrocoelites* might have survived and radiated in its immediate aftermath (49). The T-OAE coincides with an abrupt negative carbon isotopic excursion (CIE) which disrupts an overarching positive excursion (36, 50-52), reflecting enhanced burial of organic carbon and its preservation at the sea bottom. Although deoxygenation has been interpreted to increase since the Pli-Toa boundary, the negative CIE and widespread black shale deposition are still interpreted to reflect the peak-nadir of anoxia during the T-OAE (53).

Both warming events and the associated carbon cycle perturbations are considered a consequence of volcanic outgassing, either directly, by increase of $p\text{CO}_2$ in the seawater (54) or indirectly (i.e. by triggering other sources of isotopically light carbon, (19)). In distal sections, enhanced mercury deposition is interpreted to be a proxy for constraining the Karoo-Ferrar volcanism (55). Irrespective of the controversy over the usage of mercury as a proxy for volcanic outgassing (53), it is hitherto the best available proxy to temporally constrain the direct consequences of rapid rise of $p\text{CO}_2$ in the seawater, including warming, deoxygenation and acidification.

Formatted: Space Before: 12 pt, After: 8 pt

In Peniche, mercury anomalies coincide with extreme climatic changes (55), expected to result in a body-size decrease (6, 8). In contrast, extreme climatic changes can increase weathering and precipitation, causing body-size increase instead, due to the input of nutrients (4, 6). However, despite the evidence for increased weathering and influx of nutrients

in Peniche (24), no indication of marked freshwater input or productivity increase (27) have been observed and, therefore, a reduction in ~~body size~~ is expected. ~~It is however worth noting to mention that modern systems have shown us that the interaction between climate change, the nutrient input and productivity is quite a very complex and might vary regionally (e.g. (56) Moore et al. 2018) and according to the scale considered (57) (Caswell et al. 2018). Additionally, such variations might be difficult to constrain. However, these variations are probably hard to identify in long time scale in the geological record without modelling – so far only available for a limited number amount of time-slices such as at the Paleocene–Eocene Thermal Maximum (58, 59), PETM as the geological one (Wilson et al. 2018; Ilmija and Heinze 2019 refs).~~

Belemnites are coleoids closely related to extant teuthids (such as squid). They are very abundant and diverse in the fossil record and played an important role in Jurassic ecosystems (e.g. (60)). Most of the research on Lower Jurassic belemnites was focused on the geochemistry of north-western European basins (49, 61–63), temperate zones where the regional extent of anoxia was widespread (46, 60). The only study on belemnite ~~body size~~ ~~has been~~ focused solely on two genera and found an ambiguous response to crisis events (15). Therefore, a high-resolution analysis of belemnite ~~body size~~ during past episodes of environmental crisis will provide valuable insights on the response of cephalopods to the global change observed in modern marine ecosystems.

The Peniche reference section is well studied in terms of geochemistry and stratigraphy (24, 25, 55, 64, 65) and yields a highly abundant belemnite fauna, allowing for the first time the analysis of (i) the mechanisms behind belemnite ~~body size~~ fluctuations at different scales of organisation (individual ontogenetic, species and assemblage) during the Pliensbachian–Toarcian interval in subtropical latitudes and (ii) their potential environmental drivers – ~~mainly temperature, carbon cycle and changes in pCO₂ associated with volcanism~~. We hypothesise, on one hand, a within-lineage reduction (Lilliput effect *sensu stricto*) in belemnite ~~body size~~ across the Pli–Toa boundary event and, on the other hand, a ~~body size~~ increase across the T₂–OAE driven by species turnover in the Peniche section.

3. Materials and Methods

3.1. Taxonomy and ontogeny

Belemnite species identification was based on the analysis of traditional features, such as shape (outline and profile) and the presence of grooves in the apical region (33). The transverse section, the depth of penetration of the alveolus and the apical line_z were ~~all~~ observed using CT-scanning. ~~All of these features were afterwards compared with published descriptions and figures. This method also allowed us to recognise the features of each ontogenetic stage with the acquired longitudinal sections, making it possible to distinguish between adult (Neanic–Ephebic–Gerontic *sensu* (60)) and juvenile (Nepionic *sensu* (60)) specimens (Figs. S1 and S2), especially by having the possibility of observing the growth increments of the rostra.~~ The taxonomic composition of the belemnite assemblages across the studied interval is in conformity with data from contemporaneous Tethyan sections (33, 34, 45, 48, 60, 66, 67). Seven taxa compose the belemnite assemblages in the Peniche section (Figs. 1, S1 and S2).

Formatted: Not Highlight

Formatted: Not Highlight

Formatted: Not Highlight

Formatted: Not Highlight

Formatted: Not Highlight

Formatted: Not Highlight

Formatted: Not Highlight

Formatted: Not Highlight

Formatted: Not Highlight

Formatted: Font: (Default) Palatino-Roman, Font color: Auto, English (United Kingdom), Pattern: Clear

Formatted: Font: (Default) Palatino-Roman, Font color: Auto, English (United Kingdom), Pattern: Clear

Formatted: Not Highlight

Formatted: Not Highlight

Formatted: Space Before: 12 pt

Formatted: Font: Italic

Formatted: Font: Italic

Formatted: Indent: Left: 0 pi, Hanging: 3 pi

3.2. Sampling methods and belemnite ~~body-size~~ body-size proxies

The specimens were collected from 13 beds (stratigraphic horizons) sampled ~~from~~ 45 m of Upper Pliensbachian – Lower Toarcian sediments, corresponding to the Upper Emaciatum – Upper Levisoni ammonite zones (Fig. 1, approximately 2.7 Ma, (18)). As belemnites become rarer up section, the data derived from beds P9a, P9b and P9c was pooled to obtain a reasonable sample size for a quantitative analysis (Fig. 1). The same conservative approach was ~~done~~ adopted for beds P3a and P3b.

Formatted: Space After: 8 pt

We focused on quantitatively collecting from well-exposed bedding planes of limestones, marly limestones and marls. A total of 930 belemnite rostra were collected by (1) sampling all specimens within 14 one-square-meter areas and (2) by collecting at least 30 complete specimens in the remaining bed area, i.e., outside of the quadrats. This first sampling method allows the calculation of absolute abundance, taking into account both fragmented and complete specimens. The ~~body-size~~ body-size analysis was based only on the complete specimens, regardless of the sampling method. The more complete belemnites are suspected to be a less time-averaged sample (68) because a better preservation is usually an indicator of quick burial and little transport of the fossil. Conveniently, complete specimens (with at least part of the alveolar region preserved) are easier to assign taxonomically and ontogenetically (69). The fragmented specimens were only used to calculate the belemnite abundance per bed.

Formatted: Space After: 8 pt

The rostrum represents a considerable part of the belemnite animal, and it is hypothesised that it acts as a counterbalance for the soft ~~parts~~ tissue and phragmocone (70), despite the increasing ~~amount~~ number of studies suggesting an original partially porous rostrum structure (71, 72). However, in all known belemnites with soft ~~tissue~~ parts preservation, the fins attach to the rostrum and the soft-parts closely track the outline of their internal skeleton, as is the case in their extant relatives (73, 74). For these reasons, the rostrum can be considered a reasonable proxy for body size in the absence of preserved soft ~~tissue~~ parts (69). This is also supported by a comparison between hard parts and soft-tissue parts in well-known taxa showing that larger rostra correspond to with a larger mantles (75) length (Klug et al. 2018). It is noteworthy that most of the studied specimens did not bear epistrotra and therefore only the orthorostra were considered for the morphometric analysis. As for the epistrotra-bearing specimens, only the orthorostrum was measured.

Formatted: Space Before: 12 pt

Formatted: Highlight

A recent study demonstrated that different species might have different relationship rostrum/soft tissue size but further work is still necessary to assess all the relationships for all the species. (Klug et al., 2018)

Formatted: Highlight

Formatted: Highlight

Previous research has demonstrated that the geometric mean (GM), an unidimensional metric that combines of the apical height, width and length ($GM = \sqrt[3]{Dv * Dl * l}$) is the most robust proxy to compare body-size between morphologically different forms belonging to different species or ontogenetic stages within a species (69). In addition, ~~the geometric mean~~ is more comparable to proxies used in extant coleoids (76). In order to calculate the GM, 277 of the complete specimens were CT-scanned to allow the calculation of this metric in a non-destructive way, since the alveolus was filled with sediment, precluding ~~direct~~ the measuring process (see Table S7 for details on the ~~the settings~~ details). Additionally, 109 specimens were measured

Formatted: Space Before: 12 pt

Formatted: Not Highlight

with a calliper, since the alveolus was empty. The relative change in volume better reflects changes in absolute body-size proxies used in extant taxa (e.g., weight). We therefore use the volume of the rostrum (GM^3) to express relative volumetric changes. ~~volume of the rostrum can be obtained by volume = GM^3 .~~

Formatted: Not Highlight

Formatted: Not Highlight

Formatted: Superscript, Not Highlight

Formatted: Not Highlight

3.3. Mechanisms of body-size changes

The mechanisms behind the belemnite body-size fluctuations through time, from the assemblage to the individual scale of organisation, were assessed by testing a set of hierarchical hypotheses (H), modified from (2) (Fig. 2).

The first hypothesis predicts a decrease in mean body-size at the assemblage scale, regardless of the underlying mechanisms (assemblage size shift hypothesis, H_1 , Fig. 2). If H_1 is validated, there are four subsequent hypotheses that could explain this decrease (2, 8). First, an increase in the proportion of small-sized species (assemblage ~~population~~ composition shift hypothesis, H_2). Second, a decrease in median body-size within specific taxa (within-taxa size shift hypothesis, H_3), which could, in turn, be related to a decrease in adult body-size (intraspecific size shift hypothesis, H_4). The last hypothesis associates the decrease in mean body-size with an increase in the proportion of juveniles (ontogenetic stage structure shift hypothesis, H_5).

Commented [i1]: Replace by assemblage

Formatted: Font: Not Italic, Not Superscript/ Subscript

Formatted: Subscript

Formatted: Space Before: 12 pt, After: 8 pt

Formatted: Font: Not Italic

Formatted: Justified, Space Before: 12 pt

Formatted: Not Highlight

Formatted: Not Highlight

Formatted: Subscript

Formatted: Not Highlight

Formatted: Subscript

Formatted: Subscript

Commented [i2]: Replace u test by mann-whitney U test in the appendix table and check in the text '-'

Formatted: Not Highlight

Formatted: Not Highlight

Formatted: Not Highlight

Formatted: Font: (Default) Palatino-Roman, 10 pt

Formatted: Justified

Formatted: Font: (Default) Palatino-Roman, 10 pt

Formatted: Font color: Auto

Formatted: Font color: Auto

Formatted

Formatted: Font: (Default) Palatino-Roman, 10 pt

Formatted: Font color: Auto

Formatted: Font: (Default) Palatino-Roman, 10 pt

Formatted: Font color: Auto

Field Code Changed

Formatted: Not Highlight

Commented [i3]: Effect size or effect on body size??

The whole set of hypotheses could explain the Lilliput effect *sensu lato*, although only H_4 relates to the Lilliput effect *sensu stricto*, as predicted by the temperature-size rule in extant coleoids (77). Note that, in comparison with recent data, the interpretation of body-size changes based on fossil data is more complex because fossil assemblages reflect not only a past biotic community but also the mortality and preservation of the organisms that were part of it.

At the assemblage scale of organisation, H_1 was tested by assessing the size distribution of the whole assemblage through time. The proportional change in body-size was calculated between stratigraphically consecutive beds (example: bed t and bed $t+1$) in percentage of the log ratio (example: $\ln \frac{\text{median bed } t+1}{\text{median bed } t} * 100\%$). ~~The non-parametric Mann-Whitney U and Kolmogorov-Smirnov (K-S) tests were used to assess the significance of the results (Table S2). The same was done at the species scale, for each taxon.~~

The median belemnite body-size (geometric mean) was calculated for each bed. The whole distribution and the median were compared using the non-parametric Mann-Whitney U and Kolmogorov-Smirnov tests to assess the significance of the belemnite body-size differences between consecutive beds (Table S2). This was done at the assemblage and species scales (Table S2).

Furthermore,

Because changes in taxonomic composition can influence the body-size patterns by the appearance and/or disappearance of taxa, we modified the within- and among-taxa approach by Rego et al. (78). This method corresponds to a pair-to-pair analysis (comparison of stratigraphically successive beds, t and t+1), focused on the taxa identified in stratigraphically successive beds (boundary crossers). This method which allow to decomposes/division of the assemblage body-size shift (Effect size on belemnite GM, Eq. 1) into three components: a

R. Soc. open sci. article template

disappearance of taxa effect (Eq. 2), a within-lineage effect (Eq. 3) and an appearance of new taxa effect (Eq. 4).

(Eq. 1) **Assemblage size shift** = assemblage median ~~body-size~~_{body-size}^{bed 1+1} - assemblage median ~~body-size~~_{body-size}^{bed 1+}

(Eq. 2) **Disappearance of taxa effect** = ~~pre~~-boundary crossers^{bed 2+1} - all taxa^{bed 1+1}

(Eq. 3) **Within-lineage effect** = ~~post~~-boundary crossers^{bed 2+1} - ~~pre~~-boundary crossers^{bed 1+1}

(Eq. 4) **Appearance of new taxa effect** = all taxa^{bed 2+1} - ~~post~~-boundary crossers^{bed 2+1}

~~In an individual sample, appearance of taxa could relate to immigration (if known from other sections before) or origination (only appear here) while disappearance could relate to emigration (taxa are still known from other region after) or extinction (taxa do not return globally). Rare taxa might also temporary disappear or appear in larger samples. Knowing the exact reason for appearance and disappearance can only be disentangled from a global comparison in case of an ideal fossil record, so that we prefer to use the more neutral terms appearance and disappearance.~~

Finally, at the individual ontogenetic scale of organisation, the hypotheses were tested by assessing ~~body-size~~_{body-size} variation within different ontogenetic stages (adults and juveniles) for ~~particular individual~~ taxa. Variation partitioning allowed us to assess the proportion of ~~body-size~~_{body-size} (geometric mean) variation explained by ontogeny, taxonomic assignment and separation by beds, and by their joint effects. If the fraction of variation corresponding to beds separation is low, this suggests that mechanisms driving differences between samples are similar and not related to particularities of the sample (e.g., lithology, age). Variation in the ~~body-size~~_{body-size} data was partitioned using partial redundancy analysis (RDA) as implemented in the *vegan* package in R (79, 80). The significance of each fraction was assessed through ~~analysis of variance (ANOVA)~~_{ANOVA}. The whole time-series was compared with the Pli-Toa boundary event.

3.4. Environmental drivers of ~~body-size~~_{body-size} changes

In order to test our hypothesis of a relationship between belemnite ~~body-size~~_{body-size} and environmental perturbations, the distribution of the GM was compared with three main geologic proxies (Table 1). These are: brachiopod stable oxygen isotopes (23), bulk rock carbon isotopes (25) and ~~bulk rock~~ mercury concentration normalised by total organic carbon (55), used as proxies for seawater palaeotemperature, negative excursions in the carbon cycle and volcanogenic outgassing, respectively (Table 1). To account for the impact of sedimentary properties on ~~body-size~~_{body-size} patterns, we corrected for the effect of lithology and belemnite absolute abundance (mean abundance of belemnites per m², Figs. 1 and S3) by residualising (81). A simple linear regression analysis is performed between the ~~body-size~~_{body-size} and the combined effect of abundance and lithology. The residuals from this analysis are then used as outcome in the ultimate regression analysis between ~~body-size~~_{body-size} and abiotic variables. All variables of the model were continuous with the exception of lithology. We assigned our samples a categorical variable for lithology (marl to limestone) based on carbonate/clay content. Collinearity between explanatory variables was tested by using *cor* function in R (Fig. S5). Sedimentary properties as well as various other parameters linked with climate warming (such as strontium isotopes, total organic carbon and abundance of primary producers) could not be directly included in the

Formatted: Space Before: 12 pt

Formatted: Space After: 12 pt

Formatted: Space Before: 12 pt

Formatted: Highlight

Field Code Changed

Formatted: Default Paragraph Font, Font: (Default) Palatino-Roman, 10 pt

Formatted: Not Highlight

models due to their high collinearity with $\delta^{13}\text{C}_{\text{carb}}$ or the other two used parameters (Fig. S4), hence the choice of the abiotic parameters indicated in Table 1.

A multiple linear regression using generalised least squares (GLS) was performed and the models were fitted by maximising the log-likelihood (*gls* function in R). The first-order autoregressive (AR(1)) model was used, which has the property of seeking autocorrelation and of minimising the error term in a time-series (82). Seven models (see R script, supplementary material) were compared using Akaike's second order corrected information criterion (AICc scores), which corrects for small sample size. The power of the best model was assessed by means of an ANOVA test. The statistical significance of each coefficient of the best model was assessed by calculating the p-value under a t approximation. The significance level was $p < 0.05$ for all the analyses, unless stated otherwise. The analyses were performed in R (80), using the packages *nlme* (83) and *qpcR* (84).

The regression analysis only included data from Emaciatum and Polymorphum zones (beds P1 to P9), due to the lack of a representative sample size in upper Levisoni Zone (bed P10, Fig. 1).

4. Results

4.1 Belemnite body-size fluctuations

Assemblage scale

A total of seven belemnite species were identified in Peniche (Fig. 1). Most of the taxa range from the uppermost Pliensbachian to the Lower Toarcian (uppermost Polymorphum Zone). *Bairdovius* sp. A is exclusively represented in the uppermost Pliensbachian, being stratigraphically replaced by *Pseudohastites longiformis* in the assemblage. During the Toarcian, the Levisoni Zone is devoid of belemnites, with the exception of bed P10 (upper Levisoni Zone) which includes two specimens assigned to *Acrocoelites* sp. specimens (Fig. 1).

Three episodes of median rostrum body-size decrease were recognised at the assemblage scale across the studied interval: between beds P1- and P2, between bed P2- and P3, and at the Pli-Toa boundary event (beds P4- and P5). Only- with the latter being the only corresponds to a decrease with statistically significant decrease (p-value Kolmogorov-Smirnov test = 0.01; p-value Mann-Whitney U test = 0.04 Mann-Whitney U and Kolmogorov-Smirnov tests; Fig. 3B) and corresponding to a 13 % body-size decrease (Fig. 4). This decrease in the belemnite body-size proxy (GM) corresponds to a 40 % decrease of rostrum volume (Table S2).

The assemblage body-size shift at the Pli-Toa boundary event is almost exclusively caused by the boundary crossers *Ps. longiformis*, *P. bisulcata* and *Hastitidae* sp. indet. (within-taxa effect; Fig. 3B). The taxonomic composition across the boundary does not change markedly, with *Ps. Longiformis*, the most abundant taxon, comprising 61.9 % (bed P4) and 56.2 % (bed P5) of the assemblages (Figs. 1 and 4). The disappearance of *Passaloteuthis milleri* and *Parapassaloteuthis* sp. at the Pli-Toa boundary event causes 4 % of the decrease in the assemblage median body-size (disappearance of taxa effect; Fig. 3B).

Formatted: Space Before: 12 pt

Formatted: Font: Not Italic

Formatted: Highlight

In contrast, the other two episodes of body-size reduction (between bed P1 and P2 and between bed P2 and P3) are mainly related to the appearance (15.9 % and 16.2 %, respectively) and disappearance of taxa (12.1 % and 15.2 %, respectively) but also to the body-size decrease of specific taxa (within-taxa effect; Fig. 3B). The within-taxa effect on the assemblage body-size shift between beds P1 and P2 and between beds P2 and P3 (xx72 %; xxx69 %) is however much lower than at the Pli-Toa boundary (xxx96 %, Fig. 3).

At the assemblage scale, an increase in median body-size is observed between beds P3 and P4, P5 and P6, P7 and P8, P8 and P9 and between P9 and P10. However, only the bed pairs P8-P9 and beds P9-P10 correspond to significantly different body-sizes (p-value = xxx Fig. 3B).

In summary, across the studied interval at the assemblage scale, a significant decrease in belemnite body-size is observed across the Pli-Toa boundary (beds P4 and P5). After that, from bed P5 to bed P9, a general increase in belemnite body-size is observed, followed by a belemnite gap, from bed P9 to bed P10 (Figs. 3B and 5), during which no belemnites occur.

The remaining pairs of beds correspond to an increasing median body size at the assemblage scale, with P8-P9 and P9-P10 being statistically significant (Fig. 3B).

A total of seven belemnite species were identified in Peniche (Fig. 1). Most of the taxa range from the uppermost Pliensbachian to the Lower Toarcian (uppermost Polymorphum Zone). *Bairdowius* sp. A is exclusively represented in the uppermost Pliensbachian, being stratigraphically replaced by *Pseudohastites longiformis* in the assemblage. During the Toarcian, the Levisoni Zone is devoid of belemnites, with the exception of bed P10 (upper Levisoni Zone) which includes two *Aerocoelites* sp. specimens (Fig. 1).

The assemblage body size shift at the Pli-Toa boundary event is almost exclusively caused by a and therefore mainly driven by the boundary crossers (*Ps. longiformis*, *P. biculcata* and *Hastitidae* sp. indet.), within taxa effect (Fig. 3B) and therefore mainly driven by the boundary crossers (*Ps. longiformis*, *P. biculcata* and *Hastitidae* sp. indet.). The taxonomic composition across the boundary does not change markedly, with *Ps. longiformis* the most abundant taxon, comprising 61.9 % (P4) and 56.2 % (P5) of the assemblages (Figs. 1 and 4). The disappearance of taxa effect (*Passaloteuthis milleri* and *Parapassaloteuthis* sp.) across the Pli-Toa boundary event, causes 4 % of the decrease in the assemblage median body size (Fig. 3B).

In contrast, the other two episodes of body size reduction (P1-P2 and P2-P3) are characterised by higher effects of appearance (15.9 % and 16.2 %, respectively) and disappearance of taxa (12.1 % and 15.2 %, respectively), with the within-taxa effect (Fig. 3B) being comparatively lower than at the boundary (Fig. 3B).

From P5 to P9, an increase in belemnite body size at the assemblage scale is observed, followed by a belemnite gap, from P9 to P10 (Figs. 3B and 5).

Species scale

Formatted

Formatted: Font: Palatino-Roman, 10 pt, Font color: Text 1, Not Highlight

Formatted: Justified

Formatted

Formatted: Underline, Font color: Auto

Formatted: Indent: Hanging: 0.05 pi

Formatted: Font: Underline, Font color: Auto, Expanded by 0.75 pt

Formatted: Font: Underline, Expanded by 0.75 pt

Formatted

Formatted: Font: Not Italic, Underline, Font color: Auto, Expanded by 0.75 pt

Formatted: Font: Underline, Font color: Auto, Expanded by 0.75 pt

Formatted

Formatted

Commented [i4]: How much lower?

Formatted

Formatted: Font: Underline, Expanded by 0.75 pt

Formatted: Justified

Formatted: Font: (Default) Palatino-Roman, 10 pt, Underline, Expanded by 0.75 pt

Formatted: Normal, Justified

Formatted: Font: Palatino-Roman, 10 pt, Underline, Not Strikethrough

Three taxa (*Ps. longiformis*, *P. bisulcata* and Hastitidae sp. indet.) are recorded immediately before and immediately after the Pli–Toa boundary event (boundary crossers; Fig. 1). The most significant and largest median size-reduction in size is observed in the most abundant taxon, *Ps. longiformis*, (17 % decrease in GM and 51 % in rostrum volume, Fig. 6)–the most abundant taxon (17 % decrease in GM and 51 % in rostrum volume; Fig. 6).

Formatted: Font: Not Italic

Despite the lack of statistical significance, *P. bisulcata*, the second most abundant taxon, also decreases in size (7 % in GM and 21 % in volume, Fig. 6) across the Pli–Toa boundary event, while Hastitidae sp. indet., on the contrary, slightly increases in size (8 %, Fig. S7). It is noteworthy that *P. bisulcata* body-size markedly decreases in body-size (109 %, Fig. 6) immediately after the Pli–Toa boundary event, from bed P5 to bed P6 (lowermost Polymorphum Zone), which is coincident with the extirpation of Hastitidae sp. indet. (Fig. 1).

Formatted: Font: Not Italic

Formatted: Font: Not Italic

Individual ontogenetic scale

Considering the whole time-series, the RDA-variation partitioning analysis results revealed that taxonomic composition (47 %) and ontogeny (17 %) are responsible for most of the variation in belemnite body-size. The separation by bed explains only 2 % of the body-size variation (Fig. 3A, values between parentheses in brackets??).

Formatted: Not Highlight

At the Pli–Toa boundary event, the body-size-variation partitioning into taxonomy and ontogeny is similar to what is observed for the whole time-series (49 % and 11 % respectively). However, the joint effect of taxonomy and ontogeny explains 20 % of the variation in body-size, in contrast with the 6 % observed for the whole time-series (Fig. 3A, values without parentheses brackets). This probably suggests that across the Pli–Toa boundary event, ontogenetic stages behave more similarly among taxa in explaining body-size variation.

Formatted: Highlight

Formatted: Highlight

At the individual ontogenetic scale of organisation, focusing on *Ps. longiformis*, only the adult specimens of *Ps. longiformis* recorded a show a significant (Kolmogorov–Smirnov test p-value = $2.61E-6$; Mann–Whitney U test p value = $3.98E-7$) body-size decrease across the Pli–Toa boundary event (21 %, Table S2 Fig. 6). The ratio adult vs juveniles of *Ps. longiformis* does not change significantly across the Pli–Toa boundary event (31.25 %, Fig. 6).

Formatted: Highlight

In *P. bisulcata*, a reduction of 7 % (although not significant) in adult body-size is observed in *P. bisulcata* specimens, across the Pli–Toa boundary event, together with a slight increased percentage of juveniles of the same species (Fig. 6). Immediately after the boundary, from bed P5 to bed P6, a marked reduction (109 %) in *P. bisulcata* body-size is recognised mainly among the juvenile specimens, in tandem together with an increase in their proportion (from 75 % to 68.75 %, Fig. 6).

Formatted: Font: Not Italic

4.2 Relationship with environmental parameters

After correcting for the effects of sedimentary properties (lithology and fossil abundance), the results of the GLS-regression analysis revealed that the variation in belemnite body-size

size is best correlated with the variation in $\delta^{18}\text{O}$ (Table 2, model no. 5), at the assemblage scale. The overall model is significant (Table S3), suggesting a body-size reduction with increasing $\delta^{18}\text{O}$ values/decreasing seawater temperature values (Fig. S2). However, the models combining $\delta^{18}\text{O}$ with $\delta^{13}\text{C}_{\text{carb}}$ (model no. 2, Table 2) and combining Hg/TOC with $\delta^{13}\text{C}_{\text{carb}}$ (model no. 3, Table 2) could not be rejected, according to the AICc scores ($\Delta < 2$). If the data is not corrected for the effects of the sedimentary properties the same results are obtained, despite minor differences in significance (Table S5).

After correcting for the effects of sedimentary properties, when considering the most abundant taxon, *Ps. longiformis*, the best model explaining the body-size variation includes solely Hg/TOC (model no. 6, Table 2). However, the effects of $\delta^{18}\text{O}$, $\delta^{13}\text{C}_{\text{carb}}$ and Hg/TOC cannot be discarded, since statistically the models n. 5, 2 and 4 cannot be distinguished from model no. 6, according to the AICc scores. If the data is not corrected for the effects of lithology and abundance beforehand, model no.4 $-(\text{Hg}/\text{TOC} + \delta^{18}\text{O})$ best explains *Ps. longiformis* body-size variation, with highly significant results (Table S5). The role of Hg/TOC in the model is more significant than the role of $\delta^{18}\text{O}$ (Table S4b).

After correcting for the effects of lithology and fossil abundance, *P. bisulcata* body-size is best correlated with Hg/TOC (model no. 6, Table 2), as well, despite the lack of significance (Table 3). However, the effects of $\delta^{18}\text{O}$, $\delta^{13}\text{C}_{\text{carb}}$ and Hg/TOC cannot be disregarded, since statistically the models no. 5, 2 and 4 cannot be distinguished from model no. 6. If no correction is applied beforehand, the model no.4 (Hg/TOC + $\delta^{18}\text{O}$) best explains the *P. bisulcata* body-size variation, with highly significant results (Table S5).

5. Discussion

5. 1. Scales and mechanisms of body-size variation

5. 1.1. Pli-Toa boundary event

Body size reductions have been identified in several episodes of biotic/abiotic crisis, many times related to warming. Body-size fluctuations in fossil assemblages are common and can be caused by relate to a variety of mechanisms. Our goal is to test the effect of warming and related stressors in belemnite body size dynamics across the Pliensbachian-Toarcian crisis recorded in the Peniche section.

Statistical evaluation of our data has shown that size changes of belemnite rostra in the studied section at Peniche are significant for the Pliensbachian-Toarcian boundary event, i.e., the H₁ (assemblage size shift) hypothesis (H₁) was validated. This size change is almost exclusively driven by a body-size reduction of single species - Body size fluctuations in fossil assemblages are common and can relate to a variety of mechanisms. The assemblage size shift hypothesis (H₁) was validated and despite the fluctuations observed, only the reduction observed at the Pli Toa boundary event is significant. By decomposing the assemblage size shift we were able to disentangle the mechanisms of belemnite body size change, which was almost exclusively driven by a within-lineage effect.

Out of the boundary crossers, the most abundant taxon, *Ps. longiformis*, is mainly driving the observed body size reduction, allowing us to validate the within-taxa size shift hypothesis (H₃). No

Formatted: Font: Not Bold

Formatted: Font: Palatino-Roman

Formatted: Comment Text

Formatted: Font: Palatino-Roman

Formatted: Font: Palatino-Roman

Formatted: Font: Palatino-Roman

Formatted: Font: (Default) Palatino-Roman, Font color: Auto, Pattern: Clear

Formatted: Comment Text, Justified

Formatted: Font: (Default) Palatino-Roman, Font color: Auto, Pattern: Clear

Formatted: Font: (Default) Palatino-Roman, Font color: Auto, Pattern: Clear

Formatted: Subscript

Formatted: Font: (Default) Palatino-Roman, Font color: Auto, Pattern: Clear

Formatted: Font: (Default) Palatino-Roman, Font color: Auto, Pattern: Clear

Formatted: Font: Italic

Formatted: Not Superscript/ Subscript

Formatted: Font: Italic

Formatted: Font: Italic

Formatted: Font: Not Italic

Formatted: Font: Not Italic

marked changes in the proportion of ontogenetic stages of *Ps. longiformis* are observed at the Pli-Toa boundary event, meaning that the largest within-lineage body-size change is not due to changes in the ontogenetic structure of the population, but rather to an adult body-size reduction. In conclusion, at the individual ontogenetic scale, we were able to validate the intraspecific size-shift hypothesis (H_3), across the Pli-Toa boundary event, finding a 51 % decrease in rostrum volume (21 % in median GM) of adult specimens of *Ps. longiformis*. In contrast, *P. bisulcata* body-size reduction is a combination of increased percentage of juveniles (increased mortality of juveniles) and reduction of adult body-size.

Despite the fact that ontogeny was not taken into account, the Lilliput effect *sensu lato* was already identified within particular cephalopod species in the Cleveland Basin, interpreted as a response to the deteriorating environmental conditions associated with the Early Toarcian crisis (15). However, this work does not explicitly take into account ontogeny (e.g., impact of the changes in the proportion of ontogenetic stages on body size increase of juvenile stage). Similarly, among extant squid, maturation at small size and young age is interpreted as a life-history strategy to cope with warming events (9).

The differential taxa response found in this study might indicate different environmental tolerances, related to individual physiology or life history strategies (8). The only important morphological difference detectable from the fossilised remains between *Ps. longiformis* and the other taxa is the presence of an epirostrum, a calcified structure which develops in late ontogenetic stages (85) as an extension of the orthorostrum. This has been interpreted that the development of an epirostrum facilitates the to indicate a rigid tail and shift of fin to middle of the body which might the adults to streamline the body and possibly facilitate the animal's movement in the water by gliding as or counterbalance to the development of specialized reproductive organs (85, 86). (Arkipkin, 2015 #145)(Doyle, 1985 #244)(Arkipkin, 2015 #145)(Doyle, 1985 #244)other benefits for reproduction This is interpreted as a reproduction structure, similar to the mantle extension observed in extant cephalopods (ref). The increased pCO_2 that characterises the environmental crisis of the Early Toarcian in Peniche (23) could directly (through calcification) or indirectly (through nutrient availability) have affected the calcification potential of the epirostrum. However, further data needs to be assembled in order assess the physiological and environmental constraints of the development of such a structure.

According to our results, a belemnite body-size reduction is observed at the Pli-Toa boundary event in Peniche. However, no major changes in the belemnite taxonomic composition are observed (Fig. 4). Despite being considered one of the early pulses of the Early Toarcian crisis— particularly in ammonoid cephalopods (46)— particularly in ammonoid cephalopods (REF) and the observed body size reduction,

the Pli-Toa boundary event in Peniche does not record major changes in the belemnite taxonomic composition (Fig. 4). This is consistent with data available from other regions at higher latitudes (34, 45, 60, 67), which demonstrate that the T-OAE, in contrast, corresponds to an important turnover in belemnite species (45). In fact, the T-OAE has had caused greater much more impact on marine biota than the Pli-Toa boundary event (e.g. (16, 30, 39, 40), namely in the Lusitanian Basin where planktonic and benthic organisms were largely affected (27, 28, 87-89)→). This highlights the deterioration of the environmental conditions during the Early Toarcian, starting at the Pli-Toa boundary event and culminating in the T-OAE, an interval barren interval of

Formatted: Font: Not Italic

Formatted: Font: Not Italic

Formatted: Comment Text

Formatted: Comment Text

Formatted: Font: Italic

Field Code Changed

Formatted: Not Highlight

Formatted: Not Highlight

Formatted: Not Highlight

Formatted: Font: (Default) Palatino-Roman, 10 pt, Font color: Black, Pattern: Clear

Formatted: Font: (Default) Palatino-Roman, 10 pt, Font color: Black, Pattern: Clear

Formatted: Not Highlight

Formatted: Not Highlight

Formatted: Not Highlight

Formatted: Not Highlight

Formatted: Not Highlight

Formatted: Not Highlight

Formatted: Not Highlight

Field Code Changed

Formatted: Not Highlight

Formatted: Not Highlight

Formatted: Not Highlight

Formatted: Not Highlight

Formatted: Highlight

belemnites in Peniche, as indicated by our results and previous studies (65) (rocha et al., 2016).
~~our study and other studies previously reported their presence;~~

Formatted: Not Highlight

Formatted: Not Highlight

Formatted: Highlight

Formatted: Space After: 8 pt, Adjust space between Latin and Asian text, Adjust space between Asian text and numbers

Formatted: Font: Not Bold

5. 1.2. The aftermath of Pli-Toa boundary event

~~Notwithstanding the decrease of seawater temperature during the early Polymorphum Zone, this interval is characterised by palaeoenvironmental perturbations as evidenced by the carbon isotopic record (23, 25). Moreover, seawater temperature is still higher than during the Late Pliensbachian (23) (Suan et al.). The conditions are, however, not as severe as during the Pli-Toa boundary event or during the onset of the T-OAE, which is also supported by the response of other groups of marine biota in the Peniche section (27, 28, 87-89).~~

~~The marked body size decrease of *P. bisulcata* (109 %) body size decreases during the aftermath of the Pli-Toa boundary event (109 % from bed P5 to bed P6, early Polymorphum Zone) occurred in tandem with a marked due to an increase of in the relative proportion of juveniles of that species (Fig. 6). This might indicate the temporary emigration of large adults (9) (Hoving et al. 2013) and/or higher juvenile mortality (Pauly 1980; Audzijonyte et al. 2018), due to unsuitability of the habitat, as interpreted for fish and cephalopod species (90, 91). However, we also cannot rule out the existence of smaller (stunted) adults (9) (Hoving et al. 2013) which do might not develop typical adult features (92) (Neige and Boletzky 2007) but this could only be disentangled with further studies, namely with a sclerochronological analysis.~~

Formatted: Not Highlight

Field Code Changed

Commented [KD5]: Pauly, D. (1980). On the interrelationships between natural mortality, growth parameters, and mean environmental temperature in 175 fish stocks. *ICES Journal of Marine Science*, 39(2), 175-192.
 Audzijonyte should already be cited as well as Hoving et al. 2013

Formatted: Font: Not Italic

Formatted: Font: Not Italic

The decrease in *P. bisulcata* body size in the aftermath of the Pli-Toa boundary event coincides with an increase in adult body size of *Ps. longiformis*, and with the disappearance of *Hastitidae* sp. indet. It is tempting to assume that the rapid recovery of *Ps. longiformis* potentially triggered a change in the belemnite population dynamics, potentially due to competition for food resources, causing the emigration of some species (*Hastitidae* sp. indet.) and the increased mortality and stress in early growth of others (*P. bisulcata*). ~~A shift in distribution to cooler latitudes with warming is considered one of the most important response in modern marine ecosystems (93, 94) (Poloczanska et al. 2013), and other factors also have been implicated to contribute to the decrease in size within individual species to communities (2, 91) (Daufresne et al. 2009; Audzijonyte et al. 2018). This is commonly observed in marine ecosystems among this and that (refs).~~

Furthermore, the rest of the Polymorphum Zone (i.e. from bed P6 to bed P9) is characterized by an increase in *Ps. longiformis* body size and abundance, (and also and less relative mortality of juveniles) and abundance, relatively to *P. bisulcata*. The latter becomes less abundant and maintains its adult body size, emphasizing a potential competition between the *Ps. longiformis* and *P. bisulcata*. However, the fact that there is no increase in the proportion of *Ps. longiformis* comparatively to the proportion of *P. bisulcata* from bed P5 to bed P6 (Fig. 6) is not entirely compatible with this interpretation and highlights the constraints of testing the effects of competition and other biotic parameters on body size in fossil assemblages.

~~Notwithstanding the decrease of seawater temperature during the early Polymorphum Zone, this interval is characterised by palaeoenvironmental perturbations as evidenced by the carbon isotopic record (23, 25). Moreover, seawater temperature is still higher than during the Late Pliensbachian. The conditions are, however, not as severe as during the Pli-Toa boundary event or during the onset of the T OAE, which is also supported by the response of other groups of marine biota in the Peniche section (27, 28, 80-82).~~

5.1.3. T-OAE

The interval corresponding to the onset of the T-OAE in Peniche corresponds to a belemnite gap, starting around the Polymorphum–Levisoni zones boundary (95). This is coincident with the beginning of the second mercury anomaly (55) and with warming (23), similarly to the lowermost Toarcian conditions, although associated with stronger carbon cycle perturbations (Fig. 1).

The belemnite gap could reflect inhospitable conditions in the Lusitanian Basin during the onset of the T-OAE. This is not the case during the Pli-Toa boundary event, which only caused changes in the dynamics of the population, namely in the ontogenetic structure due to temporary migration, and body size changes. This might suggest that coleoids – even those with flexible life history strategies, pre-adapted to such conditions, such as *Ps. longiformis* – could not cope with deteriorating conditions in subtropical epicontinental European basins, resulting in northward migration and/or local extinction during the T-OAE. This would also be consistent with the absence of belemnites in the Riff Mountains during the Levisoni Zone. In Morocco, the interval correspondent to the T-OAE is also barren of belemnites (compare (48, 96); Sanders et al., 2014; Bardin et al. 2014).

In fact, the deterioration of water-column conditions must have been worse during the T-OAE in comparison with the Pli-Toa boundary event, namely in terms of oxygenation of the water column, as evidenced by geochemical data (25), and the negative response of marine organisms in the Lusitanian Basin (27). Moreover, a similar belemnite response was identified in coeval north-western European basins, as the belemnite gap observed in Peniche overlaps with intervals of belemnite gap or belemnite abundance decrease in the Cleveland and Cardigan Bay, and Swabo-Franconian temperate basins. This has been previously interpreted as response to unfavourable anoxic euxinic water-column conditions (45, 49, 97), interpreted as response to unfavourable anoxic euxinic water-column conditions. In the Swabo-Franconian basin the belemnite assemblage get less diverse during the T-OAE (98) (Riegraf et al. 1984). In Morocco, the interval correspondent to the T-OAE is also barren of belemnites (Sanders et al. 2014).

Interestingly, the decrease in belemnite abundance, or disappearance, happens before the most severe conditions of the T-OAE are met in subtropical Peniche section (i.e. when the temperature was not the highest, see Fig. 1) and it is not preceded by a body-size decrease. On the contrary, during the middle and upper Polymorphum Zone (i.e., from bed P5 to bed P9) belemnite body-size increases at the assemblage scale. However, this is probably related to the fact that juveniles become rare due to unsuitability of habitat in these habitats that were potentially no longer suitable for juvenile specimens. This pattern and might indicate that combination of warming and deoxygenation affected belemnite physiology, reproduction, as well as competition for resources, even before the environmental nadir of the T-OAE. This highlights the importance of climate warming, in addition to regional anoxia, which have been argued to be a major constraint on belemnites abundance and distribution in the north-western basins of the Tethys Ocean (45, 49, 97).

However, although the Mediterranean basins were not anoxic, wider more widespread anoxia in other parts of the western Tethys (e.g., NW Europe) (99), could have indirectly

Formatted: Font: Not Bold

Formatted: Font color: Auto

Formatted: Font color: Auto

Formatted: Not Highlight

Formatted: Not Highlight

Formatted: Not Highlight

Formatted: Not Highlight

Field Code Changed

Formatted: Not Highlight

Formatted: Highlight

Formatted: Highlight

Formatted: Highlight

Formatted: Highlight

Formatted: Not Highlight

Formatted: Not Highlight

Formatted: Not Highlight

Formatted: Font: Not Bold

Formatted: Font: Not Bold

Formatted: Font: Not Bold

Formatted: Font: (Default) Palatino-Roman, 10 pt, Font color: Auto, Pattern: Clear, Not Highlight

Formatted: Font: (Default) Palatino-Roman, 10 pt, Font color: Auto, Pattern: Clear, Not Highlight

Formatted: Font: (Default) Palatino-Roman, 10 pt, Font color: Auto, Pattern: Clear, Not Highlight

Field Code Changed

Formatted: Font: (Default) Palatino-Roman, 10 pt, Font color: Auto, Pattern: Clear, Not Highlight

affected belemnites in the non-anoxic Mediterranean domain (Lusitanian basin and Morocco)-by restricting migratory routes and/or food supply and even might have inhibited their ability to move to cooler refuges when temperatures began to rise.

Belemnites temporarily re-appear in bed P10 (upper Levisoni Zone), after the onset of the T₂-OAE, coinciding approximately with the end of the T₂-OAE negative CIE (Fig. 1), with a poor record of *Acrocoelites*. This genus is known to appear and radiate in the north-eastern basins of the NW Tethys (34, 45, 46, 60, 66) after the T₂-OAE, replacing the genus *Passaloteuthis*, which has been interpreted as an evolutionary adaptation in terms of ecological preferences related to the stressful conditions of the T₂-OAE (49). Moreover, the change from the Late Pliensbachian/Early Toarcian Belemnitinae-dominated assemblage to the Middle Toarcian Megateuthidinae-dominated assemblage (together with the disappearance of smaller sized species such as *Hastitidae* sp. indet., *Parapassaloteuthis* sp. A, *P. milleri*, *Ps. longiformis*), might not only be related to ecological preferences but also to selective extinction, as pointed out by (7), who states that past extinction events were either nonselective or preferentially removed smaller-bodied taxa.

Formatted: Font: Not Italic

However, the lack of a continuous belemnite record during the Levisoni Zone in Peniche hampers a more detailed assessment. Further research in other basins is necessary to disentangle the extent of this phenomenon.

5.2. The effectrelationship of sedimentary facies onto belemnite body-sizebody-size

Formatted: Font: Not Bold

Despite the fact that both the magnitude of palaeoenvironmental changes relative to comparatively the rapid nature of these crisis events (100-102) is quite high on geological timescales (103) (Kemp et al. 2015) is quite high, of the studied events are considered to be geologically comparatively rapid (85-87)- their representation in the sedimentary record is dependent oninfluenced by facies evolution. Sedimentary facies dependent of several factors such as regional tectonics, climate, and sea level changes, which the role of these parameters control the sedimentation rate, which is not constant across the studied interval. We use lithology to account for changes in the facies (related with relative sea-level changes) and belemnite abundance to account for changes in the sedimentation rate, since both are thought to have an effect on morphological patterns (104).

Formatted: Not Highlight

Accumulations with abundant belemnites, particularly observed in beds P4 and P5, could be related to changes in sedimentation rate relative to the accumulation of belemnites (105). The lowermost Polymorphum Zone in Peniche is thought to correspond to a condensed interval (65, 88, 106). Our data reveals that belemnite body-sizebody-size is larger in beds with higher belemnite abundance (Fig. S10). Abundance, however, does not markedly change across the boundary, where the most significant size change is observed. Regarding lithology, belemnite body-sizebody-size is larger in marls in comparison with limestones but these effects are not strong (Fig. S9).

Our results indicate that, When studying the effect of environmental perturbations on belemnite body-sizebody-size, at the scale of the assemblage, lithology and abundance do not affect the pattern observed. Before and after correction, temperature is always the best parameter in explaining the body-sizebody-size variation. However, the significance is higher without

correction which is not surprising considering that residualising might also filter out potential ecological signals contained within the sedimentary properties in addition to the effect of preservation and collection biases. Even with the correction for the effects of sedimentary properties, and despite the existence of a significant model, the analysis failed in revealing a single model to explain belemnite ~~body-size~~ variation. Instead, several models are very similar to the model with the best AICc score. This ~~likely relates to highlights~~ the strong interaction between abiotic parameters ~~for which we have reasonable proxies~~ in driving ~~body-size~~ patterns during episodes of climate warming.

However, when analysing the data at species level (*P. bisulcata* and *Ps. longiformis*), the best model ~~in both species~~ explaining the ~~body-size~~ variation after correction is Hg/TOC_t as opposed to temperature without correction ~~in both species~~. This might be related to the smaller datasets, especially in the case of *P. bisulcata*, which does not allow ~~us~~ to distinguish between individual models. Either way, the increased warming, at the assemblage scale, and Hg/TOC anomalies, as drivers of individual species ~~body-size~~ variation, are not incompatible – it just shows that ~~changes in~~ warming generally correlate with decreasing sizes of belemnite assemblages across the Pli–Toa boundary and that the impact of “catastrophic” environmental perturbations triggered by volcanism are the most severe and probably work in concert in their impact on ~~body-size~~ decrease in individual species.

The lowermost Toarcian (base of Polymorphum Zone) in Peniche also coincides with a marine transgression (64, 65) which might have impacted on the size distribution, but differences in lithology and sedimentation rates could not clearly explain differences in our dataset. Furthermore, deepening would rather result in finding larger species (76), which would be consistent with our observations of finding larger specimens in marls. But a ~~body-size~~ decrease is seen across the boundary. Further analyses across different parts of the ~~Lusitanian B~~ basin would be necessary to disentangle its potentially less subtle effects in shallower sections. ~~As we have no good and/or uncorrelated proxies for other factors, such as like productivity or ocean acidification, we cannot rule out that these or other factors had an important role in explaining belemnite body-size future studies might even find other models to be better in explaining the variation. However, warming is expected to have impact on all of these environmental changes-parameters.~~

5.3. Environmental drivers of ~~body-size~~ fluctuations

The Pli–Toa boundary corresponds to the first episode of stress in the Early Toarcian palaeoenvironmental crisis (23) and has been associated with the first pulse of Kar~~oo~~–Ferrar volcanism (55). This had a major effect on climate, disturbing the entire ocean–atmosphere system, by causing carbon cycle perturbations, ~~and~~ fluctuations in the seawater temperature, ~~as well as the start of more nd-widespread deoxygenation and organic matter burial~~ (107). Our results are consistent with the hypothesis that climate is an important driver of the belemnite ~~body-size~~ variation at the assemblage scale, with a decreasing ~~body-size~~ associated with increasing seawater temperature (Fig. S8). Salinity could, however, partially affect the interpretation of the oxygen isotopic data, particularly in north–western European basins, where the input of brackish and nutrient–rich Arctic surface waters seems to affect the signal (99, 108). However, decreased salinity has been deemed less significant in Peniche than warming and carbon cycle perturbations, based on phytoplankton communities (28).

Formatted: Font: Not Bold

Formatted: Font: Not Bold

Formatted: Font: (Default) Palatino-Roman, 10 pt, Font color: Auto, English (United States), Pattern: Clear

Formatted: English (United States)

R. Soc. open sci. article template

Furthermore, modelling suggests that, at these palaeocoordinates, salinity changes are close to zero (99, 108).

This reduction in adult size of *Ps. longiformis* and *P. bisulcata* – the Lilliput effect *sensu stricto* – can potentially be compared with the decrease in adult mantle length of extant squid during rapid warming events (9, 11) *interpreted to be related with accelerated life-histories of squid, increasing their growth rates and shortening their life-spans. Individual squid might also require more food per unit body-size, require more oxygen for faster metabolisms and have a reduced capacity to cope without food* (77) (Peel, 2008 #144) (Peel, 2008 #144) (Peel and Jackson 2008) *because of this and that*. To what degree the decrease in adult size relates with direct effects of warming on physiology (development rate, metabolism, respiration) or *instead rather* indirect effects of environmental changes on resource acquisitions affecting growth and early development is still under debate (8, 109).

Formatted: Font: Italic

Formatted: Highlight

Apart from increasing seawater temperature, the mercury anomalies and carbon cycle perturbations at the Pli-Toa boundary event in Peniche, also coincide with increase weathering (25), local decreased productivity (27, 28), increasing $p\text{CO}_2$ (and potentially ocean acidification) and decreasing dissolved oxygen (23). All of these factors are expected to work in concert in driving marine organisms body-size (6) and our analysis of belemnite body-size is consistent with that. This is also reflected in the strong collinearity of environmental and biotic proxies related to these effects.

Strontium isotopes from Peniche support the idea of increased weathering during the Pli-Toa boundary warming event and potentially nutrient influx (25), which is also consistent with the interpretation of enhanced mercury concentration in the sediments. However, *counterintuitively* this warming event is interpreted to lead to a decrease in primary productivity, with a decrease in *both* phytoplankton abundance (27, 28, 43). This emphasizes how warming and related stressors might *in turn* affect the size of primary consumers and potentially *then* other levels of the food chain, such as predators like cephalopods. *This is also consistent with the increased extinction risk for pelagic predators modelled for various hyperthermal events, including end-Permian mass extinction and the Early Toarcian crisis* (7, 44) (Payne et al. 2016; Dunhill et al. 2018).

Additionally, the fact that ~~Dwarfing~~ has been observed *as a* response *to starving to lack of food* in ~~the laboratory~~ *experiments supports this interpretations, but so far not in the wild cephalopods* (110). ~~In addition~~ *Nonetheless*, macroecological studies show that *spatial body-size* ~~body-size~~ distribution in extant cephalopods is better explained by seawater temperature than productivity (76, 111).

Formatted: Not Highlight

The negative CIE observed during the Early Toarcian is interpreted to be related to volcanic outgassing either directly by the rapid input of isotopically light carbon into the atmosphere-hydrosphere system (54), or indirectly by triggering various sources of isotopically light carbon (19). This would not only have caused rapid climate warming, but also increased $p\text{CO}_2$ and decreased pH of the seawater. Cephalopods are usually interpreted to be quite resistant to ocean acidification in comparison with other marine organisms – cuttlefish even increase calcification. *This is interpreted to be related to their strong acid-base regulatory abilities, and the fact that because the cuttlebone is a fully internal structure* ~~of this and that~~ (112, 113). However, the limited data available hitherto seem to indicate that the effects of acidification might be more severe in early ontogeny, resulting in pathologies (113, 114). Nonetheless, no clear signs of aberrant *early* development were observed in the studied rostra. More importantly, there are no direct proxies for marked ocean acidification during the Early Toarcian in the Lusitanian Basin

Formatted: Not Highlight

Formatted: Not Highlight

Field Code Changed

and only indirect evidences, such as calcareous nannofossils (*Schizosphaerella*) size reduction. However, this has been interpreted as an indirect consequence of $p\text{CO}_2$ increase, due to changes in the climate and sea-level (27), rather than a direct cause of high $p\text{CO}_2$.

Together with the deposition of organic-rich black shales, the Early Toarcian negative CIE has been traditionally interpreted to reflect increased anoxia in the ocean (115, 116). Irrespective of the potential local or regional overprint of increasing stratification due to basin restriction or arctic water input, the perturbations at the Pli-Toa boundary event in Peniche are thought to reflect the start of decreasing oxygenation of seawater while the T-OAE perturbations represent the global peak of anoxia (107). Nonetheless, there is no evidence for bottom-water anoxia in the Lusitanian Basin (25, 117), in contrast with the north-western European basins (118). Despite that fact, less severe deoxygenation might have played a role in belemnite body-size and distribution, since increased seawater temperature results in reduced oxygen availability relative to demand (119).

The strong collinearity between biotic and abiotic factors available in the literature for the Peniche section (Fig. S4) hampers an analysis of their individual effect on belemnite body-size. Nonetheless, this study allows investigation the relationship of belemnite body-size reduction with direct (temperature) and indirect environmental stressors (e.g. deoxygenation, ocean acidification, input of nutrients) associated with climate warming, despite the complex interaction between biotic and abiotic factors, highlighting the importance of taking into account different scales of organization.

Nonetheless, this study allows to distinguish direct and indirect effects of climate warming on belemnite body size reduction, despite the complex interaction between biotic and abiotic factors, highlighting the importance of taking into account different scales of organization.

Formatted: Font: Palatino-Roman, 10 pt, Font color: Auto, Pattern: Clear

Formatted: Font: Palatino-Roman, 10 pt, Font color: Auto, Pattern: Clear

Formatted: Font: Palatino-Roman, 10 pt, Font color: Auto, Pattern: Clear

Formatted: Font: Palatino-Roman, 10 pt, Font color: Auto, Pattern: Clear

Commented [i6]: not sure what you're considering direct vs indirect?

6. Conclusion

We document for the first time a median body-size decrease of belemnites across the Pli-Toa boundary event at different scales of palaeobiological organisation in the Peniche reference section. We find no evidence for a major taxonomic turnover (similar to results in more high-latitude sections) and the decrease is mainly driven by a decrease in adult size of the dominant taxon *Ps. longiformis* – Lilliput effect sensu stricto. This phenomenon coincides with the onset of the first pulse of the Upper Pliensbachian – Lower Toarcian palaeoenvironmental crisis, probably triggered by volcanism of the Karoo-Ferrar large igneous province. Our results indicate that climate warming best explains the body-size fluctuations observed, although the interplay with perturbations of the carbon cycle and other environmental factors possibly triggered by increased volcanogenic outgassing are evident. Our results suggest that morphological responses precede extinction pulses in belemnites (i.e. during the T-OAE) in the Lusitanian Basin and highlight that decreasing adult body-size might rather be a life-history strategy to deal with temporarily deteriorating conditions related to warming than being a result of taxonomic turnover. During/Towards the T-OAE, belemnites disappear from the Lusitanian Basin and probably subtropical basins (Morocco) suggesting their local extinction and/or migration. Furthermore, changes within the belemnite assemblage, such as competition

Formatted: Font: Italic

and emigration dynamics, seem to play a role in explaining belemnite body-size variation, albeit minor.

Acknowledgments

We thank Birgit Leipner-Mata and Manuel Blank for helping in belemnite preparation, and Benjamin Gügel, Christian Schulbert and Martina Schlott for helping with scanning the specimens. We thank Manuel Steinbauer, Carl Reddin, Wolfgang Kiessling and Vanessa Roden as well as the referees, for the valuable comments on the manuscript.

Funding Statement

This is a contribution to the DFG Research Unit FOR 2332 (grant number Ba 5148/1-1 to KDB) TERSANE and to the IGCP 655 (IUGS-UNESCO).

Data Accessibility

The studied specimens are stored in the Science Museum (Museu da Ciência, University of Coimbra, Portugal) and the 3D data are available within Zenodo (<https://doi.org/10.5281/zenodo.3459233>). Supplementary figures and tables and the R script used for the statistical analyses have been uploaded as part of the supplementary material.

The datasets supporting this article have been uploaded as part of the Supplementary Material, together with supplementary figures and tables.

The specimens are stored in the Science Museum (Museu da Ciência, University of Coimbra, Portugal) and the 3D data will be stored on Zenodo.

Formatted: Highlight

Competing Interests

We declare we have no competing interests.

Authors' Contributions

PR participated in the design of the study, carried out the data collection and analysis, interpreted the results and drafted the manuscript. KDB designed and coordinated the study, participated in the data analysis and helped interpreting the results and drafting the manuscript. The fieldwork was carried out by PR, KDB and LVD. PN helped with data collection and preliminary data analysis. LVD helped interpreting the results. RW helped with the taxonomic work. All authors contributed to the writing process and gave final approval for publication.

References

1. Schmidt DN, Lazarus D, Young JR, Kucera M. Biogeography and evolution of body size in marine plankton. *Earth-Science Reviews*. 2006;78(3):239-66.
2. Daufresne M, Lengfellner K, Sommer U. Global warming benefits the small in aquatic ecosystems. *Proceedings of the National Academy of Sciences*. 2009;106(31):12788-93.
3. Gardner JL, Peters A, Kearney MR, Joseph L, Heinsohn R. Declining body size: a third universal response to warming? *Trends in ecology & evolution*. 2011;26(6):285-91.
4. Sheridan JA, Bickford D. Shrinking body size as an ecological response to climate change. *Nature Climate Change*. 2011;1(8):401-6.

[revised manuscript text omitted]

56. Moore JK, Fu W, Primeau F, Britten GL, Lindsay K, Long M, et al. Sustained climate warming drives declining marine biological productivity. *Science (New York, NY)*. 2018;359(6380):1139-43.
57. Caswell BA, Paine M, Frid CLJ. Seafloor ecological functioning over two decades of organic enrichment. *Marine Pollution Bulletin*. 2018;136:212-29.
58. Wilson J, Monteiro F, Schmidt D, Ward B, Ridgwell A. Linking marine plankton ecosystems and climate: A new modeling approach to the warm early Eocene climate. *Paleoceanography and Paleoclimatology*. 2018;33(12):1439-52.
59. Ilyina T, Heinze M. Carbonate Dissolution Enhanced by Ocean Stagnation and Respiration at the Onset of the Paleocene-Eocene Thermal Maximum. *Geophysical Research Letters*. 2019;46(2):842-52.
60. Doyle P. The British Toarcian (lower Jurassic) Belemnites: Part 1. Monograph of the Palaeontographical Society. 1990;584:1-49.
61. Harazim D, Van De Schootbrugge BAS, Sorichter K, Fiebig J, Weug A, Suan G, et al. Spatial variability of watermass conditions within the European Epicontinental Seaway during the Early Jurassic (Pliensbachian–Toarcian). *Sedimentology*. 2013;60(2):359-90.
62. McArthur J, Doyle P, Leng M, Reeves K, Williams C, Garcia-Sanchez R, et al. Testing palaeo-environmental proxies in Jurassic belemnites: Mg/Ca, Sr/Ca, Na/Ca, $\delta^{18}\text{O}$ and $\delta^{13}\text{C}$. *Palaeogeography, Palaeoclimatology, Palaeoecology*. 2007;252(3-4):464-80.
63. van de Schootbrugge B, McArthur JM, Bailey TR, Rosenthal Y, Wright JD, Miller KG. Toarcian oceanic anoxic event: An assessment of global causes using belemnite C isotope records. *Paleoceanography*. 2005;20(3):PA3008.
64. Duarte LV. Lithostratigraphy, sequence stratigraphy and depositional setting of the Pliensbachian and Toarcian series in the Lusitanian Basin (Portugal). In: Rocha RB, editor. *The Peniche Section (Portugal) Contributions to the definition of the Toarcian GSSP: International Subcommission on Jurassic Stratigraphy*; 2007. p. 17-23.
65. Rocha RB, Mattioli E, Duarte LV, Pittet B, Elmi S, Mouterde R, et al. Base of the Toarcian Stage of the Lower Jurassic defined by the Global Boundary Stratotype Section and Point (GSSP) at the Peniche section (Portugal). *Episodes Journal of International Geoscience*. 2016;39(3):460-81.
66. Doyle P. The British Toarcian (lower Jurassic) Belemnites: Part 2. Monograph of the Palaeontographical Society. 1992;587:50-79.
67. Pinard J-D, Weis R, Neige P, Mariotti N, Di Cencio A. Belemnites from the Upper Pliensbachian and the Toarcian (Lower Jurassic) of Tournadous (Causses, France). *Neues Jahrbuch für Geologie und Paläontologie-Abhandlungen*. 2014;273(2):155-77.
68. Tomašových A, Schlögl J, Biroň A, Hudáčková N, Mikuš T. Taphonomic Clock and Bathymetric Dependence of Cephalopod Preservation in bathyal, sediment-starved environments. *PALAIOS*. 2017;32(3):135-52.
69. Rita P, De Baets K, Schlott M. Rostrum size differences between Toarcian belemnite battlefields. *Fossil Record*. 2018;21(1):171-82.
70. Monks N, Hardwick JD, Gale AS. The Function of the Belemnite Guard. *Paläontologische Zeitschrift*. 1996;70(3):425.
71. Benito MI, Reolid M, Viedma C. On the microstructure, growth pattern and original porosity of belemnite rostra: insights from calcitic Jurassic belemnites. *Journal of Iberian Geology*. 2016;42(2):201.

Formatted: Portuguese (Portugal)

Formatted: German (Germany)

72. Hoffmann R, Richter DK, Neuser RD, Jöns N, Linzmeier BJ, Lemanis RE, et al. Evidence for a composite organic–inorganic fabric of belemnite rostra: Implications for palaeoceanography and palaeoecology. *Sedimentary Geology*. 2016;341:203-15.
73. Reitner J, Ulrichs M. Echte Weichteilbelemniten aus dem Untertoarcium (Posidonienschiefer) Südwestdeutschlands. *Neues Jahrbuch für Geologie und Paläontologie*. 1983;165(3):450-65.
74. Klug C, Schweigert G, Fuchs D, Kruta I, Tischlinger H. Adaptations to squid-style high-speed swimming in Jurassic belemnitids. *Biology Letters*. 2016;12(1):20150877.
75. Jenny D, Fuchs D, Arkhipkin AI, Hauff RB, Fritschi B, Klug C. Predatory behaviour and taphonomy of a Jurassic belemnoid coleoid (Diplobelida, Cephalopoda). *Sci Rep*. 2019;9(1):7944-.
76. Rosa R, Gonzalez L, Dierssen HM, Seibel BA. Environmental determinants of latitudinal size-trends in cephalopods. *Marine Ecology Progress Series*. 2012;464:153-65.
77. Pecl GT, Jackson GD. The potential impacts of climate change on inshore squid: biology, ecology and fisheries. *Reviews in Fish Biology and Fisheries*. 2008;18(4):373-85.
78. Rego BL, Wang SC, Altiner D, Payne JL. Within- and among-genus components of size evolution during mass extinction, recovery, and background intervals: a case study of Late Permian through Late Triassic foraminifera. *Paleobiology*. 2012;38(4):627-43.
79. Oksanen J, Blanchet FG, Kindt R, Legendre P, Minchin P, O'Hara RB, et al. Vegan: community ecology package. R package version 2.5-2 (<https://CRAN.R-project.org/package=vegan>). 2018.
80. R Development Core Team. R: a language and environment for statistical computing. Vienna, Austria: R Foundation for Statistical Computing. Retrieved from <http://www.R-project.org>. 2018.
81. Jakob EM, Marshall SD, Uetz GW. Estimating fitness: a comparison of body condition indices. *Oikos*. 1996;77(1):61-7.
82. Box GEP, Jenkins GM. *Time Series Analysis: Forecasting and Control*: Holden-Day; 1994.
83. Pinheiro J BD, DebRoy S, Sarkar D, R Core Team. nlme: Linear and Nonlinear Mixed Effects Models. R package version 3.1-137 (<https://CRAN.R-project.org/package=nlme>). 2018.
84. Spiess A-N. qpcR: Modelling and Analysis of Real-Time PCR Data. R package version 1.4-1 (<https://cran.r-project.org/web/packages/qpcR/>). 2018.
85. Arkhipkin A, Weis R, Mariotti N, Shcherbich Z. 'Tailed' cephalopods. *Journal of Molluscan Studies*. 2015;81(3):345-55.
86. Doyle P. Sexual dimorphism in the belemnite *Youngibelus* from the Lower Jurassic of Yorkshire. *PALEONTOLOGY*. 1985;28(1):133-46.
87. Comas-Rengifo MJ, Duarte LV, Felix FF, Joral FG, Goy A, Rocha RB. Latest Pliensbachian–Early Toarcian brachiopod assemblages from the Peniche section (Portugal) and their correlation. *Episodes*. 2015;38(1):2-8.
88. Rita P, Reolid M, Duarte LV. Benthic foraminiferal assemblages record major environmental perturbations during the Late Pliensbachian–Early Toarcian interval in the Peniche GSSP, Portugal. *Palaeogeogr, Palaeoclimatol, Palaeoecol*. 2016;454:267-81.
89. Reolid M, Duarte LV, Rita P. Changes in foraminiferal assemblages and environmental conditions during the T-OAE (Early Jurassic) in the northern Lusitanian Basin, Portugal. *Palaeogeography, Palaeoclimatology, Palaeoecology*. 2019.
90. Pauly D. On the interrelationships between natural mortality, growth parameters, and mean environmental temperature in 175 fish stocks. *ICES Journal of Marine Science*. 1980;39(2):175-92.
91. Audzijonyte A, Richards SA. The Energetic Cost of Reproduction and Its Effect on Optimal Life-History Strategies. *The American Naturalist*. 2018;192(4):E150-E62.

Formatted: German (Germany)

Formatted: Portuguese (Portugal)

92. Neige P, Boletzky SV. Morphometrics of the shell of three *Sepia* species (Mollusca: Cephalopoda): intra- and interspecific variation. *Zoologische Beiträge*. 1997;38:137-56.
93. Poloczanska ES, Brown CJ, Sydeman WJ, Kiessling W, Schoeman DS, Moore PJ, et al. Global imprint of climate change on marine life. *Nature Climate Change*. 2013;3(10):919.
94. Reddin CJ, Kocsis ÁT, Kiessling W. Marine invertebrate migrations trace climate change over 450 million years. *Global ecology and biogeography*. 2018;27(6):704-13.
95. Duarte LV, Comas-Rengifo MJ, Hesselbo S, Mattioli E, G. Suan, Baker S, et al. The Toarcian Oceanic Anoxic Event at Peniche. An exercise in integrated stratigraphy - Stop 1.3. Field trip Guidebook: The Toarcian Oceanic Anoxic Event in the Western Iberian Margin and its context within the Lower Jurassic evolution of the Lusitanian Basin 2018. p. 54.
96. Bardin J, Rouget I, Benzaggagh M, Fürsich FT, Cecca F. Lower Toarcian (Jurassic) ammonites of the South Riffian ridges (Morocco): systematics and biostratigraphy. *Journal of Systematic Palaeontology*. 2015;13(6):471-501.
97. Xu W, Ruhl M, Jenkyns HC, Leng MJ, Huggett JM, Minisini D, et al. Evolution of the Toarcian (Early Jurassic) carbon-cycle and global climatic controls on local sedimentary processes (Cardigan Bay Basin, UK). *Earth and Planetary Science Letters*. 2018;484:396-411.
98. Riegraf W, Werner G, Lörcher F. *Der Posidonienschiefer: Biostratigraphie, Fauna und Fazies des südwestdeutschen Untertoarciums (Lias E)*. Stuttgart: Enke; 1984. 195 p.
99. Ruvalcaba Baroni I, Pohl A, van Helmond NA, Papadomanolaki NM, Coe AL, Cohen AS, et al. Ocean circulation in the Toarcian (Early Jurassic): A key control on deoxygenation and carbon burial on the European Shelf. *Paleoceanography and Paleoclimatology*. 2018;33(9):994-1012.
100. Boulila S, Galbrun B, Sadki D, Gardin S, Bartolini A. Constraints on the duration of the early Toarcian T-OAE and evidence for carbon-reservoir change from the High Atlas (Morocco). *Global and Planetary Change*. 2019;175:113-28.
101. Huang C, Hesselbo SP. Pacing of the Toarcian Oceanic Anoxic Event (Early Jurassic) from astronomical correlation of marine sections. *Gondwana Research*. 2014;25(4):1348-56.
102. Suan G, Pittet B, Bour I, Mattioli E, Duarte LV, Mailliot S. Duration of the Early Toarcian carbon isotope excursion deduced from spectral analysis: Consequence for its possible causes. *Earth and Planetary Science Letters*. 2008;267(3):666-79.
103. Kemp DB, Eichenseer K, Kiessling W. Maximum rates of climate change are systematically underestimated in the geological record. *Nature communications*. 2015;6:8890.
104. Holland SM. The quality of the fossil record: a sequence stratigraphic perspective. *Paleobiology*. 2000:148-68.
105. Doyle P, Macdonald DI. Belemnite battlefields. *Lethaia*. 1993;26(1):65-80.
106. Pittet B, Suan G, Lenoir F, Duarte LV, Mattioli E. Carbon isotope evidence for sedimentary discontinuities in the lower Toarcian of the Lusitanian Basin (Portugal): Sea level change at the onset of the Oceanic Anoxic Event. *Sedimentary geology*. 2014;303:1-14.
107. Them TR, Gill BC, Caruthers AH, Gerhardt AM, Gröcke DR, Lyons TW, et al. Thallium isotopes reveal protracted anoxia during the Toarcian (Early Jurassic) associated with volcanism, carbon burial, and mass extinction. *Proceedings of the National Academy of Sciences*. 2018;115(26):6596-601.
108. Dera G, Donnadiou Y. Modeling evidences for global warming, Arctic seawater freshening, and sluggish oceanic circulation during the Early Toarcian anoxic event. *Paleoceanography and Paleoclimatology*. 2012;27(2).

Formatted: German (Germany)

109. Audzijonyte A, Barneche DR, Baudron AR, Belmaker J, Clark TD, Marshall CT, et al. Is oxygen limitation in warming waters a valid mechanism to explain decreased body sizes in aquatic ectotherms? *Global Ecology and Biogeography*. 2019;28(2):64-77.
110. Boletzky Sv. Effets de la sous-nutrition prolongée sur le développement de la coquille de *Sepia officinalis* L.(Mollusca, Cephalopoda). *Bulletin de la Société zoologique de France*. 1974;99(4):667-73.
111. Pauly D, Kinne O. Gasping fish and panting squids: oxygen, temperature and the growth of water-breathing animals: International Ecology Institute Oldendorf/Luhe, Germany; 2010.
112. Gutowska MA, Melzner F, Pörtner HO, Meier S. Cuttlebone calcification increases during exposure to elevated seawater pCO₂ in the cephalopod *Sepia officinalis*. *Marine Biology*. 2010;157(7):1653-63.
113. Dorey N, Melzner F, Martin S, Oberhänsli F, Teyssié J-L, Bustamante P, et al. Ocean acidification and temperature rise: effects on calcification during early development of the cuttlefish *Sepia officinalis*. *Marine Biology*. 2013;160(8):2007-22.
114. Lacoue-Labarthe T, Reveillac E, Oberhänsli F, Teyssié J-L, Jeffree R, Gattuso J. Effects of ocean acidification on trace element accumulation in the early-life stages of squid *Loligo vulgaris*. *Aquatic Toxicology*. 2011;105(1-2):166-76.
115. Ikeda M, Hori RS, Ikehara M, Miyashita R, Chino M, Yamada K. Carbon cycle dynamics linked with Karoo-Ferrar volcanism and astronomical cycles during Pliensbachian-Toarcian (Early Jurassic). *Global and Planetary Change*. 2018;170:163-71.
116. Jenkyns HC. Geochemistry of oceanic anoxic events. *Geochemistry, Geophysics, Geosystems*. 2010;11(3).
117. Rodríguez-Tovar FJ, Miguez-Salas O, Duarte LV. Toarcian Oceanic Anoxic Event induced unusual behaviour and palaeobiological changes in *Thalassinoides* tracemakers. *Palaeogeography, Palaeoclimatology, Palaeoecology*. 2017;485:46-56.
118. McArthur JM, Algeo TJ, van de Schootbrugge B, Li Q, Howarth RJ. Basinal restriction, black shales, Re-Os dating, and the Early Toarcian (Jurassic) oceanic anoxic event. *Paleoceanography*. 2008;23(4).
119. Verberk WCEP, Atkinson D. Why polar gigantism and Palaeozoic gigantism are not equivalent: effects of oxygen and temperature on the body size of ectotherms. *Functional Ecology*. 2013;27(6):1275-85.
120. Cárdenas AL, Harries PJ. Effect of nutrient availability on marine origination rates throughout the Phanerozoic eon. *Nature Geoscience*. 2010;3(6):430.

Formatted: Portuguese (Portugal)

Tables

Table 1

Geologic proxy [source]	Environmental variable	Basis	Interpretation in the context of the LB	Theoretical controls on body-size size
Brachiopod stable oxygen isotope ratio ($\delta^{18}O$) [(23)]	Dominantly bottom seawater temperature	Temperature-dependent isotopic fractionation between carbonate minerals and seawater (120)	High temperatures during negative excursions	Increasing temperature through various mechanisms including increase of both growth and development rates are expected to lead to a smaller adult body-size size (8)
$\delta^{13}C_{carb}$ [(25)]	Carbon cycle perturbations related with anoxia	Isotopic fractionation between photosynthesisers and seawater. Photosynthesisers remove light C^{12} from the seawater and this is incorporated into the sea bottom sediment after burial as organic carbon (116).	Negative excursions during the Early Toarcian reflect enhanced burial of organic carbon during the zenith of anoxia (107). They could also be influenced by increased primary production but are interpreted to reflect rather widespread oxygen depletion as primary productivity is interpreted to have dropped during these intervals (27, 28)	At the edge of an organism's temperature range, growth is usually impaired by insufficient energy or oxygen supply, decreasing both growth rate and body-size size at any developmental stage (8).
Hg/TOC ratio [(55)]	Volcanogenic outgassing	Volcanism represents a source of mercury to the atmosphere. Due to rain or runoff, mercury moves from the terrestrial realm/ atmosphere to the marine realm in mineral form or adsorbed in detrital organics. Hg burial is limited by low abundance and/or burial of organic matter or sulphids that would scavenge aqueous Hg (53, 55)	Hg anomalies (high Hg/TOC ratio) in the sediment are interpreted to represent markers of volcanism in distal sections. They have been interpreted to reflect volcanic outgassing, but their interpretation in some distal sections might be more complex (53).	Increasing pCO_2 , combined with increasing temperature, causes a decrease in both dissolved oxygen and pH of the seawater. In association with other factors related to rapid warming, such as weathering and eutrophication, these factors are expected to lead to a body-size size decrease due to their direct effects on the availability of food resources (6).

Table 2

Model no.	Δ	AIC c scores	ANOVA p-value
-----------	----------	--------------	---------------

Assemblage				
GM ~ 1	null	5.09	1921.87	–
GM ~ $\delta^{18}\text{O}$	5	0	1920.92	0.0849
GM ~ $\delta^{18}\text{O} + \delta^{13}\text{C}_{\text{carb}}$	2	1.99	1922.91	
GM ~ Hg/TOC + $\delta^{13}\text{C}_{\text{carb}}$	3	1.91	1922.84	
Ps. longiformis				
GM ~ 1	null	2.51	671.37	–
GM ~ Hg/TOC	6	0	668.87	0.0327
GM ~ $\delta^{18}\text{O} + \delta^{13}\text{C}_{\text{carb}}$	2	0.87	669.74	
GM ~ Hg/TOC + $\delta^{18}\text{O}$	4	1.31	670.17	
GM ~ $\delta^{18}\text{O}$	5	0.50	669.37	
P. bisulcata				
GM ~ 1	null	-1.46	607.97	–
GM ~ Hg/TOC	6	0	609.43	0.4306
GM ~ $\delta^{13}\text{C}_{\text{carb}} + \text{Hg/TOC}$	3	1.87	611.30	
GM ~ $\delta^{18}\text{O}$	5	0.35	609.78	
GM ~ $\delta^{13}\text{C}_{\text{carb}}$	7	0.58	610.01	

Table 3

scale	model no.	coefficients	value	std.error	t-value	p-value	degrees of freedom	residual
Assemblage	5	Intercept	2.44	1.48	1.64	0.1019	340	338
		$\delta^{18}\text{O}$	2.04	1.19	1.72	0.061	–	
Ps. longiformis	15	Intercept	0.31	0.23	1.32	0.1904	160	158
		Hg/TOC	-2	0.93	-2.14	0.0335	–	
P. bisulcata	5	Intercept	0.43	0.94	0.46	0.6458	101	99
		Hg/TOC	-3.15	4.02	-0.78	0.4358	–	

Figures

Figure 1

Figure 2

Figure 3

Figure 4

Figure 5

Figure 6

Figure and Table captions

Figure 1 –Variation of belemnite ~~body size~~body-size (GM, assemblage scale), absolute abundance (no. of belemnites/m²) and stratigraphic ranges of species, compared with the variation of -the analysed geologic proxies (mercury concentration, Hg/TOC (24), carbon and oxygen isotopes, $\delta^{13}\text{C}_{\text{carb}}$ and $\delta^{18}\text{O}$ (22, 23)) and sequence stratigraphy data (50) from the Upper Pliensbachian-Lower Toarcian of Peniche. The shaded area highlights the Pli-Toa boundary event. The bed numbers in square brackets correspond to the beds that were merged due to sample size constraints, regarding the belemnite body-size analysis. See Fig. S2 for details on belemnite abundance. See Fig. 5 for more information on lithology. Ammonite Zones: Emaciatum Zone = *Emaciatoceras emaciatum*; Polymorphum Zone = *Dactyloceras polymorphum*; Levisoni Zone = *Hildaites levisoni*. BG=belemnite gap; GM = geometric mean. T=Transgressive; R=Regressive.

Figure 2– Conceptual scheme depicting the tested hypotheses regarding the mechanisms behind belemnites ~~body size~~body-size reduction, at different palaeobiological scales of organisation.

Figure 3 – A: Venn diagram depicting the partition of belemnites ~~body size~~body-size variation between taxonomic composition, bed separation and ontogeny. The values between parentheses correspond to the whole time-series and the values without parentheses correspond to the Pli-Toa boundary. See “materials and methods section” for more details on the variation partitioning methodology. B: Effect on ~~body size~~body-size change at the assemblage and species scale of organisation. The bed pairs in the horizontal axis represent to pairwise comparisons. See Table S6 for details. For a correct interpretation of the right-side scale, the bottom of the arrowhead should be considered, rather than the tip of the arrow. Note that the x-axis depicts comparisons of consecutive pairs of beds (not to scale). BG = belemnite gap; U test= Mann-Whitney U test; K-S test = Kolmogorov-Smirnov test.

Figure 4 – Relative frequency of the different taxa comprising the Upper Pliensbachian-Lower Toarcian Peniche belemnite assemblage. Note that determinable incomplete specimens were also taken into account and so sample size (n) differs from Figs. 4 and 5. The error bars correspond to the 95 % confidence interval of the relative frequency of the species *Ps. longiformis*/*Bairdowius* sp. A, and *P. bisulcata*, the most abundant taxa.

Figure 5 –A: Lithology, belemnite ~~body size~~body-size variation (violin plots) (GM, geometric mean) and , proportional ~~body size~~body-size change across the studied interval in Peniche. B: and Relative frequency of ontogenetic stages at the assemblage scale across the studied interval in Peniche. Note that the stratigraphic log is not drawn to scale, for real thickness of beds, see Fig. 1. The error bars on the right side graph correspond to the 95 % confidence interval of the juveniles and adults ratio. BG = belemnite gap; GM = geometric mean.

Figure 6 – Belemnite ~~body size~~body-size variation (GM, geometric mean), proportional body-size change, relative frequency of ontogenetic stages and sample size (n) across the Upper Pliensbachian-Lower Toarcian of Peniche at species scale (*P. bisulcata* and *Ps. longiformis*). *Passaloteuthis* genus was used to calculate the relative frequency of ontogenetic stages of *P. bisulcata* due to the difficulty of a species level classification of the juvenile specimens of *Passaloteuthis* genus. The error bars correspond to the 95 % confidence interval. The error bars correspond to the 95 % confidence interval of the juveniles and adults ratio. The error bars correspond to the 95 % confidence interval. For sample size, see Fig. S6. GM = geometric mean.

Formatted: Font: Not Bold

Formatted: Portuguese (Portugal)

Formatted: Font: (Default) Palatino-Roman, Not Italic, Font color: Auto, Portuguese (Portugal), Pattern: Clear

Formatted: Portuguese (Portugal)

Formatted: Font: Italic, Portuguese (Portugal)

Formatted: Portuguese (Portugal)

Formatted: Font: Not Italic, Font color: Auto

Formatted: Font: Not Italic

Formatted: Not Highlight

Formatted: Font: Italic

Formatted: Font: Italic

Formatted: Font: Not Italic, Font color: Auto

Formatted: Font: Not Italic, Font color: Auto

Table 1 – Environmental constraints (abiotic parameters) based on their geological proxies and their theoretical role in mediating body-size changes and their interpretation in the context of the Lusitanian Basin (LB).

Table 2 – AICc ranking of models describing the effect of palaeotemperature ($\delta^{18}\text{O}$), carbon cycle perturbations ($\delta^{13}\text{C}_{\text{carb}}$) and volcanism (Hg/TOC) on belemnites body-size (GM) for the assemblage and species scale (*Ps. longiformis* and *P. bisulcata*), corrected for the effects of sedimentary properties (lithology and belemnite abundance). Only the models with $\Delta < 2$ are listed. See TS4 for the full list of GLS models.

Table 3 – Details of the selected GLS models comparing belemnite body-size (GM) with palaeotemperature ($\delta^{18}\text{O}$), carbon cycle perturbations ($\delta^{13}\text{C}_{\text{carb}}$) and volcanism (Hg/TOC), corrected for the effects of sedimentary properties (lithology and belemnite abundance). GM= geometric mean.

Supplementary figure and table captions

Table S1 – Measured body-size parameters on belemnite specimens from the Upper Pliensbachian-Toarcian interval of the Peniche section and selected environmental parameters from the literature (ESM file 1).

Table S2 – Log ratio, sample size and results of the statistical tests on the differences in the median of pairs of beds. The significance level was $p < 0.05$. U test= Mann-Whitney U test; K-S test = Kolmogorov-Smirnov test. (ESM file 2)-

Table S3 – Partitioning analysis results highlighting the relative variation fractions (proportions) of belemnites body-size explained by taxonomic composition, beds and ontogeny for the whole time-series and for the pair-to-pair analysis. Bold indicates a minimum of 90 % statistical significance level. (ESM file 3)-

Table S4 – Details of GLS regression analysis comparing belemnite ~~body-size~~ body-size and the environmental parameters: a – details of the selected GLS models, after correcting for lithology and abundance; b – Details of the selected GLS models, without correcting for lithology and abundance (raw data); c– Full list of the analysed models for assemblage and species scales, with AICc scores and delta values indicated, together with the p-values for the models with the smaller AICc score. (ESM file 4)-

Table S5 – AICc ranking of models describing the effect of palaeotemperature ($\delta^{18}\text{O}$), carbon cycle perturbations ($\delta^{13}\text{C}_{\text{carb}}$) and volcanism(Hg/TOC) on belemnites ~~body-size~~ body-size for the assemblage and species scale (*Ps. longiformis* and *P. bisulcata*), not corrected for the effects of sedimentary properties (lithology and belemnite abundance). Only the models with $\Delta < 2$ are listed. See TS4 for the full list of GLS model. (ESM file 5)-

Table S6 – Detailed values of the components of the assemblage size shift calculated with Rego et al. method (disappearance of taxa effect, appearance of new taxa effect and within-lineage effect). (see ESM file 6)-The statistical significance of each coefficient of the best model was assessed by calculating the p-value under a t approximation. Significance level = $p < 0.05$.

~~Table S7 – Micro-CT phoenix v|tome|x s 240 (Research Edition) scanner settings for the scanned belemnite specimens. The reconstruction was made with the CEDatos|x 2.4 software. Subsequent image stack processing (e.g. subsampling), as well as the measurements and volume acquisition, was derived using Studio Volume Graphics Max™ v 3.0 software (Heidelberg) (ESM file 6)-~~

Table S7 – Micro-CT phoenix v|tome|x s 240 (Research Edition) scanner settings for the scanned belemnite specimens. The reconstruction was made with the GEDatos|x 2.4 software. Subsequent image stack processing (e.g. subsampling), as well as the measurements and volume acquisition, was derived using Studio Volume Graphics Max™ v 3.0 software (Heidelberg). (ESM file 7)-

Figure S1 – Selected belemnites from the Upper Pliensbachian-Lower Toarcian of Peniche: A1 - *Bairstowius* sp. A (adult from bed P1); A2- *Bairstowius* sp. A (adult from bed P2); A3- *Bairstowius* sp. A (juvenile from bed P1); B1- *Pseudohastites longiformis* (adult from bed P4); B2 and B3 - *Pseudohastites longiformis* (juveniles from bed P4); C1 and C2- *Hastitidae* sp. indet. (from bed P5); D1 and D2 - *Passaloteuthis milleri* (adults from bed P2). The left side corresponds to lateral view and right side corresponds to dorsal/ventral view. (ESM file 8)-

Formatted: Font: Not Italic

Figure S2 – Selected belemnites from the Upper Pliensbachian-Lower Toarcian of Peniche: E1- *Parapassaloteuthis* sp. A (adult from bed P3); E2 - *Parapassaloteuthis* sp. A (juvenile from bed P3); F - *Passaloteuthis* sp. juv. (from bed P1); G1-G2 - *Passaloteuthis bisulcata* (adults from bed P5); G3 - *Passaloteuthis bisulcata* (adult from bed P2). The left side corresponds to lateral view and right side corresponds to dorsal/ventral view. (ESM file 8)-

Figure S3 – Belemnite absolute abundance (no. of belemnites collected per m²). The interrupted black line connects the mean (black dots) for the beds where more than one quadrat were analysed. Each red dot represents a quadrat of 1 m². The error bars represent the standard deviation of the mean. (ESM file 8)-

Figure S4 – Correlation matrix depicting the relation between the different environmental variables available in the literature for the Peniche section. Note that only $\delta^{18}\text{O}$, $\delta^{13}\text{C}_{\text{carb}}$, Hg/TOC, lithology and abundance were included as explanatory variables in the regression analysis due to the high collinearity among the remaining ones. See TS1 for details. (ESM file 8)-

Figure S5 – Correlation matrix depicting the relation between the different explanatory variables used in the performed regression analysis between belemnite body-size and abiotic parameters. (ESM file 8)-

Figure S6 – Belemnite ~~body-size~~body-size variation (geometric mean, GM), proportional body-size change, relative frequency of ontogenetic stages and sample size (n) across the Upper Pliensbachian-Lower Toarcian of Peniche at species scale (*P. bisulcata* and *Ps. longiformis*). *Passaloteuthis* genus was used to calculate the relative frequency of ontogenetic stages of *P. bisulcata* due to the difficulty of a species level classification of the juvenile specimens of *Passaloteuthis* genus. The error bars correspond to the 95 % confidence interval. Note that the sample size (n) corresponds to species level. (ESM file 8)-

Formatted: Font: 9 pt, Not Italic

Formatted: Font: 9 pt

Figure S7 – ~~Body-size~~Body-size variation (geometric mean, GM) and sample size (n) of Pliensbachian-Toarcian boundary-crossers (*P. bisulcata*, *Ps. longiformis* and *Hastitidae* sp. indet.) of the Peniche section. Although *Parapassaloteuthis* sp. A also crosses the Pli-Toa boundary, we have no complete specimens across this interval and therefore, no accurate estimate of ~~body-size~~body-size. (ESM file 8)-

Formatted: Font: 9 pt, Not Italic

Formatted: Font: Not Italic

Figure S8 – Relationship between the seawater palaeotemperature proxy ($\delta^{18}\text{O}$) and belemnite ~~body-size~~body-size (geometric mean, GM) during the Upper Pliensbachian-Lower Toarcian of Peniche. The grey area corresponds to the 95 % confidence interval. (ESM file 8)-

Figure S9 – Relationship between belemnite ~~body-size~~body-size (geometric mean, GM) and lithology for the studied interval in Peniche. Adjusted R²=-0.0009292; p-value= 0.4083. The grey area corresponds to the 95 % confidence interval. (ESM file 8)-

Figure S10 – Relationship between belemnite ~~body-size~~body-size (geometric mean, GM) and belemnite absolute abundance (no. of belemnites/ m²) for the studied interval in Peniche. The grey area corresponds to the 95 % confidence interval. Adjusted R²=9.356e-05; p-value=0.3105. (ESM file 8)-